# Improving radar-based rainfall nowcasting by a nearest neighbour approach: Part I – Storm Characteristics

Bora SHEHU[1], Uwe HABERLANDT[1]

[1]Institute for Hydrology and Water Resources Management, Leibniz University Hannover, Germany

*Correspondence to: Bora Shehu (shehu@iww.uni-hannover.de)*

**Abstract.**

The nowcast of rainfall storms at fine temporal and spatial resolutions is quite challenging due to the unpredictable nature of rainfall at such scales. Typically, rainfall storms are recognized by weather radar, and extrapolated in the future by the Lagrangian persistence. However, storm evolution is much more dynamic and complex than the Lagrangian persistence, leading to short forecast horizons especially for convective events. Thus, the aim of this paper is to investigate the improvement that past similar storms can introduce to the object-oriented radar based nowcast. Here we propose a nearest neighbour approach that measures first the similarity between the "to-be-nowcasted" storm and past observed storms, and later uses the behaviour of the past most similar storms to issue either a single nowcast (by averaging the 4 most similar storm-responses) or an ensemble nowcast (by considering 30 most similar storm-responses). Three questions are tackled here: i) what features should be used to describe storms in order to check for similarity? ii) how to measure similarity between past storms? and iii) is this similarity useful for object-oriented nowcast? For this purpose, individual storms from 110 events in the period 2000-2018 recognized within the Hannover Radar Range (R~115km$^2$), Germany, are used as a basis for investigation. A "leave-one-event-out" cross-validation is employed to test the nearest neighbour approach for the prediction of the area, mean intensity, the x and y velocity components, and the total lifetime of the "to-be-nowcasted" storm for lead times from +5min up to + 3 hours. Prior to the application, two importance analyses methods (Pearson correlation and partial information correlation) are employed to identify the most important predictors. The results indicate that most of storms behave similarly, and the knowledge obtained from such similar past storms helps to capture better the storm dissipation, and improves the nowcast compared to the Lagrangian persistence especially for convective events (storms shorter than 3 hours) and longer lead times (from 1 to 3 hours). The main advantage of the nearest neighbour approach is seen when applied in a probabilistic way (with the 30 closest neighbours as ensembles) rather than in a deterministic way (averaging the response from 4 closest neighbours). The probabilistic approach seems promising, especially for convective storms, and it can be further improvement by either increasing the sample size, employing more suitable methods for the predictor identification, or selecting physical predictors.

**Keywords:**

Rainfall nowcast, Lagrangian persistence, probabilistic nowcast, similar storms, nearest neighbour

**1. Introduction**

Urban pluvial floods are caused by short, local and intense rainfall convective storms, that overcome rapidly the drainage capacity of the sewer network and lead to surface inundations. These types of floods are becoming more relevant with time due to the expansion of urban areas worldwide (Jacobson, 2011; United, 2018), and the potential of such storms getting more extreme under the changing global climate (Van Dijk et al., 2014). Because of the high economical, and even human losses associated with these floods, modelling and forecasting becomes crucial for impact-based early warnings (i.e. July 2008 in Dortmund (Grünewald, 2009), August 2008 in Tokyo (A. Kato & Maki, 2009)). However, one of the main challenges in the urban pluvial flood forecasting, remains the accurate estimation of rainfall intensities at very fine scales. Since the urban area responds fast and locally to the rainfall (due to the sealed surfaces and the artificial deviation of watercourse), the Quantitative Precipitation Forecasts (QPFs) fed into the urban models should be provided at very fine temporal (1-5min) and spatial ($100m^2 - 1km^2$) scales (Berne et al., 2004). The Numerical Weather Prediction Models (NWP) are typically used in hydrology for weather forecast to several days ahead, nevertheless they are not suitable for urban modelling as they still cannot produce reliable and accurate intensities for spatial scales smaller than $10km^2$ and temporal time steps shorter than an hour (R. Kato et al., 2017; Surcel et al., 2015). Ground rainfall measurements (rain-gauges) are considered the true observation of rainfall but they are as well not adequate for QPFs because, due to the sparsity of the existing rain-gauge networks, they cannot capture the spatial structure of rainfall. Therefore, the only product useful in providing QPFs for urban pluvial floods remains the weather radar. The weather radar can measure indirectly the rainfall intensities at high spatial ($\sim 1km^2$) and temporal ($\sim 5min$) resolutions by capturing the reflected energy from the water droplets in the atmosphere. The rainfall structures and their evolution in time and space can be easily identified by the radar and hence serve as a basis for issuing QPFs at different forecast horizons. One of the main drawbacks of radar-based forecast, is that a rainfall structure has to be first identified in order to be extrapolated in the future. In other words, rainfall cannot be predicted before it has started anywhere in the region, only the movement can be predicted. As already discussed in Bowler et al., (2006) and Jensen et al. (2015), these initialization errors cause the radar forecast to be used only for short forecast horizons (up to 3 hours), and that is why are typically referred to as nowcasts. For longer lead times a blending between NWP and radar based nowcasts should be used instead (Codo & Rico-Ramirez, 2018; Foresti et al., 2016; Jasper-Tönnies et al., 2018). Nonetheless, for short forecast horizons up to 2-3h , the radar nowcast remains the best product for pluvial flood simulations as it outperforms the NWP one (Berenguer et al., 2012; Jensen et al., 2015; Lin et al., 2005; Zahraei et al., 2012).

Two approaches can be distinguished on the radar based QPFs depending on how the rainfall structures are identified, tracked and extrapolated into the future: object-oriented nowcasting (herein as object-based to avoid the confusion with the programming term) and field-based nowcasting. The object-based nowcast treats rainfall structures as objects, each object is regarded as a storm and is defined as a set of radar grid cells that moves together as a unit (Dixon & Wiener, 1993). The field-based approach considers the rainfall as an continuous field inside a given domain, and through methods like optical flow, tracks and extrapolates how the intensity is moving from one pixel to the other inside this domain (Ruzanski et al., 2011; Zahraei et al., 2012). Convective storms have been proven to have a unique movement from nearby storms (Moseley et al., 2013), thus are thought to be better nowcasted with object-based approach (Kyznarová & Novák, 2009). On the other hand, the field-based approach with an optical flow solution, tracks and extrapolates rainfall structures inside a region together as a unit with a constant velocity (Lucas & Kanade, 1981) and are considered more suitable for major scale events, i.e. stratiform storms, as they are widespread in the radar image and exhibit more uniform movements (Han et al., 2009). Even though the field-based approached has gained popularity recently (Ayzel et al., 2020; Imhoff et al., 2020) they still have trouble nowcasting convective storms. Thus, the focus in

this study is on object-based nowcasts as they are more convenient for convective storms that typically cause urban pluvial floods.

**Figure 1** illustrates the three main steps performed in an object-based nowcast: a) first the storm is identified –a group of grid cells with intensity higher than a threshold is recognized in the radar image at time $t_0$, b) the storm identified is then tracked for the time $t_0+\Delta t$ (where $\Delta t$ is the temporal resolution of the radar data) and velocities are assigned from consecutive storm objects, and finally c) the storm as lastly observed at time t (when the nowcast is issued) is extrapolated at a specific lead time (the time in the future when the forecast is needed) $t_{+LT}$, with the last observed velocity vector. This is a linear extrapolation of the storm structure in the future, considering the spatial structure and the movement of the storm as constant in time - also referred to as Lagrangian Persistence (Germann et al., 2006). Applications of such storm-based nowcast are common in literature like TITAN, HyRaTrac, Konrad etc. (Han et al., 2009; Hand, 1996; Krämer, 2008; Lang, 2001; C. E. Pierce et al., 2004).

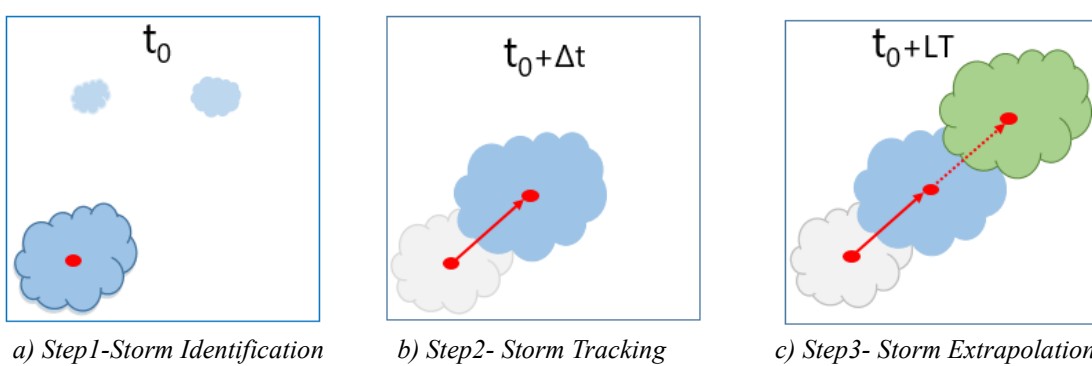

| a) Step1-Storm Identification | b) Step2- Storm Tracking | c) Step3- Storm Extrapolation |

*Figure 1 The main steps of an object-based radar nowcast. Blue indicates the current state of the storm at any time t, grey indicates the past states of the storm (at $t_0+\Delta t$), and green indicates the future states of the storm ($t_{0+LT}$) (Shehu, 2020)*

Apart from the initialization errors mentioned before, other error sources in the object-based nowcast can be attributed to storm identification, storm tracking and Lagrangian extrapolation (L. Foresti & Seed, 2015; C. Pierce et al., 2012; Rossi et al., 2015). Many works have been already conducted to investigate the role of different intensity thresholds on the storm identification, or of different storm tracking algorithms on the nowcasting results (Goudenhoofdt & Delobbe, 2013; Han et al., 2009; Hou & Wang, 2017; Jung & Lee, 2015; Kober & Tafferner, 2009). Very high intensity thresholds may be suitable for convective storms, however can cause false splitting of the storms and which can affect negatively the tracking algorithm. Thus, one has to be careful in adjusting the intensity threshold dynamically over the radar field and type of storm. Storm tracking algorithm can be improved if certain relationships are learned from past observed dataset (like a Fuzzy approach in Jung & Lee (2015) or a tree-based structure in Hou & Wang (2017)), but there is still a limit that the tracking improvement cannot surpass due to the implementation of the Lagrangian persistence (Hou & Wang, 2017). These errors due to the Lagrangian persistence are particularly high for convective events at longer lead times (past 1 hour) as the majority of convective storms dissipate within 60 minutes (Goudenhoofdt & Delobbe, 2013; Wilson et al., 1998). At these lead times, the persistence fails to predict the dissipation of these storm cells, while for shorter lead times it fails to represent the growing/decaying rate and the changing movement of a storm cell (Germann et al., 2006). For stratiform events, since they are more persistent in nature, Lagrangian persistence can give reliable results up to 2 or 3 hours lead time (Krämer, 2008). Nevertheless studies have found that for fine spatial ($1km^2$) and temporal (5min) scales, the Lagrangian Persistence can yield reliable results up to 20-30 min lead time, which is also known in the literature as the predictability limit of rainfall at such scales (Grecu & Krajewski, 2000; R. Kato et al., 2017; Ruzanski et al., 2011). In object-based radar nowcast, this predictability limit can be extended up to 1 hour for stratiform events and up to 30-

45min for convective events if a better radar product (merged with rain gauge data) is fed into the nowcast model (Shehu & Haberlandt, 2021). Past these lead times, the errors due to the growth/decay and dissipation of the storms dominate.

The predictability of convective storms can be extended, if instead of the Lagrangian persistence, one estimates these non-linear processes (growth/decay/dissipation) by utilizing storm life characteristics analysed from past observations (Goudenhoofdt & Delobbe, 2013; Zawadzki, 1973). For instance, (Kyznarová & Novák, 2009) used the CellTrack algorithm to derive life cycle characteristics of convective storms and observed that there is a dependency between storm area, maximum intensity, life phase and height of 0°C isotherm level. Similar results were also found by (Moseley et al., 2013) which concluded that convective storms show a clear life cycle with the peak occurring at 1/3 of total storm duration, a strong dependency on the temperature and increasing average intensity with longer durations. In case of extreme convective storms, earlier peaks are more obvious causing a steeper increase to maximum intensity. A later study by (Moseley et al., 2019) found that the longest and most intense storms were expected in the late afternoon hours in Germany. Thus, it is to be expected that an extensive observation of past storm behaviours can be very useful in creating and establishing new nowcasting rules (Wilson et al., 2010) that can outperform the Lagrangian persistence. An implementation of such learning from previous observed storms (with focus only on the object-based nowcast and not the field-based one) is for instance shown by (Hou & Wang, 2017) where a Fuzzy classification scheme was implemented to improve the tracking and matching of storms which resulted in an improved nowcast, and Zahraei et al. (2013) where a Self-Organizing-Maps (SOM) algorithm was used to predict the initialization and dissipation of storms on coarse scales extending the predictability of storms by 20%. These studies suggest that past observed relationships may be useful in extending the predictability limit of the convective storms. Under this context, a nearest neighbour method (k-NN) may be developed at the storm scale and used to first recognize similar storms in the past, and then assign their behaviours to the "to-be-nowcasted" storm. The nearest neighbour method has been used in the field of hydrology mainly for classification , regression or resampling purposes (e.g. Lall & Sharma (1996)) but there are some examples of prediction as well (Galeati, 1990). The assumption of this method is that similar events are described by similar predictors, and if one identifies the predictors successfully, similar events that behave similarly can be identified. For a new event, the respective response is then obtained by averaging the responses of past k – most similar storms. The k-value can be optimized by minimizing a given cost function. Because of the averaging, the response obtained, will be a new one, satisfying thus the condition that nature doesn't repeat itself, but nevertheless it is confined within the limits of the observed events (therefore is unable to predict extreme behaviours outside of the observed range).

Similar approaches are implemented in field-based nowcast (referred to as analogue events), where past similar radar fields are selected based on weather conditions and radar characteristics i.e. in NORA nowcast by (Panziera et al., 2011) mainly for orographic rainfall , or in the multi-scaled analogues nowcast model by (Zou et al., 2020). Panziera et al. 2011 showed that there is a strong dependency between air-mass stability, wind speed and direction and the rainfall patterns observed from the radar data, and that the NORA nowcast can improve the hourly nowcasts of orographic rain up to 1 hour when compared to Eulerian Persistence and up to 4 hours when compared with the COSMO2 NWP. Improvement of predictability through a multi-scaled analogues nowcast was also reported by (Zou et al., 2020), which identified neighbours first by accounting similar meteorological conditions and then the spatial information from radar data. However, both of these studies show the applicability of the method on rainfall types that tend to repeat the rainfall patterns; i.e. the orographic forcing in the case of Panziera et al. (2011) and winter stratiform events in the case of Zou et al. (2020). So far, to the authors knowledge, such application of the k-NN has not been applied for convective events. This application seems reasonable as an extension of the object-based radar nowcast, in order to treat each convective storm independently. It can be used instead of the Lagrangian persistence in step 3 in **Figure** 1-c, for the extrapolation of rainfall storms into the future. Moreover, the benefit of the k-NN application is that one can either give a single or an

ensemble nowcast; since k-neighbours can be selected as similar to a storm at hand, a probability based on the similarity rank, can be issued at each of the past storm, providing so an ensemble of responses, which are more preferred compared to the deterministic nowcast due to the high uncertainty associated with rainfall predictions at such fine scales (Germann & Zawadzki, 2004). Thus, it is the aim of this study to investigate the suitability of the k-NN application to substitute the Lagrangian Persistence in the nowcasting of mainly convective events that have the potential to cause urban pluvial floods.

We would like to achieve this by first investigating if a K-NN is able to nowcast successfully storm characteristics like Area, Intensity, Movement and Total Lifetime for different life cycles and lead times. Based on the observed dependency of the storm characteristics on the life cycle, it would be interesting to see if the morphological features are enough to describe the evolution of the convective storms. Therefore, the focus is here only of the features recognized by the radar data, and further works will include as well the use of meteorological factors. To reach our aim, the suitability of the k-NN approach is studied as an extension of the existing object-based nowcast algorithm HyRaTrac developed from Krämer (2008). Before such an application, questions that arise are I) what features are more important when describing a storm, II) how to evaluate similarity between storms and III) how to use their information for nowcasting the storm at hand. The paper is organized as follows: first in Section 2 the study area is described, following with the structure of the k-NN method in Section 3.1 where: the generation of the storm database is discussed in Section 3.1.1, the predictors selected and target variables are given in in Section 3.1.2, the methods used for predictor identification in Section 3.1.3, and different application of the k-NN in Section 3.1.4. The optimization and the performance criteria are shown in Section 3.2 followed by the results in Section 4 separated into predictors influence (Section 4.1), deterministic k-NN (Section 4.2), probabilistic k-NN performance (Section 4.3), and the nowcasting of unmatched storms (Section 4.4). Finally, the study is closed with conclusions and outlook in Section 5.

**2. Study Area and Data**

The study area is located in northern Germany, and lies within the Hannover Radar Range as illustrated in **Figure 2**. The radar station is situated at the Hannover Airport, and it covers an area with a radius of 115 km. The Hannover radar data are C-band data (single-pol) provided by German Weather Service (DWD), and measure the reflectivity at an azimuth angle of 1° and at 5 min scans (Winterrath et al., 2012). The reflectivity is converted to intensity according to Marshall-Palmer relationship with the coefficients a=256 and b=1.42 (Bartels et al., 2004). The radar data are corrected from the static clutters and erroneous beams and then converted to Cartesian Coordinate system (1 km² and 5 min) as described in

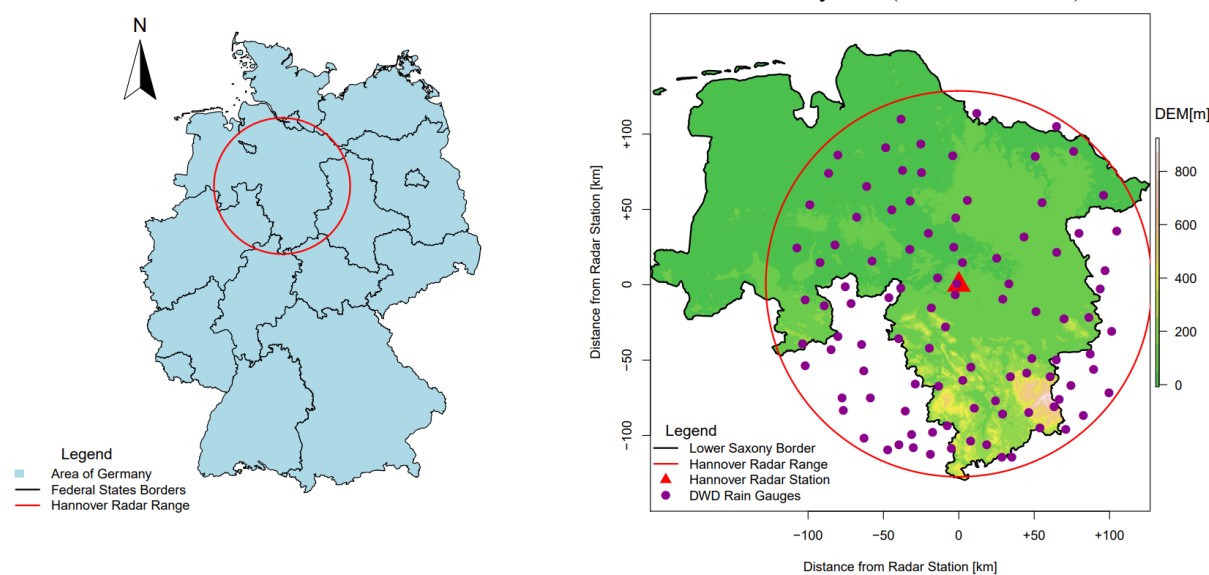

*Figure 2 The location of the study area left) within Germany and right) with the corresponding elevation and boundaries, and as well with the available recording rain gauges (purple) and radar (red) station. The DEM is short for Digital Elevation Model (adapted from Shehu and Haberlandt, 2021).*

(Berndt et al., 2014), while the rain-gauges measure the rainfall intensities at 1min temporal resolution but are aggregated to 5min time steps. Additionally, following the results from Shehu & Haberlandt (2021), a conditional merging between the radar data and 100 rain-gauge recording (see **Figure** 2 -right) with the radar range at 5 min time steps is performed. The conditional merging aims to improve the kriging interpolation of the gauge recordings by adding the spatial variability and maintaining the storm structures as recognized by the radar data. In case a radar image is missing, the kriging interpolation of the gauge recordings is taken instead.

The period from 2000 to 2018 is used as a basis for this investigation, from which 110 events with different characteristics were extracted (see Shehu & Haberlandt (2021) or Shehu (2020)). These events were selected for urban flood purposes, and contain mainly convective events and few stratiform ones. Here, rainfall events are referred to a time period when rainfall has been observed inside the radar range and at least at one rain gauge has registered an extreme rainfall volume (return period higher than 5 years) for durations varying from 5 min to 1 day. The start and the end of the rainfall event is determined when areal mean radar intensity is higher/lower than 0.05mm for more than 4 hours. Within a rainfall event many rainfall storms, at different times and locations, can be recognized. **Figure** 3-a shows a simple illustration to distinguish between the rainfall event and rainfall storm concepts employed in this study.

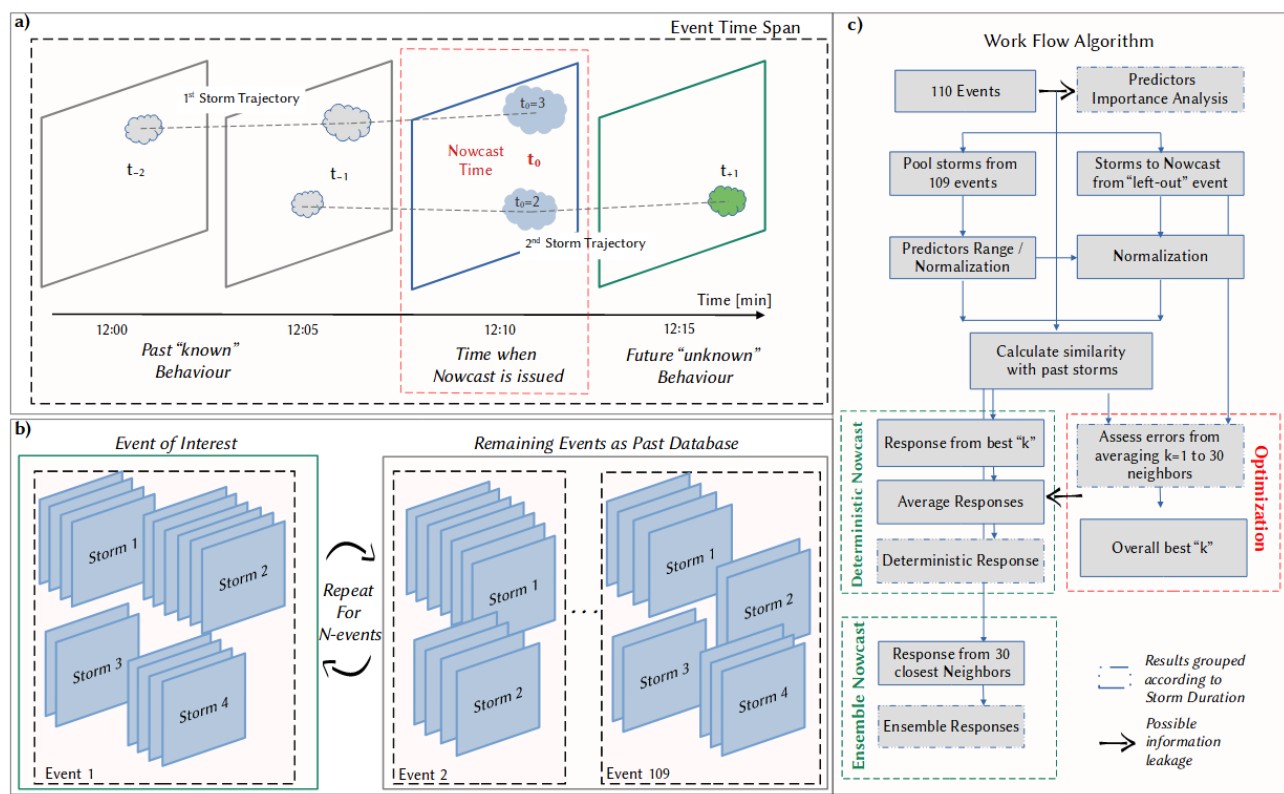

*Figure 3 Illustration of concepts and workflows in this study a) an event contains many rainfall storms inside the radar range which are tracked and nowcasted: the dashed grey lines indicate the movements of storms in space-time within the radar event and the event time span. b) The "leave-one-out-event cross-validation" – the storms of the event of interest are removed from the past database, and the nowcast of these storms is issued based on the past database. This process is repeated 110 times (once for each event). c) the workflow implemented here for the optimization and application of the k-NN approach.*

### 3. Methods

#### 3.1 Developing the k-NN model

##### 3.1.1 Generating the storm database

Each of the selected events contains many storms, whose identification and tracking was performed on the basis of the HyRaTrac algorithm in the hindcast mode (Krämer, 2008; Schellart et al., 2014). A storm is initialized if a group of spatially connected radar grid cells (> 64) has a reflectivity higher than Z=20dBz, while storms are recognized as convective – if a group bigger than 16 radar grid cells has an intensity higher than 25 dBz, and as stratiform – if a group bigger than 128 radar grid cells has an intensity higher than 20 dBz. Typically, higher values (40dBz) are used to identify the core of convective storms (as in E-Titan), but to avoid false splitting of convective storms and to test the methodology on all types of storms, these identification thresholds were kept low (following as well the studies from Moseley et al. 2013). Once storms at different time steps are recognized, they are matched as evolution of a single storm, if the centre of intensity of storm at t=0 falls within the boundary box of the storm at t-5 min. The tracking of individual storms in consecutive images is done by the cross-correlation optimization between the last 2 images (t=0 and t-5 min), and local displacement vectors for each storm are calculated. In case a storm is just recognized (the storm does not yet have previous history), then global displacement vectors based on cross-correlation of the entire radar image are assigned to them. It is usually the case, that two storms merge together at a certain time, or a single storm splits between several daughter storms. The splitting and merging of the storms is considered here if two criteria are met: a) the minimum distance between the storms that have splatted or merged is smaller than the perimeter of the merged or that-is-splitting storm, and b) the position of the centre of intensity of former/latter storms is within the boundaries of the latter/former storm.

Thus, a dataset with several types of storms is built and saved. The storms are saved with an ID based on the starting time and location, and for each time step of the storm evolution the spatial information is saved and various features are calculated. Here the features computed from the spatial information of the rainfall inside the storm boundaries at a given time step (in 5min) of the storms' life, is referred to as the "state" of the storm. A storm that has been observed for 15 minutes, consists of three "states" each occurring at a 5 min time step. For each of the storm states an ellipsoid is fitted to the intensities in order to calculate the major and minor axis and the orientation angle of the major axis. This storm database is the basis for developing the k-NN method and for investigating the similarity between storms. Some characteristics of the identified storms like duration (or also total lifetime of the storm), mean area, maximum intensity, number of splits/merges, local velocity components, and ellipsoidal features, are shown in the **Figure 4**. These storms characteristics were obtained by an hindcast analysis run of all 110 events with the HyRaTrac algorithm which resulted in around 5200 storms. The local velocities in x and y direction are obtained by a cross-correlation optimization within the storm boundary. The duration of the storm is then the lifetime of the radar pixels group as dictated by the threshold used to recognize them and the tracking algorithm that decides if the same storm is observed at continuous time steps. For more information about the tracking and identification algorithm, reader is directed to Krämer (2008).

As seen from the number of storms for each duration in **Figure 4**, the unmatched storm cells make the majority of the storms recognized. These are storms that last just 5 min (one-time step) as the algorithm fails to track them at consecutive time steps. These "storms" can either be dynamic clutter from the radar measurement, as they are characterized by small area, circular shapes (small ratio of minor and major axis) and by very high velocities, or artefacts created by low intensity thresholds used for the storm identification, or finally produced by the unrepresentativeness of the volume captured by the radar station. Another thing to keep in mind, is that merged radar are fed to the algorithm for storm recognition, and this affect the storm structures particularly when the radar data is missing. In such case, the ordinary kriging interpolation of rain gauges is given as input, which is well known to smooth the spatial distribution of rainfall

and hence resulting in a short storm characterized by a very large area. Since the "not" matched storms can either be
dynamic clutter or artefacts, they are left outside of the k-NN application. Nonetheless, they are treated shortly in section
4.5.

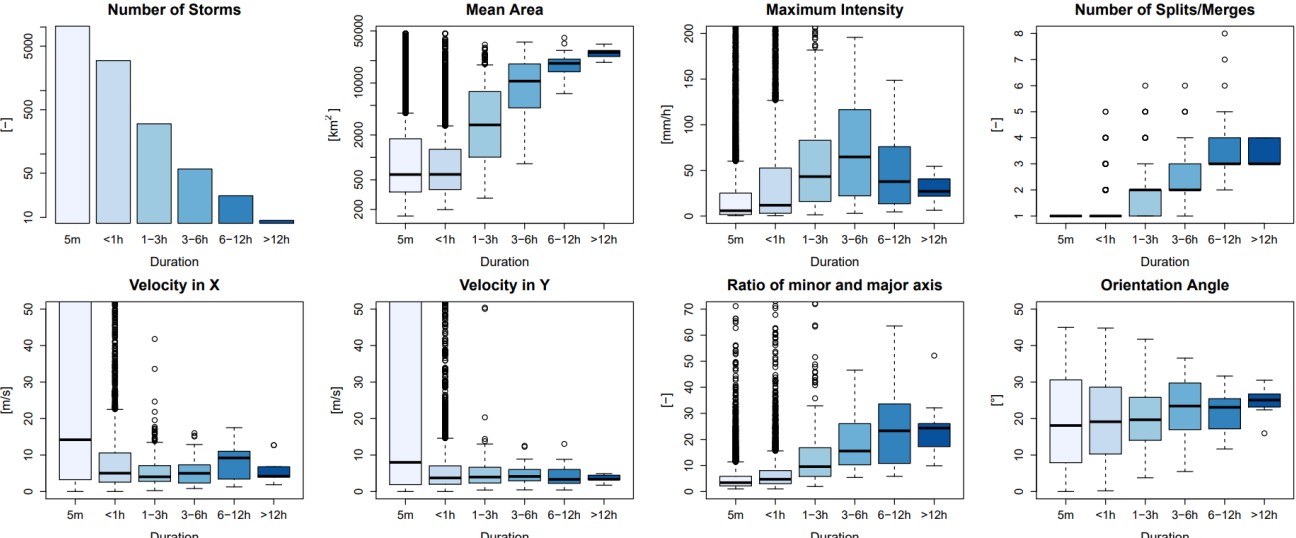

*Figure 4 Different properties of the storms recognized from 110 events separated into 6 groups according to their duration (shown in different shades of blue).*

Apart from the unmatched storms, the majority of the remaining storms are of convective nature: storms with
short duration (shorter than 6 hours), high intensity and low areal coverage. Here two types of convective storms are
distinguished: local convective with very low coverage (on average lower than 1000 km$^2$) and low intensity (on average
~ 5 mm/h), and mesoscale convective which are responsible for floods (with intensity up to 100 mm/h or more) and have
a larger coverage (on average lower than 5000 km$^2$). The stratiform storms characterized by large area, long duration and
low intensities, as well as meso-$\gamma$ scale convective events with duration up to 6 hours, are not very well represented by
the dataset as only a few of them are present in the selected events (respectively circa 20 and 50 storms). Therefore, it is
to be expected for the k-NN approach not to yield very good results for such storms due to the low representativeness.
From the characteristics of the storms illustrated in **Figure 4**, it can be seen that for stratiform storms that live longer than
twelve hours the variance of the characteristics is quite low (when compared to the rest of the storms) which can either
be attributed to the persistence of such storms or to the low representativeness in the database. Even though the data size
for stratiform is quite small, the k-NN may still deliver good results as characteristics of such storms are more similar.
Nevertheless, the stratiform storms are typically nowcasted well by the Lagrangian persistence (specially by a field-
oriented approach) as they are wide-spread and persistent. Hence the value of the k-NN is primarily seen for convective
storms and not for stratiform ones.
*3.1.2 Selecting features for similarity and target variables*

At first storms are treated like objects that manifest certain features (predictors) like area, intensity, lifetime etc.,
at each state of the storms' life until the storm dissipates (and the predictors are all set to zero). The features of the objects
are categorized into present and past features, as illustrated in **Figure 5** (shown respectively in blue and grey). The present
features describe the current state of the storm at the time of nowcast (denoted with $t_0$ in **Figure 5**), and are calculated
from one state of the storm. To compute certain features, an ellipsoid is fitted to each state of the storm. The past features,
on the other hand, describe the predictors of the past storm states (denoted with $t_{-1}$, $t_{-2}$ in **Figure 5**) and their change over
the past life of the storm. For example, the average area from time $t_{-2}$ to $t_{-1}$ is a past feature. A pre-analysis of important
predictors showed that the average features over the last 30 minutes are more suitable as past predictors than the averages
over last 15 or 60 min or than the calculation of past changing rates. Therefore, averages over past 30 minutes are
computed here:
$\qquad P_{30} = \sum_{i=t_0}^{t-30min} P_i / 7$ , $\hfill (1)$
where $P_i$ is the predictors value at time $i$, and $P_{30}$ the average value of the predictor over last 30min. In case of missing
values, the remaining time steps are used for averaging. The selected features (both present and past) that are used here
to describe storms as objects, and hence tested as predictors, are shown in **Table 1**. The present features help to recognize
storms that are similar at the given state when the nowcast is issued (blue storm in **Figure 5**) and the past ones give
additional information about the past evolution of the storm (average of grey storms in **Figure 5**). The aim of these features
is to recognize the states of previously observed storms that are most similar to the current one (shown in blue in **Figure
5**) of the "to-be-nowcasted" storm. Once the most similar past storm states are recognized, their respective future states
at different lead times can be assigned as the future behaviour (shown in green in **Figure 5**) of the current state of the "to-
be-nowcasted" storms. Since the storms are regarded as objects with specific features, future behaviours at different lead
times are determined by four target variables: area ($A_{+LT}$), mean intensity ($I_{+LT}$) and velocity in X ($Vx_{+LT}$) and Y ($Vy_{+LT}$)
direction. Additionally, the total lifetime of the storm is considered as a fifth target ($L_{tot}$). Theoretically, the total lifetime
is predicted indirectly when any of the first four targets is set to zero, however here it is considered as an independent
variable in order to investigate if similar storms have similar lifetime durations.

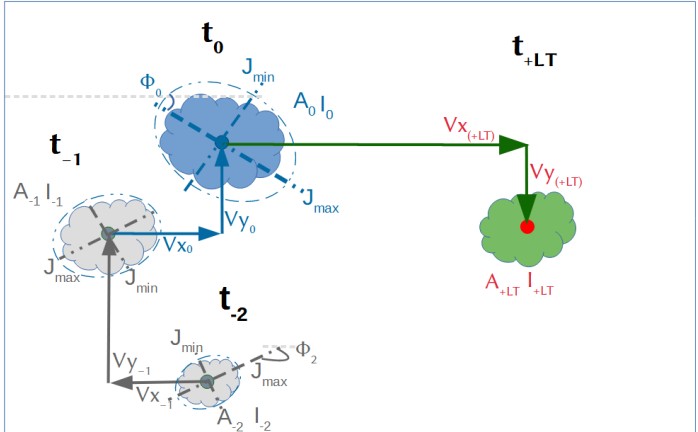

**Figure 5** *The features describing the past (grey) and present (blue) states of the storm used as predictors to nowcast the future states of the storm (green) at a specific lead time ($T_{+LT}$) that are described by 4 target variables (in red). The nowcast is issued at time $t_0$. A full description of these predictors and target variables is given in **Table 1**.*

$\qquad$ For each state of each observed storm in the database, the past and present features of that state with its' respective
future states of the five target variables from +5min to +180min (every 5 min) lead times are saved together and form the
predictor-target database that is used for the development of the k-NN nowcast model. A summary of the predictors and
target variables calculated per state is given in **Table 1**. Before optimizing and validating the k-NN method (advise **Figure
3- c**), an importance analysis is performed for each of the target variables in order to recognize the most important
predictors. As the predictors have different ranges, prior to the importance analysis and the k-NN application, they are
normalized according to their median and range between the 0.05 and 0.95 quantiles:
$\qquad normP_i = \frac{P_i - Q_{Pi}^{0.5}}{Q_{Pi}^{0.95} - Q_{Pi}^{0.05}}$ , $\hfill (2)$
where $P$ is the actual value, *normP* the normalized value, and $Q_{Pi}^{0.5}, Q_{Pi}^{0.95}, Q_{Pi}^{0.05}$ the quantiles 0.5, 0.05 and 0.95 of the $i^{th}$
predictors' vector. The reason why these quantiles were used for the normalization instead of the typical mean and
maximum to minimum range, is that some outliers are present in the data. For instance, very high and unrealistic velocities
are present in some convective storms where the tracking algorithm fails to capture adequate velocities (Han et al., 2009).
Thus, to avoid the influence of these outliers, the given range is employed.

### 3.1.3 Selection of most relevant predictors

The application of the k-NN method can be relevant if there is a clear connection between the target variable and
the features describing this target variable. For instance, in the case of Galeati (1990), a physical background backed up
the connection between target variable (discharge) and the features (daily rainfall volume and mean temperature). In the
case of the storms at such fine temporal and spatial scales, due to the erratic nature of the rainfall itself, there are no
physical related information that can be extracted from radar data. Different features of the storm itself can be investigated
for their importance to the target variable. Nevertheless, the identification of such features (referred here as predictors) is
difficult because it is bounded to the set of the available data and the relationships considered. Commonly a strong Pearson
correlation between the predictors selected and the target variable is used as an indicator of a strong linear relationship
between them. Here, the Pearson correlation absolute values are used directly as predictors weights in the k-NN
application. However, the relationship between predictors and target variables may still be of non-linear nature, thus
another predictor importance analysis should be advised when selecting the predictors. Sharma & Mehrotra (2014)
proposed a new methodology, designed specifically for the k-NN approach, where no prior assumption about the system
type is required. The method is based on a metric called the Partial Information Correlation and is computed from the
Partial Information as:
$$PIC = \sqrt{(1 - \exp(-2PI))} \ \ with \ \ PI = \int f_{X,P|Z}(x,p|z) \log\left[\frac{f_{X|Z,P|Z}(x,p|z)}{f_{X|Z}(x|z) \, f_{P|Z}(p|z)}\right] dx dp dz \ \ , \tag{3}$$
where *PIC* is the Partial Information Correlation, *PI* is the Partial Information which represents the partial dependence of
$X$ on $P$ conditioned to the presence of a predictor $Z$. The Partial Information itself is a modification of the Mutual
Information in order to measure partial statistical dependency between the predictors ($P$) and the target variable ($X$), by
adding predictors one at a time ($Z$) (step-wise procedure). The evaluation of PIC needs a pre-existing identified predictor
from which the computation can start. If the pre-defined predictor is correctly selected, then through the Equation (3), the
method is able to recognize and leave out the new predictors which are not related to the response and which don't bring
additional value to the existing relationship between the current predictors and target variable. Relative weights for the k-
NN regression application can be derived for each predictor, as a relationship between the PIC metric and the associated
partial correlation:
$$\alpha_j = PIC_{X,Zj|Z(-j)} \frac{S_{X|Z(-j)}}{S_{Zj|Z(-j)}} , \tag{4}$$
where $X$ is the target response, Zj is the added predictor from the step-wise procedure, Z(-j) previous predictor vector
excluding the predictor Zj, $S_{X|Z(-j)}$ the scaled conditional standard deviations between target (x) and predictor vector Z(-j),
$S_{Zj|Z(-j)}$ the scaled conditional standard deviations between the additional predictor (Zj) and the first predictor vector Z(-j),
and the $\alpha_j$ denotes the predictors weight. The R package NPRED was used for the investigation of the PIC derived
importance weights (Sharma et al., 2016).
Here in this study, these two importance analyses are used to determine the most important predictors and their
respective weights in the k-NN similarity calculation. For each target variable the most important predictor identified from
Pearson Correlation, is given to the PIC metric as the first predictor. The analysis is complex due to the presence of several
predictors, 38 states of future behaviour for each target variable (for each 5min between +5min to +180 min lead times),
and different nowcast times; the weights were calculated first for three lead times +15min, +60min and +180 min, and for
three storm groups separated according to their duration <60min, 60min-180min, and > 3 hours. Here the average weights
over these groups and lead times are calculated and used as a reference for each importance analysis. The k-NN errors
with these average weights are compared in Section 4.1.

*Table 1 List of all the past and present features of the storms that are investigated for their importance as predictors,*
*and the respective target variables calculated for different lead times.*

| | Features | Symbol |
|---|---|---|
| **Present Features** | number of storm cells within the storm region | Cells [-] |
| | current storm lifetime at time of nowcast | $L_{now}$ [min] |
| | area of the storm | A [km$^2$] |
| | mean spatial intensity | $I_{ave}$ [mm/h] |
| | maximum spatial intensity | $I_{max}$ [mm/h] |
| | standard deviation of the spatial intensities | $I_{sd1}$ [-] |
| | standard deviation of intensities groups inside the storm | $I_{sd2}$ [-] |
| | global velocity of the entire radar image | $V_g$ [m/s] |
| | x and y component of the local velocity of the storm region | $V_x$, $V_y$ [m/s] |
| | major and minor axis of the ellipsoid and their ratio | $J_{max}$, $J_{min}$[km], $J_r$ [-] |
| | orientation angle of the major axis of the ellipsoid | $\Phi$ [°] |
| **Past Features** | average area over the last 30 min of storm existence | $A_{30}$ [km$^2$] |
| | average mean intensity over the last 30 min of storm existence | $Iave_{30}$ [mm/h] |
| | average maximum intensity over the last 30 min of storm existence | $Imax_{30}$ [mm/h] |
| | average standard deviation of intensity over the last 30 min of storm existence | $Isd1_{30}$ [-] |
| | average standard deviation of intensity groups over the last 30 min of storm existence | $Isd2_{30}$ [-] |
| | average global velocity over the last 30 min of storm existence | $Vg_{30}$ [m/s] |
| | average x and y component of the local velocity over the last 30 min of storm existence | $Vx_{30}$, $Vy_{30}$ [m/s] |
| | average value of the major and minor axis of the ellipsoid and their ratio over the last 30 min of storm existence | $Jmax_{30}$, $Jmin_{30}$ [km] $Jr_{30}$ [-] |
| | average major axis orientation of the ellipsoid over the last 30 min of storm existence | $\Phi_{30}$ [°] |
| **Target Variables** | Total lifetime of the storm | $L_{tot}$ [min] |
| | Estimated Area and Intensity at LT from +5min to +180min | $A_{+LT}$ [km$^2$], $Iave_{+LT}$ [mm/h], |
| | Estimated Velocity X and Y at LT from +5min to +180min | $Vx_{+LT}$, $Vy_{+LT}$ [m/s] |

### 3.1.4 Developing the k-NN structure

The structure of the proposed k-NN approach at the storm scale is illustrated at **Figure 6** - left) the current "to-
be-nowcasted" storm is shown, while at – right) the past observed storms. First in Step 1, the Euclidean distance between
the most important predictors (either present or past predictors), of past storm states and the current one is calculated to
identify the most-similar states of the past storms (distance between the blue shapes at left and right side of **Figure 6**):

$$E_d = \sqrt{\sum_{i=1}^{N} w_i \cdot (X_i - Y_i)^2} , \qquad\qquad (5)$$

where $w$ is the weight of the respective $i^{th}$ predictor as dictated by the importance analysis (results are shown in **Table 3**),
$X$ the predictor of the "to-be-nowcasted" storm, $Y$ the predictor of a past observed storm, $N$ the total number of predictors
used and $E_d$ the Euclidian distance between the "to-be-nowcasted" and a past observed storm. The assumption made here
is that the smaller the distance, the higher the similarity of future behaviour between the selected storms and the "to-be-
nowcasted" storm. Therefore, in **Step 2** these distances are ranked in an ascending order and 30 past storm states with the

smallest distance are selected (***Step 3***). Once the similar past storm states have been recognized (the blue-shape in **Figure 6** - right), the future states of these storms (the green-shapes in **Figure 6** - right, each for a specific lead time from the occurrence of the selected similar blue-state), are treated as future states (the green-shape in **Figure 6** - left) of the "to-be-nowcasted" storm. In ***Step 4***, either a single (deterministic) or an ensemble (probabilistic) nowcast is issued. If a single nowcast is selected, then the green-instances of the k-neighbours are averaged with weights for each lead time:

$$R_{new} = \sum_{i=1}^{k} Pr_i \cdot R_i \,, \tag{6}$$

where $k$ is the number of neighbours obtained from optimization, $R_i$ and $Pr_i$ (from Equation 7) are respectively the response and weight of the $i^{th}$ neighbour and the $R_{new}$ the response of the "to-be-nowcasted" storm as averaged from $k$ neighbours. The response $R$ refers to each of the 5 target variables: Area, Intensity, Velocity in X and Y direction, and Total Lifetime. Contrary, if a probabilistic nowcast is selected, 30- members ensembles are selected from the closest 30 storms where each member is assigned a probability according to the rank of the respective neighbour storm:

$$Pr_i = \frac{(1/Rank_i)}{\sum_{i=1}^{k}(1/Rank_i)} \,, \tag{7}$$

where $k$ is the selected number of neighbours and *Rank* and *Pr* are respectively the rank and the probability weights of the $i^{th}$ neighbour/ensemble member. An ensemble member is then selected randomly based on the given probability weights. These probability weights calculated here are as well used for computation of the single nowcast in Equation (6).

Since the performance of the single k-NN nowcast is highly dependent on the number of k – neighbours used for the averaging, a prior optimization should be done in order to select the right k-neighbours that yield the best performance (as illustrated in **Figure 3**-c). The application of the k-NN can either be done per each target variable independently, or for all target variables grouped together. In the first approach, the dependency of the target variables between one another is not assured, they are predicted independently from one another. This is referred here as the target-based k-NN and is denoted in the results as VS1. The main advantage of this application is that, since the relationship between the target

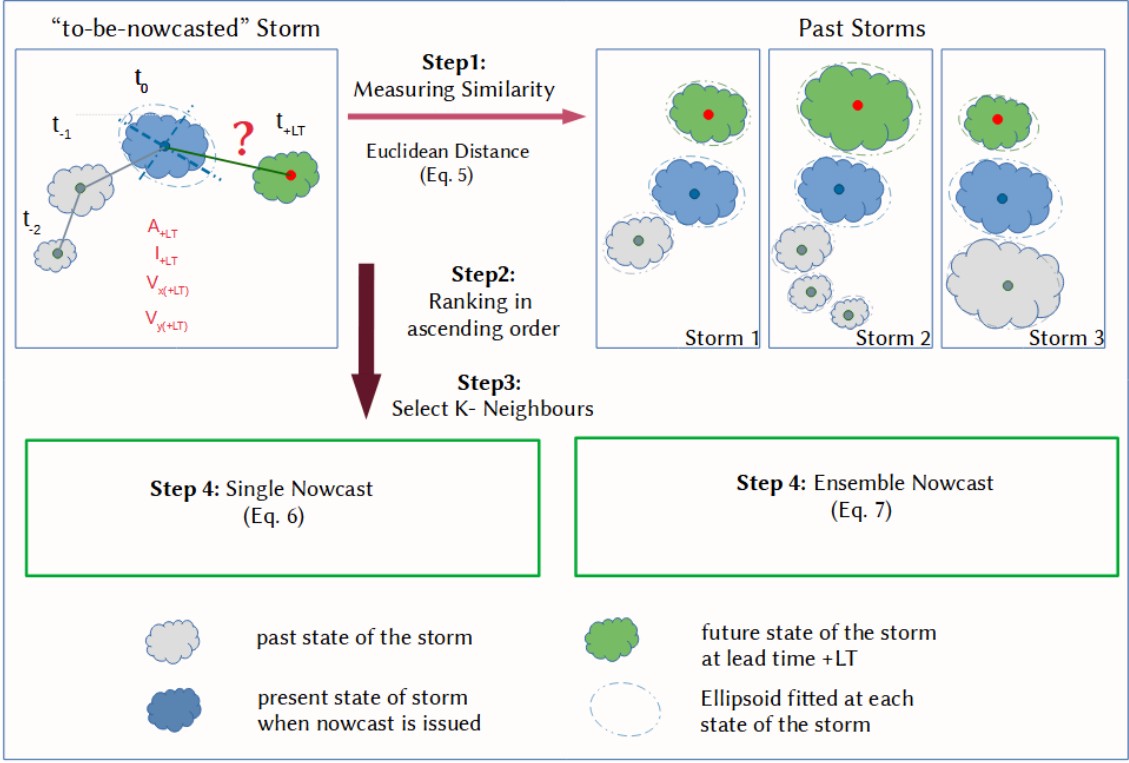

**Figure 6** *The main steps involved in the k-NN based nowcast with the estimation of similar storms (Step 1 to 3) and assigning the future responses of past storm as the new response of the "to-be-nowcasted" storm either in a deterministic nowcast (Step4-left) or in a probabilistic nowcast (Step4-right).*

variables are not kept, new storms can be generated. Theoretically, the predicted variables should have a lower error since
the application is done separately per each variable, nevertheless this approach doesn't say much if similar storms behave
similarly. Therefore, it is used here as a benchmark for best possible optimization that can be reached by the k-NN with
the current selected predictor set. In the second approach, the relationships between target variables as exhibited by
previous storms are kept. The storm structure and the relationship between features are maintained as observed and thus
the question if similar storms behave similarly can be answered. This is referred here as the storm-based k-NN and is
denoted in the results as VS2. In this study the two approaches are used (respectively called VS1 and VS2) to understand
the potential and the actual improvement that the k-NN can bring to the storm nowcast.
**3.2 Application of the k-NN and performance assessment**
*3.2.1 Optimizing the deterministic k-NN nowcast*
The optimization of the k-NN is done based on the 5189 storms extracted from 110 events on a "leave-one-out"
cross-validation. Since the "not" matched storms can either be dynamic clutter or artefacts of the tracking algorithm, they
are left outside of the k-NN optimization and validation. The assumption is here that an improvement of the radar data or
tracking algorithm would eliminate the "not" matched storms, hence the focus is only on the improvement that the k-NN
can introduce to the matched storms. "Leave-one-event-out" cross-validation means here that the storms of each event
have to be nowcasted by considering as a past database the storms from the remaining 109 events (a detailed visualization
is given in Figure 3-b). The objective function is the minimization of the mean absolute error (Equation 8) and of the
absolute mean error (Equation 9) between predicted and observed target variables at lead times from +5min to +180 min:
$$MAE_{target} = \sum_{i=1}^{N} (|Pred_{i,+LT} - Obs_{i,+LT}|)/N \,,$$
(8)

$$ME_{target} = |\sum_{i=1}^{N} (Pred_{i,+LT} - Obs_{i,+LT})/N| \,,$$
(9)

where the *Pred* is the predicted response, *Obs* the observed response for the $i^{th}$ storm, $+LT$ the lead time and *N* the number
of storms considered inside an event. The results of the storms' nowcast are also dependent on the nowcast time in respect
to the storms' life (time step of the storm existence when the nowcast is issued – refer to Figure 3-a). If the nowcast time
is 5min, only the present predictors are used for the calculation of storm similarity, and as higher the nowcast time, as
more predictors are available for the similarity calculation. It is expected for the nowcast to perform worse at the first
5min of the storm existence, as the velocities are not assigned properly to the storm region and the past predictors are not
yet calculated. Therefore, the optimization is done separately for three different groups of nowcast times, in order to
achieve a proper application of the k-NN model: Group 1 – Nowcast issued at $1^{st}$ timestep of storm recognition, Group 2
– Nowcast issued between 30min to 1 hour of storm evolution, and Group 3 – Nowcast issued between 2 and 3 hours of
storm evolution. The k-number with the lowest absolute error averaged over all the events for most of the lead times (as
median of MAE from Equation (9) and ME from Equation (9) over all events) is selected as a representative for the
deterministic nowcast.
*3.2.2 Validating the k-NN deterministic and probabilistic nowcast*
Once the important predictors are identified and the k-NN has been optimized, the performance of both
deterministic and probabilistic k-NN is assessed also in a "leave-one-event-out" cross-validation mode. Two performance
criteria are used to assess the performance:
i) absolute error per lead time and target variable computed for each event and a specific selected nowcast time
$$MAE_{target} = \sum_{i=1}^{N} (|Pred_{i,+LT} - Obs_{i,+LT}|)/N \,,$$
(10)

where the *Pred* is the predicted response, *Obs* the observed response for the $i^{th}$ storm, $+LT$ the lead time and *N* the number
of storms considered inside an event.
ii) the improvement (%) per each lead time and target variable that the k-NN approach introduces to the nowcast
(for a specific selected nowcast time) when compared to the Lagrangian persistence in object-based approach;
$$Error_{impr} \,[\%] = 100 \cdot \frac{(|Error_{ref}| - |Error_{new}|)}{|Error_{ref}|}, \tag{11}$$
where the $Error_{new}$ is the event error manifested by the k-NN, the $Error_{ref}$ the event error manifested by the Lagrangian
persistence and the $Error_{impr}$ the improvement in reducing the error per each lead time. For improvements higher than
100% or lower than -100%, the values are reassigned to the limits respectively 100% and -100%. Here the Lagrangian
persistence refers to as persistence of the storm characteristics (Area, Intensity, Velocity in X and Y Direction) as last
observed and constant for all lead times.
For the probabilistic approach, the Continuous Rank Probability Score (CRPS) as shown in Equation (12) is computed.
$$CRPS(F, y) = \int_{-\infty}^{\infty} (F(x) - 1\{y \le x\})^2 dx = E_F|Y - y| - \frac{1}{2} E_F|Y - Y'| \tag{12}$$
where $F$ is a probabilistic forecast, $y$ the observed value, $Y$ and $Y'$ independent random variables with CDF of $F$ and finite
first moment $E$ (Gneiting & Katzfuss, 2014). The CRPS is a generalization of the mean absolute error, thus if a single
nowcast is given, it is reduced to the mean absolute error (Equation 10). This enables a direct comparison between the
probabilistic and deterministic nowcast and to investigate the advantages of the probabilistic one. As in Equation (8), the
values obtained in Equation (10), (11) and (12) are averaged per each of the 110 events.
As stated earlier the results depend on the nowcast time and also storm duration (in regard to available storms).
Therefore, the performance criteria for both k-NN nowcasts were computed separately for different storm durations and
nowcast times as illustrated in **Table 2**. It is important to mention as well, that since one event may contain many storms
of similar nature, when leaving one event out for the cross-validation, the number of available storms is actually lower
than the numbers given in **Table 2**. This is particularly affecting the performance of the storms longer than 6 hours, as the
"leave-one-event-out" cross-validation leaves fewer available storms for the similarity computation. Lastly, it is important
to notice, that the performance criteria can be calculated even for nowcast times longer than the storm lifetime, if the
nowcast fails to capture the dissipation of the storms. In this case, Area, Intensity, Velocity in X and Y Direction are
compared against zero, while the Total Lifetime against the total observed lifetime of the storms.

*Table 2 The selected storm durations and nowcast times for the performance calculation of the deterministic and probabilistic nowcast and the respective number of storms for each case.*

| Storm living less than 30 min | | Storms living within 0.5 - 3 hours | | Storms living longer than 3 hours | |
|---|---|---|---|---|---|
| Nowcast Time | No. Storms | Nowcast Time | No. Storms | Nowcast Time | No. Storms |
| 5 min | 4106 | 5 min | 994 | 5min | 89 |
| 15 min | 2265 | 1h | 370 | 2h | 89 |
| 30 min | 271 | 3h | 6 | 6h | 33 |

## 4. Results:

### 4.1 Predictors Importance Analysis

**Table 3** illustrates the results of the two important analysis methods (Pearson correlation and partial information correlations - PIC) for each of the target variable and their average over the 5 variables. The stronger the shade of the green colour, the more important is the predictor for the target variable. The weights given here are averaged from the weights calculated at three different lead times and storm durations (see **Appendix 11.1** and **11.2** for more detailed information about the calculated weights). First the Pearson Correlation weights are advised for the identification of the most important predictors. From the results it is clear that the autocorrelation has a higher influence, as the target variables are mostly correlated with their respective past and present values. This influence logically is higher for the shorter lead times and smaller for the longer lead times. For longer lead times the importance of other predictors, that are not related directly with the target variable, increases. Similar patterns can be observed among the Area, Intensity and Total Lifetime target variables, indicating that these three variables may be dependent on each other, and on similar predictors like: current lifetime, area, standard deviation of intensity, the major and minor ellipsoidal axis and the global velocity. This conclusion agrees well with the life cycle characteristics of convective storms reported in the literature review. On the other hand, are the velocity components, which seem to be highly dependent on the autocorrelation and slightly correlated to area and ellipsoidal axes. It has to be mentioned that apart for the standard deviation intensities also the mean, median, and maximum spatial intensities were investigated. Nevertheless, it was found that the $I_{sd1}$ and $I_{sd2}$ had the higher correlation weights, and since there is a high collinearity between these intensity predictors, they were left out of the predictor's importance analysis.

*Table 3 Strength of relationship between the selected predictors and the target variables averaged for three lead times and storm duration groups (original weights can be seen in the Appendix 11.1 and 11.2) based on two predictors identification methods: upper –correlation, and lower –PIC weights. The green shade indicates the strength of the relationship: with 0 for no relationship at all, and 1 for highest dependency.*

| Method | Target | Present Predictors | | | | | | | | | | | | Past Predictors - averaged from last 30 min | | | | | | | | | |
|---|---|---|---|---|---|---|---|---|---|---|---|---|---|---|---|---|---|---|---|---|---|---|---|
| | | Cells | $L_{now}$ | A | Isd1 | Isd2 | Vg | Vx | Vy | Jmax | Jmin | Jr | Φ | $A_{30}$ | $Isd1_{30}$ | $Isd2_{30}$ | $Vg_{30}$ | $Vx_{30}$ | $Vy_{30}$ | $Jmax_{30}$ | $Jmin_{30}$ | $Jr_{30}$ | $Φ_{30}$ |
| Pearson Correlation | A | 0.09 | 0.18 | 0.67 | 0.15 | 0.48 | 0.05 | 0.00 | 0.00 | 0.50 | 0.49 | 0.09 | 0.00 | 0.65 | 0.17 | 0.00 | 0.07 | 0.00 | 0.06 | 0.51 | 0.49 | 0.12 | 0.00 |
| | I | 0.00 | 0.07 | 0.11 | 0.36 | 0.14 | 0.04 | 0.00 | 0.00 | 0.12 | 0.12 | 0.00 | 0.04 | 0.10 | 0.33 | 0.13 | 0.00 | 0.00 | 0.05 | 0.12 | 0.11 | 0.05 | 0.04 |
| | Vx | 0.00 | 0.00 | 0.10 | 0.02 | 0.04 | 0.16 | 0.21 | 0.00 | 0.08 | 0.00 | 0.00 | 0.03 | 0.09 | 0.00 | 0.00 | 0.18 | 0.28 | 0.00 | 0.09 | 0.00 | 0.00 | 0.00 |
| | Vy | 0.00 | 0.05 | 0.00 | 0.00 | 0.05 | 0.00 | 0.00 | 0.15 | 0.04 | 0.00 | 0.00 | 0.00 | 0.00 | 0.00 | 0.05 | 0.00 | 0.04 | 0.22 | 0.05 | 0.04 | 0.00 | 0.00 |
| | Ltot | 0.00 | 0.11 | 0.36 | 0.10 | 0.22 | 0.09 | 0.00 | 0.00 | 0.22 | 0.20 | 0.05 | 0.05 | 0.34 | 0.00 | 0.21 | 0.10 | 0.00 | 0.00 | 0.22 | 0.20 | 0.08 | 0.07 |
| | Average | 0.00 | 0.08 | 0.25 | 0.13 | 0.18 | 0.07 | 0.10 | 0.10 | 0.19 | 0.16 | 0.05 | 0.04 | 0.24 | 0.10 | 0.08 | 0.07 | 0.10 | 0.10 | 0.19 | 0.17 | 0.05 | 0.02 |
| Partial Information Correlation | A | 0.00 | 0.08 | 0.15 | 0.00 | 0.00 | 0.22 | 0.00 | 0.00 | 0.00 | 0.00 | 0.00 | 0.00 | 0.01 | 0.00 | 0.00 | 0.33 | 0.00 | 0.07 | 0.00 | 0.00 | 0.33 | 0.00 |
| | I | 0.00 | 0.00 | 0.00 | 0.00 | 0.00 | 0.00 | 0.00 | 0.00 | 1.00 | 0.00 | 0.00 | 0.00 | 0.00 | 0.00 | 0.00 | 0.00 | 0.00 | 0.00 | 0.00 | 0.00 | 0.00 | 0.00 |
| | Vx | 0.00 | 0.00 | 0.00 | 0.00 | 0.00 | 0.00 | 0.00 | 0.00 | 0.00 | 0.00 | 0.00 | 0.00 | 0.00 | 0.00 | 0.00 | 0.00 | 1.00 | 0.00 | 0.00 | 0.00 | 0.00 | 0.00 |
| | Vy | 0.00 | 0.00 | 0.00 | 0.00 | 0.00 | 0.00 | 0.00 | 0.00 | 0.00 | 0.00 | 0.00 | 0.00 | 0.00 | 0.00 | 0.00 | 0.00 | 0.00 | 1.00 | 0.00 | 0.00 | 0.00 | 0.00 |
| | Ltot | 0.00 | 0.15 | 0.13 | 0.00 | 0.00 | 0.24 | 0.00 | 0.00 | 0.00 | 0.00 | 0.00 | 0.00 | 0.00 | 0.00 | 0.00 | 0.33 | 0.00 | 0.00 | 0.00 | 0.11 | 0.33 | 0.00 |
| | Average | 0.00 | 0.05 | 0.06 | 0.00 | 0.00 | 0.09 | 0.00 | 0.00 | 0.20 | 0.00 | 0.00 | 0.00 | 0.00 | 0.00 | 0.00 | 0.13 | 0.20 | 0.01 | 0.20 | 0.02 | 0.13 | 0.00 |

The application of the PIC analyses requires that the most important predictors should be introduced to the analysis first. Hence based on the Pearson correlation values from **Table 3** the following most important predictors were selected: Area –A (as maximum correlation value from first row), Intensity –$PI_{sd1}$ (as maximum correlation value from second row), - Velocity X – $Vx_{30}$ (as maximum correlation value from third row), Velocity Y –$Vy_{30}$ (as maximum correlation value from fourth row), Total Lifetime – A (as maximum correlation value from fifth row). The results of the PIC analysis are shown in the lower row of **Table 3** and **Appendix 11.2**. For storm duration lower than 3 hours, where a lot of zeros are present, the PIC methods seems to be unable to converge to stable results or to identify important predictors. For the intensity and velocity components, the PIC identifies only 1 important predictor which, in the case of the Intensity and Velocity in the Y direction, does not correspond with the most important predictor fed first in the analysis. Contrary

for Total Lifetime and Area, only for storms that last longer than 3 hours, the method is able to converge and give the
most important predictors; for Area - A, Vg, past $Vy_{30}$ and the $L_{now}$, while for Total Lifetime - A, $Vel_g$, $L_{now}$ and $Jmin_{30}$. At
the moment it is unclear why the PIC method is unable to perform well for all of the target variables and storm groups.
One reason might be that only the Area and Total Lifetime are dependent on the chosen target variables. Another most
probable reason might be that for the other target variables the heavy-tail of the probability distribution and the high zero
sample size may influence the calculation of the joint and mutual probability distribution. The Total Lifetime is an easier
target to be analysed, which means the values are not zero and its distribution is not as heavy tailed as the distribution of
the other variables. The other variables, depending on the lead time, have more zeros included and have an asymptotic
density function. It seems that, whenever zeros are not present, like in the case of storms lasting longer than 3 hours, the
PIC is able to represent quite well the important predictors. However, the reason why this method is performing poorly
for the application at hand, even though developed specifically for the k-NN application, is not completely understood
and is not investigated further on for the time being since it is outside the scope of this paper.

Overall, the results from the Pearson correlation seem more robust and stable (throughout the lead times and
storm groups) than the PIC method (refer to **Appendix 11.1** and **11.2**); the importance weights increase with the lifetime
of the storm and decrease with higher lead time. These behaviours are expected as with increasing lead time the
uncertainty becomes bigger and with increasing lifetime the storm dynamic becomes more persistent (due to the large
scales and the stratiform movements involved). Moreover, the important predictors do not change drastically from one
lead time or storm group to the other, as seen in the PIC. Therefore, the predictors estimated from the correlation with the
given weights in **Table 3** are used as input to the k-NN application. In order to make sure that the predictor set from the
Pearson correlation was the right one, the improvement in the single k-NN training error of using these predictors instead
of the ones from PIC are shown in **Figure 7**. The results shown in this figure are computed according to the Equation (11)
(where "new" is k-NN with correlation weights, and "ref" is the k-NN with PIC weights) for the target-based k-NN
approach (solid lines) and storm-based k-NN approach (dashed lines) and are averaged for three groups of nowcast times
as indicated in the optimization of k-NN (Section 3.2.3) and as well in the legend of **Figure 7**.

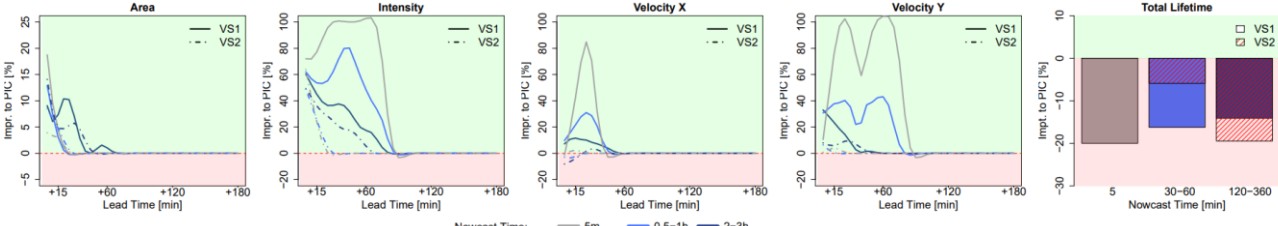

***Figure 7*** *The median Mean Absolute Error (MAE) improvement per lead time and target variable from applying the k-NN (VS1 target-based, VS2 storm-based) with the predictors and weights derived by the Pearson correlation instead of PIC. The improvements are averaged for different times of nowcast. The green plot region indicates a positive improvement of the correlation predictors in comparison to the PIC, and the red region indicates a deterioration.*

The results from **Figure 7** indicate that for the Area, Intensity, and Velocity components, the Pearson correlation
weights improve the performance of target-based k-NN from 5 up to 100% compared to the PIC weights. This happens
mainly for the short lead times (LT<+60min) throughout the three groups of nowcast times. For longer lead times there
seems to be no significant difference between the predictors sets. The same cannot be said for the Total Lifetime as a
target variable, here the Pearson correlation weights do not give the best results for all the nowcast times. In fact, here the
k-NNs based on the PIC weights seem to be more appropriate and yielded better results. However, as the other 4 target
variables are better for the Pearson correlation, this predictor set was selected for all applications of the k-NN (with
different weights according to **Table 3**) to keep the results consistent with one another. A further analysis was done that
proved that the application of the correlation weights produces lower errors than the non-weighted k-NN application (all
weights are assigned to 1 to the most important predictors from Pearson correlation).
Lastly, it should be emphasized that for the computation of predictors weights, all the events were grouped
together, and thus when applying the k-NN nowcast in the cross-validation mode, there is a potential that the information
leaks from the importance analysis to the performance of the k-NN (also illustrated in **Figure 3**-c). In other words, the
performance of the k-NN will be better, because the weights were derived from all the events grouped together. Typically,
in modelling applications, the optimization dataset should be clearly separated by the validating one, in order to remove
the effect of such information leakage. For this purpose, the correlation weights were computed 110 times, on a "leave-
one-event-out" cross-sampling, in order to investigate their dependence on the event database. The results of such cross-
sampling are visualized in **Appendix 11.3** and indicate a very low deviation of the predictors weights (lower than 0.01)
over all the target variables. The shown low variability of the Pearson Correlation weights justifies the decision to estimate
the weights from the whole database, as the potential information leakage is not likely affecting the results of the k-NN
performance. This is another reason favouring the calculation of the predictor's weights based on the Pearson Correlation.
On the other hand, the weights from the PIC analysis are changing very drastically depending on the dataset and hence
the effect of the information leakage would be much larger in the k-NN developed from PIC weights. Moreover, a
sensitivity analysis as done in **Appendix 11.3** cannot be performed for the PIC analysis because it would be extremely
time consuming.

### 4.2 Optimizing the deterministic k-NN nowcast

Once the most important predictors and their weights are determined, the optimization of the single k-NN
nowcast for the two k-NN applications (storm-based and target-based) was performed. The optimal k-value obtained from
minimizing the mean absolute error (MAE) produced by k-NN are shown in **Figure 8-upper row**. The results are
computed for the given nowcast times, lead times and target variables for both k-NN applications (VS1 target-based and
VS2 storm-based). For the 4 target variables Area, Intensity and Velocity in X/Y direction, the number of optimal values
decreases quasi exponentially for lead time up to 1 hour. After these lead time, when the majority of the storms are

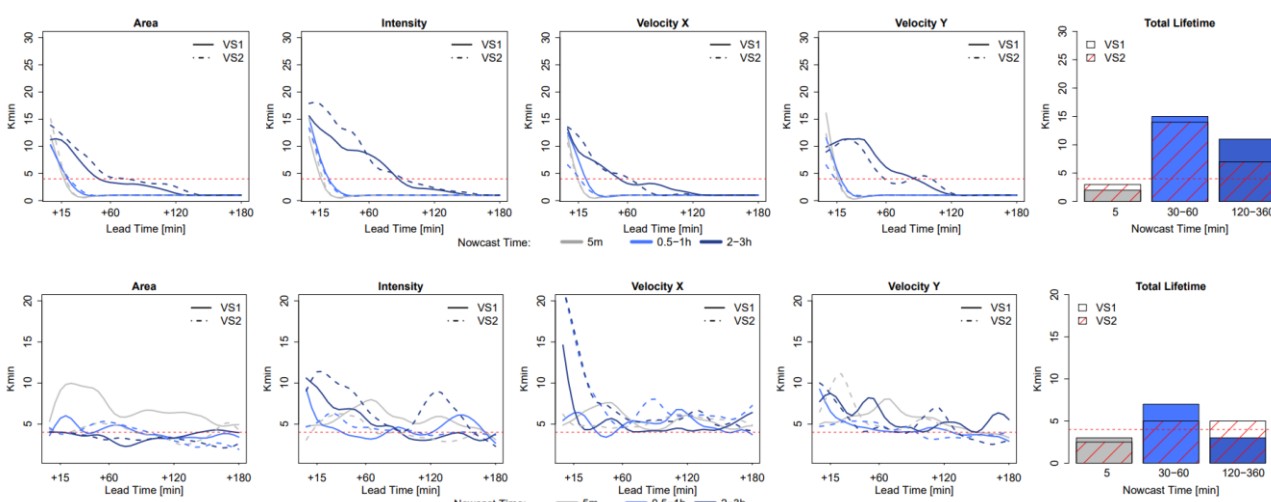

**Figure 8** *The optimization of the k-NN per target variable based on predictors and weights derived from Pearson correlation analysis: the median optimal selected "k" neighbours yielding the lowest absolute errors over the 110 events. Two k-NN applications are shown here – VS1 in solid line and VS2 in dashed line: first row – The optimal neighbour is found from minimizing the MAE for given group of nowcast times per event, second row – The optimal neighbour is found from minimizing the absolute mean error (ME) for the given group of nowcast times per event. The red dashed horizontal line indicates the k=4 that is chosen in this study for the deterministic k-NN application.*

dissipated, the optimal k-number converges at 1, meaning that the closest neighbour is enough to predict the dissipation of the storms. Contrary, for the very short lead times, the closest identified neighbour is unable to capture the growth/decay processes of the storms, thus the response has to be average from k-neighbours, with k depending strongly on target variable, nowcast time, lead time, and total lifetime. This seems to be the case also for the Total Lifetime, where averages between 3-15 neighbours are computed as $K_{min}$. Overall the k=1 seems to yield the lowest MAE for the majority of the lead time, nowcast times and target variables, and therefore is selected to continue further on with the analyses. However, selecting the first neighbour does not satisfy the requirement that the nature doesn't repeat itself, and ideally a k>1 should be achieved such that the responses from similar neighbour can be averaged to create a new response. For this purpose, the optimal K were additionally obtained by minimizing the absolute mean error (ME) and are shown in **Figure 8 -lower row**. Here the overestimation and underestimation of different storms balance one another, and the results seem to converge when averaging 3-5 neighbours. A direct comparison of the MAE for k~2-5 and k=1 was performed in order to understand if a higher k will benefit the application of both k-NN versions. The median improvements of using neighbours from 2-5 instead of 1 (over the selected groups of nowcast times) are shown only for the Total Lifetime in **Table 4**. The other target variables are left outside this analysis as the improvements averaged over all the lead times are very close to zero, as the dissipation of storms is captured well by all the 5 closest neighbours. From the results of the **Table 4** it is visible that k=4 brings the most advantages and hence was selected for both applications as a better compromise. The selection of k=4 is not an optimization per se, as it was not learned with artificial intelligence, instead was selected based on human intuition, and it does not represent the best possible training of the $K_{min}$. For a more complex optimization, the machine learning can be employed in the future to learn the parameters of the exponential relationship between $K_{min}$, lead time, nowcast time and target variable. In that case a proper splitting of the database intro training and validation should be done in order to avoid, information being leaked from the optimization to the validation of the k-NN. In our case, the effect of the information leakage at this stage (also illustrated in **Figure 3-c**) is minimized by obtaining the $K_{min}$ on a cross-sampling of the events, and averaged over the events, lead times and nowcast times.

**Table 4** *The median improvement of the total lifetime MAE when using k= 2- 5 instead of k=1 over the three selected groups of nowcast times.*

|  | k=2 | k=3 | k=4 | k=5 |
|---|---|---|---|---|
| Storm-based | 9.09% | 10.74% | 13.09% | 11.94% |
| Target-based | 3.40% | 5.89% | 6.54% | 6.02% |

### 4.3 Results of the deterministic 4-NN nowcast

The median MAE of the 4-NN determinist nowcast over all the events, run for both target- and storm-based approaches are shown in **Figure 9** for each lead time and target variable. The results are grouped according to the storm duration; i) upper row – for storms that live 30min, ii) middle row – for storms that live up to 3 hours and iii) lower row – for storms that live longer than 3 hours, and are averaged per nowcast times given in **Table 2**. As shown as well in the optimization of the 4-NN, the target-based k-NN exhibits lower Area, Intensity and Velocity errors than the storm-based 4-NN. **Table 5-a** illustrates the median deterioration (-) or improvement (+) in percent (%) over all lead times that the storm-based 4-NN can reach when compared to the target-based one.

**Table 5** *Median Deterioration (-) or Improvement (+) of k-NN storm-based (VS2) compared to target-based (VS1) over all lead times according to the storm duration and nowcast times (shown in %). Equation 11 is used here, where "ref" – is the target-based and "new" is the storm-based k-NN.*

**a) deterministic comparison of median MAE from storm-based to target - based**

| Storm Duration | Nowcast Time | Area | Intensity | Velocity X | Velocity Y | Total Lifetime |
|---|---|---|---|---|---|---|
| 5-30min | 5min | -3% | 8% | -8% | -27% | 2% |
| 5-30min | 15min | 0% | 7% | -14% | -38% | -7% |
| 5-30min | 30min | 0% | 0% | 0% | 0% | 13% |
| 0.5-3h | 5min | -0.21% | -2% | 3% | -1% | -4% |
| 0.5-3h | 60min | 30.00% | 2% | -5% | 23% | -11% |
| 0.5-3h | 180min | -15% | -28% | -100% | -100% | 28% |
| >3h | 5min | -9% | 0% | 2% | -1% | -1% |
| >3h | 120min | -10% | -7% | -3% | -10% | 2% |
| >3h | 360min | -10% | -8% | -8% | 21% | 18% |

**b) probabilistic comparison of median MAE from storm-based to target - based**

| Storm Duration | Nowcast Time | Area | Intensity | Velocity X | Velocity Y | Total Lifetime |
|---|---|---|---|---|---|---|
| 5-30min | 5min | -19% | 7% | -12.75% | -50.00% | 0.50% |
| 5-30min | 15min | 3% | 4% | -6.95% | -58% | -9.61% |
| 5-30min | 30min | 30.23% | 29.62% | 35% | 40.18% | 3.45% |
| 0.5-3h | 5min | -3.00% | 0.00% | -0.50% | -14.40% | -4.02% |
| 0.5-3h | 60min | 11.58% | -0.23% | -11.60% | 15.37% | 3% |
| 0.5-3h | 180min | -8% | -4% | -100% | -88% | 5% |
| >3h | 5min | -9.30% | -4.24% | 1.10% | -0.67% | -9.27% |
| >3h | 120min | -5% | 2% | -4% | -5% | 4.79% |
| >3h | 360min | -3.50% | -0.42% | -16% | 11% | 5.14% |

**c) deterministic comparison of median improvement towards Langrangean Persistence from storm-based to target-based**

| Storm Duration | Nowcast Time | Area | Intensity | Velocity X | Velocity Y | Total Lifetime |
|---|---|---|---|---|---|---|
| 5-30min | 5min | -1% | 0% | 0% | -1% | |
| 5-30min | 15min | 0% | 0% | 0% | -1% | |
| 5-30min | 30min | 0% | 0% | 0% | 0% | |
| 0.5-3h | 5min | 0.00% | 0% | 1% | 0% | |
| 0.5-3h | 60min | 12.53% | 1% | -1% | 3% | |
| 0.5-3h | 180min | 0% | 0% | 1% | 0% | |
| >3h | 5min | -26% | 0% | 6% | -8% | |
| >3h | 120min | -31% | -22% | -29% | -39% | |
| >3h | 360min | -5% | -21% | -8% | 89% | |

**d) probabilistic comparison of median improvement towards Langrangean Persistence from storm-based to target-based**

| Storm Duration | Nowcast Time | Area | Intensity | Velocity X | Velocity Y | Total Lifetime |
|---|---|---|---|---|---|---|
| 5-30min | 5min | -1% | 0% | 0% | 0% | |
| 5-30min | 15min | 0% | 0% | 0% | 0% | |
| 5-30min | 30min | 0% | 0% | 0% | 0% | |
| 0.5-3h | 5min | -1.40% | 0% | 0% | -1% | |
| 0.5-3h | 60min | 2.50% | 0% | 0% | 0% | |
| 0.5-3h | 180min | -1% | 0% | 0% | -1% | |
| >3h | 5min | -44% | -11% | 5% | -3% | |
| >3h | 120min | -20% | 11% | -5% | -6% | |
| >3h | 360min | -8% | 0% | -24% | 6% | |

For storm living less than 30 minutes, the MAE is decreasing with the lead time and past LT+30 min is mostly zero, as the dissipations of the storms have been captured successfully. The Total Lifetime of the majority of the storms can be captured with ~ 15 min over-/underestimation regardless of the nowcast time. The errors for the 4 target variables (except Total Lifetime) are lower for the later nowcast times than for the earlier ones (as expected). The difference between the storm- and target-based 4-NN is very small for Area, Intensity and Total Lifetime, but much higher for the velocity components (with storm-based exhibiting up to 40% higher errors than the target-based). The biggest difference seems to be for shorter lead times (LT < +1h). For the storms living up to three hours, the same behaviour is, more or less, observed. The only difference is for nowcasts issued at 3$^{rd}$ hour of the storm existence (last moment the storm is observed). Here it is clear that the 4-NN fails to capture the dissipation of the storms that live exactly three hours, however this is attributed to the number of available storms with duration of 3 hours (median over 6 storms available). Since the Area, Intensity and Total Lifetime are overestimated and not converging to zero for high lead times, it is clear that the nearest neighbours are being selected from the longer storms that do not dissipate within the next 3 hours. The differences between the two 4-NN approaches are visible mainly for lead times up to 30 min (except the nowcast at 3$^{rd}$ hour of storms life), afterwards the errors are relatively converging to each other. The storm-based 4-NN produces circa 10-20% higher errors than the target-based one for the nowcast times lower than 3hours, while for nowcast time of 3 hours, the errors are up to 100% higher than the target- based one. At these storms as well, the higher discrepancy between the two versions of 4-NN is seen at the Velocity components.

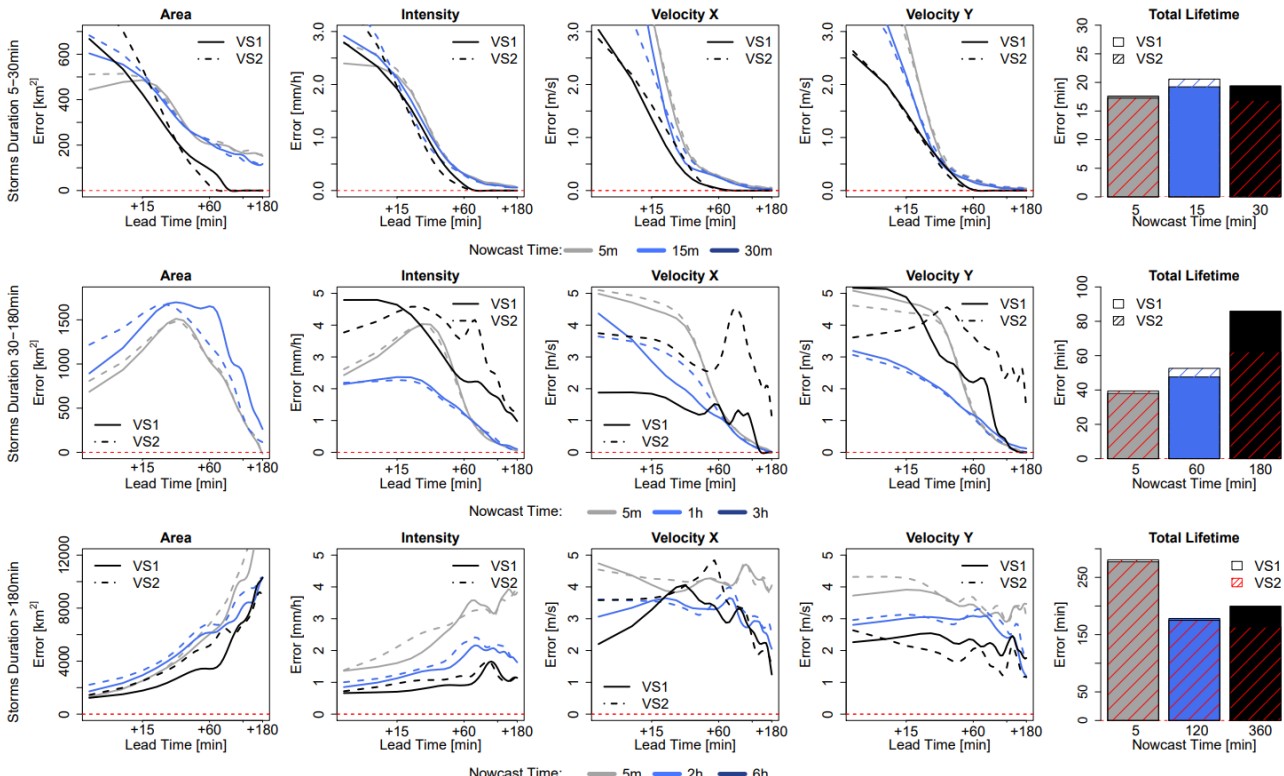

*Figure 9* *The median mean absolute error (MAE) over all the events, for each target variable (Area, Intensity, Velocity in X and Y direction and Total Lifetime) based on two 4-NN applications: -VS1 in solid and VS2 in dashed lines. The performance is shown for storms that are: shorter than 30 min (upper row), than 3 hours (middle row), and longer than 3 hours (lower row), and over the selected nowcast times. Nowcast time dictates when the nowcast is issued relative to storm initiation*

For the storms that live longer than 3 hours (under 100 storms available) the same problem, as in the nowcast time of 3 hours seen before, is present. The Total Lifetime is clearly underestimated (up to 100min) as due to database the information is taken from shorter storms. It is important to notice here, that although 70 storms are present, because of the "leave-one-event-out" validation, the storm database is actually smaller. Nevertheless, the error is manifested here differently: as the long storms are more persistent in their features: Area, Intensity and Velocity components are captured better for the short lead times with the error increasing at higher lead times. Here as well the nowcast issued at the earlier stages of the storm's life exhibit higher errors than in the later stages. Especially for the nowcast at the 6[th] hour of the storm's existence, the errors are quite low for all 5 target variables due to the persistence of the stratiform storms. For this group of long storms, the storm-based nowcast yields up to 10% higher errors than the target-based one, with only few exceptions depending on the time of nowcast and variable. It is clear that the storm-based 4-NN is more influenced by the number of available storms than the target-based approach.

**Figure 10** shows the improvement that the 4-NN introduces to the nowcast when compared to the Lagrangian persistence (either target- or storm-based) and are averaged per lead time for each of the three group of storms and the respective times of nowcast. Since the Lagrangian Persistence doesn't issue a Total Lifetime nowcast, only the four target variables (Area, Intensity and Velocity components) are considered. The green area indicates the percent of improvement from the application of the 4-NN approach, and the red area indicates the percent of deterioration from the 4-NN application (Lagrangian persistence is better). Additionally, median improvements (+) or deterioration (-) over all lead times of the storm-based compared to target-based 4-NN approach in respect to the Lagrangian Persistence are illustrated in **Table** 5-c. For the 30min storms, the 4-NN approach (both target- and storm-based) are considerably better than the

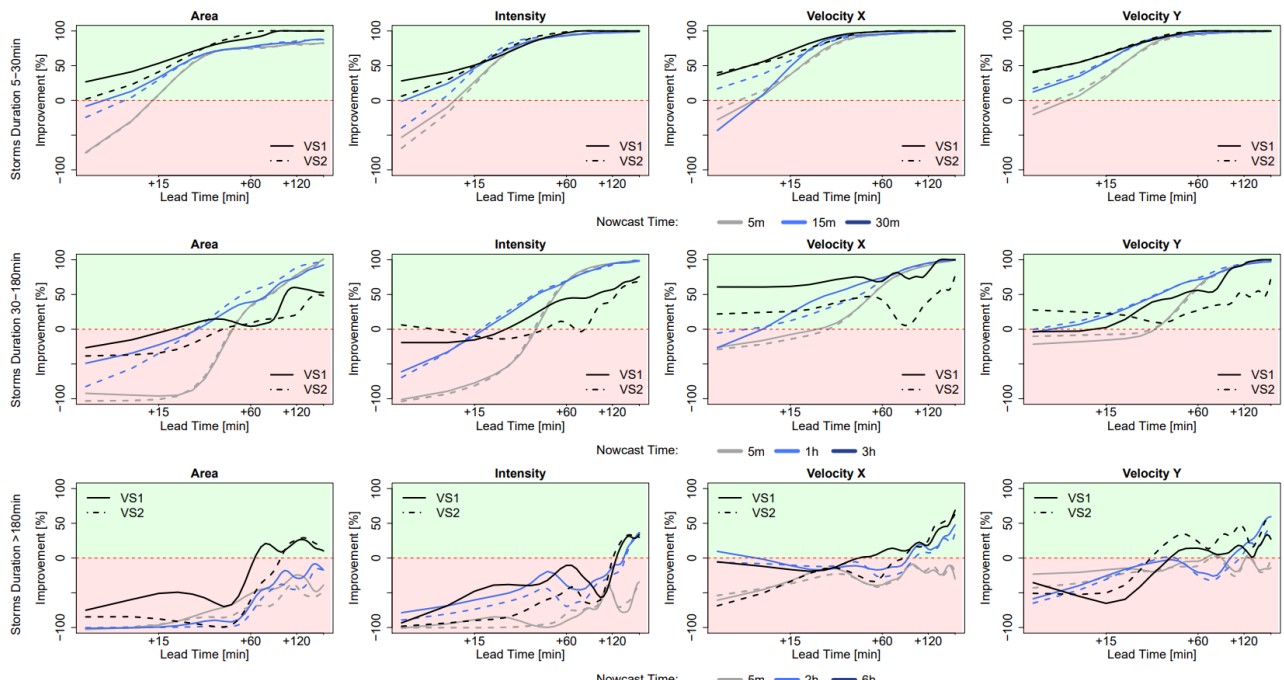

***Figure 10*** *The median improvements over all the events that the single 4-NN application can introduce in the nowcast of the target variables (Area, Intensity, Velocity in X and Y direction) in comparison to the Lagrangian persistence. The results are shown for each 4-NN application: VS1 in solid and VS2 in dashed lines and are calculated separately for storms that live shorter than 30 min (upper row), shorter than 3 hours (middle row) and longer than 3 hours (lower row), and for the respective nowcast times. The green region of the plot indicates a positive improvement (better nowcast by the 4-NN application) and the red region indicates a deterioration (better nowcast by the Lagrangian persistence).*

Lagrangian persistence: improvement is higher than 50% from the LT+15min and up to 100% from LT+60min. The
improvement is greater for nowcast at the 15th min of storm existence (when the persistence predictors are considered). It
is clear than due to the autocorrelation, the Lagrangian persistence is more reliable for the short lead times and for earlier
nowcast times. However, after LT+15min and for nowcast times near to the dissipation of the storms, where the non-
linear relationships govern, the improvements from the nearest neighbour are more significant. The target-based 4-NN
results in slightly higher improvements than the storm-based one only for lead time up to 30min, past this lead time the
improvements from both versions are converging. For the storms that live between 30 min to 3 hours, the improvements
are introduced first after LT+15 or +30 min depending on the nowcast time: with increasing nowcast time increases the
improvement as well. The only exception is for the nowcast of Area and Intensity on the 3rd hour of the storm existence,
where no clear improvement of the 4-NN approaches could be seen before LT+30min or LT+1h. This low improvement
for the nowcast time of 3 hours was expected following the poor performance of the 4-NN shown in **Figure 9**. It seems
like the Lagrangian persistence is particularly good for predicting the Area and Intensity at very short lead times (up to
LT+20min). Here, for nowcast times of 5min, the Lagrangian Persistence is 100% better than any of the 4-NN approaches.
But not the same is true for the Velocity Components, with the Lagrangian Persistence exhibiting very low advantages
against the 4-NN for the short lead times. Regarding the difference of the two 4-NN approaches, with few exceptions, the
storm-based nowcast exhibits similar improvements as the target-based. Another exception is the nowcast time of 3 hours,
where the storm-based improvements are clearly lower, especially for the higher lead times, than the target-based (up to
40%). For storms living longer than 3 hours, the improvements are present for lead times higher than 2 hours. Since the
features of the long storms (mostly of stratiform nature) are persistent in time, is understandable for the Lagrangian
Persistence to deliver better nowcast up to LT+2h. Past this lead time non-linear transformations should be considered.
Here, even though the storm database is small, the non-linear predictions based on the 4-NN capture better these
transformations than the persistence. The improvement introduced by the storm-based are generally from 20-30% lower
than the improvements introduced from the target based.
To conclude, the 4-NN deterministic nowcast brings up to 100% improvements for lead times higher than the
predictability limit of the Lagrangian persistence and depend mainly on the storm type and the size of database. Overall,
for all storms the improvement is mainly at the high lead times and later times of nowcast, as the 4-NN is capturing
particularly well the dissipation of the storms. The results from the long events are suffering the most from the small size
of the database. This was anticipated, as the events were mainly selected from convective events that have the potential
to cause urban floods. A bigger database, with more stratiform events included, can introduce a higher improvement to
the Lagrangian persistence. These improvements are expected to be higher for lead times longer than 2 hours, but is yet
to be seen if a larger database can as well behave better than the persistence even for lead times shorter than the
predictability limit. Regarding the two different 4-NN approaches, the storm-based performs 0-40% worse than the target-
based nowcast, introducing generally 40% lower improvements to the Lagrangian persistence. The main differences
between these two approaches lie between the growth/decay processes, which the target-based 4-NN can capture better.
Also, these differences are particularly larger for the Velocity Components and for the Total Lifetime, than in the Area
and Intensity as target variables. Furthermore, it seems that the storm-based 4-NN is more susceptible to the size of the
database than the target-based one. Nevertheless, there are some cases where the storm-based behaves better than the
target-based nowcast (as illustrated with green in **Table 5** -a) even though the target-based approach should be profiting
more from the selected predictors and their respective weights. A better optimized $K_{min}$ for each lead time and nowcast
time, may actually improve further on the results of both 4-NN versions, and give the advantages mainly to target-based
nowcast.

### 4.4 Results of the ensemble 30-NN nowcast

The median CRPS over all the events for the probabilistic 30NNs (in solid lines) together with the
median MAE for the deterministic 4-NN (in dashed lines), are illustrated respectively for storm-based approach in **Figure
11** and for target-based approach approaches in **Figure 12.** The results are shown as in the previous figures per each lead
time and target variable, for storms divided into 3 groups according to their duration and averaged depending on the time
of nowcast. Additionally, the median improvements (+) or deterioration (-) of storm-based CRPS values in comparison
with the target-based are given in **Table** 5-b. For the 30min long storms, the errors of the probabilistic nowcast are
typically lower than the single 4-NN nowcast for all the variables, lead times and nowcast times, independent of the
30NNs approach (either storm- or target-based). In contrast to the deterministic 4-NN, the probabilistic 30NNs
performance is very little dependent on the nowcast time (mainly for Area, Intensity and Total Lifetime). The storm-based
30NNs has up to 50% higher errors than the target-based, but on the other side can have up to 40% lower errors than the
target-based for nowcast times of 30min. This suggests that storms in this duration behave similarly and their dissipation
can be predicted adequately by the storm-based approach with more than 4 similar neighbours. For storms that live shorter
than 3 hours, the same performance is as well exhibited: the probabilistic 30NNs has lower errors than the deterministic
4-NN. The difference between the target- and storm-based nowcasts is within the range of the single 4-NN nowcast for
the first 4 target variables, with storm-based 30NNs having 15% higher errors in the first 30 min of the nowcast than the
target-based. For Intensity and the Total Lifetime, both of the 30NNs exhibit very similar errors for most of the nowcast
times. It is worth mentioning here, that for the nowcast at the 3[rd] hour of storms' existence the errors are much lower than
the single 4-NN nowcast. This proves that the most similar storms are within the 30 members, but not within the first 4
neighbours selected in the case of the single 4-NN nowcast. Due to the unrepresentativeness in the database, the errors of

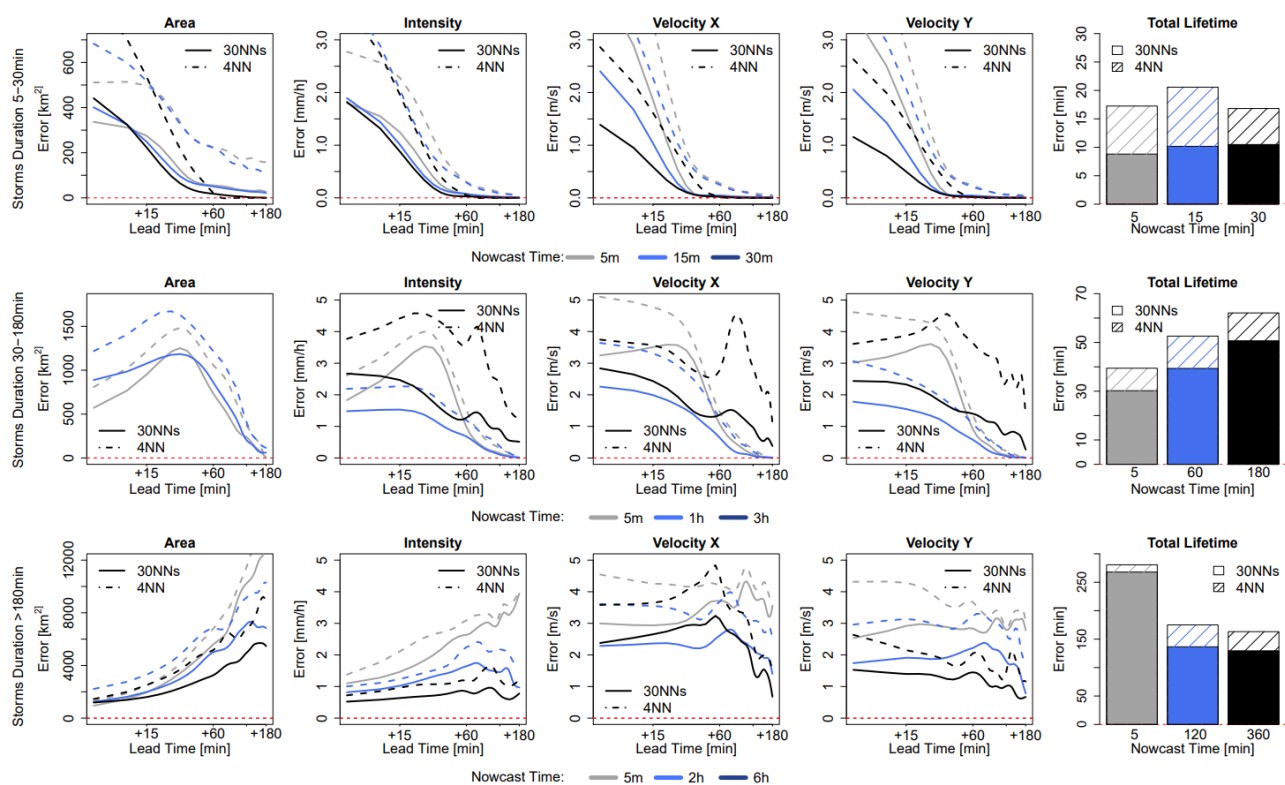

**Figure 11** *The median CRPS over all the events for each target variable (Area, Intensity, Velocity in X and Y direction and Total Lifetime) on the storm-based applications: 4-NN (deterministic) in dashed and 30NNs (probabilistic) in solid lines. The performance is computed over storms that are: shorter than 30 min (upper row), than 3 hours (middle row), and longer than 3 hours (lower row), and over the selected nowcast times.*

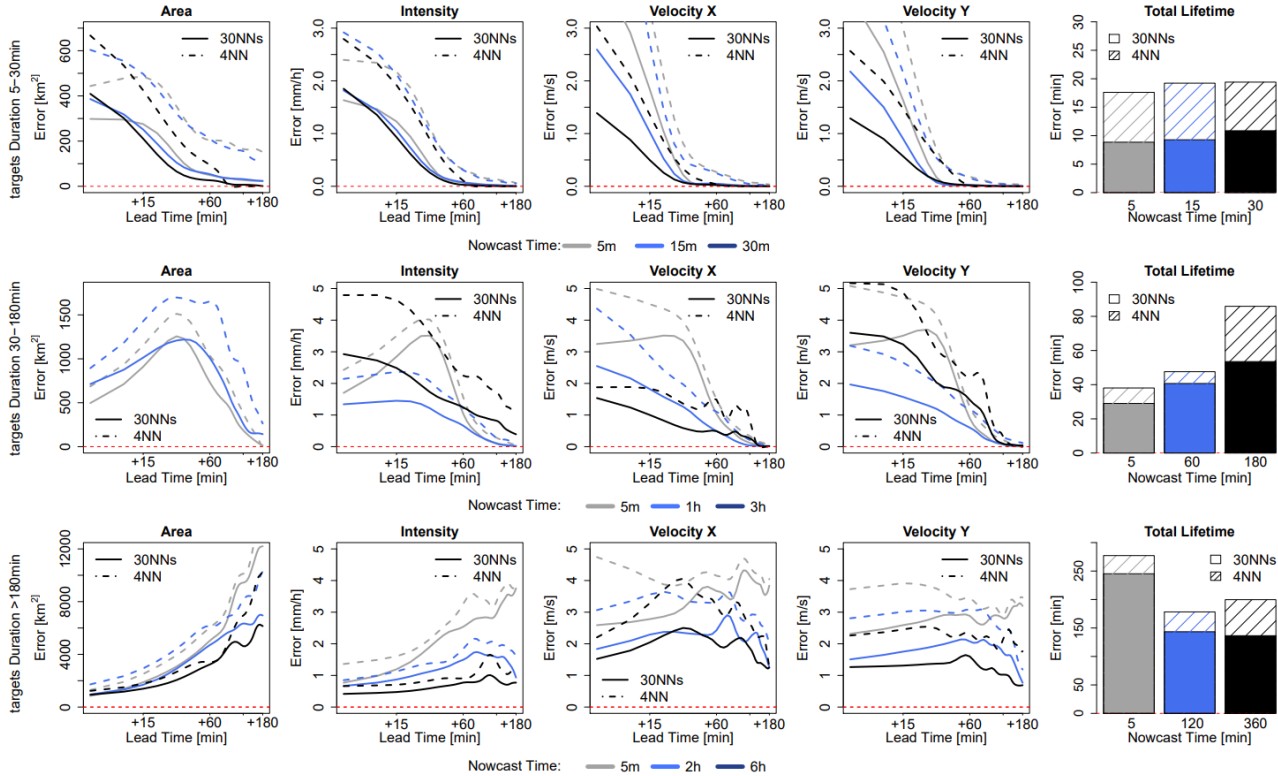

**Figure 12** *The median CRPS over all the events for each target variable (Area, Intensity, Velocity in X and Y direction and Total Lifetime) on the target-based applications: 4-NN (deterministic) in dashed and 30NNs (probabilistic) in solid lines. The median errors are computed over storms that are: shorter than 30 min (upper row), than 3 hours (middle row), and longer than 3 hours (lower row), and over the selected nowcast times.*

the longer storms are considerably higher than the other storm groups, and the errors of the first 4 target variables are
increasing with the lead time and decreasing with the nowcast time, as in the case of the deterministic 4-NN nowcasts.
However here unlike the other storm groups, the differences between the storm-based and target-based approach are
visible past 30 min lead time, with the storm-based errors being up to 15% higher than the target-based.

Overall the ensemble results are clearly better than the single 4-NN nowcast, suggesting that the best responses

are obtained by singular neighbours (either the closest one or within the 30 neighbours) and not by averaging. Thus, there
is still room for improving the single 4-NN nowcast by selecting better the important predictors and their weights or
averaging differently the nearest neighbours. Nevertheless, the results from **Figure 11** and **Figure 12** emphasize that
similar storms do behave similarly, and that the developed k-NN on the given database with 30 ensembles gives
satisfactory results. Compared to the deterministic 4NNs it has the advantage that no k-optimization is needed, and the
two approaches (storm- and target-based) have less discrepancies with one another.

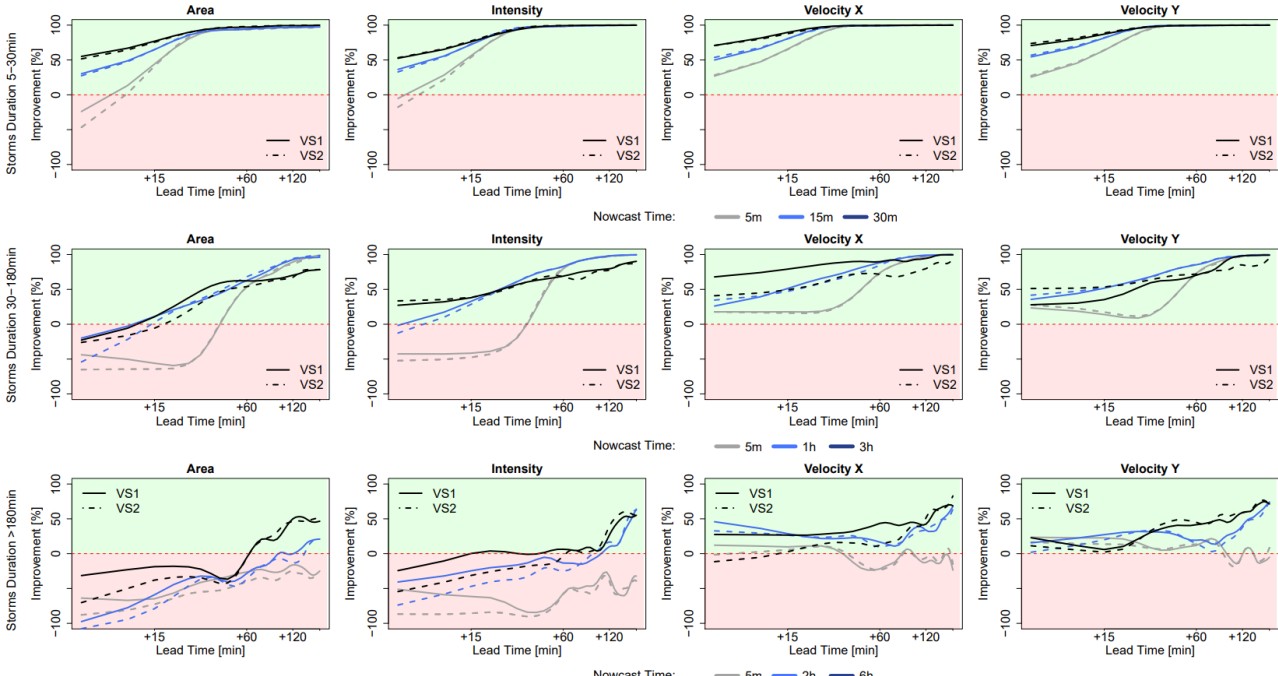

***Figure 13*** *The median improvements over all events, that the 30NNs nowcast can introduce in the nowcast of the target*
*variables (Area, Intensity, Velocity in X and Y direction) in comparison to the Lagrangian persistence. The results are*
*shown for each 30NNs application: VS1 in solid and VS2 in dashed lines and are calculated separately for storms that live*
*shorter than 30 min (upper row), shorter than 3 hours (middle row) and longer than 3 hours (lower row), and for the*
*respective nowcast times. The green region of the plot indicates a positive improvement (better nowcast by the 4-NN*
*application) and the red region indicates a deterioration (better nowcast by the Lagrangian persistence).*

**Figure 13** demonstrates the improvement of the probabilistic 30NNs when compared to the Lagrangian

persistence (storm-based in dashed line, and target-based in solid line). As before the median improvement over the events
is computed and shown for each storm duration group, nowcast time, lead time and target variables (expect for the Total
Lifetime). For all the three groups it is visible that performance increases considerably with the lead time – suggesting
that the ensemble predictions are particularly useful for the longer lead times where the single nowcast is not able to
capture the storm evolution. For short storms (duration shorter than 30min) the Lagrangian persistence is only better for
the Area and Intensity at 5min nowcast time and for very short lead times (up to 10min). However, past this lead time,
the probabilistic 30NNs has the clear advantage with improvements up to 100%. Past LT+30min, which coincides with
the predictability limit of the Lagrangian persistence at such scales, there is no difference between the nowcast time and

30NNs approach (less than 1% for all target variables and nowcast times). For storms that live shorter than 3 hours, the results are slightly worse than the very short storms., but still exhibit the same patterns. Here as well the main improvements of the 30NNs probabilistic approach is seen between LT+15min to LT+30min for all the target variables. Interesting in this storm group are the results from the nowcast time of 3 hours that exhibit different behaviours than the deterministic approach. This is expected as the Lagrangian persistence performs particularly poorly because it cannot model the storms dissipations. The difference between the two types of 30-NN is insignificant, although a bit higher than for the very short storms (~2.5% difference). For the longer storms the benefit of the probabilistic 30NNs is seen mainly for LT+60min to LT+120min, but still not as high as in the other storm groups. The worse performance is at nowcast time of 5min, where the 30NNs fails to bring any advantage to the prediction of Area and Intensity when compared to the Lagrangian Persistence. Interesting from these storms, is that the improvement is more significant at the Velocity Components than in the Area and Intensity predictions. This suggest the velocity components are more persistent (see **Figure 4**) and easier to be predicted from similar storms.

As a conclusion the probabilistic nowcasts are better than the Lagrangian Persistence mainly for convective storms that last shorter than 3 hours and lead times higher than LT+15min. Of course, there is still room for improving the 30NNs application by increasing the size of the past database. Overall, it seems that the velocity components can be captured much better by the 30NNs application than the Lagrangian Persistence, while the Lagrangian Persistence is more suitable for long persistent storms and for nowcast times of 5min where not enough information is available to select similar storms. An increase in the database, with more stratiform storms, may improve the performance of the 30NNs and its advantage over the Lagrangian Persistence. However, the value of the probabilistic 30NNs relies mainly in the nowcasting of convective events. Moreover, the possibility of merging Lagrangian Persistence with a probabilistic 30NNs approach should be explored and further investigated; the Lagrangian Persistence should be implemented for very short lead times (up to 30min) and for the first nowcast times where the predictors are not enough to select similar past storms.

### 4.5 Nowcasting the unmatched storms

For the optimization and testing of the k-NN approaches, the unmatched storms from the tracking algorithm were left outside of the database. Nevertheless, in an online application (operational nowcast), when the storm is recognized for the first time, one cannot predict if the storm is an artefact, or it will not be matched by the tracking algorithm. Therefore, it is important to investigate how the developed k-NN deals with these unmatched storms. **Figure 14** illustrates the median performance over the 110 events of the developed target-based (upper row) and storm-based (lower row) k-NN when predicting the target variables of the unmatched storms from a past database of only matched storms (storms with duration equal or longer than 10min). As in the previous results, the 30NNs probabilistic application yields better errors than the deterministic one, causing an overestimation of these storms for the first 10-20min for the target-based approach and 15-30min for the storm-based one. A direct comparison of these errors with the Lagrangian Persistence is shown in **Figure 15**, with the deterministic 4-NN in the upper row and the probabilistic 30NNs in the lower row. As expected the probabilistic 30NNs brings the most improvement when compared to the Lagrangian Persistence for all lead times and target variables. Thus, even though, most of these unmatched storms will be overestimated in their duration, the 30NNs will capture their dissipation much better than either the deterministic 4-NN or the Lagrangian Persistence.

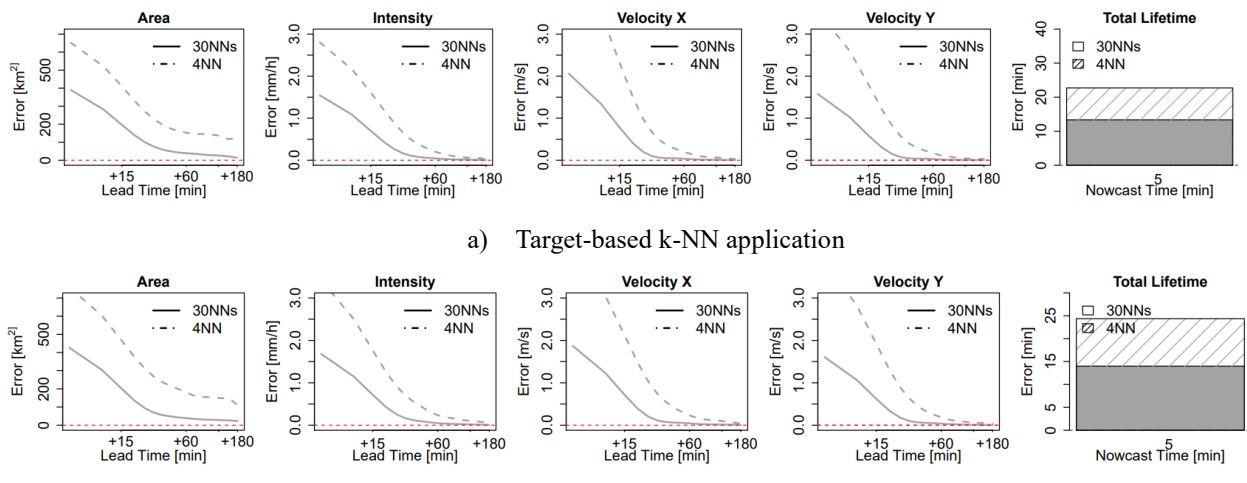

a) Target-based k-NN application

b) Storm-based k-NN application

*Figure 14* *Median CRPS error over the 110 events for each of the target variables nowcasted from 4-NN deterministic (in dashed lines) and 30NNs probabilistic (in solid lines) applications for both target- (upper row) and storm-based (lower row) approaches. The results shown here are from the "unmatched storms" when the nowcast time is 5 min.*

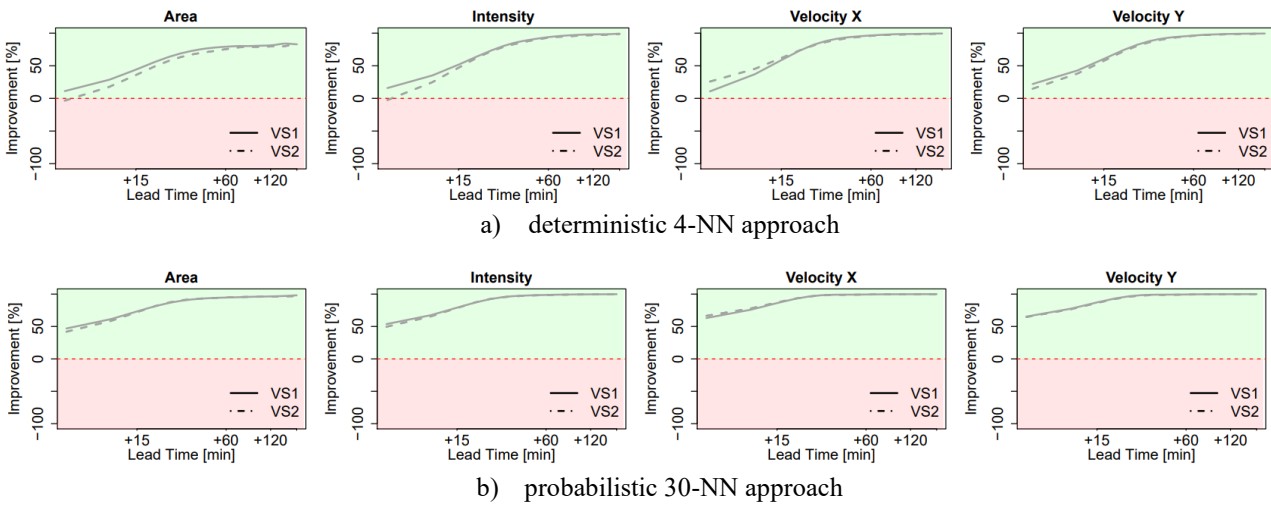

a)    deterministic 4-NN approach

b)    probabilistic 30-NN approach

*Figure 15* *Median performance improvement over the 110 events for each of the target variables nowcasted from 4-NN deterministic (upper row) and 30NNs probabilistic (lower row) applications when compared to the Lagrangian Persistence, for both target- (dashed line) and storm-based (solid line) approaches. The results shown here are from the "unmatched storms" when nowcast time is 5min.*

**5. Conclusions**

Accurate predictions of rainfall storms at fine temporal and spatial scales (5min, 1km$^2$) based on radar data are quite challenging to achieve. The errors associated with the radar measurements, identification and tracking of the storms, and more importantly the extrapolation of the storms in the future based on the Lagrangian persistence, are limiting the forecast horizons of such object-oriented radar based nowcasts to 30-45 min for convective storms and to 1 hour for stratiform events (Shehu & Haberlandt, 2021). The focus of this paper was the improvement of the storm-oriented radar based nowcasts by considering other non-linear behaviours for future extrapolation instead of the Lagrangian persistence. For this purpose, a nearest neighbour approach was proposed that predicts future behaviours based on past observed behaviours of similar storms. The method was developed and validated for the Hannover Radar Range where storms from 110 events were pooled together and used in a "leave-one-event-out" cross-validation. From 110 events a total of around 5200 storms with different morphology were identified and tracked with HyRaTrac in order to build up the database for the k-NN implementation. The storms were treated as ellipses and for each state of the storms' evolution different features (describing both present and past states) were computed. The k-NN approach was developed on these features to predict the behaviour of the storms in the future (for lead times up to 3 hours) through 5 target variables (Area, Intensity, Velocity in X and Y direction and Total Lifetime).

First an importance analysis was performed in order to recognize the most important predictors for each target variable. Two different approaches were employed for this purpose: Pearson correlation, and Partial Information Correlation (PIC). A comparison of these two methods revealed that for the application at hand the Pearson Correlation is more reliable at determining important predictors, and delivers 5%-30% better results than the PIC method. However, the PIC seems promising mainly for determining the most important predictors of the Area and Total Lifetime for storms longer than 3 hours, and is still recommended for investigation in the future. The Area, Intensity and Total Lifetime of the storms seem to be co-dependent on one another and on the features that describe their evolution. In particularly the variance of the spatial intensity is an important predictor for the three of them. On the other hand, the velocity components are dependent as well more on features that describe their evolution. Nevertheless, there is still a dependency of the area and velocity components, and should be included when predicting each other mainly for long lead times.

The weights derived from the Pearson correlation were used for the similarity estimation of different storms based on the Euclidian distance. Two k-NN approaches were developed on two similarity metrics: a) target-based approach – similarity was computed for each target independently and indicates the best performance possible by the given predictors and weights, and b) storm-based approach – similarity was computed for each storm keeping the relationship between the target variables. For the two approaches a deterministic (averaging the 4 closest neighbours) and a probabilistic (with 30 nearest neighbours) nowcast were issued for all of the storms in "leave-one-event-out" cross-validation mode. In the deterministic nowcast the difference between the two remains mainly at short lead times (up to 30 min) and at the Velocity Components, with the storm-based results yielding up to 40% higher errors than the target-based ones. However, at higher lead times the difference between the two became insignificant, as the dissipation processes were captured well for the majority of the storms. The same behaviours were observed as well in the ensemble nowcast, with target-based ensembles being slightly better than the storm-based nowcast. Overall the storm-based approach seems reasonable for Area-Intensity and Total Lifetime, as they are co-dependent and their relationship should be maintained for each storm, while target-based approach captures better the velocity components. A combination of both approaches, may results in better nowcasting of storms' characteristics.

To investigate what value each of the two k-NN approaches introduces to the nowcast, their errors (for both deterministic and probabilistic nowcast) were compared to the errors produced by the Lagrangian persistence. For both of the approaches the improvement was more than 50% for convective storms for lead times higher than 15 min, and for

mesoscale storms for lead times higher than 2 hours. The results were particularly good for the small convective storms due to the high number of storms available in the database. For the mesoscale storms (with duration longer than 3 hours) the improvements were not satisfactory due to the small sample size of such long storms. Increasing the sample size is expected to improve the performance of the k-NN for these storms as well. However, when consulting the probabilistic k-NN application it seems that, even for these storms and the given database, there are enough similar members in the 30 neighbours that are better than the Lagrangian persistence. This emphasizes that the probabilistic nowcast is less affected by the sample size than the deterministic 4-NN. Moreover, the differences between the storm-based and target-based approaches, become smaller in the probabilistic approach than the deterministic ones. Lastly, the optimization of the adequate neighbours for the deterministic approach is far more complex than implemented here, but when issuing the probabilistic nowcast there is no need to optimize the k – number. It is clear that the probabilistic application of the k-NN outperforms the deterministic ones, and has more potential for future works.

Overall the results suggest that if the database is big enough, storms that behave similarly can be recognized by their features, and their responses are useful in improving the nowcast up to 3 hours lead times. We recommend the use of the nearest neighbour in a probabilistic application (30NNs) to capture better the storm characteristics at different lead times. A merging with the Lagrangian Persistence for short lead times (up to 15min) and early nowcast times can be as well implemented. Further improvements can be achieved if the predictors importance is estimated better (i.e. Monte Carlo approach, or neural networks) or if additional predictors are included from other data sources like: cloud information from satellite data, temperature, convective available potential energy (CAPE) and convective inhibition (CIN) from Numerical Weather Prediction Models, lightning flash activity, additional measurements from Doppler or dual polarized radar data (like phase shift, doppler velocity, vertical profile at different elevation angles), various geographical information (as distance from heavy urbanized areas, mountains or water bodies) and so on. The main benefit of the probabilistic 30NNs is mainly seen for convective events and creating new nowcasting rules based on the predicted storm characteristics.

Improving the nowcasting of storm characteristics is the first step in improving rainfall nowcasting at fine temporal and spatial scales. On a second step, the knowledge about the storm characteristics (as nowcasted by the 30NNs) should be implemented on the spatial structure of the storms to estimate rainfall intensities at fine scales ($1km^2$ and 5min). There are two options to deal with the spatial distribution of the rainfall intensities inside the storm region (which is so far not treated in this study): 1. Increase/Reduce the area by the given nowcasted area (as target variable) for each lead time, scale the average intensity with the nowcasted intensity, and move the position of the storm in the future with the nowcasted velocity in x and y direction. 2. Take the spatial information of the selected neighbours, perform an optimisation in space (such that present storm and the neighbour's storms locations match) and assign this spatial information to the present storm for each lead time. The former is an extension of the target-based 30NNs, while the later an extension of the storm-based 30NNs. So far, the comparison between these two versions, showed that the target-based approach is better suited mainly to nowcast the velocity components, thus a merging of the two could also be reasonable: the storm-based approach is used for nowcasting Area-Intensity-Total Lifetime (features that are co-dependent based on the life cycle characteristics of convective storms), and the target-based approach for the nowcasting of the velocity components. Future works (Part II – Local Intensities) will include the integration of the developed 30NNs application in the object-oriented radar based nowcast to extend the rainfall predictability limit at fine spatial and temporal scales ($1km^2$ and 5min). The main focus of the Part II is to investigate if the methodology applied here can introduce improvements as well at the local scale, i.e. validation with the measurements from the rain gauge observations.

**6. Data Availability**

All data and R-codes can be provided by the corresponding authors upon request.

**7. Authors Contribution**

Study conception and design B.S, and U.H., methodology: B.S. and U.H., software and data collection: B.S., analysis and interpretation of results: B.S., writing-review and editing: B.S. and U.H., supervision and funding acquisition: U.H.

**8. Competing Interest**

The authors declare that they have no conflict of interest.

**9. Funding**

This research was funded by the German Federal Ministry of Education and Research (BMBF) grant number Förderkennzeichen 03G0846B.

**10. Acknowledgements**

The results presented in this study are part of the research project "Real-time prediction of pluvial floods and induced water contamination in urban areas (EVUS)", funded by the German Federal Ministry of Education and Research (Bundesministerium für Bildung und Forschung BMBF) who are gratefully acknowledged. We are also thankful for the provision and right to use the data from the German National Weather Service (Deutscher Wetterdienst DWD), the HyRaTrac program from Dr. Stefan Krämer, and to the Open Access fund of Leibniz Universität Hannover for funding the publication of this article. Lastly, we would like to acknowledge as well the useful comments from the three reviewers Seppo Pulkkinen, Ruben Imhoff and Georgy Ayzel that have improved the final version of the manuscript.

**11. Appendix**

*Appendix 11-1 Strength of relationship between the selected predictors and the target variables averaged for three lead times and storm duration groups based on correlation values. The green shade indicates the strength of the relationship: with 0 for no relationship at all, and 1 for highest dependency. The averaged computed values for each target variable (last row) are used as bases for **Table 3**. The correlation weights are absolute values of the correlation values between the predictors at specific lead times and target variables.*

**Area**

| Durations | Lead Time | Cells | Lnow | A | Iave | Imax | Isd1 | Isd2 | Vg | Vx | Vy | Jmax | Jmin | Jr | Φ | A30 | Iave30 | Imax30 | Isd130 | Isd230 | Vg30 | Vx30 | Vy30 | Jmax30 | Jmin30 | Jr30 | Φ30 |
|---|---|---|---|---|---|---|---|---|---|---|---|---|---|---|---|---|---|---|---|---|---|---|---|---|---|---|---|
| <1hr | 15min | 0.19 | 0.22 | 0.81 | 0.06 | 0.05 | 0.05 | 0.59 | 0.06 | 0.02 | 0.01 | 0.58 | 0.61 | 0.01 | 0.00 | 0.79 | 0.06 | 0.05 | 0.05 | 0.61 | 0.06 | 0.02 | 0.02 | 0.60 | 0.62 | 0.01 | 0.00 |
| <1hr | 60min | 0.09 | 0.27 | 0.90 | 0.19 | 0.07 | 0.20 | 0.64 | 0.05 | 0.05 | 0.03 | 0.65 | 0.68 | 0.04 | 0.02 | 0.88 | 0.21 | 0.08 | 0.22 | 0.67 | 0.08 | 0.08 | 0.07 | 0.68 | 0.70 | 0.08 | 0.02 |
| <1hr | 180min | 0.12 | 0.19 | 0.94 | 0.18 | 0.23 | 0.29 | 0.68 | 0.05 | 0.04 | 0.05 | 0.72 | 0.70 | 0.28 | 0.00 | 0.92 | 0.21 | 0.25 | 0.31 | 0.69 | 0.05 | 0.05 | 0.09 | 0.73 | 0.69 | 0.38 | 0.02 |
| <3hr | 15min | 0.12 | 0.19 | 0.61 | 0.04 | 0.03 | 0.04 | 0.45 | 0.00 | 0.01 | 0.00 | 0.43 | 0.49 | 0.01 | 0.01 | 0.60 | 0.05 | 0.02 | 0.04 | 0.46 | 0.00 | 0.02 | 0.02 | 0.45 | 0.50 | 0.01 | 0.01 |
| <3hr | 60min | 0.04 | 0.25 | 0.72 | 0.13 | 0.01 | 0.13 | 0.48 | 0.01 | 0.03 | 0.03 | 0.55 | 0.55 | 0.03 | 0.01 | 0.69 | 0.15 | 0.03 | 0.15 | 0.51 | 0.02 | 0.05 | 0.05 | 0.56 | 0.55 | 0.07 | 0.02 |
| <3hr | 180min | 0.09 | 0.13 | 0.80 | 0.16 | 0.20 | 0.25 | 0.58 | 0.10 | 0.01 | 0.05 | 0.63 | 0.57 | 0.24 | 0.00 | 0.77 | 0.20 | 0.23 | 0.28 | 0.57 | 0.11 | 0.00 | 0.09 | 0.61 | 0.55 | 0.32 | 0.02 |
| >3hr | 15min | 0.05 | 0.13 | 0.32 | 0.04 | 0.00 | 0.03 | 0.22 | 0.00 | 0.01 | 0.01 | 0.24 | 0.25 | 0.01 | 0.01 | 0.31 | 0.04 | 0.01 | 0.04 | 0.21 | 0.00 | 0.02 | 0.03 | 0.25 | 0.25 | 0.01 | 0.02 |
| >3hr | 60min | 0.03 | 0.14 | 0.42 | 0.13 | 0.09 | 0.14 | 0.27 | 0.07 | 0.02 | 0.02 | 0.32 | 0.26 | 0.02 | 0.02 | 0.39 | 0.15 | 0.10 | 0.16 | 0.27 | 0.07 | 0.02 | 0.05 | 0.32 | 0.25 | 0.05 | 0.02 |
| >3hr | 180min | 0.06 | 0.07 | 0.53 | 0.17 | 0.19 | 0.22 | 0.39 | 0.16 | 0.05 | 0.07 | 0.41 | 0.34 | 0.16 | 0.06 | 0.50 | 0.20 | 0.22 | 0.25 | 0.38 | 0.18 | 0.07 | 0.11 | 0.40 | 0.32 | 0.20 | 0.08 |
| **Average** | | **0.09** | **0.18** | **0.67** | **0.12** | **0.10** | **0.15** | **0.48** | **0.05** | **0.03** | **0.03** | **0.50** | **0.49** | **0.09** | **0.02** | **0.65** | **0.14** | **0.11** | **0.17** | **0.48** | **0.07** | **0.04** | **0.06** | **0.51** | **0.49** | **0.12** | **0.02** |

**Intensity**

| Durations | Lead Time | Cells | Lnow | A | Iave | Imax | Isd1 | Isd2 | Vg | Vx | Vy | Jmax | Jmin | Jr | Φ | Area | meanPi | maxPi | sdPI1 | sdPI2 | Velg | Velx | Vely | Jx | Jy | Jr | phi |
|---|---|---|---|---|---|---|---|---|---|---|---|---|---|---|---|---|---|---|---|---|---|---|---|---|---|---|---|
| <1hr | 15min | 0.02 | 0.05 | 0.00 | 0.55 | 0.52 | 0.54 | 0.11 | 0.06 | 0.03 | 0.00 | 0.02 | 0.02 | 0.00 | 0.01 | 0.00 | 0.52 | 0.50 | 0.52 | 0.11 | 0.06 | 0.03 | 0.01 | 0.02 | 0.02 | 0.00 | 0.01 |
| <1hr | 60min | 0.04 | 0.01 | 0.12 | 0.70 | 0.61 | 0.69 | 0.06 | 0.04 | 0.01 | 0.07 | 0.00 | 0.01 | 0.02 | 0.02 | 0.14 | 0.64 | 0.59 | 0.65 | 0.03 | 0.05 | 0.01 | 0.09 | 0.02 | 0.02 | 0.02 | 0.01 |
| <1hr | 180min | 0.03 | 0.13 | 0.11 | 0.81 | 0.68 | 0.77 | 0.13 | 0.09 | 0.09 | 0.03 | 0.14 | 0.15 | 0.05 | 0.06 | 0.11 | 0.76 | 0.68 | 0.75 | 0.13 | 0.13 | 0.11 | 0.04 | 0.14 | 0.15 | 0.06 | 0.10 |
| <3hr | 15min | 0.02 | 0.10 | 0.11 | 0.15 | 0.22 | 0.17 | 0.14 | 0.04 | 0.02 | 0.01 | 0.08 | 0.10 | 0.01 | 0.00 | 0.10 | 0.14 | 0.20 | 0.16 | 0.13 | 0.04 | 0.02 | 0.01 | 0.08 | 0.09 | 0.02 | 0.01 |
| <3hr | 60min | 0.01 | 0.06 | 0.01 | 0.31 | 0.45 | 0.37 | 0.10 | 0.02 | 0.02 | 0.03 | 0.07 | 0.07 | 0.02 | 0.05 | 0.01 | 0.28 | 0.43 | 0.34 | 0.09 | 0.03 | 0.02 | 0.06 | 0.06 | 0.05 | 0.04 | 0.05 |
| <3hr | 180min | 0.01 | 0.06 | 0.10 | 0.43 | 0.50 | 0.47 | 0.20 | 0.08 | 0.06 | 0.01 | 0.25 | 0.22 | 0.09 | 0.09 | 0.08 | 0.42 | 0.47 | 0.44 | 0.19 | 0.10 | 0.08 | 0.01 | 0.24 | 0.21 | 0.12 | 0.10 |
| >3hr | 15min | 0.03 | 0.11 | 0.12 | 0.02 | 0.08 | 0.03 | 0.11 | 0.02 | 0.01 | 0.01 | 0.08 | 0.10 | 0.01 | 0.01 | 0.11 | 0.01 | 0.06 | 0.02 | 0.10 | 0.02 | 0.01 | 0.01 | 0.08 | 0.10 | 0.01 | 0.02 |
| >3hr | 60min | 0.02 | 0.06 | 0.08 | 0.07 | 0.17 | 0.11 | 0.09 | 0.02 | 0.00 | 0.03 | 0.06 | 0.04 | 0.02 | 0.04 | 0.05 | 0.06 | 0.16 | 0.10 | 0.08 | 0.02 | 0.00 | 0.05 | 0.05 | 0.03 | 0.04 | 0.04 |
| >3hr | 180min | 0.01 | 0.05 | 0.36 | 0.10 | 0.10 | 0.06 | 0.31 | 0.03 | 0.02 | 0.10 | 0.38 | 0.35 | 0.11 | 0.05 | 0.34 | 0.07 | 0.05 | 0.02 | 0.30 | 0.05 | 0.05 | 0.16 | 0.36 | 0.33 | 0.15 | 0.05 |
| **Average** | | **0.02** | **0.07** | **0.11** | **0.35** | **0.37** | **0.36** | **0.14** | **0.04** | **0.03** | **0.03** | **0.12** | **0.12** | **0.03** | **0.04** | **0.10** | **0.32** | **0.35** | **0.33** | **0.13** | **0.05** | **0.04** | **0.05** | **0.12** | **0.11** | **0.05** | **0.04** |

**Velocity X**

| Durations | Lead Time | Cells | Lnow | A | Iave | Imax | Isd1 | Isd2 | Vg | Vx | Vy | Jmax | Jmin | Jr | Φ | A30 | Iave30 | Imax30 | Isd130 | Isd230 | Vg30 | Vx30 | Vy30 | Jmax30 | Jmin30 | Jr30 | Φ30 |
|---|---|---|---|---|---|---|---|---|---|---|---|---|---|---|---|---|---|---|---|---|---|---|---|---|---|---|---|
| <1hr | 15min | 0.04 | 0.02 | 0.09 | 0.01 | 0.01 | 0.00 | 0.06 | 0.14 | 0.17 | 0.02 | 0.06 | 0.05 | 0.01 | 0.02 | 0.08 | 0.01 | 0.01 | 0.00 | 0.08 | 0.13 | 0.18 | 0.02 | 0.14 | 0.07 | 0.02 | 0.02 |
| <1hr | 60min | 0.03 | 0.03 | 0.12 | 0.03 | 0.02 | 0.02 | 0.04 | 0.31 | 0.37 | 0.06 | 0.10 | 0.03 | 0.01 | 0.03 | 0.11 | 0.04 | 0.02 | 0.03 | 0.04 | 0.33 | 0.52 | 0.09 | 0.15 | 0.04 | 0.00 | 0.03 |
| <1hr | 180min | 0.04 | 0.01 | 0.06 | 0.05 | 0.04 | 0.05 | 0.00 | 0.27 | 0.32 | 0.05 | 0.12 | 0.06 | 0.01 | 0.06 | 0.07 | 0.04 | 0.03 | 0.04 | 0.00 | 0.35 | 0.42 | 0.05 | 0.16 | 0.07 | 0.01 | 0.05 |
| <3hr | 15min | 0.03 | 0.06 | 0.10 | 0.02 | 0.01 | 0.01 | 0.06 | 0.03 | 0.07 | 0.01 | 0.05 | 0.04 | 0.01 | 0.02 | 0.08 | 0.02 | 0.00 | 0.02 | 0.05 | 0.03 | 0.06 | 0.01 | 0.15 | 0.03 | 0.02 | 0.03 |
| <3hr | 60min | 0.06 | 0.06 | 0.15 | 0.03 | 0.03 | 0.02 | 0.06 | 0.20 | 0.30 | 0.06 | 0.11 | 0.05 | 0.01 | 0.04 | 0.14 | 0.05 | 0.03 | 0.03 | 0.05 | 0.25 | 0.42 | 0.07 | 0.16 | 0.04 | 0.01 | 0.04 |
| <3hr | 180min | 0.04 | 0.01 | 0.10 | 0.03 | 0.02 | 0.02 | 0.02 | 0.27 | 0.26 | 0.04 | 0.13 | 0.07 | 0.00 | 0.06 | 0.10 | 0.02 | 0.02 | 0.02 | 0.02 | 0.29 | 0.38 | 0.05 | 0.18 | 0.07 | 0.00 | 0.05 |
| >3hr | 15min | 0.04 | 0.06 | 0.10 | 0.02 | 0.01 | 0.01 | 0.04 | 0.02 | 0.05 | 0.02 | 0.04 | 0.01 | 0.01 | 0.02 | 0.09 | 0.02 | 0.00 | 0.02 | 0.04 | 0.02 | 0.05 | 0.01 | 0.15 | 0.01 | 0.01 | 0.02 |
| >3hr | 60min | 0.04 | 0.04 | 0.05 | 0.04 | 0.03 | 0.04 | 0.04 | 0.07 | 0.16 | 0.05 | 0.00 | 0.04 | 0.00 | 0.02 | 0.04 | 0.05 | 0.03 | 0.04 | 0.05 | 0.08 | 0.23 | 0.07 | 0.15 | 0.05 | 0.02 | 0.02 |
| >3hr | 180min | 0.03 | 0.02 | 0.10 | 0.00 | 0.02 | 0.02 | 0.03 | 0.15 | 0.17 | 0.03 | 0.11 | 0.04 | 0.01 | 0.03 | 0.10 | 0.01 | 0.02 | 0.02 | 0.03 | 0.16 | 0.24 | 0.03 | 0.15 | 0.05 | 0.01 | 0.04 |
| **Average** | | **0.04** | **0.03** | **0.10** | **0.03** | **0.02** | **0.02** | **0.04** | **0.16** | **0.21** | **0.04** | **0.08** | **0.04** | **0.01** | **0.03** | **0.09** | **0.03** | **0.02** | **0.02** | **0.04** | **0.18** | **0.28** | **0.04** | **0.15** | **0.05** | **0.01** | **0.03** |

**Velocity Y**

| Durations | Lead Time | Cells | Lnow | A | Iave | Imax | Isd1 | Isd2 | Vg | Vx | Vy | Jmax | Jmin | Jr | Φ | A30 | Iave30 | Imax30 | Isd130 | Isd230 | Vg30 | Vx30 | Vy30 | Jmax30 | Jmin30 | Jr30 | Φ30 |
|---|---|---|---|---|---|---|---|---|---|---|---|---|---|---|---|---|---|---|---|---|---|---|---|---|---|---|---|
| <1hr | 15min | 0.04 | 0.04 | 0.04 | 0.02 | 0.05 | 0.03 | 0.06 | 0.03 | 0.02 | 0.15 | 0.03 | 0.03 | 0.01 | 0.00 | 0.04 | 0.02 | 0.04 | 0.03 | 0.07 | 0.04 | 0.03 | 0.17 | 0.04 | 0.03 | 0.01 | 0.00 |
| <1hr | 60min | 0.00 | 0.04 | 0.02 | 0.08 | 0.09 | 0.08 | 0.00 | 0.03 | 0.05 | 0.22 | 0.00 | 0.00 | 0.01 | 0.02 | 0.03 | 0.08 | 0.09 | 0.08 | 0.00 | 0.01 | 0.06 | 0.33 | 0.01 | 0.00 | 0.02 | 0.02 |
| <1hr | 180min | 0.03 | 0.08 | 0.05 | 0.02 | 0.03 | 0.03 | 0.05 | 0.00 | 0.04 | 0.27 | 0.07 | 0.01 | 0.02 | 0.00 | 0.06 | 0.02 | 0.03 | 0.03 | 0.06 | 0.01 | 0.05 | 0.41 | 0.08 | 0.02 | 0.01 | 0.02 |
| <3hr | 15min | 0.01 | 0.06 | 0.06 | 0.03 | 0.07 | 0.04 | 0.07 | 0.01 | 0.01 | 0.05 | 0.03 | 0.05 | 0.00 | 0.01 | 0.05 | 0.03 | 0.06 | 0.04 | 0.06 | 0.02 | 0.00 | 0.04 | 0.02 | 0.00 | 0.00 | 0.00 |
| <3hr | 60min | 0.01 | 0.06 | 0.02 | 0.04 | 0.10 | 0.06 | 0.03 | 0.01 | 0.06 | 0.18 | 0.02 | 0.05 | 0.01 | 0.01 | 0.00 | 0.04 | 0.11 | 0.06 | 0.03 | 0.00 | 0.07 | 0.26 | 0.01 | 0.04 | 0.01 | 0.01 |
| <3hr | 180min | 0.00 | 0.06 | 0.03 | 0.00 | 0.00 | 0.00 | 0.06 | 0.01 | 0.03 | 0.22 | 0.07 | 0.04 | 0.01 | 0.01 | 0.04 | 0.01 | 0.00 | 0.00 | 0.07 | 0.00 | 0.04 | 0.33 | 0.07 | 0.05 | 0.01 | 0.01 |
| >3hr | 15min | 0.01 | 0.07 | 0.03 | 0.00 | 0.03 | 0.01 | 0.03 | 0.01 | 0.01 | 0.04 | 0.00 | 0.01 | 0.01 | 0.03 | 0.02 | 0.00 | 0.02 | 0.00 | 0.02 | 0.01 | 0.01 | 0.03 | 0.00 | 0.00 | 0.01 | 0.03 |
| >3hr | 60min | 0.03 | 0.02 | 0.02 | 0.01 | 0.03 | 0.01 | 0.07 | 0.00 | 0.04 | 0.09 | 0.09 | 0.08 | 0.01 | 0.01 | 0.04 | 0.01 | 0.03 | 0.00 | 0.09 | 0.01 | 0.07 | 0.16 | 0.10 | 0.10 | 0.01 | 0.00 |
| >3hr | 180min | 0.00 | 0.01 | 0.01 | 0.05 | 0.04 | 0.04 | 0.04 | 0.01 | 0.02 | 0.14 | 0.09 | 0.08 | 0.01 | 0.02 | 0.02 | 0.05 | 0.04 | 0.04 | 0.05 | 0.02 | 0.01 | 0.22 | 0.05 | 0.08 | 0.00 | 0.00 |
| **Average** | | **0.01** | **0.05** | **0.03** | **0.03** | **0.05** | **0.03** | **0.05** | **0.01** | **0.03** | **0.15** | **0.04** | **0.04** | **0.01** | **0.01** | **0.03** | **0.03** | **0.05** | **0.03** | **0.05** | **0.01** | **0.04** | **0.22** | **0.05** | **0.04** | **0.01** | **0.01** |

**Duration**

| Durations | Lead Time | Cells | Lnow | A | Iave | Imax | Isd1 | Isd2 | Vg | Vx | Vy | Jmax | Jmin | Jr | Φ | A30 | Iave30 | Imax30 | Isd130 | Isd230 | Vg30 | Vx30 | Vy30 | Jmax30 | Jmin30 | Jr30 | Φ30 |
|---|---|---|---|---|---|---|---|---|---|---|---|---|---|---|---|---|---|---|---|---|---|---|---|---|---|---|---|
| Dur <1hr | | 0.06 | 0.16 | 0.31 | 0.02 | 0.04 | 0.00 | 0.22 | 0.03 | 0.03 | 0.00 | 0.19 | 0.24 | 0.00 | 0.03 | 0.30 | 0.02 | 0.03 | 0.01 | 0.21 | 0.03 | 0.03 | 0.01 | 0.19 | 0.25 | 0.01 | 0.04 |
| Dur <3hr | | 0.00 | 0.16 | 0.35 | 0.10 | 0.04 | 0.10 | 0.22 | 0.08 | 0.02 | 0.04 | 0.23 | 0.21 | 0.02 | 0.11 | 0.32 | 0.12 | 0.05 | 0.11 | 0.22 | 0.08 | 0.03 | 0.07 | 0.23 | 0.21 | 0.05 | 0.15 |
| Dur >3hr | | 0.07 | 0.02 | 0.43 | 0.20 | 0.18 | 0.21 | 0.21 | 0.15 | 0.04 | 0.04 | 0.25 | 0.16 | 0.14 | 0.01 | 0.40 | 0.22 | 0.20 | 0.23 | 0.20 | 0.18 | 0.07 | 0.07 | 0.23 | 0.14 | 0.18 | 0.01 |
| **Average** | | **0.04** | **0.11** | **0.36** | **0.11** | **0.09** | **0.10** | **0.22** | **0.09** | **0.03** | **0.03** | **0.22** | **0.20** | **0.05** | **0.05** | **0.34** | **0.12** | **0.09** | **0.12** | **0.21** | **0.10** | **0.04** | **0.05** | **0.22** | **0.20** | **0.08** | **0.07** |

*Appendix 11-2* *Strength of relationship between the selected predictors and the target variables averaged for three lead times and storm duration groups based on PIC method. The green shade indicates the strength of the relationship: with 0 for no relationship at all, and 1 for highest dependency. The averaged computed values for each target variable (last row) are used as bases for* **Table 3**. *For intensity, velocity in x and y direction, since the PIC recognized only one predictor as important, the average values is given as 1 for the selected respective predictor.*

**Area**

| Durations | Lead Time | Cells | Lnow | A | Iave | Imax | Isd1 | Isd2 | Vg | Vx | Vy | Jmax | Jmin | Jr | Φ | $A_{30}$ | $Iave_{30}$ | $Imax_{30}$ | $Isd1_{30}$ | $Isd2_{30}$ | $Vg_{30}$ | $Vx_{30}$ | $Vy_{30}$ | $Jmax_{30}$ | $Jmin_{30}$ | $Jr_{30}$ | $\Phi_{30}$ |
|---|---|---|---|---|---|---|---|---|---|---|---|---|---|---|---|---|---|---|---|---|---|---|---|---|---|---|---|
| <1hr | 15min | 0.00 | 0.00 | 0.00 | 0.00 | 0.00 | 0.00 | 0.00 | 0.00 | 0.00 | 0.00 | 0.00 | 0.00 | 0.00 | 0.00 | 0.00 | 0.00 | 0.00 | 0.00 | 0.00 | 0.00 | 0.00 | 0.00 | 0.00 | 0.00 | 1.00 | 0.00 |
| | 60min | 0.00 | 0.00 | 0.00 | 0.00 | 0.00 | 0.00 | 0.00 | 0.00 | 0.00 | 0.00 | 0.00 | 0.00 | 0.00 | 0.00 | 0.00 | 0.00 | 0.00 | 0.00 | 0.00 | 0.00 | 0.00 | 0.00 | 0.00 | 0.00 | 1.00 | 0.00 |
| | 180min | 0.00 | 0.00 | 0.00 | 0.00 | 0.00 | 0.00 | 0.00 | 0.00 | 0.00 | 0.00 | 0.00 | 0.00 | 0.00 | 0.00 | 0.00 | 0.00 | 0.00 | 0.00 | 0.00 | 0.00 | 0.00 | 0.00 | 0.00 | 0.00 | 1.00 | 0.00 |
| <3hr | 15min | 0.00 | 0.00 | 0.00 | 0.00 | 0.00 | 0.00 | 0.00 | 0.00 | 0.00 | 0.00 | 0.00 | 0.00 | 0.00 | 0.00 | 0.00 | 0.00 | 0.00 | 0.00 | 1.00 | 0.00 | 0.00 | 0.00 | 0.00 | 0.00 | 0.00 | 0.00 |
| | 60min | 0.00 | 0.00 | 0.00 | 0.00 | 0.00 | 0.00 | 0.00 | 0.00 | 0.00 | 0.00 | 0.00 | 0.00 | 0.00 | 0.00 | 0.00 | 0.00 | 0.00 | 0.00 | 1.00 | 0.00 | 0.00 | 0.00 | 0.00 | 0.00 | 0.00 | 0.00 |
| | 180min | 0.00 | 0.00 | 0.00 | 0.00 | 0.00 | 0.00 | 0.00 | 0.00 | 0.00 | 0.00 | 0.00 | 0.00 | 0.00 | 0.00 | 0.00 | 0.00 | 0.00 | 0.00 | 1.00 | 0.00 | 0.00 | 0.00 | 0.00 | 0.00 | 0.00 | 0.00 |
| >3hr | 15min | 0.00 | 0.10 | 0.25 | 0.00 | 0.00 | 0.00 | 0.00 | 0.57 | 0.00 | 0.00 | 0.00 | 0.00 | 0.00 | 0.00 | 0.08 | 0.00 | 0.00 | 0.00 | 0.00 | 0.00 | 0.00 | 0.00 | 0.00 | 0.00 | 0.00 | 0.00 |
| | 60min | 0.00 | 0.29 | 0.67 | 0.00 | 0.00 | 0.00 | 0.00 | 0.66 | 0.00 | 0.00 | 0.00 | 0.00 | 0.00 | 0.00 | 0.00 | 0.00 | 0.00 | 0.00 | 0.00 | 0.00 | 0.00 | 0.33 | 0.00 | 0.00 | 0.00 | 0.00 |
| | 180min | 0.00 | 0.30 | 0.40 | 0.00 | 0.00 | 0.00 | 0.00 | 0.72 | 0.00 | 0.00 | 0.00 | 0.00 | 0.00 | 0.00 | 0.00 | 0.00 | 0.00 | 0.00 | 0.00 | 0.00 | 0.00 | 0.28 | 0.00 | 0.00 | 0.00 | 0.00 |
| **Average** | | 0.00 | 0.08 | 0.15 | 0.00 | 0.00 | 0.00 | 0.00 | 0.22 | 0.00 | 0.00 | 0.00 | 0.00 | 0.00 | 0.00 | 0.01 | 0.00 | 0.00 | 0.00 | 0.33 | 0.00 | 0.00 | 0.07 | 0.00 | 0.00 | 0.33 | 0.00 |

**Intensity**

| Durations | Lead Time | Cells | Lnow | A | Iave | Imax | Isd1 | Isd2 | Vg | Vx | Vy | Jmax | Jmin | Jr | Φ | $A_{30}$ | $Iave_{30}$ | $Imax_{30}$ | $Isd1_{30}$ | $Isd2_{30}$ | $Vg_{30}$ | $Vx_{30}$ | $Vy_{30}$ | $Jmax_{30}$ | $Jmin_{30}$ | $Jr_{30}$ | $\Phi_{30}$ |
|---|---|---|---|---|---|---|---|---|---|---|---|---|---|---|---|---|---|---|---|---|---|---|---|---|---|---|---|
| <1hr | 15min | 0.00 | 0.00 | 0.00 | 0.00 | 0.00 | 0.00 | 0.00 | 0.00 | 0.00 | 0.00 | 0.00 | 0.00 | 0.00 | 0.00 | 0.00 | 0.00 | 0.00 | 0.00 | 0.00 | 0.00 | 0.00 | 0.00 | 0.00 | 0.00 | 0.00 | 0.00 |
| | 60min | 0.00 | 0.00 | 0.00 | 0.00 | 0.00 | 0.00 | 0.00 | 0.00 | 0.00 | 0.00 | 0.00 | 0.00 | 0.00 | 0.00 | 0.00 | 0.00 | 0.00 | 0.00 | 0.00 | 0.00 | 0.00 | 0.00 | 0.00 | 0.00 | 0.00 | 0.00 |
| | 180min | 0.00 | 0.00 | 0.00 | 0.00 | 0.00 | 0.00 | 0.00 | 0.00 | 0.00 | 0.00 | 0.00 | 0.00 | 0.00 | 0.00 | 0.00 | 0.00 | 0.00 | 0.00 | 0.00 | 0.00 | 0.00 | 0.00 | 0.00 | 0.00 | 0.00 | 0.00 |
| <3hr | 15min | 0.00 | 0.00 | 0.00 | 0.00 | 0.00 | 0.00 | 0.00 | 0.00 | 0.00 | 0.00 | 1.00 | 0.00 | 0.00 | 0.00 | 0.00 | 0.00 | 0.00 | 0.00 | 0.00 | 0.00 | 0.00 | 0.00 | 0.00 | 0.00 | 0.00 | 0.00 |
| | 60min | 0.00 | 0.00 | 0.00 | 0.00 | 0.00 | 0.00 | 0.00 | 0.00 | 0.00 | 0.00 | 1.00 | 0.00 | 0.00 | 0.00 | 0.00 | 0.00 | 0.00 | 0.00 | 0.00 | 0.00 | 0.00 | 0.00 | 0.00 | 0.00 | 0.00 | 0.00 |
| | 180min | 0.00 | 0.00 | 0.00 | 0.00 | 0.00 | 0.00 | 0.00 | 0.00 | 0.00 | 0.00 | 1.00 | 0.00 | 0.00 | 0.00 | 0.00 | 0.00 | 0.00 | 0.00 | 0.00 | 0.00 | 0.00 | 0.00 | 0.00 | 0.00 | 0.00 | 0.00 |
| >3hr | 15min | 0.00 | 0.00 | 0.00 | 0.00 | 0.00 | 0.00 | 0.00 | 0.00 | 0.00 | 0.00 | 1.00 | 0.00 | 0.00 | 0.00 | 0.00 | 0.00 | 0.00 | 0.00 | 0.00 | 0.00 | 0.00 | 0.00 | 0.00 | 0.00 | 0.00 | 0.00 |
| | 60min | 0.00 | 0.00 | 0.00 | 0.00 | 0.00 | 0.00 | 0.00 | 0.00 | 0.00 | 0.00 | 1.00 | 0.00 | 0.00 | 0.00 | 0.00 | 0.00 | 0.00 | 0.00 | 0.00 | 0.00 | 0.00 | 0.00 | 0.00 | 0.00 | 0.00 | 0.00 |
| | 180min | 0.00 | 0.00 | 0.00 | 0.00 | 0.00 | 0.00 | 0.00 | 0.00 | 0.00 | 0.00 | 1.00 | 0.00 | 0.00 | 0.00 | 0.00 | 0.00 | 0.00 | 0.00 | 0.00 | 0.00 | 0.00 | 0.00 | 0.00 | 0.00 | 0.00 | 0.00 |
| **Average** | | 0.00 | 0.00 | 0.00 | 0.00 | 0.00 | 0.00 | 0.00 | 0.00 | 0.00 | 0.00 | 1.00 | 0.00 | 0.00 | 0.00 | 0.00 | 0.00 | 0.00 | 0.00 | 0.00 | 0.00 | 0.00 | 0.00 | 0.00 | 0.00 | 0.00 | 0.00 |

**Velocity X**

| Durations | Lead Time | Cells | Lnow | A | Iave | Imax | Isd1 | Isd2 | Vg | Vx | Vy | Jmax | Jmin | Jr | Φ | $A_{30}$ | $Iave_{30}$ | $Imax_{30}$ | $Isd1_{30}$ | $Isd2_{30}$ | $Vg_{30}$ | $Vx_{30}$ | $Vy_{30}$ | $Jmax_{30}$ | $Jmin_{30}$ | $Jr_{30}$ | $\Phi_{30}$ |
|---|---|---|---|---|---|---|---|---|---|---|---|---|---|---|---|---|---|---|---|---|---|---|---|---|---|---|---|
| <1hr | 15min | 0.00 | 0.00 | 0.00 | 0.00 | 0.00 | 0.00 | 0.00 | 0.00 | 0.00 | 0.00 | 0.00 | 0.00 | 0.00 | 0.00 | 0.00 | 0.00 | 0.00 | 0.00 | 0.00 | 0.00 | 0.00 | 0.00 | 0.00 | 0.00 | 0.00 | 0.00 |
| | 60min | 0.00 | 0.00 | 0.00 | 0.00 | 0.00 | 0.00 | 0.00 | 0.00 | 0.00 | 0.00 | 0.00 | 0.00 | 0.00 | 0.00 | 0.00 | 0.00 | 0.00 | 0.00 | 0.00 | 0.00 | 0.00 | 0.00 | 0.00 | 0.00 | 0.00 | 0.00 |
| | 180min | 0.00 | 0.00 | 0.00 | 0.00 | 0.00 | 0.00 | 0.00 | 0.00 | 0.00 | 0.00 | 0.00 | 0.00 | 0.00 | 0.00 | 0.00 | 0.00 | 0.00 | 0.00 | 0.00 | 0.00 | 0.00 | 0.00 | 0.00 | 0.00 | 0.00 | 0.00 |
| <3hr | 15min | 0.00 | 0.00 | 0.00 | 0.00 | 0.00 | 0.00 | 0.00 | 0.00 | 0.00 | 0.00 | 0.00 | 0.00 | 0.00 | 0.00 | 0.00 | 0.00 | 0.00 | 0.00 | 0.00 | 0.00 | 1.00 | 0.00 | 0.00 | 0.00 | 0.00 | 0.00 |
| | 60min | 0.00 | 0.00 | 0.00 | 0.00 | 0.00 | 0.00 | 0.00 | 0.00 | 0.00 | 0.00 | 0.00 | 0.00 | 0.00 | 0.00 | 0.00 | 0.00 | 0.00 | 0.00 | 0.00 | 0.00 | 1.00 | 0.00 | 0.00 | 0.00 | 0.00 | 0.00 |
| | 180min | 0.00 | 0.00 | 0.00 | 0.00 | 0.00 | 0.00 | 0.00 | 0.00 | 0.00 | 0.00 | 0.00 | 0.00 | 0.00 | 0.00 | 0.00 | 0.00 | 0.00 | 0.00 | 0.00 | 0.00 | 1.00 | 0.00 | 0.00 | 0.00 | 0.00 | 0.00 |
| >3hr | 15min | 0.00 | 0.00 | 0.00 | 0.00 | 0.00 | 0.00 | 0.00 | 0.00 | 0.00 | 0.00 | 0.00 | 0.00 | 0.00 | 0.00 | 0.00 | 0.00 | 0.00 | 0.00 | 0.00 | 0.00 | 0.00 | 0.00 | 0.00 | 0.00 | 0.00 | 0.00 |
| | 60min | 0.00 | 0.00 | 0.00 | 0.00 | 0.00 | 0.00 | 0.00 | 0.00 | 0.00 | 0.00 | 0.00 | 0.00 | 0.00 | 0.00 | 0.00 | 0.00 | 0.00 | 0.00 | 0.00 | 0.00 | 0.00 | 0.00 | 0.00 | 0.00 | 0.00 | 0.00 |
| | 180min | 0.00 | 0.00 | 0.00 | 0.00 | 0.00 | 0.00 | 0.00 | 0.00 | 0.00 | 0.00 | 0.00 | 0.00 | 0.00 | 0.00 | 0.00 | 0.00 | 0.00 | 0.00 | 0.00 | 0.00 | 0.00 | 0.00 | 0.00 | 0.00 | 0.00 | 0.00 |
| **Average** | | 0.00 | 0.00 | 0.00 | 0.00 | 0.00 | 0.00 | 0.00 | 0.00 | 0.00 | 0.00 | 0.00 | 0.00 | 0.00 | 0.00 | 0.00 | 0.00 | 0.00 | 0.00 | 0.00 | 0.00 | 1.00 | 0.00 | 0.00 | 0.00 | 0.00 | 0.00 |

**Velocity Y**

| Durations | Lead Time | Cells | Lnow | A | Iave | Imax | Isd1 | Isd2 | Vg | Vx | Vy | Jmax | Jmin | Jr | Φ | $A_{30}$ | $Iave_{30}$ | $Imax_{30}$ | $Isd1_{30}$ | $Isd2_{30}$ | $Vg_{30}$ | $Vx_{30}$ | $Vy_{30}$ | $Jmax_{30}$ | $Jmin_{30}$ | $Jr_{30}$ | $\Phi_{30}$ |
|---|---|---|---|---|---|---|---|---|---|---|---|---|---|---|---|---|---|---|---|---|---|---|---|---|---|---|---|
| <1hr | 15min | 0.00 | 0.00 | 0.00 | 0.00 | 0.00 | 0.00 | 0.00 | 0.00 | 0.00 | 0.00 | 0.00 | 0.00 | 0.00 | 0.00 | 0.00 | 0.00 | 0.00 | 0.00 | 0.00 | 0.00 | 0.00 | 0.00 | 0.00 | 0.00 | 0.00 | 0.00 |
| | 60min | 0.00 | 0.00 | 0.00 | 0.00 | 0.00 | 0.00 | 0.00 | 0.00 | 0.00 | 0.00 | 0.00 | 0.00 | 0.00 | 0.00 | 0.00 | 0.00 | 0.00 | 0.00 | 0.00 | 0.00 | 0.00 | 0.00 | 0.00 | 0.00 | 0.00 | 0.00 |
| | 180min | 0.00 | 0.00 | 0.00 | 0.00 | 0.00 | 0.00 | 0.00 | 0.00 | 0.00 | 0.00 | 0.00 | 0.00 | 0.00 | 0.00 | 0.00 | 0.00 | 0.00 | 0.00 | 0.00 | 0.00 | 0.00 | 0.00 | 0.00 | 0.00 | 0.00 | 0.00 |
| <3hr | 15min | 0.00 | 0.00 | 0.00 | 0.00 | 0.00 | 0.00 | 0.00 | 0.00 | 0.00 | 0.00 | 0.00 | 0.00 | 0.00 | 0.00 | 0.00 | 0.00 | 0.00 | 0.00 | 0.00 | 0.00 | 0.00 | 0.00 | 0.00 | 0.00 | 0.00 | 0.00 |
| | 60min | 0.00 | 0.00 | 0.00 | 0.00 | 0.00 | 0.00 | 0.00 | 0.00 | 0.00 | 0.00 | 0.00 | 0.00 | 0.00 | 0.00 | 0.00 | 0.00 | 0.00 | 0.00 | 0.00 | 0.00 | 0.00 | 0.00 | 0.00 | 0.00 | 0.00 | 0.00 |
| | 180min | 0.00 | 0.00 | 0.00 | 0.00 | 0.00 | 0.00 | 0.00 | 0.00 | 0.00 | 0.00 | 0.00 | 0.00 | 0.00 | 0.00 | 0.00 | 0.00 | 0.00 | 0.00 | 0.00 | 0.00 | 0.00 | 0.00 | 0.00 | 0.00 | 0.00 | 0.00 |
| >3hr | 15min | 0.00 | 0.00 | 0.00 | 0.00 | 0.00 | 0.00 | 0.00 | 0.00 | 0.00 | 0.00 | 0.00 | 0.00 | 0.00 | 0.00 | 0.00 | 0.00 | 0.00 | 0.00 | 0.00 | 0.00 | 0.00 | 0.00 | 1.00 | 0.00 | 0.00 | 0.00 |
| | 60min | 0.00 | 0.00 | 0.00 | 0.00 | 0.00 | 0.00 | 0.00 | 0.00 | 0.00 | 0.00 | 0.00 | 0.00 | 0.00 | 0.00 | 0.00 | 0.00 | 0.00 | 0.00 | 0.00 | 0.00 | 0.00 | 0.00 | 1.00 | 0.00 | 0.00 | 0.00 |
| | 180min | 0.00 | 0.00 | 0.00 | 0.00 | 0.00 | 0.00 | 0.00 | 0.00 | 0.00 | 0.00 | 0.00 | 0.00 | 0.00 | 0.00 | 0.00 | 0.00 | 0.00 | 0.00 | 0.00 | 0.00 | 0.00 | 0.00 | 1.00 | 0.00 | 0.00 | 0.00 |
| **Average** | | 0.00 | 0.00 | 0.00 | 0.00 | 0.00 | 0.00 | 0.00 | 0.00 | 0.00 | 0.00 | 0.00 | 0.00 | 0.00 | 0.00 | 0.00 | 0.00 | 0.00 | 0.00 | 0.00 | 0.00 | 0.00 | 0.00 | 1.00 | 0.00 | 0.00 | 0.00 |

**Duration**

| Durations | Lead Time | Cells | Lnow | A | Iave | Imax | Isd1 | Isd2 | Vg | Vx | Vy | Jmax | Jmin | Jr | Φ | $A_{30}$ | $Iave_{30}$ | $Imax_{30}$ | $Isd1_{30}$ | $Isd2_{30}$ | $Vg_{30}$ | $Vx_{30}$ | $Vy_{30}$ | $Jmax_{30}$ | $Jmin_{30}$ | $Jr_{30}$ | $\Phi_{30}$ |
|---|---|---|---|---|---|---|---|---|---|---|---|---|---|---|---|---|---|---|---|---|---|---|---|---|---|---|---|
| Dur <1hr | | 0.00 | 0.00 | 0.00 | 0.00 | 0.00 | 0.00 | 0.00 | 0.00 | 0.00 | 0.00 | 0.00 | 0.00 | 0.00 | 0.00 | 0.00 | 0.00 | 0.00 | 0.00 | 0.00 | 0.00 | 0.00 | 0.00 | 0.00 | 0.00 | 1.00 | 0.00 |
| Dur <3hr | | 0.00 | 0.00 | 0.00 | 0.00 | 0.00 | 0.00 | 0.00 | 0.00 | 0.00 | 0.00 | 0.00 | 0.00 | 0.00 | 0.00 | 0.00 | 0.00 | 0.00 | 0.00 | 0.00 | 1.00 | 0.00 | 0.00 | 0.00 | 0.00 | 0.00 | 0.00 |
| Dur >3hr | | 0.00 | 0.45 | 0.40 | 0.00 | 0.00 | 0.00 | 0.00 | 0.72 | 0.00 | 0.00 | 0.00 | 0.00 | 0.00 | 0.00 | 0.00 | 0.00 | 0.00 | 0.00 | 0.00 | 0.00 | 0.00 | 0.00 | 0.00 | 0.33 | 0.00 | 0.00 |
| **Average** | | 0.00 | 0.15 | 0.13 | 0.00 | 0.00 | 0.00 | 0.00 | 0.24 | 0.00 | 0.00 | 0.00 | 0.00 | 0.00 | 0.00 | 0.00 | 0.00 | 0.00 | 0.00 | 0.00 | 0.33 | 0.00 | 0.00 | 0.00 | 0.11 | 0.33 | 0.00 |

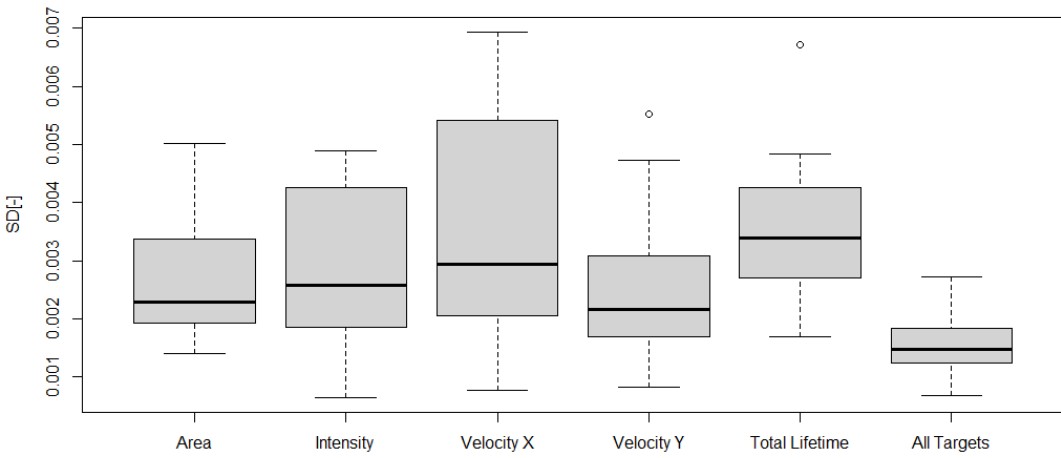

**795**

***Appendix 11-3*** *The standard deviation of the Pearson Correlation Weights between predictors and target variables obtained from a cross-sampling of the events (leave one event at a time out). The boxplot for each target variable describes the spread of the standard deviation over all selected predictors.*

**796**   **12. References**

**797**   Ayzel, G., Scheffer, T., & Heistermann, M. (2020). RainNet v1.0: A convolutional neural network for radar-based

**798**       precipitation nowcasting. *Geoscientific Model Development*, *13*(6). https://doi.org/10.5194/gmd-13-2631-2020

**799**   Bartels, H., Weigl, E., Reich, T., Lang, P., Wagner, A., Kohler, O., Gerlach, N., & MeteoSolutions GmbH. (2004).

**800**       *Projekt RADOLAN - Routineverfahren zur Online-Aneichung der Radarniederschlagsdaten mit Hilfe von*

**801**       *automatischen Bodenniederschlagsstationen (Ombrometer).*

**802**   Berenguer, M., Surcel, M., Zawadzki, I., Xue, M., & Kong, F. (2012). The diurnal cycle of precipitation from

**803**       continental radar mosaics and numerical weather prediction models. Part II: Intercomparison among numerical

**804**       models and with Nowcasting. *Monthly Weather Review*, *140*(8), 2689–2705. https://doi.org/10.1175/MWR-D-11-

**805**       00181.1

**806**   Berndt, C., Rabiei, E., & Haberlandt, U. (2014). Geostatistical merging of rain gauge and radar data for high temporal

**807**       resolutions and various station density scenarios. *Journal of Hydrology*, *508*, 88–101.

**808**       https://doi.org/10.1016/j.jhydrol.2013.10.028

**809**   Berne, A., Delrieu, G., Creutin, J. D., & Obled, C. (2004). Temporal and spatial resolution of rainfall measurements

**810**       required for urban hydrology. *Journal of Hydrology*, *299*, 166–179. https://doi.org/10.1016/S0022-

**811**       1694(04)00363-4

**812**   Bowler, N. E., Pierce, C. E., & Seed, A. W. (2006). STEPS: A probabilistic precipitation forecasting scheme which

**813**       merges an extrapolation nowcast with downscaled NWP. *Quarterly Journal of the Royal Meteorological Society*,

**814**       *132*, 2127–2155. https://doi.org/10.1256/qj.04.100

**815**   Codo, M., & Rico-Ramirez, M. A. (2018). Ensemble radar-based rainfall forecasts for urban hydrological applications.

**816**       *Geosciences (Switzerland)*, *8*(8), 297. https://doi.org/10.3390/geosciences8080297

**817**   Dixon, M., & Wiener, G. (1993). TITAN: thunderstorm identification, tracking, analysis, and nowcasting - a radar-based

**818**       methodology. *Journal of Atmospheric & Oceanic Technology*, *10*(6), 785–797. https://doi.org/10.1175/1520-

**819**       0426(1993)010<0785:TTITAA>2.0.CO;2

**820**   Foresti, L., Reyniers, M., Seed, A., & Delobbe, L. (2016). Development and verification of a real-time stochastic

precipitation nowcasting system for urban hydrology in Belgium. *Hydrology and Earth System Sciences*, *20*(1), 505-527. https://doi.org/10.5194/hess-20-505-2016

Galeati, G. (1990). A comparison of parametric and non-parametric methods for runoff forecasting. *Hydrological Sciences Journal*, *35*(1), 79–94. https://doi.org/10.1080/02626669009492406

Germann, U., & Zawadzki, I. (2004). Scale Dependence of the Predictability of Precipitation from Continental Radar Images. Part II: Probability Forecasts. *Journal of Applied Meteorology*, *43*(1), 74–89. https://doi.org/10.1175/1520-0450(2004)043<0074:SDOTPO>2.0.CO;2

Germann, U., Zawadzki, I., & Turner, B. (2006). Predictability of precipitation from continental radar images. Part IV: Limits to prediction. *Journal of the Atmospheric Sciences*, *63*(8), 2092–2108. https://doi.org/10.1175/JAS3735.1

Gneiting, T., & Katzfuss, M. (2014). Probabilistic Forecasting. *Annual Review of Statistics and Its Application*, *1*(1), 125–151. https://doi.org/10.1146/annurev-statistics-062713-085831

Goudenhoofdt, E., & Delobbe, L. (2013). Statistical characteristics of convective storms in belgium derived from volumetric weather radar observations. *Journal of Applied Meteorology and Climatology*. https://doi.org/10.1175/JAMC-D-12-079.1

Grecu, M., & Krajewski, W. F. (2000). A large-sample investigation of statistical procedures for radar based short-term quantitative precipitation forecasting. *Journal of Hydrology*, *239*(1–4), 69–84. https://doi.org/10.1016/S0022-1694(00)00360-7

Grünewald, U. (2009). *Zu Entstehung und Verlauf des extremen Niederschlags-Abfluss-Ereignisses am 26.07.2008 im Stadtgebiet von Dortmund*.

Haberlandt, U. (2015). Stochastic simulation of daily lows for reservoir planning considering climate change . *Hydrologie Und Wasserbewirtschaftung*, *59*(5), 247–254. https://doi.org/10.5675/HyWa-2015,5-5

Han, L., Fu, S., Zhao, L., Zheng, Y., Wang, H., & Lin, Y. (2009). 3D convective storm identification, tracking, and forecasting - An enhanced TITAN algorithm. *Journal of Atmospheric and Oceanic Technology*, *26*(4), 719-732. https://doi.org/10.1175/2008JTECHA1084.1

Hand, W. H. (1996). An object-oriented technique for nowcasting heavy showers and thunderstorms. *Meteorological Applications*, *3*, 31–41. https://doi.org/10.1002/met.5060030104

Hou, J., & Wang, P. (2017). Storm tracking via tree structure representation of radar data. *Journal of Atmospheric and Oceanic Technology*, *34*, 729–747. https://doi.org/10.1175/JTECH-D-15-0119.1

Imhoff, R. O., Brauer, C. C., Overeem, A., Weerts, A. H., & Uijlenhoet, R. (2020). Spatial and Temporal Evaluation of Radar Rainfall Nowcasting Techniques on 1,533 Events. *Water Resources Research*, *56*(8), 1–22. https://doi.org/10.1029/2019WR026723

Jacobson, C. R. (2011). Identification and quantification of the hydrological impacts of imperviousness in urban catchments: A review. *Journal of Environmental Management*, *92*(6), 1438–1448. https://doi.org/10.1016/j.jenvman.2011.01.018

Jasper-Tönnies, A., Hellmers, S., Einfalt, T., Strehz, A., & Fröhle, P. (2018). Ensembles of radar nowcasts and COSMO-DE-EPS for urban flood management. *Water Science and Technology*, *2017*(1), 27–35. https://doi.org/10.2166/wst.2018.079

Jensen, D. G., Petersen, C., & Rasmussen, M. R. (2015). Assimilation of radar-based nowcast into a HIRLAM NWP model. *Meteorological Applications*, *22*(3), 485–494. https://doi.org/10.1002/met.1479

Johnson, J. T., Mackeen, P. L., Witt, A., Mitchell, E. D., Stumpf, G. J., Eilts, M. D., & Thomas, K. W. (1998). The storm cell identification and tracking algorithm: An enhanced WSR-88D algorithm. *Weather and Forecasting*, *13*(2), 263–276. https://doi.org/10.1175/1520-0434(1998)013<0263:TSCIAT>2.0.CO;2

Jung, S. H., & Lee, G. (2015). Radar-based cell tracking with fuzzy logic approach. *Meteorological Applications*, *22*,
716–730. https://doi.org/10.1002/met.1509

Kato, A., & Maki, M. (2009). Localized heavy rainfall near Zoshigaya, Tokyo, Japan on 5 August 2008 observed by X-
band polarimetric radar - Preliminary analysis. *Scientific Online Letters on the Atmosphere*, *5*(1), 89–92.
https://doi.org/10.2151/sola.2009-023

Kato, R., Shimizu, S., Shimose, K. I., Maesaka, T., Iwanami, K., & Nakagaki, H. (2017). Predictability of meso-γ-scale,
localized, extreme heavy rainfall during the warm season in Japan using high-resolution precipitation nowcasts.
*Quarterly Journal of the Royal Meteorological Society*, *153*(704), 1406–1420. https://doi.org/10.1002/qj.3013

Kober, K., & Tafferner, A. (2009). Tracking and nowcasting of convective cells using remote sensing data from radar
and satellite. *Meteorologische Zeitschrift*, *18*(1), 75–84. https://doi.org/10.1127/0941-2948/2009/359

Krämer, S. (2008). *Quantitative Radardatenaufbereitung für die Niederschlagsvorhersage und die*
*Siedlungsentwässerung*. Leibniz Universität Hannover.

Kyznarová, H., & Novák, P. (2009). CELLTRACK - Convective cell tracking algorithm and its use for deriving life
cycle characteristics. *Atmospheric Research*, *93*(1–3), 317–327. https://doi.org/10.1016/j.atmosres.2008.09.019

L. Foresti, & Seed, A. (2015). On the spatial distribution of rainfall nowcasting errors due to orographic forcing.
*Meteorological Applications*, *22*(1), 60–74.

Lall, U., & Sharma, A. (1996). A Nearest Neighbor Bootstrap For Resampling Hydrologic Time Series. *Water*
*Resources Research*, *32*, 679–693. https://doi.org/10.1029/95WR02966

Lang, P. (2001). Cell tracking and warning indicators derived from operational radar products. *Proceedings of the 30th*
*International Conference on Radar Meteorology, Munich, Germany*, *i*, 245–247.
http://link.springer.com/10.1007/s00376-014-0003-z

Lin, C., Vasić, S., Kilambi, A., Turner, B., & Zawadzki, I. (2005). Precipitation forecast skill of numerical weather
prediction models and radar nowcasts. *Geophysical Research Letters*, *32*, L14801.
https://doi.org/10.1029/2005GL023451

Lucas, B. ., & Kanade, T. (1981). Iterative Technique of Image Registration and Its Application to Stereo. *Proceedings*
*of the International Joint Conference on Neural Networks*, 674–679.

Moseley, C., Berg, P., & Haerter, J. O. (2013). Probing the precipitation life cycle by iterative rain cell tracking. *Journal*
*of Geophysical Research Atmospheres*, *118*(24), 13,361-13,370. https://doi.org/10.1002/2013JD020868

Moseley, C., Henneberg, O., & Haerter, J. O. (2019). A Statistical Model for Isolated Convective Precipitation Events.
*Journal of Advances in Modeling Earth Systems*, *11*(1). https://doi.org/10.1029/2018MS001383

Panziera, L., Germann, U., Gabella, M., & Mandapaka, P. V. (2011). NORA-Nowcasting of Orographic Rainfall by
means of analogues. *Quarterly Journal of the Royal Meteorological Society*, *137*(661), 2106–2123.
https://doi.org/10.1002/qj.878

Pierce, C. E., Ebert, E., Seed, A. W., Sleigh, M., Collier, C. G., Fox, N. I., Donaldson, N., Wilson, J. W., Roberts, R., &
Mueller, C. K. (2004). The nowcasting of precipitation during Sydney 2000: An appraisal of the QPF algorithms.
*Weather and Forecasting*, *19*(1), 7–21. https://doi.org/10.1175/1520-0434(2004)019<0007:TNOPDS>2.0.CO;2

Pierce, C., Seed, A., Ballard, S., Simonin, D., & Li, Z. (2012). *Nowcasting. Doppler Radar Observations - Weather*
*Radar, Wind Profiler, Ionospheric Radar, and Other Advanced Applications* (J. Bech and J. L. Chau (ed.); pp. 97–
142). https://doi.org/10.5772/39054

Rossi, P. J., Chandrasekar, V., Hasu, V., & Moisseev, D. (2015). Kalman filtering-based probabilistic nowcasting of
object-oriented tracked convective storms. *Journal of Atmospheric and Oceanic Technology*, *32*(3), 461–477.
https://doi.org/10.1175/JTECH-D-14-00184.1

Ruzanski, E., Chandrasekar, V., & Wang, Y. (2011). The CASA nowcasting system. *Journal of Atmospheric and*
*Oceanic Technology*, *28*(5), 640–655. https://doi.org/10.1175/2011JTECHA1496.1

Schellart, A., Liguori, S., Krämer, S., Saul, A., & Rico-Ramirez, M. A. (2014). Comparing quantitative precipitation
forecast methods for prediction of sewer flows in a small urban area. *Hydrological Sciences Journal*, *59*(7), 1418–
1436. https://doi.org/10.1080/02626667.2014.920505

Sharma, A., & Mehrotra, R. (2014). An information theoretic alternative to model a natural system using observational
information alone. *Water Resources Research*, *50*(1), 650–660.
https://doi.org/https://doi.org/10.1002/2013WR013845

Sharma, A., Mehrotra, R., Li, J., & Jha, S. (2016). A programming tool for nonparametric system prediction using
Partial Informational Correlation and Partial Weights. *Environmental Modelling & Software*, *83*, 271–275.
https://doi.org/https://doi.org/10.1016/j.envsoft.2016.05.021

Shehu, B. (2020). *Improving the rainfall nowcasting for fine temporal and spatial scales suitable for urban hydrology*.
Leibniz Universität Hannover.

Shehu, B., & Haberlandt, U. (2021). Relevance of merging radar and rainfall gauge data for rainfall nowcasting in urban
hydrology. *Journal of Hydrology*, *594*, 125931. https://doi.org/https://doi.org/10.1016/j.jhydrol.2020.125931

Surcel, M., Zawadzki, I., & Yau, M. K. (2015). A study on the scale dependence of the predictability of precipitation
patterns. *Journal of the Atmospheric Sciences*, *72*(1), 216–235. https://doi.org/10.1175/JAS-D-14-0071.1

United, N. (2018). World Urbanization Prospects. In *Demographic Research*.
file:///C:/Users/rocey/Downloads/WUP2018-Report.pdf

Van Dijk, E., Van Der Meulen, J., Kluck, J., & Straatman, J. H. M. (2014). Comparing modelling techniques for
analysing urban pluvial flooding. *Water Science and Technology*, *69*(2), 305. https://doi.org/10.2166/wst.2013.699

Wilson, J. W., Crook, N. A., Mueller, C. K., Sun, J., & Dixon, M. (1998). Nowcasting Thunderstorms: A Status Report.
*Bulletin of the American Meteorological Society*, *79*(10), 2079–2099. https://doi.org/10.1175/1520-
0477(1998)079<2079:NTASR>2.0.CO;2

Wilson, J. W., Feng, Y., Chen, M., & Roberts, R. D. (2010). Nowcasting challenges during the Beijing olympics:
Successes, failures, and implications for future nowcasting systems. *Weather and Forecasting*, *25*(6), 1691–1714.
https://doi.org/10.1175/2010WAF2222417.1

Winterrath, T., Rosenow, W., & Weigl, E. (2012). On the DWD quantitative precipitation analysis and nowcasting
system for real-time application in German flood risk management. *IAHS-AISH Publication*, *351*, 323–329.

Zahraei, A., Hsu, K. lin, Sorooshian, S., Gourley, J. J., Hong, Y., & Behrangi, A. (2013). Short-term quantitative
precipitation forecasting using an object-based approach. *Journal of Hydrology*, *483*, 1–15.
https://doi.org/10.1016/j.jhydrol.2012.09.052

Zahraei, A., Hsu, K. lin, Sorooshian, S., Gourley, J. J., Lakshmanan, V., Hong, Y., & Bellerby, T. (2012). Quantitative
Precipitation Nowcasting: A Lagrangian Pixel-Based Approach. *Atmospheric Research*, *118*, 418–434.
https://doi.org/10.1016/j.atmosres.2012.07.001

Zawadzki, I. I. (1973). Statistical Properties of Precipitation Patterns. *Journal of Applied Meteorology*, *12*(3), 459–472.
https://doi.org/10.1175/1520-0450(1973)012<0459:spopp>2.0.co;2

Zou, X., Dai, Q., Wu, K., Yang, Q., & Zhang, S. (2020). An empirical ensemble rainfall nowcasting model using multi-
scaled analogues. *Natural Hazards*, *103*(1), 165–188. https://doi.org/10.1007/s11069-020-03964-3