# Peer review of "Improving radar-based rainfall nowcasting by a nearest neighbour approach: Part I – Storm Characteristics"

_Hydrology and Earth System Sciences, 2021_

## Referee Comment (RC1)

Review of: **Improving object-oriented radar based nowcast by a nearest neighbor approach** by Bora Shehu and Uwe Haberlandt

Ruben Imhoff

Ruben.Imhoff@deltares.nl

May 27, 2021

**Summary**

The authors present a novel method to improve object-oriented radar rainfall nowcasts by making use of a k-nearest neighbor approach. The method uses already existing methods and on top of that, it replaces the generally used Lagrangian persistence for storm cell movement and tracking with a k-nearest neighbor algorithm. The results are promising and clearly outperform the benchmark (an object-oriented radar rainfall nowcasting algorithm that uses Lagrangian persistence for storm advection), especially for storms of short durations. These are the more convective storms, that often lead to flooding, and that are also harder to predict by other nowcasting methods. Hence, I think that this work is relevant and an interesting new approach. What I particularly have liked about the approach is the more physical look on using machine learning for a particular part in the nowcasting chain. The choice for a k-nearest neighbor approach makes sense from that perspective, and the results remain explainable (and thus also open for improvements).

After going through the manuscript, I have quite some suggestions and questions left, but I hope that these suggestions will help in further improving the manuscript. I would like to thank the authors for this interesting work and I look forward to seeing the revised version of it.

**General comments**

**Methods**

The unmatched storm cells are left out of the analyses. I do agree with this choice, but I wonder what the effect is on the algorithm performance once the 'normal' radar data, with these unmatched storm cells / artefacts, are fed into the algorithm. Can the authors comment on this?

What is the algorithm performance for extreme events or events that were not part of the training data? In addition, can the authors say some more about the size (memory) of the training dataset, the computation times and how big the training set should be for adequate use? From both an operational and reproducibility perspective, this would be very relevant information.

The authors often mention timestep of nowcast or nowcast time. Up to the end of the manuscript, I have found the terminology and meaning confusing (it may just be me..). Do the authors mean the issue time of the nowcast since the start/evolution of the storm with this? If so, I would recommend changing this for clarity.

I appreciate the use of an adequate benchmark (another object-oriented nowcasting method using Lagrangian persistence for the movement of storm cells) in this study. However, can the authors

comment on other works that take next to Lagrangian persistence also other processes into account, e.g. splits and mergers (e.g. Dixon & Wiener, 1993; Han et al., 2009), and the rate of growth and dissipation (e.g. Pulkkinen et al., 2020 – although not necessarily constructed for object-oriented nowcasts)? Thus, how would this work relate to or even improve such methods? An analysis comparing such a method to this method is of course outside the scope of this paper, but it would be great if the authors can at least comment on it.

*Ensemble approach and analysis*

The construction of the ensembles is interesting, but the description in the methodology was not always clear to me (see the comments mentioned later in this document). Can I ask the authors to describe a little more elaborately how the ensemble members are constructed and how different weights are applied for example?

The ensemble validation (e.g. lines 306 – 311) is useful and is clearly tailored toward showing that the target value is present in the training dataset. Besides the focus on separate ensemble members, it may be interesting to also plot the ensemble spread vs. the error to get a more statistical indication of whether the observation falls within the ensemble spread or not (for the full ensemble). Note that this is to a certain extent already present in e.g. Fig. 13. Hence, see this rather as a suggestion than a must. For inspiration, see for instance Fig. 9 in Foresti et al. (2016) or supplement Fig. S6 in Imhoff et al. (2020) as spread vs error examples for ensemble nowcasting.

*Results*

Starting with the figures, I did like the analyses chosen by the authors, but the labels, legend and text in the figures was often hard to read. Zooming in is luckily possible, but can I ask the authors to make the font sizes of the figures bigger? Besides, the schematic figures describing the method are clear and are very nice figures (no changes needed there).

One figure that I did miss at the start of the results, is an example nowcast for several lead times with the k-NN methods, the Lagrangian persistence and the observations (the radar images, I suppose). This can directly show what to expect and visualize why we see certain results in the subsequent figures. Can I ask the authors to make such a figure, possibly one for each class (duration), so likely a small-scale convective event, a mesoscale convective event and perhaps a stratiform event?

*Reproducibility*

Are any of the data, scripts, etc. publicly available? That would increase the reproducibility, but also the impact of this very interesting work.

**Specific comments**

Title: Perhaps good to mention here radar-based rainfall nowcasts, to clarify the focus on rainfall forecasts.

Lines 33 – 43: Could you explain why the focus is on object-oriented nowcasting of primarily convective events here (so, why object-oriented and why convective events as focus)? I suppose this is because the authors focus on mostly convective events that can or have resulted in flooding.

Line 118: Is the C-band radar single or dual-pol?

Lines 122 – 123: Can I ask the authors to elaborate a bit on this merging method. It can be brief, the rest is mentioned in the paper of course.

Line 124 – "110 events": What is the definition of an event in this study and did you systematically look for certain event characteristics? Seeing the following lines, the authors have chosen to focus on mostly convective events. How were those selected?

Line 146 – "unmatched storm cells": What have you done with these unmatched storm cells? Are they left out of the method and analysis or not? I see the answer now in Sec. 3.2.3 Perhaps good to very briefly mention this (or point to Sec. 3.2.3) here.

Lines 146 – 152: How is the storm duration defined?

Lines 158 – 161: I agree! Nowcasting, especially Lagrangian persistence, either in an object-oriented or intermittent field based approach, works quite well for these stratiform events. It may be worth mentioning that this is not where the k-NN method will provide a lot of improvement (at least, that is what I expect) compared to already existing methods. Hence, it is no problem that the sample size is leaning more heavily on the convective events.

Line 164 / Eq. 1: What if one radar image is missing or another problem occurs for a certain time step? How will this be treated? In case of a missing radar image, would it be safer to divide by the number of used predictors in equation 1 (max. 6, but possibly less) instead of by 6? In addition, there are seven 5-min steps from t=0 to t=t-30, assuming that t=0 and t=t-30 are included. Should you divide by seven or is step t-30 not included?

Lines 242 – 243 and lines 253 – 255: How are the weights determined? I see that the weight of eq. 5 is determined from the results (Fig. 6), perhaps good to refer to this.

Line 253 / Eq. 6: In the estimated response of the to-be-nowcasted storm, are the total lifetime, area and intensity simply the weighted average of the k nearest neighbors? What about the location, are $V_{X+LT}$ and $V_{Y+LT}$ used to displace the location of the current state of the storm at $t_0$? It may be clear for future readers to specify this a bit in more detail.

Lines 255 – 261: I'm a bit lost with the way the ensemble nowcast is constructed. Are 30 individual members issued by taking every time a different selection of neighbors? Or just by one neighbor out the top 30?

Lines 277 – 298: Is there a difference in the training between VS1 and VS2, mainly w.r.t. the error per target or for all targets together? Or is the training identical for both approaches?

Line 304 – "Lagrangian persistence": Just to be sure, Lagrangian persistence in object-oriented nowcasting, right?

Lines 314 – 316: Do you have an indication how much less this was (in the worst-case scenarios)?

Lines 367 – 369: Do you have any idea why the result is different for the Total Lifetime?

Lines 380 – 385 and Fig. 8: How large is the spread in the results (the optimal k-value) per class? I can imagine that this does not make the figure clearer when added (as IQR for example), but perhaps the authors have an indication of this.

Lines 392 – 401: Is the decreasing error with increasing lead time (mainly visible in the top row of Fig. 9) a result of storms dissipating sooner than 30 min, which is then forecast well? Could the

authors comment on that? It would otherwise be unexpected to see the performance improve with increasing lead time (you would expect the opposite).

Lines 402 – 412: I agree with the reasoning for the 36[th] time step. However, the y-axis of Fig. 9 (middle row) is clearly scaled to this time step, which makes it hard to distinguish what happens for the other two lines. Could you change the axis scale and just describe why the 36[th] time step falls outside this scale (which is already described now)? Another question regarding the 36[th] time step, because I'm not sure I understand the times of nowcast well here: isn't the time of nowcasts the same as the issue time, so the nowcast starts 3h after the evolution of the storm. In the class here, you are only considering storms that last maximum 3h. Hence, aren't we looking at storms that should have died already?

Lines 452 – 453: I think this conclusion needs the nuance that this is the case for the shorter storm durations, whereas for longer durations this is not or only to a lesser extent the case.

Line 492 – "Overall the ensemble results are better than the single 4-NN nowcast": Based on the results and the shown figure, I think you can only state that the best ensemble member performs better than the single 4-NN nowcast.

Lines 574 – 576: There should be some nuance here. Although the results are very promising and often outcompete Lagrangian persistence, these high improvement numbers are generally reached for short-living storms, while the improvement is less (sometimes even worse) for longer-living storms.

Line 586 – "additional predictors": For the interested reader(s), can you say more about which predictors from those sources you think would be feasible for this?

Figure 4 caption: Perhaps refer to Table 1 for the meaning of the symbols (the predictors) in the figure.

**Technical corrections**

Figure 2: As I hope that this interesting paper will be read by people from all over the world, it may be good to add a small subfigure indicating where this region is located in Germany or even in NW-Europe. Besides, the indicated coordinates seem to be in a local coordinate system. Can I ask the authors to mention this in the caption or, perhaps even better, to place lat-lon coordinates on the map?

Figure 3: What is the last group (duration) on the x-axis?

Figures 7, 8 and 9: The light grey color for the 5-min class is not easy to distinguish (especially on a colored background). Could you make the grey a little darker or use a different color?

Line 30 – "short-term rainfall nowcast": perhaps say short-term rainfall forecasting?

Lines 32 – 33: A minor detail about the storm and intermittent field references: the list of references can be almost endless here. It is not necessary to cite all of them, but perhaps you could say (e.g. + references) to indicate that this is just a sample of all studies to these topics.

Line 84 – "show for instance Hou & Wang (2017)": is for instance shown by Hou & Wang (2017).

Line 211 – "predictor important analysis": should this be importance instead of important?

Line 391 – "event-based 4-NN": For consistency, did you mean storm-based 4-NN?

Line 463 – "same results": similar results.

Line 561 – "Person": Pearson.

Line 562 – "two measurement": Two measurements.

Line 570 – "the death processes": Perhaps better to use dissipation processes of storms?

**References**

Dixon, M., & Wiener, G. (1993). TITAN: Thunderstorm identification, tracking, analysis, and nowcasting—A radar-based methodology. *Journal of Atmospheric and Oceanic Technology*, 10(6), 785–797. https://doi.org/10.1175/1520-0426(1993)010<0785:TTITAA>2.0.CO;2

Foresti, L., Reyniers, M., Seed, A., & Delobbe, L. (2016). Development and verification of a real-time stochastic precipitation nowcasting system for urban hydrology in Belgium. *Hydrology and Earth System Sciences*, 20(1), 505–527. https://doi.org/10.5194/hess-20-505-2016

Han, L., Fu., S., Zhao, L., Zheng, Y., Wang, H. & Lin, Y. (2009). 3D convective storm identification, tracking and forecasting – An enhanced TITAN algorithm. *Journal of Atmospheric and Oceanic Technology*, 26(4), 719 – 732, https://doi.org/10.1175/2008JTECHA1084.1

Imhoff, R. O., Brauer, C. C., Overeem, A., Weerts, A. H., & Uijlenhoet, R. (2020). Spatial and temporal evaluation of radar rainfall nowcasting techniques on 1,533 events. *Water Resources Research*, 56, e2019WR026723. https://doi.org/10.1029/2019WR026723

Pulkkinen, S., Chandrasekar, V., Von Lerber, A. & Harri, A-M (2020). Nowcasting of convective rainfall using volumetric radar observations. *IEEE Transactions on Geoscience and Remote Sensing*, 1 – 15, https://doi.org/10.1109/TGRS.2020.2984594

---

## Referee Comment (RC2)

**Review of "Improving object-oriented radar based nowcast by a nearest neighbor approach" by Bora Shehu and Uwe Haberlandt**

Seppo Pulkkinen
seppo.pulkkinen@fmi.fi
July 4 2021

**Summary**

The authors propose a methodology for predicting the evolution of storm cells by comparing them to past storms. This is done by using a nearest neighbor approach. Overall, I find this a highly relevant and well-written paper with scientifically significant results. The proposed approach has a high novelty value. I did not find any fundamental flaws in the methodology. The paper is also well within the scope of HESS. However, I have several questions and comments about the data selection, how the verification is done and some requests for clarification. The literature review is somewhat lacking to put the work into a broader context. There is also room for improvement in how the results are presented and the language needs some polishing. However, all these are minor details that can be improved with a small amount of additional work.

I'm looking forward to see the revised paper. I can recommend it for publication once the following concerns have been addressed.

**General comments**

- To make the title better reflect the content, you could add the word "rainfall" because the paper is about rainfall nowcasting.
- Before going directly to the matter, it could be worthwhile to add one general paragraph about nowcasting. Like why nowcasting is done, its societal need and what kind of hazards can be prevented with reliable rainfall nowcasts.
- In the beginning of the introduction, the authors should make a more clear distinction between the two nowcasting approaches (field- and object-based) and add more description about what purposes they are used for. For instance, mention that field-based methods are well-suited for predicting large-scale stratiform precipitation systems but cell-based methods are best-suited for predicting the motion of intense convective cells.
- To put their work into a broader context, the authors could mention in the introduction that the proposed approach is conceptually similar to the so-called analogue-based nowcasting. The idea of this approach is to look for similar events from a large sample of archived radar data. See, for instance:

  *L. Panziera, U. Germann, M. Gabella and P. Mandapaka: NORA–Nowcasting of Orographic Rainfall by means of Analogues, Quarterly Journal of the Royal Meteorological Society, 137(661), 2106-2123, 2011.*

  There are a number of others, so I recommend the authors to do a literature review. However, all the previous studies I know attempt to find analogs from full radar images, not from individual cells or their features. This is a novel aspect, which should be clearly pointed out in the manuscript.
- I have concerns about the choice of the predictors. The proposed methodology opens the possibility to use a large number of different predictors (and targets).

However, the set chosen in the study is in my opinion quite limited and the capability of the model is thus not fully utilized. In addition, they are more or less correlated with each other, and also with the target variables, which the authors admit. I think that using the following additional predictors could reveal the full potential of the model:

- convective available potential energy (CAPE)
- convective inhibition (CIN)
- signatures from radar-measured Doppler and polarimetric parameters, as well as vertical profile information obtained by using all elevation angles
- lightning flash density
- geographical features like terrain altitude or proximity of water bodies

These are probably beyond the scope of this study, but I encourage the authors to include them in a follow-up paper. In addition, the authors could replace the generic description of additional predictors in the last paragraph of Section 5 by specifically mentioning some of the above.

Note that a relationship between CAPE and CIN and the life cycle of convective cells is suggested in:

*C. Moseley, O. Henneberg and J. O. Harter: A Statistical Model for Isolated Convective Precipitation Events, Journal of Advances in Modeling Earth Systems, 11(1), 360-375, 2019.*

- A fundamental reason why similar storm cells behave similarly is that their life cycles follow characteristic patterns. For instance, the areal extent and intensity of storm cells are related to each other and the storm lifetime. In particular, the future behavior of cells depends on what stage they are in their life cycle. I think this aspect needs to be discussed more in the paper with literature references to put the research in a broader context. To this end, the authors could study the following papers:

  *H. Kyznarova and P. Novak: CELLTRACK-Convective cell tracking algorithm and its use for deriving life cycle characteristics, Atmospheric Research, 93(1-3), 317-327, 2009*

  *C. Moseley, P. Berg and J. O. Haerter: Probing the precipitation life cycle by iterative rain cell tracking, JGR: Atmospheres, 118(24), 361-370, 2013*

  There is also a large amount of meteorological literature, where the life cycles of convective storms are discussed.

- A major limitation of the k-neighbors approach is that it cannot generalize beyond the training data. How would the proposed method perform for extreme events that have a very limited number or no training samples? Please add more discussion or analysis about this.

- In many places, the authors are describing results that are not shown anywhere, so the reader cannot verify the validity of the claims. An example of this can be seen at lines 362-374. Could you include some of the not shown results that are discussed in the text in an appendix or in supplementary material?

**Specific comments**

- Line 101: What does "step 3" refer to? Storm extrapolation in Figure 1?
- Figure 2:
  - What do the numbers represent in the x- and y-tick labels? It think it would be more informative to show the distance from the radar in kilometers.
  - Does DEM mean altitude obtained from a digital elevation model?
- Lines 131-137: What is the justification for these threshold choices?
- Lines 139-140: What is the "spatial rainfall intensity of a storm". Is it some kind of average or maximum value taken inside the storm object?
- Line 142: I'm curious how the ellipsoid is fitted. Please provide a more detailed explanation (though no need to include this in the paper).
- Line 145: Please give a more detailed description about how the storm velocities are estimated?
- Line 145 and Figure 3: The merges are mentioned in the text but not shown in the figure.
- Figure 3:
  - The very high velocities of 5-minute storms look suspicious to me. How can you even estimate the velocity of a storm if its duration is only 5 minutes (i.e. one time step)?
  - Is there a reason for specifying the duration intervals in inclusive way? I would use separate intervals (i.e. 0-1h, 1-3h, 3-6h, 6-12h).
  - I'm very surprised to see how the 5-minute storms have such a large area. I would expect all storms having area over 500 $km^2$ to have lifetime longer than 5 minutes. Can you explain this?
- Line 152: Please give numbers describing "high intensity" and "low areal coverage".
- Lines 153-154: What is the evidence for making this conclusion? At least this cannot be seen from Figure 3.
- Line 210 onwards: It is not obvious to me how the partial information correlation is better able to capture non-linear behavior than the Pearson correlation coefficient. Can you add more discussion about this?
- Equation (3): How is PI defined?
- Equation (4): The notation is confusing. What does X(-j) mean? Maybe you should use subscripts for j and -j instead.
- Table 1: I would not use the word "predicted" for the target variables. I thought that they are obtained form observations, not predictions. For the velocities, I would use the word "estimated" since they are not directly observed but estimated by using some method.
- Table 1: You could attempt to eliminate the dependency of the standard deviation on the mean value by using the coefficient of variation instead (i.e. standard deviation divided by the mean).
- Lines 255-262: The actual procedure for generating the ensemble is not well-described. Are the ensemble members somehow randomly assigned based on their probabilities?
- Line 261: What is the justification for choosing the value 0.5? Is the model sensitive to this value?

- Equations (6) and (7): To me it appears that the same symbol R is used for two different purposes: response and rank. Could you use different symbols?
- Equation (8): Should the summation terms be taken their absolute values? To me summation over the differences does not make much sense if it's used as the objective function.
- Line 304: Please define precisely the concept of Lagrangian persistence in this context. It can be defined in many different ways depending on the type of the nowcast (i.e. grid- or object-based). Here it means that all storm attributes (not only the shape) are taken from the most recent values and they remain constant for all lead times. Right?
- Figure 6:
  - In Table 1, the target variables A, I, $V_x$ and $V_y$ have the subscript denoting lead time. However, these are is omitted in Figure 6. Thus, it is not clear to me what lead times do the correlations shown in Figure 6 represent. The text is just saying that the values are averaged from three different lead times.
  - The correlations depend on the lead time. Would it make sense to show the correlations separately for each of the chosen lead times instead of averaging over different lead times?
- Lines 365-366: This is difficult to follow. It is confusing that the authors mention both mean and median but are not showing the latter anywhere. In addition, you should clearly state in the caption of Figure 7 that it shows the mean.
- Lines 391 and 397: The authors are using a confusing term "event-based" that is not defined previously. Does this mean storm-based?
- Figure 7: The interpretation of the Total Lifetime figure on the right was not immediately clear to me. In particular, the connection of the black and red lines labeled as VS1 and VS2 to the boxes shown in the figure could be more clear.
- Figure 9:
  - What is the "Timestep of nowcast"? This should be clearly explained in the caption text. Now I found it from line 403 only after reading through the main text.
  - As in Figure 7, it was not immediately obvious to me how to interpret the right pane.
- Figure 10:
  - Again, clarify the meaning of the "Timestep of nowcast". Please explain it in the figure caption.
  - To me it is striking that in the worst case the 4-NN nowcast can perform more than 100% worse than the Lagrangian persistence. This is lacking discussion in the text that focuses mainly on the improvement from the Lagrangian persistence.
  - My advice here is to explore, or at least mention the possibility of blending the Lagrangian persistence and the 4-NN nowcast by using weights that depend on the lead time. This would combine the strengths of both approaches.
- Figure 12: It could be more informative to compute instead the fraction of verifying observations within (or outside) the ensemble and average this statistic over the events. The "% of timesteps" statistic gives no information about this fraction.
- Sections 4.2 and 4.3: The authors use the terms lead time and timestep interchangeably. When reading the text, their correspondence is not immediately clear to the reader (until the reader goes to Section 2 to recall that the time step is 5 minutes). Could you use only one of them?

- Section 4.4: I have some doubts whether the "best ensemble member" or "% of ensemble members better than Lagrangian persistence" verification approach is meaningful. These are not standard verification metrics. In practice, one does not know a priori which ensemble members to choose to obtain the best forecast skill. There are more elaborate ways of showing the advantages of ensemble-based predictions over deterministic ones. For instance, you can compute the continuous ranked probability score (CRPS), which is a generalization of mean absolute error (MAE) for deterministic nowcasts. The CRPS of the ensemble nowcast should be lower than the MAE, which indicates the added value of the ensemble nowcast.
- Line 511 onwards: I guess that the authors mean individual ensemble members, not whole ensembles?
- Lines 538-539: Can you give some numbers to describe what are the fine spatial and temporal scales?
- Line 546: Where does the number 5200 come from? It is mentioned for the first time in the conclusions. Perhaps it should be mentioned in Section 2 as well.

**Technical corrections**

- I'm not sure if it's proper to use the word "object-oriented". It refers to programming terminology. Could you use object-based instead?
- Figure 2: Should the legend read "Lower Saxony border"?
- Line 211: important $\leftarrow$ importance?
- Lines 226-227: "the $\alpha_j$ the predictors weight" $\leftarrow$ "$\alpha_j$ denote the predictors weight"?
- Line 256: 30-ensembles $\leftarrow$ 30-member ensembles?
- Line 447: persistence $\leftarrow$ persistent?
- Line 544: behaviours $\leftarrow$ behaviour
- Line 578: An increment in the sample size $\leftarrow$ Increase in the sample size?
- Figure 6: Should this be titled as a figure or a table?

---

## Referee Comment (RC3)

**The review on the manuscript "Improving object-oriented radar based nowcast by a nearest neighbour approach" by Shehu and Haberlandt**

The presented manuscript aims to benchmark the nearest neighbor approach for object-based storm nowcasting in Hannover Radar Range, Germany. The study utilizes the database of storm events that have been compiled with a focus on urban hydrological applications. Hence, it is of particular interest for radar-based precipitation nowcasting, urban hydrology, and water management. The declared scientific question is straightforward, the proposed workflow is reliable, and the obtained results are well delivered and discussed. Still, some comments and questions need clarification and, probably, would require additional experiments and analysis from the authors.

**Major comments**

**Research aim**
The decision to predict individual storm characteristics, i.e., area, mean intensity, x and y components of velocity, and lifetime, instead of predicting the entire storm evolution as an integral object should be elaborated. In general, I understand the utility of predicting individual characteristics. However, in this way, we miss the detailed information about storm event spatiotemporal evolution and could not precisely estimate neither location nor intensity-related errors. For example, the spatial structure and distribution of rainfall intensities within the storm event are particularly relevant for urban applications. The example of the possible utilization of the predicted (individual) properties could help to clarify their choice.

**Dataset**
The authors declare that the compiled dataset includes outliers (L198). That leads to the mixed-use of mean or probability-based (e.g., median) statistics. It is pretty hard to recognize and remember where and why the mean or median statistics are used. Moreover, the authors often describe the need to use mean/median for (not) accounting outliers but rarely communicate the obtained results based on that choice. Thus, it is interesting what is the proportion of outliers in the compiled database and could they be removed for the sake of consistency of mean/median statistics throughout the manuscript.

The compiled database of storm events is based on the open data provided by the German Weather Service. Is it possible to share it? It would serve both manuscript's reproducibility and community interests in the field of storm tracking and prediction.

**Baseline**
The utilization of Lagrangian persistence as a baseline is reliable, and obtained results are interesting to compare. I recommend authors provide additional information about it (how nowcasted is computed etc.) to account for inexperienced readers.

In Section 4.4., the authors compare the closest single neighbor and 30-member ensemble approach. Do the authors mind finding the single nearest neighbor as a more advanced baseline compared with four and 30-member ensemble solutions? It would then pose an additional research question (partially touched in Sect. 4.4) of an added value of ensemble approach compared to the single neighbor.

**Information leakage**
In modeling studies, it is particularly relevant to isolate calibration, validation, and test datasets to prevent so-called information leakage -- the situation when the information outside the

calibration set is used for model calibration (training). In the presented study, I suspect four procedures that may lead to information leakage:

1. Normalization of events characteristics.
2. Importance analysis and weights calculation.
3. Optimization of the number of nearest neighbors.
4. Splitting into different event groups.

The authors state that normalization and importance analysis have been done "Before training and validating the k-NN method" (L191). In this way, there is an evident information leakage that connects calibration and validation datasets. Also, it is not clear how calibration and validation datasets have been isolated to find the optimal number of nearest neighbors. I do not think that addressing data leakage would change the results much, but it is vital to ensure methodological reliability.

Splitting the database into three groups according to their duration (L312-317, Table 2) was done before the modeling. In general (and in practice), we do not know a priori if the recently appeared storm will be sporadic or last for a couple of hours or more. So, in my opinion, in making predictions, we should use all the examples from the database to find closer candidates to be used for predictions, not only those from the group of a similar duration. That also would open the new directions of analysis, e.g., how would closest examples change with the storm's evolution. Is the more mature storm similar to storms with comparable duration, or is there some skew in characteristics similarity?

The minor but also critical comment here is about the research code availability. For sure, open code would ensure research reproducibility and provide information on particular details of the computational workflow.

**Training, learning, and cross-validation**
The authors use terms of training and cross-validation, but, in my opinion, the presented manuscript does not involve both procedures. The nearest-neighbor model is not trained per se; it only uses a bag with historical examples to find one closer to the "storm-to-be-predicted" based on the similarity metric. In this way, the nearest-neighbor model also does not learn anything as it has no parameters to learn. The only parameter here is the number of the nearest neighbors to use for predictions. However, the choice of that number is entirely subjective (see comment above) and is independent of both the "storm-to-be-predicted" and the available examples and their characteristics.

I would also question the use of the term cross-validation. There is no numerical model to validate as the nearest neighbor approach is instead a database search method than a "pure" numerical model.

The authors explicitly communicate the aim of the study as "... to investigate if non-linear relationships learned from past observed storms can surpass the Lagrangian persistence and extend the predictability limit of different storms." However, as I mentioned above, the nearest neighbor model does not learn anything. It is also an open question if there are non-linear relationships (and what kind of relationships).

**Minor comments**

- "Birth" → "initialization"?
- "Death" → "dissipation"?

- L98: "k-NN." The first appearance needs transcription.
- The orientation feature is in degrees. I wonder how the difference between 1 and 359 degrees is considered.
- Interestingly, the area and number of storm cells do not show similar behavior in importance analysis, but they are highly correlated.
- L261-262: "Only neighbours that display a distance lower than 0.5 are selected for both single and ensemble nowcast in order to minimize the influence of non-similar storms." Any statistics of that?
- Figure 6: "The weights given here are averaged from the weights calculated at three different lead times and storm durations." However, the authors then mention (L341-342): "Contrary for Total Lifetime and Area, only for storms that last longer than 3 hours, the method is able to converge and give the most important predictors." However, we cannot see these results.
- L348-349: "is not completely understood and is not investigated further on for the time being since it is outside the scope of this paper." But, from the abstract and introduction: "i) what features should be used to describe storms in order to check for similarity?" Thus, it is probably in the scope of the paper.
- L354-355: "Moreover, the important predictors do not change drastically from one lead time or storm group to the other, as seen in the PIC" Could we see it from any table or figure?
- Figures: larger fonts and more vertical space between different types of events would be appreciated.

---

## Author Comment (AC1)

Dear Ruben,

thank you for your comments and suggestions. Please find below the answers or our comments on the questions you have raised. Following your suggestions and those of another review, one of the main changes that I have done, was to update the results based on absolute error and computed the Continuous Rank Probability Score for the probabilistic nowcast. Thus, the results plots have changed accordingly, and are given in the response to this review. Please find below the answers and our comments on the questions/issues you have raised (given in blue below each point). Please note that some changes have been already done in the updated version of the manuscript (these are given in quotation) while other changes are yet to be done and thus I refer to as "will be included" in the updated version of the manuscript. The updated plots and tables are given in the attached document at the end of the response.

1. Methods:
- The unmatched storm cells are left out of the analyses. I do agree with this choice, but I wonder what the effect is on the algorithm performance once the 'normal' radar data, with these unmatched storm cells / artefacts, are fed into the algorithm. Can the authors comment on this?

     Yes, indeed the unmatched storm-cells affect the nowcast, especially when the nowcast in run with the Lagrangian Persistence, because they will be extrapolated in the future for the next 3 hours without change in area, intensity or movement. For the k-NN application, however, it seems that the dissipation of some unmatched storm-cells can be predicted based on their characteristics. We have run the proposed k-NN for unmatched storms with past database only from the matched storms, to see what is the error that we should expect at the nowcast time 5min (when the kNN doesn't now that they will dissipate in the next 5min). The results are shown for both target-and storm-based kNN in the Figure 14 in the attached pdf.

     Both applications of the kNN (as target based and storm based) nowcast that the storm will last 10 -15 min approximately on average, leading to overestimation of the 4 other target variables (Area, Intensity, Velocity in X and Y direction) for short lead times; up to 30 min for both deterministic and probabilistic nowcast. So, when running a nowcast online with normal nowcast data, one would expect an overestimation of the rainfall storms for the short lead times (up to 30 min). Alternatively, the "un-matched storms" can be part of the past-database, such that the kNN can predict their behaviour better. For both of kNN application (target-and storm-based) for both deterministic and ensemble nowcast, the median error for all storms lasting only 5 minute (and nowcast issue at nowcast time 5min) becomes zero independent of the lead time. This means that, they can be predicted well by the kNN, most probably because clutter is exhibited similarly from one event to the other (note that the kNN has the potential to also identify clutters). However, we do not suggest this as the presence of the "un-matched" storms in the past database can affect the nowcasting of the other storms, especially for long storms. The long storms are affected, because due to the merging techniques between the radar and the gauge, some "unmatched storms" have very large areas, thus causing the kNN to falsely choose the neighbours from these "unmatched storms". If the merging method is improved (for instance by an advection correction so it doesn't take the information only from gauge data when the radar data is missing), and a better tracking is implemented, the kNN can be then run on the full past database.

- What is the algorithm performance for extreme events or events that were not part of the training data? From both an operational and reproducibility perspective, this would be very relevant information.

     The performance shown in this paper is calculated leaving each event outside and predicting it with the rest of the dataset (so in cross-validation mode). So, each time the errors are

calculated for a single event, that event has been taken out of the training data (hence was not part of the database). Regarding the extreme events, please keep in mind that the events selected here are already extreme events – events that have a return period higher than 5 years for durations varying from 1 hour to 1 day (based on Intensity-Duration-Frequency Curves). But we suppose you are referring to rare events of a very high return period – say 50 or 100 years? We will include an example of a single extreme event in the updated manuscript to discuss how the KNN behaves in such cases.

- In addition, can the authors say some more about the size (memory) of the training dataset, the computation times and how big the training set should be for adequate use?

  For the selected events the size of the characteristics that is save for all storms in all events is not very big (~20 MB). The computation time to extract the characteristics in hindcast mode is less than one day for all events and to train the k-NN (finding the best neighbour) based on 110 events takes in total less than 2 hours. While to issue nowcast for all 110 events once the number of K is selected is less than 3 min. Please note that this is the time of the kNN application only, an operational application together with radar data, merging with station data, tracking the storms and the nearest neighbour, has not been yet tested, thus we cannot say for sure how long are the computational times. The size of the available dataset plays an important role, but at the moment we can not conclude what is the adequate dataset for issuing reliable k-NN nowcast. This will be the subject of our future research. We will include this small discussion in the updated version of the manuscript.

- The authors often mention timestep of nowcast or nowcast time. Up to the end of the manuscript, I have found the terminology and meaning confusing (it may just be me..). Do the authors mean the issue time of the nowcast since the start/evolution of the storm with this? If so, I would recommend changing this for clarity.

  Thank you for the feedback. Apparently, it is also confusing to the other reviewers, so we will try to make it clearer. Yes, timestep of nowcast and nowcast time are referring to the same thing which is what you describe: the time (in 5min timesteps) when the nowcast is issued compared to the first identification of the storm. We have included a new figure in the manuscript to explain the nowcast time and we have used in the manuscript only the term "nowcast time". Please refer to the Figure 3 in the attached pdf.

- I appreciate the use of an adequate benchmark (another object-oriented nowcasting method using Lagrangian persistence for the movement of storm cells) in this study. However, can the authors comment on other works that take next to Lagrangian persistence also other processes into account, e.g. splits and mergers (e.g. Dixon & Wiener, 1993; Han et al., 2009), and the rate of growth and dissipation (e.g. Pulkkinen et al., 2020 – although not necessarily constructed for object-oriented nowcasts)? Thus, how would this work relate to or even improve such methods? An analysis comparing such a method to this method is of course outside the scope of this paper, but it would be great if the authors can at least comment on it.

  Yes, after following the comments of another referee (Seppo Pulkkinen) we will first improve the introduction to the rainfall nowcast and mention the work done recently on dissipation of the rainfall intensities (although on a different approach). Regarding the work done by Dixon& Wiener (1993) and Han et al. (2009), they have improved considerably the tracking of the storms – they recognize better if the storms are one unit (how they track them and how the velocities are assigned to them), however to our knowledge they still apply the Lagrangian persistence for the future extrapolation. It here that we see the value of our studies. Of course, it may be that the tracking algorithm is not working as good as in the reference studies,

however it shows the potential of the kNN application to recognize the recognize the dissipation of the convective storms. The application of the kNN on their tracking algorithm may lead to better results. Regarding the work done by Pulkkinen (2020), a kNN application may be useful to maintain the storm properties of the convective storms.

2. Ensemble approach:

- The construction of the ensembles is interesting, but the description in the methodology was not always clear to me (see the comments mentioned later in this document). Can I ask the authors to describe a little more elaborately how the ensemble members are constructed and how different weights are applied for example?

Yes of course. In theory the kNN for both deterministic and ensemble application is very similar. In both applications, first the 30 most similar neighbours are recognized, ranking them from the most similar (lowest Euclidian distance) to the least similar (largest Euclidian distance). In the case of the ensemble application, the probability of their occurrence will be given by their rank, so the first neighbour (the most similar and the first ensemble member) will have a higher probability than the last neighbour (the least similar and the 30$^{th}$ ensemble member). Then an ensemble member is issued randomly based on these rank probabilities. To make this clearer the following changes have been made in the text:

"Contrary, if a probabilistic nowcast is selected, 30-member ensembles are selected from the closest 30 storms where each member is assigned a probability according to the rank of the respective neighbour storm with the "to-be-nowcasted" storm:

$$Pr_i = \frac{(1/Rank_i)}{\sum_{i=1}^{k}(1/R_{anki})},$$ (1)

where k is the selected number of neighbours and Rank and Pr are respectively the rank and the probability weights of the i$^{th}$ neighbour/ensemble member. An ensemble member is then chosen randomly according to their probability weights. The probability weights calculated here are as well used for computation of the single nowcast in Equation (6)."

- The ensemble validation (e.g. lines 306 – 311) is useful and is clearly tailored toward showing that the target value is present in the training dataset. Besides the focus on separate ensemble members, it may be interesting to also plot the ensemble spread vs. the error to get a more statistical indication of whether the observation falls within the ensemble spread or not (for the full ensemble). Note that this is to a certain extent already present in e.g. Fig. 13. Hence, see this rather as a suggestion than a must. For inspiration, see for instance Fig. 9 in Foresti et al. (2016) or supplement Fig. S6 in Imhoff et al. (2020) as spread vs error examples for ensemble nowcasting.

We followed the advice from Seppo Pulkkinen, to compute the Continuous Ranked Probability Score (CRPS) for the ensemble nowcast, and the mean absolute error for the deterministic nowcast. The Figures below show the ensemble and deterministic results for both k-NN approach; the storm based and target based. Please refer to Figure 11 and 12 in the attached pdf.

3. Results:

- Starting with the figures, I did like the analyses chosen by the authors, but the labels, legend and text in the figures was often hard to read. Zooming in is luckily possible, but can I ask the authors to make the font sizes of the figures bigger? Besides, the schematic figures describing the method are clear and are very nice figures (no changes needed there).

Noted! We have increased the size of the labels, ticks and legends.

- One figure that I did miss at the start of the results, is an example nowcast for several lead times with the k-NN methods, the Lagrangian persistence and the observations (the radar images, I suppose). This can directly show what to expect and visualize why we see certain results in the subsequent figures. Can I ask the authors to make such a figure, possibly one for each class (duration), so likely a small-scale convective event, a mesoscale convective event and perhaps a stratiform event?

  Such a figure would be possible however it is misleading. So far, we have considered the storms as objects and we want to predict good their characteristics and not the spatial distribution of the storms (the image of the storm at different times as captured by the radar). As noted in the manuscript, another step should be implemented to transfer these predicted characteristics to the spatial distribution of the storm intensities: this can be done in two ways 1) consider the storm as an ellipse and increase/decrease the ellipse shape and intensity according to the Area and mean Intensity predicted (assumption about the internal distribution should be taken) and displace with the given velocity vector, or 2) get the information of the spatial intensity distribution from similar storms. However, in the latter case, one has to make sure that the storms have similar orientation and thus a space optimizer is needed. We would like to continue our work here with the second approach, and to publish a continuation paper with the space optimizer. But if you are interested about this, you can have a look at my phd thesis where these examples are given in the appendix (Shehu,2020). But nevertheless, I would like to avoid to show the figures that you mention because the method is not adequate for this yet.

  4. Reproducibility:
- Are any of the data, scripts, etc. publicly available? That would increase the reproducibility, but also the impact of this very interesting work

  No so far, no data or scripts are made available. However, I can share the storms characteristics database, and part of the script that applies the nearest neighbour approach. However, the tracking algorithm I can not share because it is not developed by me, and the radar data is extremely memory consuming, so I would like to avoid that.

  Specific comments:

- Title: Perhaps good to mention here radar-based rainfall nowcasts, to clarify the focus on rainfall forecasts

  Noted. We will discuss with the editor to see if the title can be changed as following to make clear that this is the application for the storm's characteristics, and that a second paper will follow to illustrate the suitability for the point scale.

  "Improving radar-based rainfall nowcast by a nearest neighbour approach: Part I – Storm Characteristics"

- Lines 33 – 43: Could you explain why the focus is on object-oriented nowcasting of primarily convective events here (so, why object-oriented and why convective events as focus)? I suppose this is because the authors focus on mostly convective events that can or have resulted in flooding.

  The focus was to improve the nowcasting of urban pluvial floods which are typically caused by convective events. Object-oriented can capture better the dynamics of these events because they treat them as independent objects rather, and can have independent movements from each other. The field-oriented approach, is more appropriate for wide spread events (stratiform events) because they have similar velocities at the whole radar image that can be

captured well by the optical flow method. In optical flow methods, regions inside the radar image have different velocity fields, however may still not capture the convective storms uniqueness. A study done by Ruzanksi et al (2011) (also implemented by Pulkkinnen et al. 2020) shows that velocity can be captured by regions in the radar data through the DARTS methods, but one needs to select prior an empirical value. Thus, while the optical field is the most used approach recently, it requires specific parameters based on the region and expert knowledge, that the researcher may not be fully able to understand or capture well. That is why the objective oriented is used mainly in convective storms.

Following your suggestion, we will include a short explanation in the updated version of the manuscript about the overall aim and the use of object-oriented in general. Please note that the introduction will be updated, to introduce the motivation better (as suggested by another review).

Ruzanski, E., Chandrasekar, V., Yanting W. (2011). The CASA Nowcasting System. Journal of Atmospheric and Oceanic Technology,28(),640-655.

- Line 118: Is the C-band radar single or dual-pol?

  The radar is single pol C-band. We don't have any phase shift (k) information, which would be interesting to use as an additional predictor to the k-NN application.

- Lines 122 – 123: Can I ask the authors to elaborate a bit on this merging method. It can be brief, the rest is mentioned in the paper of course.

  The following text was added in the manuscript:

  "The conditional merging aims to improve the kriging interpolation of the gauge recordings by adding the spatial variability and maintaining the storm structures as recognized by the radar data. In case a radar image is missing, the kriging interpolation of the gauge recordings is taken instead."

- Line 124: "110 events": What is the definition of an event in this study and did you systematically look for certain event characteristics? Seeing the following lines, the authors have chosen to focus on mostly convective events. How were those selected?

  An event is defined as time period where the radar recognizes continuous rainfall in the whole study area (so within the radar range). In an event many storms can be recognized within the radar range (also at different location and at different starting time). See Figure 3 below for a better explanation of the difference between event and storm. The events were selected based on two criteria a) on strong rainfall intensities captured by the rain gauges inside the radar range (for different durations the intensity exceeds the 2 years return period from the KOSTRA IDF relationship), b) a dry spell duration of 4 hours in the whole radar image is used to separate the events (so start and ending time). This work was done under the concern of pluvial floods in urban areas, so the focus is primarily at events that might cause such floods, thus very high intensities in a short duration. The following lines were added in the manuscript together with Figure 3 (see appendix).

  "Here, rainfall events are referred to a time period when rainfall has been observed inside the radar range and at least at one rain gauge has registered an extreme rainfall volume (return period higher than 5 years) for durations varying from 5minutes to 1day. The start and the end of the rainfall event is determined when areal mean radar intensity is lower than 0.1mm for

more than 4 hours. Within a rainfall events many rainfall storms can be recognized at different times and locations inside the radar range. Figure 3-a shows a simple illustration to distinguish between the rainfall event and rainfall storm concepts employed in this study."

- Line 146 – "unmatched storm cells": What have you done with these unmatched storm cells? Are they left out of the method and analysis or not? I see the answer now in Sec. 3.2.3 Perhaps good to very briefly mention this (or point to Sec. 3.2.3) here

  The following changes were done in the text:

  "Since the "not" matched storms can either be dynamic clutter, they are left outside of the k-NN training and validation (see section 3.2.3). The section 4.4.4 discusses shortly the influence of the "unmatched storms" in the performance of k-NN approach."

- Lines 146 – 152: How is the storm duration defined?

  The storm is a group of radar pixels that have an intensity higher than a fixed threshold. The duration of the storm is then the lifetime of the radar pixels group as dictated by the threshold used to recognize them and the tracking algorithm that decides if the same storm is observed at continuous time steps.

  "Some characteristics of the identified storms like duration (or also total lifetime of the storm), mean area, maximum intensity, number of splits/merges, local velocity components, and ellipsoidal features, are shown in the Figure 4."

- Lines 158 – 161: I agree! Nowcasting, especially Lagrangian persistence, either in an object-oriented or intermittent field-based approach, works quite well for these stratiform events. It may be worth mentioning that this is not where the k-NN method will provide a lot of improvement (at least, that is what I expect) compared to already existing methods. Hence, it is no problem that the sample size is leaning more heavily on the convective events.

  The following lines were added in the text:

  "Nevertheless, the stratiform storms are typically nowcasted well by the Lagrangian persistence (specially by a field-oriented approach) as they are wide-spread and persistent. Hence the value of the k-NN is primarily seen for convective storms and not for stratiform ones."

- Line 164 / Eq. 1: What if one radar image is missing or another problem occurs for a certain time step? How will this be treated? In case of a missing radar image, would it be safer to divide by the number of used predictors in equation 1 (max. 6, but possibly less) instead of by 6? In addition, there are seven 5-min steps from t=0 to t=t-30, assuming that t=0 and t=t-30 are included. Should you divide by seven or is step t-30 not included?

  Here one should distinguish two different missing values: 1) when radar time steps are missing – then because of the merging between radar and station data, the ordinary kriging on station data will supply the spatial information that will be fed in the algorithm. This is not the optimal solution because will affect the tracking algorithm as the kriging information is usually smoothen in space and will result in very big areas with rainfall. This can be seen in the unmatched storms that have very big areas (this is the influence of missing radar and getting the information from the kriging interpolation on gauge data). This problem can either be fixed by implementing higher thresholds for the storm recognition, or by performing an interpolation in 1min of the information to ensure that the storm characteristics are maintained (As mentioned in Seo and Krajewski (2015) as advection correction). 2) The

tracking algorithm fails to capture or associate values for the velocity or other characteristics, which means the predictor value for a specific time step is missing. In this case the predictors are calculated by excluding the missing value and calculating the mean accordingly. Regarding the Eq.1 thank you for noticing. Of course, we meant 7 so including the whole-time steps from t=0 to t-30.

The following lines were added in the text:

"Another thing to keep in mind, is that merged radar are fed to the algorithm for storm recognition, and this affect the storm structures particularly when the radar data is missing. In such case, the ordinary kriging interpolation of rain gauges is given as input, which is well known to smoothen the spatial distribution of rainfall and hence resulting in a short storm characterized by a very large area."

"$P_{30} = \sum_{i=t_0}^{t-30min} P_i / 7$ ,

where Pi is the predictors value at time i, and P30 the average value of the predictor over last 30min. In case of missing values, the remaining time steps are used for averaging."

Line 242-243 and lines 253-255: we will follow your comment and refer to the equation here.

Lines 242 – 243 and lines 253 – 255: How are the weights determined? I see that the weight of eq. 5 is determined from the results (Fig. 6), perhaps good to refer to this

Regarding the predictors weight the following change was done to clarify this issue:

"$E_d = \sqrt{\sum_{i=1}^{N} w_i \cdot (X_i - Y_i)^2}$ ,

where w is the weight of the respective ith predictor as dictated by the importance analysis (results are shown in Figure 7), X the predictor of the "to-be-nowcasted" storm, Y the predictor of a past observed storm, N the total number of predictors used and Ed the Euclidian distance between the "to-be-nowcasted" and a past observed storm."

- Line 253 / Eq. 6: In the estimated response of the to-be-nowcasted storm, are the total lifetime, area and intensity simply the weighted average of the k nearest neighbours? What about the location, are VX+LT and VY+LT used to displace the location of the current state of the storm at t0? It may be clear for future readers to specify this a bit in more detail.

  Yes the area, total lifetime and the intensity are weighted averages from the 4 closest neighbours. So the first neighbour will have the 0.48, the second 0.24, the third 0.16 and the last 0.12 – these weights results from the ranking (1/(1:4))/sum(1/(1:4)). The same averaging is done also for the Vx and the Vy, which dictate in the displacement of the storm from the current state to the state at a specific lead times.

  The following line was added in the text:

  "The response R refers to each of the 4 target variables: Area, Intensity, Velocity in X and Y direction."

- Lines 255 – 261: I'm a bit lost with the way the ensemble nowcast is constructed. Are 30 individual members issued by taking every time a different selection of neighbors? Or just by one neighbor out the top 30?

Every time a nowcast is issued for a specific storm, based on the Euclidian distance the 30 most similar storms are selected. Each of these 30 most similar storms is than considered an ensemble member. The probability of each ensemble member (past similar storm) is calculated from the rank between the 30 members. One of these members is randomly selected based on their probability.

The following explanation was added in the text:

"Contrary, if a probabilistic nowcast is selected, 30- members ensembles are selected from the closest 30 storms where each member is assigned a probability according to the rank of the respective neighbour storm with the "to-be-nowcasted" storm:

$$Pr_i = \frac{(1/Rank_i)}{\sum_{i=1}^{k}(1/R_{anki})},$$  (2)

where k is the selected number of neighbours and Rank and Pr are respectively the rank and the probability weights of the ith neighbour/ensemble member. An ensemble member is then selected randomly based on the given probability weights. These probability weights calculated here are as well used for computation of the single nowcast in Equation (6). "

- Lines 277 – 298: Is there a difference in the training between VS1 and VS2, mainly w.r.t. the error per target or for all targets together? Or is the training identical for both approaches?

    The training procedure is the same for both approaches, but is done independently for both of them. In the end k=4 was implemented for both of them, to see if it is better to predict target independently or considering inter-dependency between them from the storm evolution.

- Line 304 – "Lagrangian persistence": Just to be sure, Lagrangian persistence in object-oriented nowcasting, right?

    Noted, it has been changed to Lagrangian persistence in object oriented.

    "ii) the improvement (%) per each lead time and target variable that the k-NN approach introduces to the nowcast when compared to the Lagrangian persistence in object-based approach;"

- Lines 314 – 316: Do you have an indication how much less this was (in the worst-case scenarios)?

    This is particularly affecting the stratiform events (events longer than 6 hours) because there are not many events that fall in this category. So, in total there are 33 storms that have a lifetime longer than 6 hours, which come from 28 different events. The worst case here would be that for an event only 25 storms are available to choose the nearest neighbour.

- Lines 367 – 369: Do you have any idea why the result is different for the Total Lifetime?

    My understanding, is that the duration is an easier target to be analysed, which means the values are not zero (because we consider here the total lifetime) and its distribution is not as heavy tailed as the distribution of the other variables. The other variables, depending on the lead time, have more zeros included and have an asymptotic density function. On personal experience, when zeros are not present (or at least not in the frequency of the 4 variables here) the PIC is able to represent quite well the important predictors.

- Lines 380 – 385 and Fig. 8: How large is the spread in the results (the optimal k-value) per class? I can imagine that this does not make the figure clearer when added (as IQR for example), but perhaps the authors have an indication of this.

  The IQR for each of the classes is quite high ~ 20 neighbours, sometimes up to 25 neighbours. This emphasizes again the complex relationship between predictors, time of nowcast and storm type.

- Lines 392 – 401: Is the decreasing error with increasing lead time (mainly visible in the top row of Fig. 9) a result of storms dissipating sooner than 30 min, which is then forecast well? Could the authors comment on that? It would otherwise be unexpected to see the performance improve with increasing lead time (you would expect the opposite).

  Yes, indeed one would expect the error to increase with the lead time as in the case on storms that live longer than 3 hours in Figure 9 for nowcast time 5 min. However, because the dissipation is captured well by the k4-NN, the errors are decreasing with the lead time. In the Lagrangian application, after the dissipation of the storm, the errors would be constant with the lead time.

- Lines 402 – 412: I agree with the reasoning for the 36th time step. However, the y-axis of Fig. 9 (middle row) is clearly scaled to this time step, which makes it hard to distinguish what happens for the other two lines. Could you change the axis scale and just describe why the 36th time step falls outside this scale (which is already described now)? Another question regarding the 36th time step, because I'm not sure I understand the times of nowcast well here: isn't the time of nowcasts the same as the issue time, so the nowcast starts 3h after the evolution of the storm. In the class here, you are only considering storms that last maximum 3h. Hence, aren't we looking at storms that should have died already?

  Yes we can use another range for the graph and mention that the 36th step is outside of this range. For the other case, the area errors of nowcast at 1st hour become zero after LT15min, with errors smaller than 100km2 for shorter lead times. While the area errors of nowcast at 5min become zero after LT50min. Regarding the nowcast at the 36th time step, yes the nowcast time is the issue time in respect to the initiation of the storm (when storm was first identified). The nowcast at the 36th time step, represent the nowcast issued at the last time step when the storm if observed, this means moment before complete dissipation (or how I call it death of the storm). At this point the viewer or the nowcast, has no information that the storm is going to stop is the next 5 minutes, and hence here the errors are the highest.

- Lines 452 – 453: I think this conclusion needs the nuance that this is the case for the shorter storm durations, whereas for longer durations this is not or only to a lesser extent the case.
  Line 492 – "Overall the ensemble results are better than the single 4-NN nowcast": Based on the results and the shown figure, I think you can only state that the best ensemble member performs better than the single 4-NN nowcast.
  Lines 574 – 576: There should be some nuance here. Although the results are very promising and often outcompete Lagrangian persistence, these high improvement numbers are generally reached for short-living storms, while the improvement is less (sometimes even worse) for longer-living storms

  Your comments are noted, and the lines will be changed accordingly.

- Line 586 – "additional predictors": For the interested reader(s), can you say more about which predictors from those sources you think would be feasible for this?

  Some ideas that I think may be helpful in describing the storms better, are i) the phase shift "k" from dual polarized radars, the CAPE value from a numerical weather prediction NWP, the circulation patterns that are associated with certain events, the lightening activity that can be captured by remote sensing, or other mentioned by Seppo Pulkkinen in his comments: the convective inhibition (CN), geographical features for instance as terrain altitude and proximity to water bodies, or to dense urban areas. We have added the following lines in the manuscript:

  "Further improvements can be achieved if the predictors importance is estimated better (i.e. Monte Carlo approach, or neural networks) or if additional predictors are included from other data sources like: cloud information from satellite data, convective available potential energy (CAPE) and convective inhibition (CIN) from Numerical Weather Prediction Models, lightening flash activity, additional measurements from Doppler or dual polarized radar data (like phase shift, doppler velocity, vertical profile at different elevation angles), various geographical information (as distance from heavy urbanized areas, mountains or water bodies) etc."

- Figure 4 caption: Perhaps refer to Table 1 for the meaning of the symbols (the predictors) in the figure

  Noted and updated!

**Technical corrections**

Regarding the technical correction, thank you for the feedback, they have been updated following your comments.

- Figure 2: As I hope that this interesting paper will be read by people from all over the world, it may be good to add a small subfigure indicating where this region is located in Germany or even in NWEurope. Besides, the indicated coordinates seem to be in a local coordinate system. Can I ask the authors to mention this in the caption or, perhaps even better, to place lat-lon coordinates on the map?

  Your suggestions were accepted, please refer to Figure 2 in the attached pdf.

- Figure 3: What is the last group (duration) on the x-axis?

  The last duration on the x-axis is storms longer than 12 hours. Following the comments of another reviewer the Figure has been updated below. Please note that the following changes were done to the Figure 4 (see attached pdf):

  The maximum intensity has been updated. Unfortunately, before the standard deviation was showing instead of the maximum intensity.

  The outliers of Maximum Intensity, Ratio of minor and major axis and the Orientation angle are included in the plot.

- Figures 7, 8 and 9: The light grey color for the 5-min class is not easy to distinguish (especially on a colored background). Could you make the grey a little darker or use a different color?

  Noted and implemented!

- Line 30 – "short-term rainfall nowcast": perhaps say short-term rainfall forecasting?

  Noted and changed!

- Lines 32 – 33: A minor detail about the storm and intermittent field references: the list of references can be almost endless here. It is not necessary to cite all of them, but perhaps you could say (e.g. + references) to indicate that this is just a sample of all studies to these topics.

  Noted and will be implemented in the updated version of the manuscript. Following the comments of another review we will extend the introduction and the literature review on the topic.

- Line 84 – "show for instance Hou & Wang (2017)": is for instance shown by Hou & Wang (2017).

  Noted and changed!

- Line 211 – "predictor important analysis": should this be importance instead of important?

  Noted and changed!

- Line 391 – "event-based 4-NN": For consistency, did you mean storm-based 4-NN?

  Yes, we mean the storm-based 4-NN. Throughout the text, the event-based has been substituted to storm-based.

- Line 463 – "same results": similar results.

  Noted and changed!

- Line 561 – "Person": Pearson.

  Noted and changed!

- Line 562 – "two measurement": Two measurements.

  Noted and changed!

- Line 570 – "the death processes": Perhaps better to use dissipation processes of storms?

  Throughout the text, it has been changed accordingly.

[revised manuscript text omitted]

**Appendix 8.1** *Obtained Pearson Correlation predictors weight for each target variable, lead time and storm groups. The last row at each target variable (average values) are the predictors weights shown in Table 3*

**Area**

| Durations | Lead Time | Present Predictors | | | | | | | | | | | | | | | Average Past 30min Predictors | | | | | | | | | | | | |
|---|---|---|---|---|---|---|---|---|---|---|---|---|---|---|---|---|---|---|---|---|---|---|---|---|---|---|---|---|---|
| | | Cell | Life | A | avePI | medPI | maxPI | sdPI1 | sdPI2 | Vg | Vx | Vy | Jx | Jy | Jr | Φ | A | avePI | medPI | maxPI | sdPI1 | sdPI2 | Vg | Vx | Vy | Jx | Jy | Jr | Φ |
| <1hr | 15min | 0.19 | 0.22 | 0.81 | 0.06 | 0.04 | 0.05 | 0.05 | 0.59 | 0.06 | 0.02 | 0.01 | 0.58 | 0.61 | 0.01 | 0.00 | 0.79 | 0.06 | 0.04 | 0.05 | 0.05 | 0.61 | 0.06 | 0.02 | 0.02 | 0.60 | 0.62 | 0.01 | 0.00 |
| <1hr | 60min | 0.09 | 0.27 | 0.90 | 0.19 | 0.07 | 0.07 | 0.20 | 0.64 | 0.05 | 0.05 | 0.03 | 0.65 | 0.68 | 0.04 | 0.02 | 0.88 | 0.21 | 0.09 | 0.08 | 0.22 | 0.67 | 0.08 | 0.08 | 0.07 | 0.68 | 0.70 | 0.08 | 0.02 |
| <1hr | 180min | 0.12 | 0.19 | 0.94 | 0.18 | 0.08 | 0.23 | 0.29 | 0.68 | 0.05 | 0.04 | 0.05 | 0.72 | 0.70 | 0.28 | 0.00 | 0.92 | 0.21 | 0.05 | 0.25 | 0.31 | 0.69 | 0.05 | 0.05 | 0.09 | 0.73 | 0.69 | 0.38 | 0.02 |
| <3hr | 15min | 0.12 | 0.19 | 0.61 | 0.04 | 0.03 | 0.03 | 0.04 | 0.45 | 0.00 | 0.01 | 0.00 | 0.43 | 0.49 | 0.01 | 0.01 | 0.60 | 0.05 | 0.03 | 0.02 | 0.04 | 0.46 | 0.00 | 0.02 | 0.02 | 0.45 | 0.50 | 0.01 | 0.01 |
| <3hr | 60min | 0.04 | 0.25 | 0.72 | 0.13 | 0.05 | 0.01 | 0.13 | 0.48 | 0.01 | 0.03 | 0.03 | 0.55 | 0.55 | 0.03 | 0.01 | 0.69 | 0.15 | 0.07 | 0.03 | 0.15 | 0.51 | 0.02 | 0.05 | 0.05 | 0.56 | 0.55 | 0.07 | 0.02 |
| <3hr | 180min | 0.09 | 0.13 | 0.80 | 0.16 | 0.06 | 0.20 | 0.25 | 0.58 | 0.10 | 0.01 | 0.05 | 0.63 | 0.57 | 0.24 | 0.00 | 0.77 | 0.20 | 0.02 | 0.23 | 0.28 | 0.57 | 0.11 | 0.00 | 0.09 | 0.61 | 0.55 | 0.32 | 0.02 |
| >3hr | 15min | 0.05 | 0.13 | 0.32 | 0.04 | 0.03 | 0.00 | 0.03 | 0.22 | 0.00 | 0.01 | 0.01 | 0.24 | 0.25 | 0.01 | 0.01 | 0.31 | 0.04 | 0.03 | 0.01 | 0.04 | 0.21 | 0.00 | 0.02 | 0.03 | 0.25 | 0.25 | 0.01 | 0.02 |
| >3hr | 60min | 0.03 | 0.14 | 0.42 | 0.13 | 0.08 | 0.09 | 0.14 | 0.27 | 0.07 | 0.02 | 0.02 | 0.32 | 0.26 | 0.02 | 0.02 | 0.39 | 0.15 | 0.10 | 0.10 | 0.16 | 0.27 | 0.07 | 0.02 | 0.05 | 0.32 | 0.25 | 0.05 | 0.02 |
| >3hr | 180min | 0.06 | 0.07 | 0.53 | 0.17 | 0.03 | 0.19 | 0.22 | 0.39 | 0.16 | 0.05 | 0.07 | 0.41 | 0.34 | 0.16 | 0.06 | 0.50 | 0.20 | 0.06 | 0.22 | 0.25 | 0.38 | 0.18 | 0.07 | 0.11 | 0.40 | 0.32 | 0.20 | 0.08 |
| | Average | 0.09 | 0.18 | 0.67 | 0.12 | 0.05 | 0.10 | 0.15 | 0.48 | 0.05 | 0.03 | 0.03 | 0.50 | 0.49 | 0.09 | 0.02 | 0.65 | 0.14 | 0.05 | 0.11 | 0.17 | 0.48 | 0.07 | 0.04 | 0.06 | 0.51 | 0.49 | 0.12 | 0.02 |

**Intensity**

| Durations | Lead Time | No.Cells | Life.TS | Area | meanPI | medianPI | maxPI | sdPI1 | sdPI2 | GVel | VelX | VelY | Jx | Jy | J.ratio | Phi | Area | meanPi | medPI | maxPI | sdPI1 | sdPI2 | Velg | Velx | Vely | Jx | Jy | Jr | phi |
|---|---|---|---|---|---|---|---|---|---|---|---|---|---|---|---|---|---|---|---|---|---|---|---|---|---|---|---|---|---|
| <1hr | 15min | 0.02 | 0.05 | 0.00 | 0.55 | 0.41 | 0.52 | 0.54 | 0.11 | 0.06 | 0.03 | 0.00 | 0.02 | 0.02 | 0.00 | 0.01 | 0.00 | 0.52 | 0.40 | 0.50 | 0.52 | 0.11 | 0.06 | 0.03 | 0.01 | 0.02 | 0.02 | 0.00 | 0.01 |
| <1hr | 60min | 0.04 | 0.01 | 0.12 | 0.70 | 0.53 | 0.61 | 0.69 | 0.06 | 0.04 | 0.01 | 0.07 | 0.00 | 0.01 | 0.02 | 0.02 | 0.14 | 0.64 | 0.49 | 0.59 | 0.65 | 0.03 | 0.05 | 0.01 | 0.09 | 0.02 | 0.02 | 0.02 | 0.01 |
| <1hr | 180min | 0.03 | 0.13 | 0.11 | 0.81 | 0.67 | 0.68 | 0.77 | 0.13 | 0.09 | 0.09 | 0.03 | 0.14 | 0.15 | 0.05 | 0.06 | 0.11 | 0.76 | 0.62 | 0.68 | 0.75 | 0.13 | 0.13 | 0.11 | 0.04 | 0.14 | 0.15 | 0.06 | 0.10 |
| <3hr | 15min | 0.02 | 0.10 | 0.11 | 0.15 | 0.08 | 0.22 | 0.17 | 0.14 | 0.04 | 0.02 | 0.01 | 0.08 | 0.10 | 0.01 | 0.00 | 0.10 | 0.14 | 0.07 | 0.20 | 0.16 | 0.13 | 0.04 | 0.02 | 0.01 | 0.08 | 0.09 | 0.02 | 0.01 |
| <3hr | 60min | 0.01 | 0.06 | 0.01 | 0.31 | 0.18 | 0.45 | 0.37 | 0.10 | 0.02 | 0.02 | 0.03 | 0.07 | 0.07 | 0.02 | 0.05 | 0.01 | 0.28 | 0.16 | 0.43 | 0.34 | 0.09 | 0.03 | 0.02 | 0.06 | 0.06 | 0.05 | 0.04 | 0.05 |
| <3hr | 180min | 0.01 | 0.06 | 0.10 | 0.43 | 0.40 | 0.50 | 0.47 | 0.20 | 0.08 | 0.06 | 0.01 | 0.25 | 0.22 | 0.09 | 0.09 | 0.08 | 0.42 | 0.37 | 0.47 | 0.44 | 0.19 | 0.10 | 0.08 | 0.01 | 0.24 | 0.21 | 0.12 | 0.10 |
| >3hr | 15min | 0.03 | 0.11 | 0.12 | 0.02 | 0.00 | 0.08 | 0.03 | 0.11 | 0.02 | 0.01 | 0.01 | 0.08 | 0.10 | 0.01 | 0.01 | 0.11 | 0.01 | 0.01 | 0.06 | 0.02 | 0.10 | 0.02 | 0.01 | 0.01 | 0.08 | 0.10 | 0.01 | 0.02 |
| >3hr | 60min | 0.02 | 0.06 | 0.08 | 0.07 | 0.05 | 0.17 | 0.11 | 0.09 | 0.02 | 0.00 | 0.03 | 0.06 | 0.04 | 0.02 | 0.04 | 0.05 | 0.06 | 0.04 | 0.16 | 0.10 | 0.08 | 0.02 | 0.00 | 0.05 | 0.05 | 0.03 | 0.04 | 0.04 |
| >3hr | 180min | 0.01 | 0.05 | 0.36 | 0.10 | 0.18 | 0.10 | 0.06 | 0.31 | 0.03 | 0.02 | 0.10 | 0.38 | 0.35 | 0.11 | 0.05 | 0.34 | 0.07 | 0.15 | 0.05 | 0.02 | 0.30 | 0.05 | 0.05 | 0.16 | 0.36 | 0.33 | 0.15 | 0.05 |
| | Average | 0.02 | 0.07 | 0.11 | 0.35 | 0.28 | 0.37 | 0.36 | 0.14 | 0.04 | 0.03 | 0.03 | 0.12 | 0.12 | 0.03 | 0.04 | 0.10 | 0.32 | 0.26 | 0.35 | 0.33 | 0.13 | 0.05 | 0.04 | 0.05 | 0.12 | 0.11 | 0.05 | 0.04 |

**Velocity X**

| Durations | Lead Time | No.Cells | Life.TS | Area | meanPI | medianPI | maxPI | sdPI1 | sdPI2 | GVel | VelX | VelY | Jx | Jy | J.ratio | Phi | Area | meanPi | medPI | maxPI | sdPI1 | sdPI2 | Velg | Velx | Vely | Jx | Jy | Jr | phi |
|---|---|---|---|---|---|---|---|---|---|---|---|---|---|---|---|---|---|---|---|---|---|---|---|---|---|---|---|---|---|
| <1hr | 15min | 0.04 | 0.02 | 0.09 | 0.01 | 0.01 | 0.01 | 0.00 | 0.06 | 0.14 | 0.17 | 0.02 | 0.06 | 0.05 | 0.01 | 0.02 | 0.08 | 0.01 | 0.01 | 0.01 | 0.00 | 0.08 | 0.13 | 0.18 | 0.02 | 0.14 | 0.07 | 0.02 | 0.02 |
| <1hr | 60min | 0.03 | 0.03 | 0.12 | 0.03 | 0.04 | 0.02 | 0.02 | 0.06 | 0.31 | 0.37 | 0.06 | 0.10 | 0.03 | 0.01 | 0.03 | 0.11 | 0.04 | 0.04 | 0.02 | 0.03 | 0.04 | 0.33 | 0.52 | 0.09 | 0.15 | 0.04 | 0.00 | 0.03 |
| <1hr | 180min | 0.04 | 0.01 | 0.06 | 0.05 | 0.06 | 0.04 | 0.05 | 0.00 | 0.27 | 0.32 | 0.05 | 0.12 | 0.06 | 0.01 | 0.06 | 0.07 | 0.04 | 0.05 | 0.03 | 0.04 | 0.00 | 0.35 | 0.42 | 0.05 | 0.16 | 0.07 | 0.01 | 0.05 |
| <3hr | 15min | 0.03 | 0.06 | 0.10 | 0.02 | 0.01 | 0.01 | 0.01 | 0.06 | 0.03 | 0.07 | 0.01 | 0.05 | 0.04 | 0.01 | 0.02 | 0.08 | 0.02 | 0.02 | 0.00 | 0.02 | 0.05 | 0.03 | 0.06 | 0.01 | 0.15 | 0.03 | 0.02 | 0.03 |
| <3hr | 60min | 0.06 | 0.06 | 0.15 | 0.03 | 0.02 | 0.03 | 0.02 | 0.06 | 0.20 | 0.30 | 0.06 | 0.11 | 0.05 | 0.01 | 0.03 | 0.14 | 0.05 | 0.04 | 0.03 | 0.03 | 0.05 | 0.25 | 0.42 | 0.07 | 0.16 | 0.04 | 0.01 | 0.04 |
| <3hr | 180min | 0.04 | 0.01 | 0.10 | 0.03 | 0.04 | 0.02 | 0.02 | 0.02 | 0.27 | 0.26 | 0.04 | 0.13 | 0.07 | 0.00 | 0.06 | 0.10 | 0.02 | 0.04 | 0.02 | 0.02 | 0.02 | 0.29 | 0.38 | 0.05 | 0.18 | 0.07 | 0.00 | 0.05 |
| >3hr | 15min | 0.04 | 0.06 | 0.10 | 0.02 | 0.02 | 0.01 | 0.01 | 0.04 | 0.02 | 0.05 | 0.02 | 0.04 | 0.01 | 0.01 | 0.02 | 0.09 | 0.02 | 0.02 | 0.00 | 0.02 | 0.04 | 0.02 | 0.05 | 0.01 | 0.15 | 0.01 | 0.01 | 0.02 |
| >3hr | 60min | 0.04 | 0.04 | 0.05 | 0.04 | 0.04 | 0.03 | 0.04 | 0.04 | 0.07 | 0.16 | 0.05 | 0.00 | 0.00 | 0.01 | 0.02 | 0.04 | 0.05 | 0.04 | 0.03 | 0.04 | 0.05 | 0.08 | 0.23 | 0.07 | 0.15 | 0.05 | 0.02 | 0.02 |
| >3hr | 180min | 0.03 | 0.02 | 0.10 | 0.00 | 0.03 | 0.02 | 0.02 | 0.03 | 0.15 | 0.17 | 0.03 | 0.11 | 0.04 | 0.01 | 0.03 | 0.10 | 0.01 | 0.02 | 0.02 | 0.02 | 0.03 | 0.16 | 0.24 | 0.03 | 0.15 | 0.05 | 0.01 | 0.04 |
| | Average | 0.04 | 0.03 | 0.10 | 0.03 | 0.03 | 0.02 | 0.02 | 0.04 | 0.16 | 0.21 | 0.04 | 0.08 | 0.04 | 0.01 | 0.03 | 0.09 | 0.03 | 0.03 | 0.02 | 0.02 | 0.04 | 0.18 | 0.28 | 0.04 | 0.15 | 0.05 | 0.01 | 0.03 |

**Velocity Y**

| Durations | Lead Time | No.Cells | Life.TS | Area | meanPI | medianPI | maxPI | sdPI1 | sdPI2 | GVel | VelX | VelY | Jx | Jy | J.ratio | Phi | Area | meanPi | medPI | maxPI | sdPI1 | sdPI2 | Velg | Velx | Vely | Jx | Jy | Jr | phi |
|---|---|---|---|---|---|---|---|---|---|---|---|---|---|---|---|---|---|---|---|---|---|---|---|---|---|---|---|---|---|
| <1hr | 15min | 0.04 | 0.04 | 0.04 | 0.02 | 0.00 | 0.05 | 0.03 | 0.06 | 0.03 | 0.02 | 0.15 | 0.03 | 0.03 | 0.01 | 0.00 | 0.04 | 0.02 | 0.00 | 0.04 | 0.03 | 0.07 | 0.04 | 0.03 | 0.17 | 0.04 | 0.03 | 0.01 | 0.00 |
| <1hr | 60min | 0.00 | 0.04 | 0.02 | 0.08 | 0.07 | 0.09 | 0.08 | 0.00 | 0.03 | 0.05 | 0.22 | 0.00 | 0.00 | 0.01 | 0.02 | 0.03 | 0.08 | 0.07 | 0.09 | 0.08 | 0.00 | 0.01 | 0.06 | 0.33 | 0.01 | 0.00 | 0.02 | 0.02 |
| <1hr | 180min | 0.03 | 0.08 | 0.05 | 0.02 | 0.01 | 0.03 | 0.03 | 0.05 | 0.00 | 0.04 | 0.27 | 0.07 | 0.01 | 0.02 | 0.02 | 0.06 | 0.02 | 0.01 | 0.03 | 0.03 | 0.06 | 0.01 | 0.05 | 0.41 | 0.08 | 0.02 | 0.01 | 0.02 |
| <3hr | 15min | 0.01 | 0.06 | 0.06 | 0.03 | 0.02 | 0.07 | 0.04 | 0.07 | 0.01 | 0.01 | 0.05 | 0.03 | 0.05 | 0.00 | 0.01 | 0.05 | 0.03 | 0.01 | 0.06 | 0.04 | 0.06 | 0.02 | 0.00 | 0.04 | 0.02 | 0.04 | 0.00 | 0.00 |
| <3hr | 60min | 0.01 | 0.06 | 0.02 | 0.04 | 0.03 | 0.10 | 0.06 | 0.03 | 0.01 | 0.06 | 0.18 | 0.02 | 0.05 | 0.01 | 0.01 | 0.00 | 0.04 | 0.03 | 0.11 | 0.06 | 0.03 | 0.00 | 0.07 | 0.26 | 0.01 | 0.04 | 0.01 | 0.01 |
| <3hr | 180min | 0.00 | 0.06 | 0.03 | 0.00 | 0.00 | 0.00 | 0.00 | 0.06 | 0.01 | 0.03 | 0.22 | 0.07 | 0.04 | 0.01 | 0.01 | 0.04 | 0.01 | 0.01 | 0.00 | 0.00 | 0.07 | 0.00 | 0.04 | 0.33 | 0.07 | 0.05 | 0.01 | 0.01 |
| >3hr | 15min | 0.01 | 0.07 | 0.03 | 0.00 | 0.01 | 0.03 | 0.01 | 0.03 | 0.01 | 0.01 | 0.04 | 0.00 | 0.01 | 0.01 | 0.02 | 0.02 | 0.00 | 0.01 | 0.02 | 0.00 | 0.02 | 0.01 | 0.01 | 0.03 | 0.00 | 0.00 | 0.01 | 0.03 |
| >3hr | 60min | 0.03 | 0.02 | 0.02 | 0.01 | 0.01 | 0.03 | 0.01 | 0.07 | 0.00 | 0.04 | 0.09 | 0.09 | 0.08 | 0.01 | 0.00 | 0.04 | 0.01 | 0.01 | 0.03 | 0.00 | 0.09 | 0.01 | 0.07 | 0.16 | 0.10 | 0.10 | 0.01 | 0.00 |
| >3hr | 180min | 0.00 | 0.01 | 0.01 | 0.05 | 0.04 | 0.04 | 0.04 | 0.04 | 0.01 | 0.02 | 0.14 | 0.09 | 0.08 | 0.01 | 0.01 | 0.02 | 0.05 | 0.05 | 0.04 | 0.04 | 0.05 | 0.02 | 0.01 | 0.22 | 0.10 | 0.08 | 0.00 | 0.00 |
| | Average | 0.01 | 0.05 | 0.03 | 0.03 | 0.02 | 0.05 | 0.03 | 0.05 | 0.01 | 0.03 | 0.15 | 0.04 | 0.04 | 0.01 | 0.01 | 0.03 | 0.03 | 0.02 | 0.05 | 0.03 | 0.05 | 0.01 | 0.04 | 0.22 | 0.05 | 0.04 | 0.01 | 0.01 |

**Duration**

| Durations | Lead Time | No.Cells | Life.TS | Area | meanPI | medianPI | maxPI | sdPI1 | sdPI2 | GVel | VelX | VelY | Jx | Jy | J.ratio | Phi | Area | meanPi | medPI | maxPI | sdPI1 | sdPI2 | Velg | Velx | Vely | Jx | Jy | Jr | phi |
|---|---|---|---|---|---|---|---|---|---|---|---|---|---|---|---|---|---|---|---|---|---|---|---|---|---|---|---|---|---|
| Dur <1hr | | 0.06 | 0.16 | 0.31 | 0.02 | 0.02 | 0.04 | 0.00 | 0.22 | 0.03 | 0.03 | 0.00 | 0.19 | 0.24 | 0.00 | 0.03 | 0.30 | 0.02 | 0.03 | 0.03 | 0.01 | 0.21 | 0.03 | 0.03 | 0.01 | 0.19 | 0.25 | 0.01 | 0.04 |
| Dur <3hr | | 0.00 | 0.16 | 0.35 | 0.10 | 0.05 | 0.04 | 0.10 | 0.22 | 0.08 | 0.02 | 0.04 | 0.23 | 0.21 | 0.02 | 0.11 | 0.32 | 0.12 | 0.07 | 0.05 | 0.11 | 0.22 | 0.08 | 0.03 | 0.07 | 0.23 | 0.21 | 0.05 | 0.15 |
| Dur >3hr | | 0.07 | 0.02 | 0.43 | 0.20 | 0.11 | 0.18 | 0.21 | 0.21 | 0.15 | 0.04 | 0.04 | 0.25 | 0.16 | 0.14 | 0.01 | 0.40 | 0.22 | 0.14 | 0.20 | 0.23 | 0.20 | 0.18 | 0.07 | 0.07 | 0.23 | 0.14 | 0.18 | 0.01 |
| | Average | 0.04 | 0.11 | 0.36 | 0.11 | 0.06 | 0.09 | 0.10 | 0.22 | 0.09 | 0.03 | 0.03 | 0.22 | 0.20 | 0.05 | 0.05 | 0.34 | 0.12 | 0.08 | 0.09 | 0.12 | 0.21 | 0.10 | 0.04 | 0.05 | 0.22 | 0.20 | 0.08 | 0.07 |

***Appendix 8.2*** *Obtained PIC predictors weight for each target variable, lead time and storm groups. The last row at each target variable (average values) are the predictors weights shown in Table 3.*

**Area** (target variable)

| Durations | Lead Time | Cell | Life | A | avePI | medPI | maxPI | sdPI1 | sdPI2 | Vg | Vx | Vy | Jx | Jy | Jr | Φ | A | avePI | medPI | maxPI | sdPI1 | sdPI2 | Vg | Vx | Vy | Jx | Jy | Jr | phi |
|---|---|---|---|---|---|---|---|---|---|---|---|---|---|---|---|---|---|---|---|---|---|---|---|---|---|---|---|---|---|
| <1hr | 15min | 0.00 | 0.00 | 0.00 | 0.00 | 0.00 | 0.00 | 0.00 | 0.00 | 0.00 | 0.00 | 0.00 | 0.00 | 0.00 | 0.00 | 0.00 | 0.00 | 0.00 | 0.00 | 0.00 | 0.00 | 0.00 | 0.00 | 0.00 | 0.00 | 0.00 | 0.00 | 1.00 | 0.00 |
| | 60min | 0.00 | 0.00 | 0.00 | 0.00 | 0.00 | 0.00 | 0.00 | 0.00 | 0.00 | 0.00 | 0.00 | 0.00 | 0.00 | 0.00 | 0.00 | 0.00 | 0.00 | 0.00 | 0.00 | 0.00 | 0.00 | 0.00 | 0.00 | 0.00 | 0.00 | 0.00 | 0.00 | 0.00 |
| | 180min | 0.00 | 0.00 | 0.00 | 0.00 | 0.00 | 0.00 | 0.00 | 0.00 | 0.00 | 0.00 | 0.00 | 0.00 | 0.00 | 0.00 | 0.00 | 0.00 | 0.00 | 0.00 | 0.00 | 0.00 | 0.00 | 0.00 | 0.00 | 0.00 | 0.00 | 0.00 | 1.00 | 0.00 |
| <3hr | 15min | 0.00 | 0.00 | 0.00 | 0.00 | 0.00 | 0.00 | 0.00 | 0.00 | 0.00 | 0.00 | 0.00 | 0.00 | 0.00 | 0.00 | 0.00 | 0.00 | 0.00 | 0.00 | 0.00 | 0.00 | 0.00 | 1.00 | 0.00 | 0.00 | 0.00 | 0.00 | 0.00 | 0.00 |
| | 60min | 0.00 | 0.00 | 0.00 | 0.00 | 0.00 | 0.00 | 0.00 | 0.00 | 0.00 | 0.00 | 0.00 | 0.00 | 0.00 | 0.00 | 0.00 | 0.00 | 0.00 | 0.00 | 0.00 | 0.00 | 0.00 | 1.00 | 0.00 | 0.00 | 0.00 | 0.00 | 0.00 | 0.00 |
| | 180min | 0.00 | 0.00 | 0.00 | 0.00 | 0.00 | 0.00 | 0.00 | 0.00 | 0.00 | 0.00 | 0.00 | 0.00 | 0.00 | 0.00 | 0.00 | 0.00 | 0.00 | 0.00 | 0.00 | 0.00 | 0.00 | 1.00 | 0.00 | 0.00 | 0.00 | 0.00 | 0.00 | 0.00 |
| >3hr | 15min | 0.00 | 0.10 | 0.25 | 0.00 | 0.00 | 0.00 | 0.00 | 0.00 | 0.57 | 0.00 | 0.00 | 0.00 | 0.00 | 0.00 | 0.00 | 0.08 | 0.00 | 0.00 | 0.00 | 0.00 | 0.00 | 0.00 | 0.00 | 0.00 | 0.00 | 0.00 | 0.00 | 0.00 |
| | 60min | 0.00 | 0.29 | 0.67 | 0.00 | 0.00 | 0.00 | 0.00 | 0.00 | 0.66 | 0.00 | 0.00 | 0.00 | 0.00 | 0.00 | 0.00 | 0.00 | 0.00 | 0.00 | 0.00 | 0.00 | 0.00 | 0.00 | 0.00 | 0.33 | 0.00 | 0.00 | 0.00 | 0.00 |
| | 180min | 0.00 | 0.30 | 0.40 | 0.00 | 0.00 | 0.00 | 0.00 | 0.00 | 0.72 | 0.00 | 0.00 | 0.00 | 0.00 | 0.00 | 0.00 | 0.00 | 0.00 | 0.00 | 0.00 | 0.00 | 0.00 | 0.00 | 0.00 | 0.28 | 0.00 | 0.00 | 0.00 | 0.00 |
| | Average | 0.00 | 0.08 | 0.15 | 0.00 | 0.00 | 0.00 | 0.00 | 0.00 | 0.22 | 0.00 | 0.00 | 0.00 | 0.00 | 0.00 | 0.00 | 0.01 | 0.00 | 0.00 | 0.00 | 0.00 | 0.00 | 0.33 | 0.00 | 0.07 | 0.00 | 0.00 | 0.33 | 0.00 |

**Intensity** (target variable)

| Durations | Lead Time | No.Cells | Life.TS | Area | meanPI | medianPI | maxPI | sdPI1 | sdPI2 | GVel | VelX | VelY | Jx | Jy | J.ratio | Phi | Area | meanPi | medPI | maxPI | sdPI1 | sdPI2 | Velg | Velx | Vely | Jx | Jy | Jr | phi |
|---|---|---|---|---|---|---|---|---|---|---|---|---|---|---|---|---|---|---|---|---|---|---|---|---|---|---|---|---|---|
| <1hr | 15min | 0.00 | 0.00 | 0.00 | 0.00 | 0.00 | 0.00 | 0.00 | 0.00 | 0.00 | 0.00 | 0.00 | 0.00 | 0.00 | 0.00 | 0.00 | 0.00 | 0.00 | 0.00 | 0.00 | 0.00 | 0.00 | 0.00 | 0.00 | 0.00 | 0.00 | 0.00 | 0.00 | 0.00 |
| | 60min | 0.00 | 0.00 | 0.00 | 0.00 | 0.00 | 0.00 | 0.00 | 0.00 | 0.00 | 0.00 | 0.00 | 0.00 | 0.00 | 0.00 | 0.00 | 0.00 | 0.00 | 0.00 | 0.00 | 0.00 | 0.00 | 0.00 | 0.00 | 0.00 | 0.00 | 0.00 | 0.00 | 0.00 |
| | 180min | 0.00 | 0.00 | 0.00 | 0.00 | 0.00 | 0.00 | 0.00 | 0.00 | 0.00 | 0.00 | 0.00 | 0.00 | 0.00 | 0.00 | 0.00 | 0.00 | 0.00 | 0.00 | 0.00 | 0.00 | 0.00 | 0.00 | 0.00 | 0.00 | 0.00 | 0.00 | 0.00 | 0.00 |
| <3hr | 15min | 0.00 | 0.00 | 0.00 | 0.00 | 0.00 | 0.00 | 0.00 | 0.00 | 0.00 | 0.00 | 0.00 | 0.00 | 0.00 | 1.00 | 0.00 | 0.00 | 0.00 | 0.00 | 0.00 | 0.00 | 0.00 | 0.00 | 0.00 | 0.00 | 0.00 | 0.00 | 0.00 | 0.00 |
| | 60min | 0.00 | 0.00 | 0.00 | 0.00 | 0.00 | 0.00 | 0.00 | 0.00 | 0.00 | 0.00 | 0.00 | 0.00 | 0.00 | 1.00 | 0.00 | 0.00 | 0.00 | 0.00 | 0.00 | 0.00 | 0.00 | 0.00 | 0.00 | 0.00 | 0.00 | 0.00 | 0.00 | 0.00 |
| | 180min | 0.00 | 0.00 | 0.00 | 0.00 | 0.00 | 0.00 | 0.00 | 0.00 | 0.00 | 0.00 | 0.00 | 0.00 | 0.00 | 1.00 | 0.00 | 0.00 | 0.00 | 0.00 | 0.00 | 0.00 | 0.00 | 0.00 | 0.00 | 0.00 | 0.00 | 0.00 | 0.00 | 0.00 |
| >3hr | 15min | 0.00 | 0.00 | 0.00 | 0.00 | 0.00 | 0.00 | 0.00 | 0.00 | 0.00 | 0.00 | 0.00 | 0.00 | 0.00 | 1.00 | 0.00 | 0.00 | 0.00 | 0.00 | 0.00 | 0.00 | 0.00 | 0.00 | 0.00 | 0.00 | 0.00 | 0.00 | 0.00 | 0.00 |
| | 60min | 0.00 | 0.00 | 0.00 | 0.00 | 0.00 | 0.00 | 0.00 | 0.00 | 0.00 | 0.00 | 0.00 | 0.00 | 0.00 | 1.00 | 0.00 | 0.00 | 0.00 | 0.00 | 0.00 | 0.00 | 0.00 | 0.00 | 0.00 | 0.00 | 0.00 | 0.00 | 0.00 | 0.00 |
| | 180min | 0.00 | 0.00 | 0.00 | 0.00 | 0.00 | 0.00 | 0.00 | 0.00 | 0.00 | 0.00 | 0.00 | 0.00 | 0.00 | 1.00 | 0.00 | 0.00 | 0.00 | 0.00 | 0.00 | 0.00 | 0.00 | 0.00 | 0.00 | 0.00 | 0.00 | 0.00 | 0.00 | 0.00 |
| | Average | 0.00 | 0.00 | 0.00 | 0.00 | 0.00 | 0.00 | 0.00 | 0.00 | 0.00 | 0.00 | 0.00 | 0.00 | 0.00 | 1.00 | 0.00 | 0.00 | 0.00 | 0.00 | 0.00 | 0.00 | 0.00 | 0.00 | 0.00 | 0.00 | 0.00 | 0.00 | 0.00 | 0.00 |

**Velocity X** (target variable)

| Durations | Lead Time | No.Cells | Life.TS | Area | meanPI | medianPI | maxPI | sdPI1 | sdPI2 | GVel | VelX | VelY | Jx | Jy | J.ratio | Phi | Area | meanPi | medPI | maxPI | sdPI1 | sdPI2 | Velg | Velx | Vely | Jx | Jy | Jr | phi |
|---|---|---|---|---|---|---|---|---|---|---|---|---|---|---|---|---|---|---|---|---|---|---|---|---|---|---|---|---|---|
| <1hr | 15min | 0.00 | 0.00 | 0.00 | 0.00 | 0.00 | 0.00 | 0.00 | 0.00 | 0.00 | 0.00 | 0.00 | 0.00 | 0.00 | 0.00 | 0.00 | 0.00 | 0.00 | 0.00 | 0.00 | 0.00 | 0.00 | 0.00 | 0.00 | 0.00 | 0.00 | 0.00 | 0.00 | 0.00 |
| | 60min | 0.00 | 0.00 | 0.00 | 0.00 | 0.00 | 0.00 | 0.00 | 0.00 | 0.00 | 0.00 | 0.00 | 0.00 | 0.00 | 0.00 | 0.00 | 0.00 | 0.00 | 0.00 | 0.00 | 0.00 | 0.00 | 0.00 | 0.00 | 0.00 | 0.00 | 0.00 | 0.00 | 0.00 |
| | 180min | 0.00 | 0.00 | 0.00 | 0.00 | 0.00 | 0.00 | 0.00 | 0.00 | 0.00 | 0.00 | 0.00 | 0.00 | 0.00 | 0.00 | 0.00 | 0.00 | 0.00 | 0.00 | 0.00 | 0.00 | 0.00 | 0.00 | 0.00 | 0.00 | 0.00 | 0.00 | 0.00 | 0.00 |
| <3hr | 15min | 0.00 | 0.00 | 0.00 | 0.00 | 0.00 | 0.00 | 0.00 | 0.00 | 0.00 | 0.00 | 0.00 | 0.00 | 0.00 | 0.00 | 0.00 | 0.00 | 0.00 | 0.00 | 0.00 | 0.00 | 0.00 | 0.00 | 1.00 | 0.00 | 0.00 | 0.00 | 0.00 | 0.00 |
| | 60min | 0.00 | 0.00 | 0.00 | 0.00 | 0.00 | 0.00 | 0.00 | 0.00 | 0.00 | 0.00 | 0.00 | 0.00 | 0.00 | 0.00 | 0.00 | 0.00 | 0.00 | 0.00 | 0.00 | 0.00 | 0.00 | 0.00 | 1.00 | 0.00 | 0.00 | 0.00 | 0.00 | 0.00 |
| | 180min | 0.00 | 0.00 | 0.00 | 0.00 | 0.00 | 0.00 | 0.00 | 0.00 | 0.00 | 0.00 | 0.00 | 0.00 | 0.00 | 0.00 | 0.00 | 0.00 | 0.00 | 0.00 | 0.00 | 0.00 | 0.00 | 0.00 | 1.00 | 0.00 | 0.00 | 0.00 | 0.00 | 0.00 |
| >3hr | 15min | 0.00 | 0.00 | 0.00 | 0.00 | 0.00 | 0.00 | 0.00 | 0.00 | 0.00 | 0.00 | 0.00 | 0.00 | 0.00 | 0.00 | 0.00 | 0.00 | 0.00 | 0.00 | 0.00 | 0.00 | 0.00 | 0.00 | 0.00 | 0.00 | 0.00 | 0.00 | 0.00 | 0.00 |
| | 60min | 0.00 | 0.00 | 0.00 | 0.00 | 0.00 | 0.00 | 0.00 | 0.00 | 0.00 | 0.00 | 0.00 | 0.00 | 0.00 | 0.00 | 0.00 | 0.00 | 0.00 | 0.00 | 0.00 | 0.00 | 0.00 | 0.00 | 0.00 | 0.00 | 0.00 | 0.00 | 0.00 | 0.00 |
| | 180min | 0.00 | 0.00 | 0.00 | 0.00 | 0.00 | 0.00 | 0.00 | 0.00 | 0.00 | 0.00 | 0.00 | 0.00 | 0.00 | 0.00 | 0.00 | 0.00 | 0.00 | 0.00 | 0.00 | 0.00 | 0.00 | 0.00 | 0.00 | 0.00 | 0.00 | 0.00 | 0.00 | 0.00 |
| | Average | 0.00 | 0.00 | 0.00 | 0.00 | 0.00 | 0.00 | 0.00 | 0.00 | 0.00 | 0.00 | 0.00 | 0.00 | 0.00 | 0.00 | 0.00 | 0.00 | 0.00 | 0.00 | 0.00 | 0.00 | 0.00 | 0.00 | 1.00 | 0.00 | 0.00 | 0.00 | 0.00 | 0.00 |

**Velocity Y** (target variable)

| Durations | Lead Time | No.Cells | Life.TS | Area | meanPI | medianPI | maxPI | sdPI1 | sdPI2 | GVel | VelX | VelY | Jx | Jy | J.ratio | Phi | Area | meanPi | medPI | maxPI | sdPI1 | sdPI2 | Velg | Velx | Vely | Jx | Jy | Jr | phi |
|---|---|---|---|---|---|---|---|---|---|---|---|---|---|---|---|---|---|---|---|---|---|---|---|---|---|---|---|---|---|
| <1hr | 15min | 0.00 | 0.00 | 0.00 | 0.00 | 0.00 | 0.00 | 0.00 | 0.00 | 0.00 | 0.00 | 0.00 | 0.00 | 0.00 | 0.00 | 0.00 | 0.00 | 0.00 | 0.00 | 0.00 | 0.00 | 0.00 | 0.00 | 0.00 | 0.00 | 0.00 | 0.00 | 0.00 | 0.00 |
| | 60min | 0.00 | 0.00 | 0.00 | 0.00 | 0.00 | 0.00 | 0.00 | 0.00 | 0.00 | 0.00 | 0.00 | 0.00 | 0.00 | 0.00 | 0.00 | 0.00 | 0.00 | 0.00 | 0.00 | 0.00 | 0.00 | 0.00 | 0.00 | 0.00 | 0.00 | 0.00 | 0.00 | 0.00 |
| | 180min | 0.00 | 0.00 | 0.00 | 0.00 | 0.00 | 0.00 | 0.00 | 0.00 | 0.00 | 0.00 | 0.00 | 0.00 | 0.00 | 0.00 | 0.00 | 0.00 | 0.00 | 0.00 | 0.00 | 0.00 | 0.00 | 0.00 | 0.00 | 0.00 | 0.00 | 0.00 | 0.00 | 0.00 |
| <3hr | 15min | 0.00 | 0.00 | 0.00 | 0.00 | 0.00 | 0.00 | 0.00 | 0.00 | 0.00 | 0.00 | 0.00 | 0.00 | 0.00 | 0.00 | 0.00 | 0.00 | 0.00 | 0.00 | 0.00 | 0.00 | 0.00 | 0.00 | 0.00 | 0.00 | 0.00 | 0.00 | 0.00 | 0.00 |
| | 60min | 0.00 | 0.00 | 0.00 | 0.00 | 0.00 | 0.00 | 0.00 | 0.00 | 0.00 | 0.00 | 0.00 | 0.00 | 0.00 | 0.00 | 0.00 | 0.00 | 0.00 | 0.00 | 0.00 | 0.00 | 0.00 | 0.00 | 0.00 | 0.00 | 0.00 | 0.00 | 0.00 | 0.00 |
| | 180min | 0.00 | 0.00 | 0.00 | 0.00 | 0.00 | 0.00 | 0.00 | 0.00 | 0.00 | 0.00 | 0.00 | 0.00 | 0.00 | 0.00 | 0.00 | 0.00 | 0.00 | 0.00 | 0.00 | 0.00 | 0.00 | 0.00 | 0.00 | 0.00 | 0.00 | 0.00 | 0.00 | 0.00 |
| >3hr | 15min | 0.00 | 0.00 | 0.00 | 0.00 | 0.00 | 0.00 | 0.00 | 0.00 | 0.00 | 0.00 | 0.00 | 0.00 | 0.00 | 0.00 | 0.00 | 0.00 | 0.00 | 0.00 | 0.00 | 0.00 | 0.00 | 0.00 | 0.00 | 0.00 | 1.00 | 0.00 | 0.00 | 0.00 |
| | 60min | 0.00 | 0.00 | 0.00 | 0.00 | 0.00 | 0.00 | 0.00 | 0.00 | 0.00 | 0.00 | 0.00 | 0.00 | 0.00 | 0.00 | 0.00 | 0.00 | 0.00 | 0.00 | 0.00 | 0.00 | 0.00 | 0.00 | 0.00 | 0.00 | 1.00 | 0.00 | 0.00 | 0.00 |
| | 180min | 0.00 | 0.00 | 0.00 | 0.00 | 0.00 | 0.00 | 0.00 | 0.00 | 0.00 | 0.00 | 0.00 | 0.00 | 0.00 | 0.00 | 0.00 | 0.00 | 0.00 | 0.00 | 0.00 | 0.00 | 0.00 | 0.00 | 0.00 | 0.00 | 1.00 | 0.00 | 0.00 | 0.00 |
| | Average | 0.00 | 0.00 | 0.00 | 0.00 | 0.00 | 0.00 | 0.00 | 0.00 | 0.00 | 0.00 | 0.00 | 0.00 | 0.00 | 0.00 | 0.00 | 0.00 | 0.00 | 0.00 | 0.00 | 0.00 | 0.00 | 0.00 | 0.00 | 0.00 | 1.00 | 0.00 | 0.00 | 0.00 |

**Duration** (target variable)

| Durations | Lead Time | No.Cells | Life.TS | Area | meanPI | medianPI | maxPI | sdPI1 | sdPI2 | GVel | VelX | VelY | Jx | Jy | J.ratio | Phi | Area | meanPi | medPI | maxPI | sdPI1 | sdPI2 | Velg | Velx | Vely | Jx | Jy | Jr | phi |
|---|---|---|---|---|---|---|---|---|---|---|---|---|---|---|---|---|---|---|---|---|---|---|---|---|---|---|---|---|---|
| Dur <1hr | | 0.00 | 0.00 | 0.00 | 0.00 | 0.00 | 0.00 | 0.00 | 0.00 | 0.00 | 0.00 | 0.00 | 0.00 | 0.00 | 0.00 | 0.00 | 0.00 | 0.00 | 0.00 | 0.00 | 0.00 | 0.00 | 0.00 | 0.00 | 0.00 | 0.00 | 0.00 | 1.00 | 0.00 |
| Dur <3hr | | 0.00 | 0.00 | 0.00 | 0.00 | 0.00 | 0.00 | 0.00 | 0.00 | 0.00 | 0.00 | 0.00 | 0.00 | 0.00 | 0.00 | 0.00 | 0.00 | 0.00 | 0.00 | 0.00 | 0.00 | 0.00 | 1.00 | 0.00 | 0.00 | 0.00 | 0.00 | 0.00 | 0.00 |
| Dur >3hr | | 0.00 | 0.45 | 0.40 | 0.00 | 0.00 | 0.00 | 0.00 | 0.00 | 0.72 | 0.00 | 0.00 | 0.00 | 0.00 | 0.00 | 0.00 | 0.00 | 0.00 | 0.00 | 0.00 | 0.00 | 0.00 | 0.00 | 0.00 | 0.00 | 0.00 | 0.00 | 0.33 | 0.00 |
| | Average | 0.00 | 0.15 | 0.13 | 0.00 | 0.00 | 0.00 | 0.00 | 0.00 | 0.24 | 0.00 | 0.00 | 0.00 | 0.00 | 0.00 | 0.00 | 0.00 | 0.00 | 0.00 | 0.00 | 0.00 | 0.00 | 0.33 | 0.00 | 0.00 | 0.00 | 0.11 | 0.33 | 0.00 |

---

## Author Comment (AC2)

Dear Seppo,

thank you for your comments and suggestions, that have contributed in improving the manuscript and the idea after the nearest neighbour application. Following your suggestions, one of the main changes that I have done, was to update the results based on absolute error and computed the Continuous Rank Probability Score for the probabilistic nowcast. Thus, the results plots have changed accordingly, and are given in the response to this review. Please find below the answers and our comments on the questions/issues you have raised (given in blue below each point). Please note that some changes have been already done in the updated version of the manuscript (these are given in quotation) while other changes are yet to be done and thus I refer to as "will be included" in the updated version of the manuscript. The updated plots and tables are given in the attached pdf at the end of this response.

**General Comments:**

- To make the title better reflect the content, you could add the word "rainfall" because the paper is about rainfall nowcasting

    Following your suggestion and that of another review, the title has been changed accordingly. We will ask the editor if the title can be change to the following, to give the idea that this is only a first step in implementing the nearest neighbour method.
    "Improving radar-based rainfall nowcast by a nearest neighbour approach: Part I – Storm Characteristics"
    Please note that we have added "Part I – Storm Characteristics", as this paper is focusing only on 5 characteristics (Area, intensity, velocity x and y and duration). The plan is to follow this paper with a second one "Part II – Point scale Intensities" where the focus is the predictability of rainfall intensities at point scales (1km$^2$) for urban flood application.

- Before going directly to the matter, it could be worthwhile to add one general paragraph about nowcasting. Like why nowcasting is done, its societal need and what kind of hazards can be prevented with reliable rainfall nowcasts. In the beginning of the introduction, the authors should make a clearer distinction between the two nowcasting approaches (field- and object-based) and add more description about what purposes they are used for. For instance, mention that field-based methods are well-suited for predicting large-scale stratiform precipitation systems but cell-based methods are best-suited for predicting the motion of intense convective cells.

    The introduction to the rainfall nowcast topic and the literature review will be updated according to your suggestions and those of the first reviewer.

- To put their work into a broader context, the authors could mention in the introduction that the proposed approach is conceptually similar to the so-called analogue-based nowcasting. The idea of this approach is to look for similar events from a large sample of archived radar data. See, for instance: L. Panziera, U. Germann, M. Gabella and P. Mandapaka: NORA–Nowcasting of Orographic Rainfall by means of Analogues, Quarterly Journal of the Royal Meteorological Society, 137(661), 2106-2123, 2011. There are a number of others, so I recommend the authors to do a literature review. However, all the previous studies I know attempt to find analogs from full radar images, not from individual cells or their features. This is a novel aspect, which should be clearly pointed out in the manuscript.

    Thank you for your suggestion. Of course, we will take your advice and mention the analogue-approach from Panziera et al and relate with the nearest neighbour application.

- I have concerns about the choice of the predictors. The proposed methodology opens the possibility to use a large number of different predictors (and targets). However, the set chosen in the study is in my opinion quite limited and the capability of the model is thus not fully utilized. In addition, they are more or less correlated with each other, and also with the target variables, which the authors

admit. I think that using the following additional predictors could reveal the full potential of the model: ◦ convective available potential energy (CAPE) ◦ convective inhibition (CIN) ◦ signatures from radar-measured Doppler and polarimetric parameters, as well as vertical profile information obtained by using all elevation angles ◦ lightning flash density ◦ geographical features like terrain altitude or proximity of water bodies These are probably beyond the scope of this study, but I encourage the authors to include them in a follow-up paper. In addition, the authors could replace the generic description of additional predictors in the last paragraph of Section 5 by specifically mentioning some of the above. Note that a relationship between CAPE and CIN and the life cycle of convective cells is suggested in: C. Moseley, O. Henneberg and J. O. Harter: A Statistical Model for Isolated Convective Precipitation Events, Journal of Advances in Modeling Earth Systems, 11(1), 360-375, 2019.

> Regarding the choice of the predictors, I agree with your concerns, however the main idea here was to investigate if only the storm's characteristics derived from the radar data are able to give some extra information in improving the radar-based nowcast. Of course, in the future we plan to extend the methodology in conjunction with numerical weather prediction models (based on CAPE or CIN) and geographical features (although for the case study it is a bit difficult because the terrain is mainly flat). Extra information from Doppler or Dual-polarized radar data is quite promising to my opinion, because storms type could be distinguished better, but may be limited to the data availability in the study region. Lightning flash density is also another good idea and it has been proven to be correlated with rainfall intensity, but mainly for high lightening activity. For this manuscript, we have updated the possible predictors text in Section 5 with this and more possible predictors that can be used in the future works as follows:

> "Further improvements can be achieved if the predictors importance is estimated better (i.e. Monte Carlo approach, or neural networks) or if additional predictors are included from other data sources like: cloud information from satellite data, convective available potential energy (CAPE) and convective inhibition (CIN) from Numerical Weather Prediction Models, lightening flash activity, additional measurements from Doppler or dual polarized radar data (like phase shift, doppler velocity, vertical profile at different elevation angels), various geographical information (as distance from heavy urbanized areas, mountains or water bodies) etc."

- A fundamental reason why similar storm cells behave similarly is that their life cycles follow characteristic patterns. For instance, the areal extent and intensity of storm cells are related to each other and the storm lifetime. In particular, the future behavior of cells depends on what stage they are in their life cycle. I think this aspect needs to be discussed more in the paper with literature references to put the research in a broader context. To this end, the authors could study the following papers: H. Kyznarova and P. Novak: CELLTRACK-Convective cell tracking algorithm and its use for deriving life cycle characteristics, Atmospheric Research, 93(1-3), 317- 327, 2009 C. Moseley, P. Berg and J. O. Haerter: Probing the precipitation life cycle by iterative rain cell tracking, JGR: Atmospheres, 118(24), 361-370, 2013 There is also a large amount of meteorological literature, where the life cycles of convective storms are discussed.

  > We are aware that convective storms have life cycles that have some characteristic pattern – that's why the idea of the nearest neighbourhood originated. Following your comments, we will discuss it more in the paper and include some more meteorological literature.

- A major limitation of the k-neighbors approach is that it cannot generalize beyond the training data. How would the proposed method perform for extreme events that have a very limited number or no training samples? Please add more discussion or analysis about this.

  > The events that have been selected as part of the database and are investigated are so to say extreme events; where the measured rainfall at different durations (1 hour and 1 day) exceed

the return period of 5 years. So, this is already assessing the suitability of k-NN for predicting extreme events from a given database of extreme events. But of course, if there is a very extreme event with high intensity (exceeding previous observations), the k-NN will fail to capture the high intensity, but may still associate to it, the largest intensity observed so far. The results depend highly on the nowcast time; nowcasting before the peak or after peak. To discuss this more, we will include an example of an extreme event, and the response of the k-NN in comparison to the Lagrangian persistence.

- In many places, the authors are describing results that are not shown anywhere, so the reader cannot verify the validity of the claims. An example of this can be seen at lines 362-374. Could you include some of the not shown results that are discussed in the text in an appendix or in supplementary material?

    Yes sure, the results are the average improvements obtained from the Figures. To make these results clearer we will include a table with the information.

**Specific comments**

• Line 101: What does "step 3" refer to? Storm extrapolation in Figure 1?

    Yes, the step 3 is referring to the storm extrapolation Figure 1-c. The explanation has been added to the text as following:
    "The application of the k-NN seems reasonable as an extension of the object-based radar nowcast. It can be used instead of the Lagrangian persistence in step 3 in Figure 1-c, for the extrapolation of rainfall storms into the future."

• Figure 2: ◦ What do the numbers represent in the x- and y-tick labels? I think it would be more informative to show the distance from the radar in kilometers.

    The x- and y- tick labels show the coordinates in meters of the study area in UTM-Zone 32N. We would prefer to leave the UTM coordinates, so the reader can have an idea about the location of the study area. However, following the comments of reviewer one we have included a map of Germany to explain where the study area is located, and we have updated the x-y tick labels with the distance from radar centre in meter. Please refer to Figure 2 in the attached pdf.

◦ Does DEM mean altitude obtained from a digital elevation model?

    Yes, DEM refers to the Digital Elevation model. This is now included in the Figure caption (see Figure 2 in the attached pdf).

• Lines 131-137: What is the justification for these threshold choices?

    Typically, larger thresholds are used for the identification of convective storms, but however may lead to false splitting of the storms, which then might be treated independently in the k-NN. Thus, we lowered the threshold to 20dBz and 25dBz (light rain) so we could have a better overview of similar storms. Nevertheless, this is only a starting point, and more work will be done to see how a change in threshold affects the application of the kNN.

• Lines 139-140: What is the "spatial rainfall intensity of a storm". Is it some kind of average or maximum value taken inside the storm object?

    The spatial rainfall intensity of a storm- refers here to the spatial distribution of the rainfall intensities within the boundaries of the storm object at a specific time step. We have changed it to "Here the spatial distribution of rainfall intensities inside the storm boundaries at a given time step (in 5min) of the storms' life…"

• Line 142: I'm curious how the ellipsoid is fitted. Please provide a more detailed explanation (though no need to include this in the paper).

The ellipsoid fitting is done by the existing algorithm (HyRaTrac) that we use as a base for nowcast. More information is provided in the full description (although in German) of the methodology in Kraemer 2008. The ellipsoid is fitting to taking into consideration the mass centroid of the storm and the areal moments of inertia. So first the mass centroid of the storm in found ($X_s$, $Y_s$) and the areal moments of inertia in respect to the x and y coordinates of the centroid ($J_x$ and $J_y$) and the centrifugal moments ($J_{xy}$) are computed as shown in Eqn. 1. The axes of the ellipsoid are referred to as $J_{min}$ and $J_{max}$ and calculated as shown in Eqn. 2.

$$J_y = \int (y - Y_s)^2 \, dA \; , \; J_x = \int (x - X_s)^2 \, dA \quad and \quad J_{xy} = J_{yx} = -\int (y - Y_s) \times (x - X_s) \, dA \quad (1)$$

$$J_{max} = \frac{J_x + J_y}{2} + \sqrt{(\frac{J_y - J_x}{2})^2 + J_{xy}^2} \; and \; J_{min} = \frac{J_x + J_y}{2} - \sqrt{\left(\frac{J_y - J_x}{2}\right)^2 + J_{xy}^2} \; and \; tan2\Phi = \frac{2J_{xy}}{J_y - J_x} \quad (2)$$

• Line 145: Please give a more detailed description about how the storm velocities are estimated?

The following lines were added in the text:

"These storms characteristics were obtained by an hindcast analysis run with the HyRaTrac algorithm. The local velocities in x and y direction are obtained by a cross-correlation optimization within the storm boundaries."

• Line 145 and Figure 3: The merges are mentioned in the text but not shown in the figure.

The figure refers to both the number of merges and splits and has been updated accordingly. The updated Figure is shown below. Please note that the maximum intensity has been updated as well. Unfortunately, before the standard deviation was showing instead of the maximum intensity. The outliers of Maximum Intensity, Ratio of minor and major axis and the Orientation angle are included in the plot.

• Figure 3: The very high velocities of 5-minute storms look suspicious to me. How can you even estimate the velocity of a storm if its duration is only 5 minutes (i.e. one-time step)?

The velocities of 5min storms are explained shortly in the text as:

"In case a storm is just recognized, then global displacement vectors based on cross-correlation of the entire radar image are assigned to them."

For the very high velocities please check the following point.

• Is there a reason for specifying the duration intervals in inclusive way? I would use separate intervals (i.e. 0-1h, 1-3h, 3-6h, 6-12h).

We have separated the duration intervals in the following groups: a) only 5 min, b) 5min – 1h, c) 1-3h, d) 3-6 h, e) 6-12h and f) longer than 12 h. The 5min we have included to illustrate the problem with the unmatched storms, and the longer than 12 hours to demonstrate very persistent storms (see updated Figure 4 in the attached pdf).

• I'm very surprised to see how the 5-minute storms have such a large area. I would expect all storms having area over 500 km2 to have lifetime longer than 5 minutes. Can you explain this?

I understand your concern, but please remember that here merged data are fed to the algorithm and not raw radar data. In the merged data, a merging between the stations and the

radar is performed. This is affecting the structure of the storms mainly when the radar data is missing, because the information is taken from interpolating rain gauge information with ordinary kriging – which is known to smoothen the distribution of the rainfall and leading to very large areas above a threshold.  This explains why these large areas are shortly lived (only 5 min) with low intensities and very high velocities.  The following explanation was given in the text:

"Another thing to keep in mind, is that merged radar are fed to the algorithm for storm recognition, and this affect the storm structures particularly when the radar data is missing. In such case, the ordinary kriging interpolation of rain gauges is given as input, which is well known to smoothen the spatial distribution of rainfall and hence resulting in short storms characterized by a very large area."

• Line 152: Please give numbers describing "high intensity" and "low areal coverage".

The following changes where done in the text:

"Here two types of convective storms are distinguished: local convective with very low coverage (on average lower than 1000 km$^2$) and low intensity (on average ~ 5 mm/h), and mesoscale convective which are responsible for floods (with spatial mean intensity up to 25 mm/h) and have a larger coverage (on average lower than 1000 km$^2$)."

• Lines 153-154: What is the evidence for making this conclusion? At least this cannot be seen from Figure 3.

I'm a bit confused to which conclusion you refer: that the meso-scale convective are the main cause for generating flooding? Yes, this cannot be seen directly from Figure 4 (see attached pdf), but it is known that the meso-scale convective storms are the main cause of the flooding and they are characterized by higher coverage, high intensity and durations lower than 6 hours. We will add citations to back this statement up. Please note that I have updated Figure 4 (the maximum Intensity) because before mistakenly I was showing the standard deviation of the intensity instead of the maximum value.

• Line 210 onwards: It is not obvious to me how the partial information correlation is better able to capture non-linear behaviour than the Pearson correlation coefficient. Can you add more discussion about this?

Yes sure, the Pearson correlation is computed based on the linear regression between two variables; so, the main assumption is that the two variables are linearly dependable. The partial information correlation is based on mutual information which describes the statistical dependency of the variables on each other. Here the conditional probability distributions of the values are considered, and no previous assumption in the linearity is considered. Moreover, as explained in the text the PIC is a stepwise procedure, considering the interaction between more than two predictors, which is not the case for the Pearson correlation (as it considers only the dependencies between two variables). We will update the manuscript by writing clearly how the PIC captures the non-linear behaviour and what is the main difference with the Pearson correlation.

• Equation (3): How is PI defined?

The following changes were done in the text to address this question:

$$PIC = \sqrt{(1 - \exp(-2PI)} \;\; with \;\; PI = \int f_{X,P|Z}(x,p|z) \log\left[\frac{f_{X|Z,P|Z}(x,p|z)}{f_{X|Z}(x|z)\,f_{P|Z}(p|z)}\right] dx\,dp\,dz \;\;\; , \qquad (1)$$

"where PIC is the Partial Information Correlation, PI is the Partial Information which represents the partial dependence of X on P conditioned to the presence of a predictor Z. The Partial Information itself is a modification of the Mutual Information in order to measure partial statistical dependency between the predictors (P) and the target variable (X), by adding predictors one at a time (Z) (step-wise procedure)."

• Equation (4): The notation is confusing. What does X(-j) mean? Maybe you should use subscripts for j and -j instead.

We have done the following changes in the text to match as well the annotations from Eqn (3):

$$\alpha_j = PIC_{X,Zj|Z(-j)} \frac{S_{X|Z(-j)}}{S_{Zj|Z(-j)}},$$

"where X is the target response, Zj is the added predictor from the step-wise procedure, Z(-j) previous predictor vector excluding the predictor Zj, SX|Z(-j) the scaled conditional standard deviations between target (x) and predictor vector Z(-j), SZj|Z(-j) the scaled conditional standard deviations between the additional predictor (Zj) and the first predictor vector Z(-j), and the αj the predictors weight."

As explained in the paragraph before the PIC is a step-wise procedure. One needs to have a starting predictor (a known important predictor), and then add the other predictors one at a time and see if they provide more information to the dependency or not. The Zj expresses the predictor being added in this procedure and Z(-j) is the previous predictor vector. When the procedure has just started the Z(-j) is the pre-identified predictor.

• Table 1: I would not use the word "predicted" for the target variables. I thought that they are obtained form observations, not predictions. For the velocities, I would use the word "estimated" since they are not directly observed but estimated by using some method.

Noted! We have changed from predicted to estimated.

• Table 1: You could attempt to eliminate the dependency of the standard deviation on the mean value by using the coefficient of variation instead (i.e. standard deviation divided by the mean).

Thanks for the suggestion, we will consider this in the future. Nevertheless, I am not sure if there will be a big difference, because the predictor set is already normalized according to the median the range between Q95% and Q5% (see Eqn. 2 in the manuscript).

• Lines 255-262: The actual procedure for generating the ensemble is not well described. Are the ensemble members somehow randomly assigned based on their probabilities?

Yes, the ensemble members are randomly assigned by on their rank probabilities. The following description is added in the text:

"Contrary, if a probabilistic nowcast is selected, 30-ensemble members are selected from the closest 30 storms where each member is assigned a probability according to the rank of the respective neighbour storm with the "to-be-nowcasted" storm:

$$Pr_i = \frac{(1/Rank_i)}{\sum_{i=1}^{k}(1/R_{anki})}, \tag{2}$$

where k is the selected number of neighbours and Rank and Pr are respectively the rank and the probability weights of the i[th] neighbour/ensemble member. An ensemble member is then chosen randomly according to their probability weights. The probability weights calculated here are as well used for computation of the single nowcast in Equation (6)."

• Line 261: What is the justification for choosing the value 0.5? Is the model sensitive to this value?

> The value 0.5 represents the 95% quantile of the calculated distances of the 4th neighbour for all storms/events. Actually, the model is not very sensitive to this value, but to avoid confusion and to make a fair comparison with Lagrangian persistence (not to exclude the time steps where the neighbours have a higher distance than 0.5) we have excluded this limitation and show the results for all neighbours. However, the idea in the future is that when the kNN method is not able to recognize a similar neighbour (with distance below this threshold), the Lagrangian persistence should be used instead.

• Equations (6) and (7): To me it appears that the same symbol R is used for two different purposes: response and rank. Could you use different symbols?

> Noted! In Equation (7) we refer to it as *Rank* instead of *R*.

• Equation (8): Should the summation terms be taken their absolute values? To me summation over the differences does not make much sense if it's used as the objective function.

> The absolute values were not considered in order balance the over and under -estimation. However, we have changed the error calculation for the absolute values and updated the results of the deterministic nowcast. We have done so because we have calculated as well the Continuous Rank Probability Score for the ensemble nowcast, and thus we can provide a direct comparison between the deterministic and probabilistic approaches. The following change has been done in text:

$$Error_{target} = \sum_{i=1}^{N}(|Pred_{i,+LT}| - |Obs_{i,+LT}|)/N, \tag{8}$$

> Please note that because we have changed the objective functions, the results of the training have been updated as well. Please refer to Figure 7 and 8 in the attached pdf.

• Line 304: Please define precisely the concept of Lagrangian persistence in this context. It can be defined in many different ways depending on the type of the nowcast (i.e. grid- or object-based). Here it means that all storm attributes (not only the shape) are taken from the most recent values and they remain constant for all lead times. Right?

> Yes, you are right. The following changes have been done in the text:

> "where the Error$_{new}$ is the error manifested by the k-NN, the Error$_{ref}$ the error manifested by the Lagrangian persistence and the Error$_{impr}$ the improvement in reducing the error per each lead time. Here the Lagrangian persistence refers to as persistence of the storm characteristics (Area, Intensity, Velocity in X and Y Direction) as last observed and constant for all lead times."

• Figure 6: ◦ In Table 1, the target variables A, I, V_x and V_y have the subscript denoting lead time. However, these are is omitted in Figure 6. Thus, it is not clear to me what lead times do the correlations shown in Figure 6 represent. The text is just saying that the values are averaged from three different lead times. ◦ The correlations depend on the lead time. Would it make sense to show the correlations separately for each of the chosen lead times instead of averaging over different lead times?

> At the end of section 3.1.3 the averaging of the three lead times was explained:

> "Here in this study, these two importance analyses are used to determine the most important predictors and their respective weights in the k-NN similarity calculation. For each target variable the most important predictor identified from Pearson Correlation, is given to the PIC metric as the first predictor. The analysis is complex due to the presence of several predictors, 38 states of future behaviour for each target variable (for each 5min between +5min to +180

min lead times), and different times of nowcast; the weights were calculated first for three lead times +15min, +60min and +180 min, and for three storm groups separated according to their duration <60min, 60min-180min, and > 3 hours. Here the averages weights over these groups and lead times are calculated and used as a reference for each importance analysis. The k-NN errors with these average weights are compared in Section 4.1."

We have added the correlations for each of the selected lead times in the Appendixes 8.1 and 8.2 (see tables in the pdf attached). The tables for both target-based and storm-based approaches and for the two important analysis methods are given here. The reason why we took the average from these three lead times is because we wanted to have only one set of weights independent of the lead times and storm types as a starting point for the nearest neighbour approach.

• Lines 365-366: This is difficult to follow. It is confusing that the authors mention both mean and median but are not showing the latter anywhere. In addition, you should clearly state in the caption of Figure 7 that it shows the mean.

In the first manuscript draft, the mean is used only for the training of the k-NN, because it is a typically used target optimization statistics. The median is used when validating the k-NN, because we were interested to see the error if the k-NN on the majority of the cases. However, in the new updated manuscript we are showing the median for all the results (both training) and validation (see Figure 7 and 8 in the attached pdf).

• Lines 391 and 397: The authors are using a confusing term "event-based" that is not defined previously. Does this mean storm-based?

Yes we meant "storm-based" and we have changed it accordingly throughout the text.

• Figure 7: The interpretation of the Total Lifetime figure on the right was not immediately clear to me. In particular, the connection of the black and red lines labelled as VS1 and VS2 to the boxes shown in the figure could be clearer.

We have updated the legend in all the Figures to make it clearer for the reader.

• Figure 9: ◦ What is the "Timestep of nowcast"? This should be clearly explained in the caption text. Now I found it from line 403 only after reading through the main text. ◦ As in Figure 7, it was not immediately obvious to me how to interpret the right pane.

Following the confusion reported by another reviewer and we have the term to Nowcast Time throughout the text and in the figures. The nowcast time is also explained on a new figure that we have included to explain the main terms. Please refer to Figure 3 in the attached pdf.

• Figure 10: ◦ Again, clarify the meaning of the "Timestep of nowcast". Please explain it in the figure caption.

The captions of all figures were updated accordingly. Please refer to the attached pdf.

◦ To me it is striking that in the worst case the 4-NN nowcast can perform more than 100% worse than the Lagrangian persistence. This is lacking discussion in the text that focuses mainly on the improvement from the Lagrangian persistence. ◦ My advice here is to explore, or at least mention the possibility of blending the Lagrangian persistence and the 4-NN nowcast by using weights that depend on the lead time. This would combine the strengths of both approaches.

The 4-NN nowcast can perform more than 100% worse than the Lagrangian Persistence mainly for the area as target variable, for short lead times (LT<30min so within the predictability limit of the Lagrangian Persistence) and for storms living more than 0.5 hour. For the storms living between 30min and 3 hours, for nowcast times 5min the area errors of the KNN are more than 100% bigger than the Lagrangian Persistence error for lead times up to 15min, while for the storms living longer than 3 hours they are higher than the persistence for lead up 1 hour. This is expected at a certain extent, that for very low lead times, the autocorrelation governs and thus Lagrangian Persistence has very low errors. Since the errors are low and the improvement is obtained by diving with low error, the lower the error of Lagrangian the higher will be the deterioration on %. So, in my opinion, important is here than up to this lead time the given kNN cannot surpass the Lagrangian persistence, and persistence is behaving much better.

So to summarize, you are right one should merge the two models. As shown in Germann et al. 2002, for short lead times Lagrangian should be used (up to 20-30min) and after that a more complex model should be applied. The same is true here: the deterministic kNN can be employed for lead times past 30min. These results were of the deterministic nowcast, and we have included a new Figure that compares the CRPS of the ensemble with the MAE of the Lagrangian to see for what lead times the probabilistic nowcast are better (see Figure 13 in the attached pdf). The result of the ensemble follows same patterns of the deterministic 4-NN, but please pay attention that the deteriorations are not lower than 100%. This suggests that the probabilistic approach, i.e. single neighbours may be more useful than averaging through most similar neighbours (the case of the 4NNs).

• Figure 12: It could be more informative to compute instead the fraction of verifying observations within (or outside) the ensemble and average this statistic over the events. The "% of timesteps" statistic gives no information about this fraction.

The % of time steps is showing the fraction of the verifying observations within the ensemble range, but averaged for each nowcast time and storm duration groups. Do you mean to average this value for each event first and then for nowcast time and storm duration group? If so, we will update this Figure by showing the event average fraction of verifying observations that fall within the range of the ensemble members, shown for each lead time, nowcast time and storm duration group.

• Sections 4.2 and 4.3: The authors use the terms lead time and timestep interchangeably. When reading the text, their correspondence is not immediately clear to the reader (until the reader goes to Section 2 to recall that the time step is 5 minutes). Could you use only one of them?

Yes of course, we have updated the manuscript and tried to keep only the term lead time.

• Section 4.4: I have some doubts whether the "best ensemble member" or "% of ensemble members better than Lagrangian persistence" verification approach is meaningful. These are not standard verification metrics. In practice, one does not know a priori which ensemble members to choose to obtain the best forecast skill. There are more elaborate ways of showing the advantages of ensemble-based predictions over deterministic ones. For instance, you can compute the continuous ranked probability score (CRPS), which is a generalization of mean absolute error (MAE) for deterministic nowcasts. The CRPS of the ensemble nowcast should be lower than the MAE, which indicates the added value of the ensemble nowcast.

We have now computed only the Continuous rank probability score (CRPS) for the ensemble predictions, and the MAE for the deterministic nowcast. The Figures 11 and 12 in the attached pdf show the ensemble and deterministic results for both k-NN approach; the storm based and

target based. These Figures are included in the new updated version of the manuscript. As you see for both storm-based and target-based approach, the errors of the ensemble nowcast are lower than the deterministic nowcast for all nowcast times, storms groups and target variable – hence the added value of the ensemble nowcast. This suggests that the nearest neighbour approach should be better implemented in an ensemble approach and not by averaging the closest neighbours. As seen in the training of the kNN. a proper converging of the K-number was not possible (as the number of k to average is depending on the nowcast and lead time). This problem of training can be avoided on the ensemble nowcast and the result show that the errors are lower than in the case of the deterministic approach.

• Line 511 onwards: I guess that the authors mean individual ensemble members, not whole ensembles?

Yes, we were referring to individual ensemble members.

• Lines 538-539: Can you give some numbers to describe what are the fine spatial and temporal scales?

The following changes have been done in the text:

"Accurate predictions of rainfall storms at fine temporal and spatial scales (5min, 1km2) based on radar data are quite challenging to achieve."

• Line 546: Where does the number 5200 come from? It is mentioned for the first time in the conclusions. Perhaps it should be mentioned in Section 2 as well.

The following description was mentioned in Section 3.1.1:

"These storms characteristics were obtained by an hindcast analysis run of all 110 events with the HyRaTrac algorithm which resulted in around 5200 storms."

**Technical corrections**

• I'm not sure if it's proper to use the word "object-oriented". It refers to programming terminology. Could you use object-based instead?

I agree with your concern and it can also be changed to object-based. However, in some literature that we cite (Hand 1996; Rossi et al., 2015), it is already mentioned as "object-oriented", while in others as object-based (Zahrei et al., 2013). For this reason, we mention as "object-oriented" the first time in the introduction, but we clarify that in order to avoid confusion with the programming term we refer as the object-based nowcast.

Hand, W. H. (1996). An object-oriented technique for nowcasting heavy showers and

thunderstorms. *Meteorological Applications*, *3*, 31–41.

https://doi.org/10.1002/met.5060030104

Rossi, P. J., Chandrasekar, V., Hasu, V., & Moisseev, D. (2015). Kalman filtering-based

probabilistic nowcasting of object-oriented tracked convective storms. *Journal of Atmospheric*

*and Oceanic Technology*, *32*(3), 461–477. https://doi.org/10.1175/JTECH-D-14-00184.1

Zahraei, A., Hsu, K. lin, Sorooshian, S., Gourley, J. J., Hong, Y., & Behrangi, A. (2013). Short-term

quantitative precipitation forecasting using an object-based approach. *Journal of Hydrology*,

*483*, 1–15. https://doi.org/10.1016/j.jhydrol.2012.09.052

• Figure 2: Should the legend read "Lower Saxony border"?

Noted and changed!

• Line 211: important ← importance?

Noted and changed!

• Lines 226-227: "the $\alpha_j$ the predictors weight" ← "$\alpha_j$ denote the predictors weight"?

Noted and changed!

• Line 256: 30-ensembles ← 30-member ensembles?

Noted and changed!

• Line 447: persistence ← persistent?

Noted and changed!

• Line 544: behaviours ← behaviour

Noted and changed!

• Line 578: An increment in the sample size ← Increase in the sample size?

Noted and changed!

• Figure 6: Should this be titled as a figure or a table?

We changed it to a table title.

[revised manuscript text omitted]

**Appendix 8.1** *Obtained Pearson Correlation predictors weight for each target variable, lead time and storm groups. The last row at each target variable (average values) are the predictors weights shown in Table 3*

**Area**

| Durations | Lead Time | Cell | Life | A | avePI | medPI | maxPI | sdPI1 | sdPI2 | Vg | Vx | Vy | Jx | Jy | Jr | Φ | A | avePI | medPI | maxPI | sdPI1 | sdPI2 | Vg | Vx | Vy | Jx | Jy | Jr | Φ |
|---|---|---|---|---|---|---|---|---|---|---|---|---|---|---|---|---|---|---|---|---|---|---|---|---|---|---|---|---|---|
| <1hr | 15min | 0.19 | 0.22 | 0.81 | 0.06 | 0.04 | 0.05 | 0.05 | 0.59 | 0.06 | 0.02 | 0.01 | 0.58 | 0.61 | 0.01 | 0.00 | 0.79 | 0.06 | 0.04 | 0.05 | 0.05 | 0.61 | 0.06 | 0.02 | 0.02 | 0.60 | 0.62 | 0.01 | 0.00 |
| | 60min | 0.09 | 0.27 | 0.90 | 0.19 | 0.07 | 0.07 | 0.20 | 0.64 | 0.05 | 0.05 | 0.03 | 0.65 | 0.68 | 0.04 | 0.02 | 0.88 | 0.21 | 0.09 | 0.08 | 0.22 | 0.67 | 0.08 | 0.08 | 0.07 | 0.68 | 0.70 | 0.08 | 0.02 |
| | 180min | 0.12 | 0.19 | 0.94 | 0.18 | 0.08 | 0.23 | 0.29 | 0.68 | 0.05 | 0.04 | 0.05 | 0.72 | 0.70 | 0.28 | 0.00 | 0.92 | 0.21 | 0.05 | 0.25 | 0.31 | 0.69 | 0.05 | 0.05 | 0.09 | 0.73 | 0.69 | 0.38 | 0.02 |
| <3hr | 15min | 0.12 | 0.19 | 0.61 | 0.04 | 0.03 | 0.03 | 0.04 | 0.45 | 0.00 | 0.01 | 0.00 | 0.43 | 0.49 | 0.01 | 0.01 | 0.60 | 0.05 | 0.03 | 0.02 | 0.04 | 0.46 | 0.00 | 0.02 | 0.02 | 0.45 | 0.50 | 0.01 | 0.01 |
| | 60min | 0.04 | 0.25 | 0.72 | 0.13 | 0.05 | 0.01 | 0.13 | 0.48 | 0.01 | 0.03 | 0.03 | 0.55 | 0.55 | 0.03 | 0.01 | 0.69 | 0.15 | 0.07 | 0.03 | 0.15 | 0.51 | 0.02 | 0.05 | 0.05 | 0.56 | 0.55 | 0.07 | 0.02 |
| | 180min | 0.09 | 0.13 | 0.80 | 0.16 | 0.06 | 0.20 | 0.25 | 0.58 | 0.10 | 0.01 | 0.05 | 0.63 | 0.57 | 0.24 | 0.00 | 0.77 | 0.20 | 0.02 | 0.23 | 0.28 | 0.57 | 0.11 | 0.00 | 0.09 | 0.61 | 0.55 | 0.32 | 0.02 |
| >3hr | 15min | 0.05 | 0.13 | 0.32 | 0.04 | 0.03 | 0.00 | 0.03 | 0.22 | 0.00 | 0.01 | 0.01 | 0.24 | 0.25 | 0.01 | 0.01 | 0.31 | 0.04 | 0.03 | 0.01 | 0.04 | 0.21 | 0.00 | 0.02 | 0.03 | 0.25 | 0.25 | 0.01 | 0.02 |
| | 60min | 0.03 | 0.14 | 0.42 | 0.13 | 0.08 | 0.09 | 0.14 | 0.27 | 0.07 | 0.02 | 0.02 | 0.32 | 0.26 | 0.02 | 0.02 | 0.39 | 0.15 | 0.10 | 0.10 | 0.16 | 0.27 | 0.07 | 0.02 | 0.05 | 0.32 | 0.25 | 0.05 | 0.02 |
| | 180min | 0.06 | 0.07 | 0.53 | 0.17 | 0.03 | 0.19 | 0.22 | 0.39 | 0.16 | 0.05 | 0.07 | 0.41 | 0.34 | 0.16 | 0.06 | 0.50 | 0.20 | 0.06 | 0.22 | 0.25 | 0.38 | 0.18 | 0.07 | 0.11 | 0.40 | 0.32 | 0.20 | 0.08 |
| | Average | 0.09 | 0.18 | 0.67 | 0.12 | 0.05 | 0.10 | 0.15 | 0.48 | 0.05 | 0.03 | 0.03 | 0.50 | 0.49 | 0.09 | 0.02 | 0.65 | 0.14 | 0.05 | 0.11 | 0.17 | 0.48 | 0.07 | 0.04 | 0.06 | 0.51 | 0.49 | 0.12 | 0.02 |

**Intensity**

| Durations | Lead Time | No.Cells | Life.TS | Area | meanPI | medianPI | maxPI | sdPI1 | sdPI2 | GVel | VelX | VelY | Jx | Jy | J.ratio | Phi | Area | meanPI | medPI | maxPI | sdPI1 | sdPI2 | Velg | Velx | Vely | Jx | Jy | Jr | phi |
|---|---|---|---|---|---|---|---|---|---|---|---|---|---|---|---|---|---|---|---|---|---|---|---|---|---|---|---|---|---|
| <1hr | 15min | 0.02 | 0.05 | 0.00 | 0.55 | 0.41 | 0.52 | 0.54 | 0.11 | 0.06 | 0.03 | 0.00 | 0.02 | 0.02 | 0.00 | 0.01 | 0.00 | 0.52 | 0.40 | 0.50 | 0.52 | 0.11 | 0.06 | 0.03 | 0.01 | 0.02 | 0.02 | 0.00 | 0.01 |
| | 60min | 0.04 | 0.01 | 0.12 | 0.70 | 0.53 | 0.61 | 0.69 | 0.06 | 0.04 | 0.01 | 0.07 | 0.00 | 0.01 | 0.02 | 0.02 | 0.14 | 0.64 | 0.49 | 0.59 | 0.65 | 0.03 | 0.05 | 0.01 | 0.09 | 0.02 | 0.02 | 0.02 | 0.01 |
| | 180min | 0.03 | 0.13 | 0.11 | 0.81 | 0.67 | 0.68 | 0.77 | 0.13 | 0.09 | 0.09 | 0.03 | 0.14 | 0.15 | 0.05 | 0.06 | 0.11 | 0.76 | 0.62 | 0.68 | 0.75 | 0.13 | 0.13 | 0.11 | 0.04 | 0.14 | 0.15 | 0.06 | 0.10 |
| <3hr | 15min | 0.02 | 0.10 | 0.11 | 0.15 | 0.08 | 0.22 | 0.17 | 0.14 | 0.04 | 0.02 | 0.01 | 0.08 | 0.10 | 0.01 | 0.00 | 0.10 | 0.14 | 0.07 | 0.20 | 0.16 | 0.13 | 0.04 | 0.02 | 0.01 | 0.08 | 0.09 | 0.02 | 0.01 |
| | 60min | 0.01 | 0.06 | 0.01 | 0.31 | 0.18 | 0.45 | 0.37 | 0.10 | 0.02 | 0.02 | 0.03 | 0.07 | 0.07 | 0.02 | 0.05 | 0.01 | 0.28 | 0.16 | 0.43 | 0.34 | 0.09 | 0.03 | 0.02 | 0.06 | 0.06 | 0.05 | 0.04 | 0.05 |
| | 180min | 0.01 | 0.06 | 0.10 | 0.43 | 0.40 | 0.50 | 0.47 | 0.20 | 0.08 | 0.06 | 0.01 | 0.25 | 0.22 | 0.09 | 0.09 | 0.08 | 0.42 | 0.37 | 0.47 | 0.44 | 0.19 | 0.10 | 0.08 | 0.01 | 0.24 | 0.21 | 0.12 | 0.10 |
| >3hr | 15min | 0.03 | 0.11 | 0.12 | 0.02 | 0.00 | 0.08 | 0.03 | 0.11 | 0.02 | 0.01 | 0.01 | 0.08 | 0.10 | 0.01 | 0.01 | 0.11 | 0.01 | 0.01 | 0.06 | 0.02 | 0.10 | 0.02 | 0.01 | 0.01 | 0.08 | 0.10 | 0.01 | 0.02 |
| | 60min | 0.02 | 0.06 | 0.08 | 0.07 | 0.05 | 0.17 | 0.11 | 0.09 | 0.02 | 0.00 | 0.03 | 0.06 | 0.04 | 0.02 | 0.04 | 0.05 | 0.06 | 0.04 | 0.16 | 0.10 | 0.08 | 0.02 | 0.00 | 0.05 | 0.05 | 0.03 | 0.04 | 0.04 |
| | 180min | 0.01 | 0.05 | 0.36 | 0.10 | 0.18 | 0.10 | 0.06 | 0.31 | 0.03 | 0.02 | 0.10 | 0.38 | 0.35 | 0.11 | 0.05 | 0.34 | 0.07 | 0.15 | 0.05 | 0.02 | 0.30 | 0.05 | 0.05 | 0.16 | 0.36 | 0.33 | 0.15 | 0.05 |
| | Average | 0.02 | 0.07 | 0.11 | 0.35 | 0.28 | 0.37 | 0.36 | 0.14 | 0.04 | 0.03 | 0.03 | 0.12 | 0.12 | 0.03 | 0.04 | 0.10 | 0.32 | 0.26 | 0.35 | 0.33 | 0.13 | 0.05 | 0.04 | 0.05 | 0.12 | 0.11 | 0.05 | 0.04 |

**Velocity X**

| Durations | Lead Time | No.Cells | Life.TS | Area | meanPI | medianPI | maxPI | sdPI1 | sdPI2 | GVel | VelX | VelY | Jx | Jy | J.ratio | Phi | Area | meanPI | medPI | maxPI | sdPI1 | sdPI2 | Velg | Velx | Vely | Jx | Jy | Jr | phi |
|---|---|---|---|---|---|---|---|---|---|---|---|---|---|---|---|---|---|---|---|---|---|---|---|---|---|---|---|---|---|
| <1hr | 15min | 0.04 | 0.02 | 0.09 | 0.01 | 0.01 | 0.01 | 0.00 | 0.06 | 0.14 | 0.17 | 0.02 | 0.06 | 0.05 | 0.01 | 0.02 | 0.08 | 0.01 | 0.01 | 0.01 | 0.00 | 0.08 | 0.13 | 0.18 | 0.02 | 0.14 | 0.07 | 0.02 | 0.02 |
| | 60min | 0.03 | 0.03 | 0.12 | 0.03 | 0.04 | 0.02 | 0.02 | 0.06 | 0.31 | 0.37 | 0.06 | 0.10 | 0.03 | 0.01 | 0.03 | 0.11 | 0.04 | 0.04 | 0.02 | 0.03 | 0.04 | 0.33 | 0.52 | 0.09 | 0.15 | 0.04 | 0.00 | 0.03 |
| | 180min | 0.04 | 0.01 | 0.06 | 0.05 | 0.06 | 0.04 | 0.05 | 0.00 | 0.27 | 0.32 | 0.05 | 0.12 | 0.06 | 0.01 | 0.06 | 0.07 | 0.04 | 0.05 | 0.03 | 0.04 | 0.00 | 0.35 | 0.42 | 0.05 | 0.16 | 0.07 | 0.01 | 0.05 |
| <3hr | 15min | 0.03 | 0.06 | 0.10 | 0.02 | 0.01 | 0.01 | 0.01 | 0.06 | 0.03 | 0.07 | 0.01 | 0.05 | 0.04 | 0.01 | 0.02 | 0.08 | 0.02 | 0.02 | 0.00 | 0.02 | 0.05 | 0.03 | 0.06 | 0.01 | 0.15 | 0.03 | 0.02 | 0.03 |
| | 60min | 0.06 | 0.06 | 0.15 | 0.03 | 0.02 | 0.03 | 0.02 | 0.06 | 0.20 | 0.30 | 0.06 | 0.11 | 0.05 | 0.01 | 0.03 | 0.14 | 0.05 | 0.04 | 0.03 | 0.03 | 0.05 | 0.25 | 0.42 | 0.07 | 0.16 | 0.04 | 0.01 | 0.04 |
| | 180min | 0.04 | 0.01 | 0.10 | 0.03 | 0.04 | 0.02 | 0.02 | 0.02 | 0.27 | 0.26 | 0.04 | 0.13 | 0.07 | 0.00 | 0.06 | 0.10 | 0.02 | 0.04 | 0.02 | 0.02 | 0.02 | 0.29 | 0.38 | 0.05 | 0.18 | 0.07 | 0.00 | 0.05 |
| >3hr | 15min | 0.04 | 0.06 | 0.10 | 0.02 | 0.02 | 0.01 | 0.01 | 0.04 | 0.02 | 0.05 | 0.02 | 0.04 | 0.01 | 0.01 | 0.02 | 0.09 | 0.02 | 0.02 | 0.00 | 0.02 | 0.04 | 0.02 | 0.05 | 0.01 | 0.15 | 0.01 | 0.01 | 0.02 |
| | 60min | 0.04 | 0.04 | 0.05 | 0.04 | 0.04 | 0.03 | 0.04 | 0.04 | 0.07 | 0.16 | 0.05 | 0.00 | 0.04 | 0.01 | 0.02 | 0.04 | 0.04 | 0.04 | 0.03 | 0.04 | 0.05 | 0.08 | 0.23 | 0.07 | 0.15 | 0.05 | 0.02 | 0.02 |
| | 180min | 0.03 | 0.02 | 0.10 | 0.00 | 0.03 | 0.02 | 0.02 | 0.03 | 0.15 | 0.17 | 0.03 | 0.11 | 0.04 | 0.01 | 0.03 | 0.10 | 0.01 | 0.02 | 0.02 | 0.02 | 0.03 | 0.16 | 0.24 | 0.03 | 0.15 | 0.05 | 0.01 | 0.04 |
| | Average | 0.04 | 0.03 | 0.10 | 0.03 | 0.03 | 0.02 | 0.02 | 0.04 | 0.16 | 0.21 | 0.04 | 0.08 | 0.04 | 0.01 | 0.03 | 0.09 | 0.03 | 0.03 | 0.02 | 0.02 | 0.04 | 0.18 | 0.28 | 0.04 | 0.15 | 0.05 | 0.01 | 0.03 |

**Velocity Y**

| Durations | Lead Time | No.Cells | Life.TS | Area | meanPI | medianPI | maxPI | sdPI1 | sdPI2 | GVel | VelX | VelY | Jx | Jy | J.ratio | Phi | Area | meanPI | medPI | maxPI | sdPI1 | sdPI2 | Velg | Velx | Vely | Jx | Jy | Jr | phi |
|---|---|---|---|---|---|---|---|---|---|---|---|---|---|---|---|---|---|---|---|---|---|---|---|---|---|---|---|---|---|
| <1hr | 15min | 0.04 | 0.04 | 0.04 | 0.02 | 0.00 | 0.05 | 0.03 | 0.06 | 0.03 | 0.02 | 0.15 | 0.03 | 0.03 | 0.01 | 0.00 | 0.04 | 0.02 | 0.00 | 0.04 | 0.03 | 0.07 | 0.04 | 0.03 | 0.17 | 0.04 | 0.03 | 0.01 | 0.00 |
| | 60min | 0.00 | 0.04 | 0.02 | 0.08 | 0.07 | 0.09 | 0.08 | 0.00 | 0.03 | 0.05 | 0.22 | 0.00 | 0.00 | 0.01 | 0.02 | 0.03 | 0.08 | 0.07 | 0.09 | 0.08 | 0.00 | 0.01 | 0.06 | 0.33 | 0.01 | 0.00 | 0.02 | 0.02 |
| | 180min | 0.03 | 0.08 | 0.05 | 0.02 | 0.01 | 0.03 | 0.03 | 0.05 | 0.00 | 0.04 | 0.27 | 0.07 | 0.01 | 0.01 | 0.02 | 0.06 | 0.02 | 0.01 | 0.03 | 0.03 | 0.06 | 0.01 | 0.05 | 0.41 | 0.08 | 0.02 | 0.01 | 0.02 |
| <3hr | 15min | 0.01 | 0.06 | 0.06 | 0.03 | 0.02 | 0.07 | 0.04 | 0.07 | 0.01 | 0.01 | 0.05 | 0.03 | 0.05 | 0.00 | 0.01 | 0.05 | 0.03 | 0.01 | 0.06 | 0.04 | 0.06 | 0.02 | 0.00 | 0.04 | 0.02 | 0.04 | 0.00 | 0.00 |
| | 60min | 0.01 | 0.06 | 0.02 | 0.04 | 0.03 | 0.10 | 0.06 | 0.03 | 0.01 | 0.06 | 0.18 | 0.02 | 0.05 | 0.01 | 0.01 | 0.00 | 0.04 | 0.03 | 0.11 | 0.06 | 0.03 | 0.00 | 0.07 | 0.26 | 0.01 | 0.04 | 0.01 | 0.01 |
| | 180min | 0.00 | 0.06 | 0.03 | 0.00 | 0.00 | 0.00 | 0.00 | 0.06 | 0.01 | 0.03 | 0.22 | 0.07 | 0.04 | 0.01 | 0.01 | 0.04 | 0.01 | 0.01 | 0.00 | 0.00 | 0.07 | 0.00 | 0.04 | 0.33 | 0.07 | 0.05 | 0.01 | 0.01 |
| >3hr | 15min | 0.01 | 0.07 | 0.03 | 0.00 | 0.01 | 0.03 | 0.01 | 0.03 | 0.01 | 0.01 | 0.04 | 0.00 | 0.01 | 0.01 | 0.02 | 0.02 | 0.00 | 0.01 | 0.02 | 0.00 | 0.02 | 0.01 | 0.01 | 0.03 | 0.00 | 0.00 | 0.01 | 0.03 |
| | 60min | 0.03 | 0.02 | 0.02 | 0.01 | 0.01 | 0.03 | 0.01 | 0.07 | 0.00 | 0.04 | 0.09 | 0.09 | 0.08 | 0.01 | 0.00 | 0.04 | 0.01 | 0.01 | 0.03 | 0.00 | 0.09 | 0.01 | 0.07 | 0.16 | 0.10 | 0.10 | 0.01 | 0.00 |
| | 180min | 0.00 | 0.01 | 0.01 | 0.05 | 0.04 | 0.04 | 0.04 | 0.04 | 0.01 | 0.02 | 0.14 | 0.09 | 0.08 | 0.01 | 0.01 | 0.02 | 0.05 | 0.05 | 0.04 | 0.04 | 0.05 | 0.02 | 0.01 | 0.22 | 0.10 | 0.08 | 0.00 | 0.00 |
| | Average | 0.01 | 0.05 | 0.03 | 0.03 | 0.02 | 0.05 | 0.03 | 0.05 | 0.01 | 0.03 | 0.15 | 0.04 | 0.04 | 0.01 | 0.01 | 0.03 | 0.03 | 0.02 | 0.05 | 0.03 | 0.05 | 0.01 | 0.04 | 0.22 | 0.05 | 0.04 | 0.01 | 0.01 |

**Duration**

| Durations | Lead Time | No.Cells | Life.TS | Area | meanPI | medianPI | maxPI | sdPI1 | sdPI2 | GVel | VelX | VelY | Jx | Jy | J.ratio | Phi | Area | meanPi | medPI | maxPI | sdPI1 | sdPI2 | Velg | Velx | Vely | Jx | Jy | Jr | phi |
|---|---|---|---|---|---|---|---|---|---|---|---|---|---|---|---|---|---|---|---|---|---|---|---|---|---|---|---|---|---|
| Dur <1hr | | 0.06 | 0.16 | 0.31 | 0.02 | 0.02 | 0.04 | 0.00 | 0.22 | 0.03 | 0.03 | 0.00 | 0.19 | 0.24 | 0.00 | 0.03 | 0.30 | 0.02 | 0.03 | 0.03 | 0.01 | 0.21 | 0.03 | 0.03 | 0.01 | 0.19 | 0.25 | 0.01 | 0.04 |
| Dur <3hr | | 0.00 | 0.16 | 0.35 | 0.10 | 0.05 | 0.04 | 0.10 | 0.22 | 0.08 | 0.02 | 0.04 | 0.23 | 0.21 | 0.02 | 0.11 | 0.32 | 0.12 | 0.07 | 0.05 | 0.11 | 0.22 | 0.08 | 0.03 | 0.07 | 0.23 | 0.21 | 0.05 | 0.15 |
| Dur >3hr | | 0.07 | 0.02 | 0.43 | 0.20 | 0.11 | 0.18 | 0.21 | 0.21 | 0.15 | 0.04 | 0.04 | 0.25 | 0.16 | 0.14 | 0.01 | 0.40 | 0.22 | 0.14 | 0.20 | 0.23 | 0.20 | 0.18 | 0.07 | 0.07 | 0.23 | 0.14 | 0.18 | 0.01 |
| | Average | 0.04 | 0.11 | 0.36 | 0.11 | 0.06 | 0.09 | 0.10 | 0.22 | 0.09 | 0.03 | 0.03 | 0.22 | 0.20 | 0.05 | 0.05 | 0.34 | 0.12 | 0.08 | 0.09 | 0.12 | 0.21 | 0.10 | 0.04 | 0.05 | 0.22 | 0.20 | 0.08 | 0.07 |

Appendix 8.2 Obtained PIC predictors weight for each target variable, lead time and storm groups. The last row at each target variable (average values) are the predictors weights shown in Table 3.

**Area (target variable)**

| Durations | Lead Time | Cell | Life | A | avePI | medPI | maxPI | sdPI1 | sdPI2 | Vg | Vx | Vy | Jx | Jy | Jr | Φ | A | avePI | medPI | maxPI | sdPI1 | sdPI2 | Vg | Vx | Vy | Jx | Jy | Jr | phi |
|---|---|---|---|---|---|---|---|---|---|---|---|---|---|---|---|---|---|---|---|---|---|---|---|---|---|---|---|---|---|
| | | | | | | | | | | | | | | | | *(Present Predictors)* | | | | | | | | | | | | *(Average Past 30min Predictors)* | |
| <1hr | 15min | 0.00 | 0.00 | 0.00 | 0.00 | 0.00 | 0.00 | 0.00 | 0.00 | 0.00 | 0.00 | 0.00 | 0.00 | 0.00 | 0.00 | 0.00 | 0.00 | 0.00 | 0.00 | 0.00 | 0.00 | 0.00 | 0.00 | 0.00 | 0.00 | 0.00 | 0.00 | 1.00 | 0.00 |
| <1hr | 60min | 0.00 | 0.00 | 0.00 | 0.00 | 0.00 | 0.00 | 0.00 | 0.00 | 0.00 | 0.00 | 0.00 | 0.00 | 0.00 | 0.00 | 0.00 | 0.00 | 0.00 | 0.00 | 0.00 | 0.00 | 0.00 | 0.00 | 0.00 | 0.00 | 0.00 | 0.00 | 1.00 | 0.00 |
| <1hr | 180min | 0.00 | 0.00 | 0.00 | 0.00 | 0.00 | 0.00 | 0.00 | 0.00 | 0.00 | 0.00 | 0.00 | 0.00 | 0.00 | 0.00 | 0.00 | 0.00 | 0.00 | 0.00 | 0.00 | 0.00 | 0.00 | 0.00 | 0.00 | 0.00 | 0.00 | 0.00 | 1.00 | 0.00 |
| <3hr | 15min | 0.00 | 0.00 | 0.00 | 0.00 | 0.00 | 0.00 | 0.00 | 0.00 | 0.00 | 0.00 | 0.00 | 0.00 | 0.00 | 0.00 | 0.00 | 0.00 | 0.00 | 0.00 | 0.00 | 0.00 | 0.00 | 1.00 | 0.00 | 0.00 | 0.00 | 0.00 | 0.00 | 0.00 |
| <3hr | 60min | 0.00 | 0.00 | 0.00 | 0.00 | 0.00 | 0.00 | 0.00 | 0.00 | 0.00 | 0.00 | 0.00 | 0.00 | 0.00 | 0.00 | 0.00 | 0.00 | 0.00 | 0.00 | 0.00 | 0.00 | 0.00 | 1.00 | 0.00 | 0.00 | 0.00 | 0.00 | 0.00 | 0.00 |
| <3hr | 180min | 0.00 | 0.00 | 0.00 | 0.00 | 0.00 | 0.00 | 0.00 | 0.00 | 0.00 | 0.00 | 0.00 | 0.00 | 0.00 | 0.00 | 0.00 | 0.00 | 0.00 | 0.00 | 0.00 | 0.00 | 0.00 | 1.00 | 0.00 | 0.00 | 0.00 | 0.00 | 0.00 | 0.00 |
| >3hr | 15min | 0.00 | 0.10 | 0.25 | 0.00 | 0.00 | 0.00 | 0.00 | 0.00 | 0.57 | 0.00 | 0.00 | 0.00 | 0.00 | 0.00 | 0.00 | 0.08 | 0.00 | 0.00 | 0.00 | 0.00 | 0.00 | 0.00 | 0.00 | 0.00 | 0.00 | 0.00 | 0.00 | 0.00 |
| >3hr | 60min | 0.00 | 0.29 | 0.67 | 0.00 | 0.00 | 0.00 | 0.00 | 0.00 | 0.66 | 0.00 | 0.00 | 0.00 | 0.00 | 0.00 | 0.00 | 0.00 | 0.00 | 0.00 | 0.00 | 0.00 | 0.00 | 0.00 | 0.00 | 0.33 | 0.00 | 0.00 | 0.00 | 0.00 |
| >3hr | 180min | 0.00 | 0.30 | 0.40 | 0.00 | 0.00 | 0.00 | 0.00 | 0.00 | 0.72 | 0.00 | 0.00 | 0.00 | 0.00 | 0.00 | 0.00 | 0.00 | 0.00 | 0.00 | 0.00 | 0.00 | 0.00 | 0.00 | 0.00 | 0.28 | 0.00 | 0.00 | 0.00 | 0.00 |
| | Average | 0.00 | 0.08 | 0.15 | 0.00 | 0.00 | 0.00 | 0.00 | 0.00 | 0.22 | 0.00 | 0.00 | 0.00 | 0.00 | 0.00 | 0.00 | 0.01 | 0.00 | 0.00 | 0.00 | 0.00 | 0.00 | 0.33 | 0.00 | 0.07 | 0.00 | 0.00 | 0.33 | 0.00 |

**Intensity (target variable)**

| Durations | Lead Time | No.Cells | Life.TS | Area | meanPI | medianPI | maxPI | sdPI1 | sdPI2 | GVel | VelX | VelY | Jx | Jy | J.ratio | Phi | Area | meanPi | medPI | maxPI | sdPI1 | sdPI2 | Velg | Velx | Vely | Jx | Jy | Jr | phi |
|---|---|---|---|---|---|---|---|---|---|---|---|---|---|---|---|---|---|---|---|---|---|---|---|---|---|---|---|---|---|
| <1hr | 15min | 0.00 | 0.00 | 0.00 | 0.00 | 0.00 | 0.00 | 0.00 | 0.00 | 0.00 | 0.00 | 0.00 | 0.00 | 0.00 | 0.00 | 0.00 | 0.00 | 0.00 | 0.00 | 0.00 | 0.00 | 0.00 | 0.00 | 0.00 | 0.00 | 0.00 | 0.00 | 0.00 | 0.00 |
| <1hr | 60min | 0.00 | 0.00 | 0.00 | 0.00 | 0.00 | 0.00 | 0.00 | 0.00 | 0.00 | 0.00 | 0.00 | 0.00 | 0.00 | 0.00 | 0.00 | 0.00 | 0.00 | 0.00 | 0.00 | 0.00 | 0.00 | 0.00 | 0.00 | 0.00 | 0.00 | 0.00 | 0.00 | 0.00 |
| <1hr | 180min | 0.00 | 0.00 | 0.00 | 0.00 | 0.00 | 0.00 | 0.00 | 0.00 | 0.00 | 0.00 | 0.00 | 0.00 | 0.00 | 0.00 | 0.00 | 0.00 | 0.00 | 0.00 | 0.00 | 0.00 | 0.00 | 0.00 | 0.00 | 0.00 | 0.00 | 0.00 | 0.00 | 0.00 |
| <3hr | 15min | 0.00 | 0.00 | 0.00 | 0.00 | 0.00 | 0.00 | 0.00 | 0.00 | 0.00 | 0.00 | 0.00 | 1.00 | 0.00 | 0.00 | 0.00 | 0.00 | 0.00 | 0.00 | 0.00 | 0.00 | 0.00 | 0.00 | 0.00 | 0.00 | 0.00 | 0.00 | 0.00 | 0.00 |
| <3hr | 60min | 0.00 | 0.00 | 0.00 | 0.00 | 0.00 | 0.00 | 0.00 | 0.00 | 0.00 | 0.00 | 0.00 | 1.00 | 0.00 | 0.00 | 0.00 | 0.00 | 0.00 | 0.00 | 0.00 | 0.00 | 0.00 | 0.00 | 0.00 | 0.00 | 0.00 | 0.00 | 0.00 | 0.00 |
| <3hr | 180min | 0.00 | 0.00 | 0.00 | 0.00 | 0.00 | 0.00 | 0.00 | 0.00 | 0.00 | 0.00 | 0.00 | 1.00 | 0.00 | 0.00 | 0.00 | 0.00 | 0.00 | 0.00 | 0.00 | 0.00 | 0.00 | 0.00 | 0.00 | 0.00 | 0.00 | 0.00 | 0.00 | 0.00 |
| >3hr | 15min | 0.00 | 0.00 | 0.00 | 0.00 | 0.00 | 0.00 | 0.00 | 0.00 | 0.00 | 0.00 | 0.00 | 1.00 | 0.00 | 0.00 | 0.00 | 0.00 | 0.00 | 0.00 | 0.00 | 0.00 | 0.00 | 0.00 | 0.00 | 0.00 | 0.00 | 0.00 | 0.00 | 0.00 |
| >3hr | 60min | 0.00 | 0.00 | 0.00 | 0.00 | 0.00 | 0.00 | 0.00 | 0.00 | 0.00 | 0.00 | 0.00 | 1.00 | 0.00 | 0.00 | 0.00 | 0.00 | 0.00 | 0.00 | 0.00 | 0.00 | 0.00 | 0.00 | 0.00 | 0.00 | 0.00 | 0.00 | 0.00 | 0.00 |
| >3hr | 180min | 0.00 | 0.00 | 0.00 | 0.00 | 0.00 | 0.00 | 0.00 | 0.00 | 0.00 | 0.00 | 0.00 | 1.00 | 0.00 | 0.00 | 0.00 | 0.00 | 0.00 | 0.00 | 0.00 | 0.00 | 0.00 | 0.00 | 0.00 | 0.00 | 0.00 | 0.00 | 0.00 | 0.00 |
| | Average | 0.00 | 0.00 | 0.00 | 0.00 | 0.00 | 0.00 | 0.00 | 0.00 | 0.00 | 0.00 | 0.00 | 1.00 | 0.00 | 0.00 | 0.00 | 0.00 | 0.00 | 0.00 | 0.00 | 0.00 | 0.00 | 0.00 | 0.00 | 0.00 | 0.00 | 0.00 | 0.00 | 0.00 |

**Velocity X (target variable)**

| Durations | Lead Time | No.Cells | Life.TS | Area | meanPI | medianPI | maxPI | sdPI1 | sdPI2 | GVel | VelX | VelY | Jx | Jy | J.ratio | Phi | Area | meanPi | medPI | maxPI | sdPI1 | sdPI2 | Velg | Velx | Vely | Jx | Jy | Jr | phi |
|---|---|---|---|---|---|---|---|---|---|---|---|---|---|---|---|---|---|---|---|---|---|---|---|---|---|---|---|---|---|
| <1hr | 15min | 0.00 | 0.00 | 0.00 | 0.00 | 0.00 | 0.00 | 0.00 | 0.00 | 0.00 | 0.00 | 0.00 | 0.00 | 0.00 | 0.00 | 0.00 | 0.00 | 0.00 | 0.00 | 0.00 | 0.00 | 0.00 | 0.00 | 0.00 | 0.00 | 0.00 | 0.00 | 0.00 | 0.00 |
| <1hr | 60min | 0.00 | 0.00 | 0.00 | 0.00 | 0.00 | 0.00 | 0.00 | 0.00 | 0.00 | 0.00 | 0.00 | 0.00 | 0.00 | 0.00 | 0.00 | 0.00 | 0.00 | 0.00 | 0.00 | 0.00 | 0.00 | 0.00 | 0.00 | 0.00 | 0.00 | 0.00 | 0.00 | 0.00 |
| <1hr | 180min | 0.00 | 0.00 | 0.00 | 0.00 | 0.00 | 0.00 | 0.00 | 0.00 | 0.00 | 0.00 | 0.00 | 0.00 | 0.00 | 0.00 | 0.00 | 0.00 | 0.00 | 0.00 | 0.00 | 0.00 | 0.00 | 0.00 | 0.00 | 0.00 | 0.00 | 0.00 | 0.00 | 0.00 |
| <3hr | 15min | 0.00 | 0.00 | 0.00 | 0.00 | 0.00 | 0.00 | 0.00 | 0.00 | 0.00 | 0.00 | 0.00 | 0.00 | 0.00 | 0.00 | 0.00 | 0.00 | 0.00 | 0.00 | 0.00 | 0.00 | 0.00 | 0.00 | 1.00 | 0.00 | 0.00 | 0.00 | 0.00 | 0.00 |
| <3hr | 60min | 0.00 | 0.00 | 0.00 | 0.00 | 0.00 | 0.00 | 0.00 | 0.00 | 0.00 | 0.00 | 0.00 | 0.00 | 0.00 | 0.00 | 0.00 | 0.00 | 0.00 | 0.00 | 0.00 | 0.00 | 0.00 | 0.00 | 1.00 | 0.00 | 0.00 | 0.00 | 0.00 | 0.00 |
| <3hr | 180min | 0.00 | 0.00 | 0.00 | 0.00 | 0.00 | 0.00 | 0.00 | 0.00 | 0.00 | 0.00 | 0.00 | 0.00 | 0.00 | 0.00 | 0.00 | 0.00 | 0.00 | 0.00 | 0.00 | 0.00 | 0.00 | 0.00 | 1.00 | 0.00 | 0.00 | 0.00 | 0.00 | 0.00 |
| >3hr | 15min | 0.00 | 0.00 | 0.00 | 0.00 | 0.00 | 0.00 | 0.00 | 0.00 | 0.00 | 0.00 | 0.00 | 0.00 | 0.00 | 0.00 | 0.00 | 0.00 | 0.00 | 0.00 | 0.00 | 0.00 | 0.00 | 0.00 | 0.00 | 0.00 | 0.00 | 0.00 | 0.00 | 0.00 |
| >3hr | 60min | 0.00 | 0.00 | 0.00 | 0.00 | 0.00 | 0.00 | 0.00 | 0.00 | 0.00 | 0.00 | 0.00 | 0.00 | 0.00 | 0.00 | 0.00 | 0.00 | 0.00 | 0.00 | 0.00 | 0.00 | 0.00 | 0.00 | 0.00 | 0.00 | 0.00 | 0.00 | 0.00 | 0.00 |
| >3hr | 180min | 0.00 | 0.00 | 0.00 | 0.00 | 0.00 | 0.00 | 0.00 | 0.00 | 0.00 | 0.00 | 0.00 | 0.00 | 0.00 | 0.00 | 0.00 | 0.00 | 0.00 | 0.00 | 0.00 | 0.00 | 0.00 | 0.00 | 0.00 | 0.00 | 0.00 | 0.00 | 0.00 | 0.00 |
| | Average | 0.00 | 0.00 | 0.00 | 0.00 | 0.00 | 0.00 | 0.00 | 0.00 | 0.00 | 0.00 | 0.00 | 0.00 | 0.00 | 0.00 | 0.00 | 0.00 | 0.00 | 0.00 | 0.00 | 0.00 | 0.00 | 0.00 | 1.00 | 0.00 | 0.00 | 0.00 | 0.00 | 0.00 |

**Velocity Y (target variable)**

| Durations | Lead Time | No.Cells | Life.TS | Area | meanPI | medianPI | maxPI | sdPI1 | sdPI2 | GVel | VelX | VelY | Jx | Jy | J.ratio | Phi | Area | meanPi | medPI | maxPI | sdPI1 | sdPI2 | Velg | Velx | Vely | Jx | Jy | Jr | phi |
|---|---|---|---|---|---|---|---|---|---|---|---|---|---|---|---|---|---|---|---|---|---|---|---|---|---|---|---|---|---|
| <1hr | 15min | 0.00 | 0.00 | 0.00 | 0.00 | 0.00 | 0.00 | 0.00 | 0.00 | 0.00 | 0.00 | 0.00 | 0.00 | 0.00 | 0.00 | 0.00 | 0.00 | 0.00 | 0.00 | 0.00 | 0.00 | 0.00 | 0.00 | 0.00 | 0.00 | 0.00 | 0.00 | 0.00 | 0.00 |
| <1hr | 60min | 0.00 | 0.00 | 0.00 | 0.00 | 0.00 | 0.00 | 0.00 | 0.00 | 0.00 | 0.00 | 0.00 | 0.00 | 0.00 | 0.00 | 0.00 | 0.00 | 0.00 | 0.00 | 0.00 | 0.00 | 0.00 | 0.00 | 0.00 | 0.00 | 0.00 | 0.00 | 0.00 | 0.00 |
| <1hr | 180min | 0.00 | 0.00 | 0.00 | 0.00 | 0.00 | 0.00 | 0.00 | 0.00 | 0.00 | 0.00 | 0.00 | 0.00 | 0.00 | 0.00 | 0.00 | 0.00 | 0.00 | 0.00 | 0.00 | 0.00 | 0.00 | 0.00 | 0.00 | 0.00 | 0.00 | 0.00 | 0.00 | 0.00 |
| <3hr | 15min | 0.00 | 0.00 | 0.00 | 0.00 | 0.00 | 0.00 | 0.00 | 0.00 | 0.00 | 0.00 | 0.00 | 0.00 | 0.00 | 0.00 | 0.00 | 0.00 | 0.00 | 0.00 | 0.00 | 0.00 | 0.00 | 0.00 | 0.00 | 0.00 | 0.00 | 0.00 | 0.00 | 0.00 |
| <3hr | 60min | 0.00 | 0.00 | 0.00 | 0.00 | 0.00 | 0.00 | 0.00 | 0.00 | 0.00 | 0.00 | 0.00 | 0.00 | 0.00 | 0.00 | 0.00 | 0.00 | 0.00 | 0.00 | 0.00 | 0.00 | 0.00 | 0.00 | 0.00 | 0.00 | 0.00 | 0.00 | 0.00 | 0.00 |
| <3hr | 180min | 0.00 | 0.00 | 0.00 | 0.00 | 0.00 | 0.00 | 0.00 | 0.00 | 0.00 | 0.00 | 0.00 | 0.00 | 0.00 | 0.00 | 0.00 | 0.00 | 0.00 | 0.00 | 0.00 | 0.00 | 0.00 | 0.00 | 0.00 | 0.00 | 0.00 | 0.00 | 0.00 | 0.00 |
| >3hr | 15min | 0.00 | 0.00 | 0.00 | 0.00 | 0.00 | 0.00 | 0.00 | 0.00 | 0.00 | 0.00 | 0.00 | 0.00 | 0.00 | 0.00 | 0.00 | 0.00 | 0.00 | 0.00 | 0.00 | 0.00 | 0.00 | 0.00 | 0.00 | 0.00 | 1.00 | 0.00 | 0.00 | 0.00 |
| >3hr | 60min | 0.00 | 0.00 | 0.00 | 0.00 | 0.00 | 0.00 | 0.00 | 0.00 | 0.00 | 0.00 | 0.00 | 0.00 | 0.00 | 0.00 | 0.00 | 0.00 | 0.00 | 0.00 | 0.00 | 0.00 | 0.00 | 0.00 | 0.00 | 0.00 | 1.00 | 0.00 | 0.00 | 0.00 |
| >3hr | 180min | 0.00 | 0.00 | 0.00 | 0.00 | 0.00 | 0.00 | 0.00 | 0.00 | 0.00 | 0.00 | 0.00 | 0.00 | 0.00 | 0.00 | 0.00 | 0.00 | 0.00 | 0.00 | 0.00 | 0.00 | 0.00 | 0.00 | 0.00 | 0.00 | 1.00 | 0.00 | 0.00 | 0.00 |
| | Average | 0.00 | 0.00 | 0.00 | 0.00 | 0.00 | 0.00 | 0.00 | 0.00 | 0.00 | 0.00 | 0.00 | 0.00 | 0.00 | 0.00 | 0.00 | 0.00 | 0.00 | 0.00 | 0.00 | 0.00 | 0.00 | 0.00 | 0.00 | 0.00 | 1.00 | 0.00 | 0.00 | 0.00 |

**Duration (target variable)**

| Durations | Lead Time | No.Cells | Life.TS | Area | meanPI | medianPI | maxPI | sdPI1 | sdPI2 | GVel | VelX | VelY | Jx | Jy | J.ratio | Phi | Area | meanPi | medPI | maxPI | sdPI1 | sdPI2 | Velg | Velx | Vely | Jx | Jy | Jr | phi |
|---|---|---|---|---|---|---|---|---|---|---|---|---|---|---|---|---|---|---|---|---|---|---|---|---|---|---|---|---|---|
| Dur <1hr | | 0.00 | 0.00 | 0.00 | 0.00 | 0.00 | 0.00 | 0.00 | 0.00 | 0.00 | 0.00 | 0.00 | 0.00 | 0.00 | 0.00 | 0.00 | 0.00 | 0.00 | 0.00 | 0.00 | 0.00 | 0.00 | 0.00 | 0.00 | 0.00 | 0.00 | 0.00 | 1.00 | 0.00 |
| Dur <3hr | | 0.00 | 0.00 | 0.00 | 0.00 | 0.00 | 0.00 | 0.00 | 0.00 | 0.00 | 0.00 | 0.00 | 0.00 | 0.00 | 0.00 | 0.00 | 0.00 | 0.00 | 0.00 | 0.00 | 0.00 | 0.00 | 1.00 | 0.00 | 0.00 | 0.00 | 0.00 | 0.00 | 0.00 |
| Dur >3hr | | 0.00 | 0.45 | 0.40 | 0.00 | 0.00 | 0.00 | 0.00 | 0.00 | 0.72 | 0.00 | 0.00 | 0.00 | 0.00 | 0.00 | 0.00 | 0.00 | 0.00 | 0.00 | 0.00 | 0.00 | 0.00 | 0.00 | 0.00 | 0.00 | 0.00 | 0.00 | 0.33 | 0.00 |
| | Average | 0.00 | 0.15 | 0.13 | 0.00 | 0.00 | 0.00 | 0.00 | 0.00 | 0.24 | 0.00 | 0.00 | 0.00 | 0.00 | 0.00 | 0.00 | 0.00 | 0.00 | 0.00 | 0.00 | 0.00 | 0.00 | 0.33 | 0.00 | 0.00 | 0.00 | 0.11 | 0.33 | 0.00 |

---

## Author Comment (AC3)

Dear Georgy,

thank you for your comments and suggestions, that have contributed in improving the manuscript and the idea after the nearest neighbour application. Following the suggestions of another review, one of the main changes that I have done, was to update the results based on absolute error and computed the Continuous Rank Probability Score for the probabilistic nowcast. Thus, the results plots have changed accordingly, and are given in the response to this review. Please find below the answers and our comments on the questions/issues you have raised (given in blue below each point). Please note that some changes have been already done in the updated version of the manuscript (these are given in quotation) while other changes are yet to be done and thus I refer to as "will be included" in the updated version of the manuscript. The updated plots and tables are given in the attached pdf at the end of this response.

**Major Comments:**

***Research Aim:*** The decision to predict individual storm characteristics, i.e., area, mean intensity, x and y components of velocity, and lifetime, instead of predicting the entire storm evolution as an integral object should be elaborated. In general, I understand the utility of predicting individual characteristics. However, in this way, we miss the detailed information about storm event spatiotemporal evolution and could not precisely estimate neither location nor intensity-related errors. For example, the spatial structure and distribution of rainfall intensities within the storm event are particularly relevant for urban applications. The example of the possible utilization of the predicted (individual) properties could help to clarify their choice.

> Yes of course, your concern is right and we will discuss it better in the updated version of the manuscript. In this paper we focus only on the storm characteristics to see, first if the k-NN application is suitable (either deterministic or probabilistic). Once the storm characteristics can be properly nowcasted, there are two options to deal with the spatial distribution of the rainfall intensities inside the storm region: 1. Increase/Reduce the area by the given nowcasted area (as target variable) for each lead time, scale the average intensity with the nowcasted intensity, and move the position of the storm in the future with the nowcasted velocity in x and y direction. 2. Take the spatial information of the selected neighbours (with the method we propose here), perform an optimisation in space (such that present storm and the neighbour's storms locations match) and assign this spatial information to the present storm for each lead time. This will be done in a follow up paper. That is why we will check with the editor to see if the title can be changed to "Part I – Storm Characteristics", to give the idea that this is only the first step, and the follow up paper will be "Part II – Rainfall Intensities at 1km$^2$".

***Database***: The authors declare that the compiled dataset includes outliers (L198). That leads to the mixed-use of mean or probability-based (e.g., median) statistics. It is pretty hard to recognize and remember where and why the mean or median statistics are used. Moreover, the authors often describe the need to use mean/median for (not) accounting outliers but rarely communicate the obtained results based on that choice. Thus, it is interesting what is the proportion of outliers in the compiled database and could they be removed for the sake of consistency of mean/median statistics throughout the manuscript.

> In the first manuscript draft, the mean is used only for the training of the k-NN, because it is a typically used target optimization statistics. The median is used when validating the k-NN, because we were interested to see the error if the k-NN on the majority of the cases. However, in the new updated manuscript we are showing the median for all the results: training and validation. Please refer to the updated Figures in the attached pdf.

The compiled database of storm events is based on the open data provided by the German Weather Service. Is it possible to share it? It would serve both manuscript's reproducibility and community interests in the field of storm tracking and prediction.

> Yes, the complied database of the storm events is based on the open data from the DWD. However, the tracking algorithm used as a basis for this study (HyRaTrac) is not open for public (or at least I have to ask the person who created it). But What I can make public is database of the storm characteristics and the application of the k-NN on these storm characteristics.

**Baseline:** The utilization of Lagrangian persistence as a baseline is reliable, and obtained results are interesting to compare. I recommend authors provide additional information about it (how nowcasted is computed etc.) to account for inexperienced readers.

> Following the comments of the other reviewers as well, we have updated the introduction to the topic to include more information for the inexperienced readers.

In Section 4.4., the authors compare the closest single neighbor and 30-member ensemble approach. Do the authors mind finding the single nearest neighbor as a more advanced baseline compared with four and 30-member ensemble solutions? It would then pose an additional research question (partially touched in Sect. 4.4) of an added value of ensemble approach compared to the single neighbor.

> In section 4.4 we do not compare the closest neighbour with the 30-member ensemble approach. The ensemble member is issued randomly based on the rank probability of the 30-closest neighbour. In this section, we assess the best possible outcome from the ensembles – which is the ensemble with the lowest error. This suggests that the predictors selected and their weights are not able to select the best single neighbour at the deterministic approach, that is why an ensemble approach is better (as it is not so dependent on the number of K to average, and the predictors weights). Following the comments of another reviewer, we have calculated the continuous rank probability score (CRPS) for the ensemble members, which is a generalization of the mean absolute error and ensures a direct comparison with the deterministic approach. The Figures 11 and 12 of the CRPS together with the mean absolute error (MAE) of the deterministic 4-NN for the two cases (storm- and target – based) are shown in the attached pdf. It is clear that the errors of the ensemble members are lower than the 4-NN approach, thus it is clear that the ensemble approach is more suitable.

**Information leakage**: In modelling studies, it is particularly relevant to isolate calibration, validation, and test datasets to prevent so-called information leakage -- the situation when the information outside the calibration set is used for model calibration (training). In the presented study, I suspect four procedures that may lead to information leakage:

1. Normalization of events characteristics.
2. Importance analysis and weights calculation.
3. Optimization of the number of nearest neighbours.
 4. Splitting into different event groups.

The authors state that normalization and importance analysis have been done "Before training and validating the k-NN method" (L191). In this way, there is an evident information leakage that connects calibration and validation datasets. Also, it is not clear how calibration and validation datasets have been isolated to find the optimal number of nearest neighbours. I do not think that addressing data leakage would change the results much, but it is vital to ensure methodological reliability.

> Regarding the importance analysis and weights calculations: In this application the information leakage may occur only at the important analysis and the weights calculations. Here all the

events are grouped together to check the relationship between predictors and target variables, and the importance weights are calculated. In the case of Pearson correlation, the weights are not changing drastically when one event is left out and the weights are calculated from the remaining events. This is not the case with the Partial Information Correlation, the weights are changing from one set to the other, that is why we also preferred the Pearson Correlation and the PIC. Since we have disregarded the use of the PIC for the weight calculations, here I would like to show you the results of the Pearson Correlation weights, how much they change when one event is left out (out at a time) and the weights are calculated from the remaining events. The following events indicate the standard deviation of the predictors' weights for each of the target variable computed from leaving one event out. The boxplot represents the standard deviation of the selected predictors. As you can see the deviation of the weights between the predictors and target variables is very low (lower than 0.01). This low variability of the predictors weights justifies our decision to estimate the weights from the whole database. While this is clear for the Pearson correlation, the same can not be said for the partial information correlation. Moreover, a sensitivity analysis like this can not be performed for the PIC because is extremely time consuming.

[Figure]

Regarding normalization of event characteristics: Here no information leakage is possible, because both pseudo-training and the validation of the k-NN is done on a "leave-one-out-event" approach. This means one event at a time is taken out of the database, and the past database that is used for the prediction has the remaining 109 events. The normalization of the event characteristics is then done only based on this past database, thus for the event nowcasted the values of the characteristics can be higher than 1 if they have higher values observed then in the past database. This procedure is done 110 times.

Regarding optimization of the number of nearest neighbours: As stated before this is also done is a "leave-one-out-event" approach. So, the optimum k for that event is found by the remaining past database. The optimum k is grouped then (only the results) according to the time when the nowcast is issued (nowcast time) to see how this value of K is changing with the lead time and with the nowcast time (see Figure 8 in the attached pdf). As it visible there is no clear best optimum k, and we selected the k=4 as a first attempt to reach this optimum. An information leakage would then be if we used the optimum k for each storm duration and lead time (results of this k-NN pseudo-training).

> Regarding Splitting into different event groups: Im a bit confused about this. We do not use the information of different event groups prior, they are just used posterior. So the k-NN event is run with all durations together, and only later on the results are computed for different groups, so we can understand with groups are nowcasted better than the others.

> We have included in Figure 3 were potential information leakage may occur and we will discuss it better in the paper.

Splitting the database into three groups according to their duration (L312-317, Table 2) was done before the modeling. In general (and in practice), we do not know a priori if the recently appeared storm will be sporadic or last for a couple of hours or more. So, in my opinion, in making predictions, we should use all the examples from the database to find closer candidates to be used for predictions, not only those from the group of a similar duration. That also would open the new directions of analysis, e.g., how would closest examples change with the storm's evolution. Is the more mature storm similar to storms with comparable duration, or is there some skew in characteristics similarity?

> No, the database was not split into three groups before the modelling! Table 2 is just informative to see how many storms belong to each group, that can explain the effect of the database size on the results. K-NN doesn't know before that a storm is belongs at a particular duration, it just calculates based on the past database (all storm durations included) the most similar storms and take the response of the most similar storms. This is the case for instance of Figure 10, second row for nowcast time at $36^{th}$ time step of storms existence. The k-NN is not able to find storms that have similar durations, but finds similar storms from the ones that have a long duration, and hence it overestimates the total lifetime of the storm and the rainfall area. So, to conclude, the storms durations (these three groups) are just use to summarize the results, to see which type can be simulated better, the k-NN is performed on the past database with all the storms grouped together.

> We will try to clarify this better in the updated version of the manuscript. We have also included a new Figure (See Figure 3 in the attached pdf) that hopefully will make the work flow clearer.

The minor but also critical comment here is about the research code availability. For sure, open code would ensure research reproducibility and provide information on particular details of the computational workflow.

> I hope Figure 3-C can make the workflow clearer. Due to time constrains at the moment we can not upload the R-codes, but we will soon do this.

Training, learning, and cross-validation:

The authors use terms of training and cross-validation, but, in my opinion, the presented manuscript does not involve both procedures. The nearest-neighbour model is not trained per se; it only uses a bag with historical examples to find one closer to the "storm-to-be-predicted" based on the similarity metric. In this way, the nearest-neighbour model also does not learn anything as it has no parameters to learn. The only parameter here is the number of the nearest neighbours to use for predictions. However, the choice of that number is entirely subjective (see comment above) and is independent of both the "storm-to-be-predicted" and the available examples and their characteristics. I would also question the use of the term cross-validation. There is no numerical model to validate as the nearest neighbour approach is instead a database search method than a "pure" numerical model.

> Yes k-NN is a parsimonious model based on the past database, and is a lazy learner as it is entirely dependent on the past database. And yes, the k-value is the only parameter so to say that can be optimized or learned by having an optimization function. This is what we tried to do in the

training/learning of this k-parameter, but the response we got was very dynamic in respect to the lead time and nowcast time. A proper learning of this k-values seems not to be possible for the nowcast application as there is not a single optimum reached and there are too many degrees of freedom (lead time and nowcast time) and instead we decided on a k-value. These are the disadvantages of the deterministic application of the k-NN (and we wanted to show actually that). We could also choose to show the deterministic application with the first neighbour, nevertheless that would have been criticized because it is always arguable that the value k-could have been optimized. Instead the probabilistic application of the k-NN doesn't have this problem because it doesn't need a training, and is taking the 30 closest neighbours (and it works better than the deterministic approach). However, I agree that this is not well discuss in the paper and we are trying to make it clearer in the updated version. We could refer to training also as pseudo-training and explain it to the reader that is not a proper training as for instance in the training of an artificial neural network.

Regarding the use of the cross-validation term, I do not agree fully with you. Even though this is not a numerical model, and even in the case it doesn't have parameters, the database search has still to be tested and to be validated that it works good enough. Cross-validation is a broader term not only for numerical models, and it refers to as out-of-sample testing of any model, regardless of what the model is made of (or what inside the model). As we want to check/test how well our past database can simulate a new event outside of our sample, and repeat this for N times, I think it lies within the definition of the cross-validation.

The authors explicitly communicate the aim of the study as "... to investigate if non-linear relationships learned from past observed storms can surpass the Lagrangian persistence and extend the predictability limit of different storms." However, as I mentioned above, the nearest neighbour model does not learn anything. It is also an open question if there are non-linear relationships (and what kind of relationships).

I think you might confuse the terms a little bit. Lagrangian persistence is a linear extrapolation in the future; constant area, intensity and movement. The "non-linear relationships" referred to here refers to the ability to included other non-linear extrapolations for the future: so the area, intensity and movement are changing with the lead time. More importantly that the storms can be dissipated because in the linear extrapolation of the Lagrangian persistence this is completely ignored. This is what we refer to with non-linear relationship – a non-linear extrapolation is time that can be learned or obtained from the past database. The phrase will is also now updated to:

"… to investigate if non-linear relationships estimated from past observed storms can surpass the Lagrangian persistence and extend the predictability limit of different storms…"

In the application of the kNN, it is clear that the storm dissipation can be captured better than the Lagrangian persistence, hence there is a non-linear behaviour of the storm that can be estimated based on a past database.

**Minor comments:**

● "Birth" → "initialization"? Noted and changed!

● "Death" → "dissipation"? Noted and changed!

● L98: "k-NN." The first appearance needs transcription. Noted and changed!

● The orientation feature is in degrees. I wonder how the difference between 1 and 359 degrees is considered.

The orientation feature is expressed in degrees from 1 to 180 degrees with positive sign between 1 and 180 degrees and negative sign between 180 and 360 degrees (-180 to 0). So when considering the difference between 1° and 359° degrees (-1°), the difference will be 2 degrees.

● Interestingly, the area and number of storm cells do not show similar behaviour in importance analysis, but they are highly correlated.

I'm not sure I fully understand what you mean. The following figure shows the correlation between the predictors, and the number of storm cells are not highly correlated with the area, this is as well portrayed in the importance analysis. Please note that the number of the storm cells refers to the co-existence of storms in a split or merge situation, so there are two storms for instance but they are treated as one, and is not to be confused with the number of storms inside an event.

[Figure]

● L261-262: "Only neighbours that display a distance lower than 0.5 are selected for both single and ensemble nowcast in order to minimize the influence of non-similar storms." Any statistics of that?

The value 0.5 represents the 95% quantile of the calculated distances all the first 4 neighbours for all storms/events. Actually, the model is not very sensitive to this value, but to avoid confusion and to make a fair comparison with Lagrangian persistence (not to exclude the time steps where the neighbours have a higher distance than 0.5) we have excluded this limitation and show the results for all distances.

● Figure 6: "The weights given here are averaged from the weights calculated at three different lead times and storm durations." However, the authors then mention (L341-342): "Contrary for Total Lifetime and Area, only for storms that last longer than 3 hours, the method is able to converge and give the most important predictors." However, we cannot see these results.

At the end of section 3.1.3 the averaging of the three lead times was explained:

"Here in this study, these two importance analyses are used to determine the most important predictors and their respective weights in the k-NN similarity calculation. For each target variable the most important predictor identified from Pearson Correlation, is given to the PIC metric as the first predictor. The analysis is complex due to the presence of several predictors, 38 states of future behaviour for each target variable (for each 5min between +5min to +180 min lead times), and different times of nowcast; the weights were calculated first for three lead times +15min, +60min and +180 min, and for three storm groups separated according to their duration <60min, 60min-180min, and > 3 hours. Here the averages weights over these groups and lead times are calculated and used as a reference for each importance analysis. The k-NN errors with these average weights are compared in Section 4.1."

We have added the correlations for each of the selected lead times in the appendix. The tables for both target-based and storm-based approaches and for the two important analysis methods are given in the attached pdf (Appendix 8.1 and 8.2). The reason why we took the average from these three lead times is because we wanted to have only one set of weights independent of the lead times and storm types as a starting point for the nearest neighbour approach.

● L348-349: "is not completely understood and is not investigated further on for the time being since it is outside the scope of this paper." But, from the abstract and introduction: "i) what features should be used to describe storms in order to check for similarity?" Thus, it is probably in the scope of the paper.

The PIC was built specifically for the k-NN application, nevertheless on our experience the PIC seems not so robust; as the values were changing depending on the database or if one randomly chose the events to be feed in the importance analysis. As mentioned in the paper, one of the main reasons why it works for the duration and not the other target variables are the presence of the zero values. When too many zero are present, the distribution is skewed and affected the calculation of the partial dependency. Also, when the data set is to big, based on personal experience the PIC is very time consuming and fails to converge. On the other side, the Pearson correlation seems to yield reasonable and stable results. Thus, we continue the k-NN application with Pearson correlation instead of with the PIC application, to identify the features that best describe similar storms. To understand fully why PIC specifically behaves like it does, was not the scope of the paper. What features should be used to describe storms to check for similarity is investigated with the Pearson correlation.

● L354-355: "Moreover, the important predictors do not change drastically from one lead time or storm group to the other, as seen in the PIC" Could we see it from any table or figure?

Please refer to the table above for the PIC weights.

● Figures: larger fonts and more vertical space between different types of events would be appreciated

Noted and implemented.

[revised manuscript text omitted]

 *Obtained Pearson Correlation predictors weight for each target variable, lead time and storm groups. The last row at each target variable (average values) are the predictors weights shown in Table 3*

**Area**

| Durations | Lead Time | Cell | Life | A | avePI | medPI | maxPI | sdPI1 | sdPI2 | Vg | Vx | Vy | Jx | Jy | Jr | Φ | A | avePI | medPI | maxPI | sdPI1 | sdPI2 | Vg | Vx | Vy | Jx | Jy | Jr | Φ |
|---|---|---|---|---|---|---|---|---|---|---|---|---|---|---|---|---|---|---|---|---|---|---|---|---|---|---|---|---|---|
| <1hr | 15min | 0.19 | 0.22 | 0.81 | 0.06 | 0.04 | 0.05 | 0.05 | 0.59 | 0.06 | 0.02 | 0.01 | 0.58 | 0.61 | 0.01 | 0.00 | 0.79 | 0.06 | 0.04 | 0.05 | 0.05 | 0.61 | 0.06 | 0.02 | 0.02 | 0.60 | 0.62 | 0.01 | 0.00 |
| <1hr | 60min | 0.09 | 0.27 | 0.90 | 0.19 | 0.07 | 0.07 | 0.20 | 0.64 | 0.05 | 0.05 | 0.03 | 0.65 | 0.68 | 0.04 | 0.02 | 0.88 | 0.21 | 0.09 | 0.08 | 0.22 | 0.67 | 0.08 | 0.08 | 0.07 | 0.68 | 0.70 | 0.08 | 0.02 |
| <1hr | 180min | 0.12 | 0.19 | 0.94 | 0.18 | 0.08 | 0.23 | 0.29 | 0.68 | 0.05 | 0.04 | 0.05 | 0.72 | 0.70 | 0.28 | 0.00 | 0.92 | 0.21 | 0.05 | 0.25 | 0.31 | 0.69 | 0.05 | 0.05 | 0.09 | 0.73 | 0.69 | 0.38 | 0.02 |
| <3hr | 15min | 0.12 | 0.19 | 0.61 | 0.04 | 0.03 | 0.03 | 0.04 | 0.45 | 0.00 | 0.01 | 0.00 | 0.43 | 0.49 | 0.01 | 0.01 | 0.60 | 0.05 | 0.03 | 0.02 | 0.04 | 0.46 | 0.00 | 0.02 | 0.02 | 0.45 | 0.50 | 0.01 | 0.01 |
| <3hr | 60min | 0.04 | 0.25 | 0.72 | 0.13 | 0.05 | 0.01 | 0.13 | 0.48 | 0.01 | 0.03 | 0.03 | 0.55 | 0.55 | 0.03 | 0.01 | 0.69 | 0.15 | 0.07 | 0.03 | 0.15 | 0.51 | 0.02 | 0.05 | 0.05 | 0.56 | 0.55 | 0.07 | 0.02 |
| <3hr | 180min | 0.09 | 0.13 | 0.80 | 0.16 | 0.06 | 0.20 | 0.25 | 0.58 | 0.10 | 0.01 | 0.05 | 0.63 | 0.57 | 0.24 | 0.00 | 0.77 | 0.20 | 0.02 | 0.23 | 0.28 | 0.57 | 0.11 | 0.00 | 0.09 | 0.61 | 0.55 | 0.32 | 0.02 |
| >3hr | 15min | 0.05 | 0.13 | 0.32 | 0.04 | 0.03 | 0.00 | 0.03 | 0.22 | 0.00 | 0.01 | 0.01 | 0.24 | 0.25 | 0.01 | 0.01 | 0.31 | 0.04 | 0.03 | 0.01 | 0.04 | 0.21 | 0.00 | 0.02 | 0.03 | 0.25 | 0.25 | 0.01 | 0.02 |
| >3hr | 60min | 0.03 | 0.14 | 0.42 | 0.13 | 0.08 | 0.09 | 0.14 | 0.27 | 0.07 | 0.02 | 0.02 | 0.32 | 0.26 | 0.02 | 0.02 | 0.39 | 0.15 | 0.10 | 0.10 | 0.16 | 0.27 | 0.07 | 0.02 | 0.05 | 0.32 | 0.25 | 0.05 | 0.02 |
| >3hr | 180min | 0.06 | 0.07 | 0.53 | 0.17 | 0.03 | 0.19 | 0.22 | 0.39 | 0.16 | 0.05 | 0.07 | 0.41 | 0.34 | 0.16 | 0.06 | 0.50 | 0.20 | 0.06 | 0.22 | 0.25 | 0.38 | 0.18 | 0.07 | 0.11 | 0.40 | 0.32 | 0.20 | 0.08 |
| | Average | 0.09 | 0.18 | 0.67 | 0.12 | 0.05 | 0.10 | 0.15 | 0.48 | 0.05 | 0.03 | 0.03 | 0.50 | 0.49 | 0.09 | 0.02 | 0.65 | 0.14 | 0.05 | 0.11 | 0.17 | 0.48 | 0.07 | 0.04 | 0.06 | 0.51 | 0.49 | 0.12 | 0.02 |

**Intensity**

| Durations | Lead Time | No.Cells | Life.TS | Area | meanPI | medianPI | maxPI | sdPI1 | sdPI2 | GVel | VelX | VelY | Jx | Jy | J.ratio | Phi | Area | meanPi | medPI | maxPI | sdPI1 | sdPI2 | Velg | Velx | Vely | Jx | Jy | Jr | phi |
|---|---|---|---|---|---|---|---|---|---|---|---|---|---|---|---|---|---|---|---|---|---|---|---|---|---|---|---|---|---|
| <1hr | 15min | 0.02 | 0.05 | 0.00 | 0.55 | 0.41 | 0.52 | 0.54 | 0.11 | 0.06 | 0.03 | 0.00 | 0.02 | 0.02 | 0.00 | 0.01 | 0.00 | 0.52 | 0.40 | 0.50 | 0.52 | 0.11 | 0.06 | 0.03 | 0.01 | 0.02 | 0.02 | 0.00 | 0.01 |
| <1hr | 60min | 0.04 | 0.01 | 0.12 | 0.70 | 0.53 | 0.61 | 0.69 | 0.06 | 0.04 | 0.01 | 0.07 | 0.00 | 0.01 | 0.02 | 0.02 | 0.14 | 0.64 | 0.49 | 0.59 | 0.65 | 0.03 | 0.05 | 0.01 | 0.09 | 0.02 | 0.02 | 0.02 | 0.01 |
| <1hr | 180min | 0.03 | 0.13 | 0.11 | 0.81 | 0.67 | 0.68 | 0.77 | 0.13 | 0.09 | 0.09 | 0.03 | 0.14 | 0.15 | 0.05 | 0.06 | 0.11 | 0.76 | 0.62 | 0.68 | 0.75 | 0.13 | 0.13 | 0.11 | 0.04 | 0.14 | 0.15 | 0.06 | 0.10 |
| <3hr | 15min | 0.02 | 0.10 | 0.11 | 0.15 | 0.08 | 0.22 | 0.17 | 0.14 | 0.04 | 0.02 | 0.01 | 0.08 | 0.10 | 0.01 | 0.00 | 0.10 | 0.14 | 0.07 | 0.20 | 0.16 | 0.13 | 0.04 | 0.02 | 0.01 | 0.08 | 0.09 | 0.02 | 0.01 |
| <3hr | 60min | 0.01 | 0.06 | 0.01 | 0.31 | 0.18 | 0.45 | 0.37 | 0.10 | 0.02 | 0.02 | 0.03 | 0.07 | 0.07 | 0.02 | 0.05 | 0.01 | 0.28 | 0.16 | 0.43 | 0.34 | 0.09 | 0.03 | 0.02 | 0.06 | 0.06 | 0.05 | 0.04 | 0.05 |
| <3hr | 180min | 0.01 | 0.06 | 0.10 | 0.43 | 0.40 | 0.50 | 0.47 | 0.20 | 0.08 | 0.06 | 0.01 | 0.25 | 0.22 | 0.09 | 0.09 | 0.08 | 0.42 | 0.37 | 0.47 | 0.44 | 0.19 | 0.10 | 0.08 | 0.01 | 0.24 | 0.21 | 0.12 | 0.10 |
| >3hr | 15min | 0.03 | 0.11 | 0.12 | 0.02 | 0.00 | 0.08 | 0.03 | 0.11 | 0.02 | 0.01 | 0.01 | 0.08 | 0.10 | 0.01 | 0.01 | 0.11 | 0.01 | 0.01 | 0.06 | 0.02 | 0.10 | 0.02 | 0.01 | 0.01 | 0.08 | 0.10 | 0.01 | 0.02 |
| >3hr | 60min | 0.02 | 0.06 | 0.08 | 0.07 | 0.05 | 0.17 | 0.11 | 0.09 | 0.02 | 0.00 | 0.03 | 0.06 | 0.04 | 0.02 | 0.04 | 0.05 | 0.06 | 0.04 | 0.16 | 0.10 | 0.08 | 0.02 | 0.00 | 0.05 | 0.05 | 0.03 | 0.04 | 0.04 |
| >3hr | 180min | 0.01 | 0.05 | 0.36 | 0.10 | 0.18 | 0.10 | 0.06 | 0.31 | 0.03 | 0.02 | 0.10 | 0.38 | 0.35 | 0.11 | 0.05 | 0.34 | 0.07 | 0.15 | 0.05 | 0.02 | 0.30 | 0.05 | 0.05 | 0.16 | 0.36 | 0.33 | 0.15 | 0.05 |
| | Average | 0.02 | 0.07 | 0.11 | 0.35 | 0.28 | 0.37 | 0.36 | 0.14 | 0.04 | 0.03 | 0.03 | 0.12 | 0.12 | 0.03 | 0.04 | 0.10 | 0.32 | 0.26 | 0.35 | 0.33 | 0.13 | 0.05 | 0.04 | 0.05 | 0.12 | 0.11 | 0.05 | 0.04 |

**Velocity X**

| Durations | Lead Time | No.Cells | Life.TS | Area | meanPI | medianPI | maxPI | sdPI1 | sdPI2 | GVel | VelX | VelY | Jx | Jy | J.ratio | Phi | Area | meanPi | medPI | maxPI | sdPI1 | sdPI2 | Velg | Velx | Vely | Jx | Jy | Jr | phi |
|---|---|---|---|---|---|---|---|---|---|---|---|---|---|---|---|---|---|---|---|---|---|---|---|---|---|---|---|---|---|
| <1hr | 15min | 0.04 | 0.02 | 0.09 | 0.01 | 0.01 | 0.01 | 0.00 | 0.06 | 0.14 | 0.17 | 0.02 | 0.06 | 0.05 | 0.01 | 0.02 | 0.08 | 0.01 | 0.01 | 0.01 | 0.00 | 0.08 | 0.13 | 0.18 | 0.02 | 0.14 | 0.07 | 0.02 | 0.02 |
| <1hr | 60min | 0.03 | 0.03 | 0.12 | 0.03 | 0.04 | 0.02 | 0.02 | 0.04 | 0.31 | 0.37 | 0.06 | 0.10 | 0.03 | 0.01 | 0.03 | 0.11 | 0.04 | 0.04 | 0.02 | 0.03 | 0.04 | 0.33 | 0.52 | 0.09 | 0.15 | 0.04 | 0.00 | 0.03 |
| <1hr | 180min | 0.04 | 0.01 | 0.06 | 0.05 | 0.06 | 0.04 | 0.05 | 0.00 | 0.27 | 0.32 | 0.05 | 0.12 | 0.06 | 0.01 | 0.06 | 0.07 | 0.04 | 0.05 | 0.03 | 0.04 | 0.00 | 0.35 | 0.42 | 0.05 | 0.16 | 0.07 | 0.01 | 0.05 |
| <3hr | 15min | 0.03 | 0.06 | 0.10 | 0.02 | 0.01 | 0.01 | 0.01 | 0.06 | 0.03 | 0.07 | 0.01 | 0.05 | 0.04 | 0.01 | 0.02 | 0.08 | 0.02 | 0.02 | 0.00 | 0.02 | 0.05 | 0.03 | 0.06 | 0.01 | 0.15 | 0.03 | 0.02 | 0.04 |
| <3hr | 60min | 0.06 | 0.06 | 0.15 | 0.03 | 0.02 | 0.03 | 0.02 | 0.06 | 0.20 | 0.30 | 0.06 | 0.11 | 0.05 | 0.01 | 0.04 | 0.14 | 0.05 | 0.04 | 0.03 | 0.03 | 0.05 | 0.25 | 0.42 | 0.07 | 0.16 | 0.04 | 0.01 | 0.04 |
| <3hr | 180min | 0.04 | 0.01 | 0.10 | 0.03 | 0.04 | 0.02 | 0.02 | 0.02 | 0.27 | 0.26 | 0.04 | 0.13 | 0.07 | 0.00 | 0.06 | 0.10 | 0.02 | 0.04 | 0.02 | 0.02 | 0.02 | 0.29 | 0.38 | 0.05 | 0.18 | 0.07 | 0.00 | 0.05 |
| >3hr | 15min | 0.04 | 0.06 | 0.10 | 0.02 | 0.02 | 0.01 | 0.01 | 0.04 | 0.02 | 0.05 | 0.02 | 0.04 | 0.01 | 0.01 | 0.02 | 0.09 | 0.02 | 0.02 | 0.00 | 0.02 | 0.04 | 0.02 | 0.05 | 0.01 | 0.15 | 0.01 | 0.01 | 0.02 |
| >3hr | 60min | 0.04 | 0.04 | 0.05 | 0.04 | 0.04 | 0.03 | 0.04 | 0.04 | 0.07 | 0.16 | 0.05 | 0.00 | 0.04 | 0.00 | 0.02 | 0.04 | 0.05 | 0.04 | 0.03 | 0.04 | 0.05 | 0.08 | 0.23 | 0.07 | 0.15 | 0.05 | 0.02 | 0.02 |
| >3hr | 180min | 0.03 | 0.02 | 0.10 | 0.00 | 0.03 | 0.02 | 0.02 | 0.03 | 0.15 | 0.17 | 0.03 | 0.11 | 0.04 | 0.01 | 0.03 | 0.10 | 0.01 | 0.02 | 0.02 | 0.02 | 0.03 | 0.16 | 0.24 | 0.03 | 0.15 | 0.05 | 0.01 | 0.04 |
| | Average | 0.04 | 0.03 | 0.10 | 0.03 | 0.03 | 0.02 | 0.02 | 0.04 | 0.16 | 0.21 | 0.04 | 0.08 | 0.04 | 0.01 | 0.03 | 0.09 | 0.03 | 0.03 | 0.02 | 0.02 | 0.04 | 0.18 | 0.28 | 0.04 | 0.15 | 0.05 | 0.01 | 0.03 |

**Velocity Y**

| Durations | Lead Time | No.Cells | Life.TS | Area | meanPI | medianPI | maxPI | sdPI1 | sdPI2 | GVel | VelX | VelY | Jx | Jy | J.ratio | Phi | Area | meanPi | medPI | maxPI | sdPI1 | sdPI2 | Velg | Velx | Vely | Jx | Jy | Jr | phi |
|---|---|---|---|---|---|---|---|---|---|---|---|---|---|---|---|---|---|---|---|---|---|---|---|---|---|---|---|---|---|
| <1hr | 15min | 0.04 | 0.04 | 0.04 | 0.02 | 0.00 | 0.05 | 0.03 | 0.06 | 0.03 | 0.02 | 0.15 | 0.03 | 0.03 | 0.01 | 0.00 | 0.04 | 0.02 | 0.00 | 0.04 | 0.03 | 0.07 | 0.04 | 0.03 | 0.17 | 0.04 | 0.03 | 0.01 | 0.00 |
| <1hr | 60min | 0.00 | 0.04 | 0.02 | 0.08 | 0.07 | 0.09 | 0.08 | 0.00 | 0.03 | 0.05 | 0.22 | 0.00 | 0.00 | 0.01 | 0.02 | 0.03 | 0.08 | 0.07 | 0.09 | 0.08 | 0.00 | 0.01 | 0.06 | 0.33 | 0.01 | 0.00 | 0.02 | 0.02 |
| <1hr | 180min | 0.03 | 0.08 | 0.05 | 0.02 | 0.01 | 0.03 | 0.03 | 0.05 | 0.00 | 0.04 | 0.27 | 0.07 | 0.01 | 0.02 | 0.02 | 0.06 | 0.02 | 0.01 | 0.03 | 0.03 | 0.06 | 0.01 | 0.05 | 0.41 | 0.08 | 0.02 | 0.01 | 0.02 |
| <3hr | 15min | 0.01 | 0.06 | 0.06 | 0.03 | 0.02 | 0.07 | 0.04 | 0.07 | 0.01 | 0.01 | 0.05 | 0.03 | 0.05 | 0.01 | 0.00 | 0.05 | 0.03 | 0.01 | 0.06 | 0.04 | 0.06 | 0.02 | 0.00 | 0.04 | 0.02 | 0.04 | 0.00 | 0.00 |
| <3hr | 60min | 0.01 | 0.06 | 0.02 | 0.04 | 0.03 | 0.10 | 0.06 | 0.03 | 0.01 | 0.06 | 0.18 | 0.02 | 0.05 | 0.01 | 0.01 | 0.00 | 0.04 | 0.03 | 0.11 | 0.06 | 0.03 | 0.00 | 0.07 | 0.26 | 0.01 | 0.04 | 0.01 | 0.01 |
| <3hr | 180min | 0.00 | 0.06 | 0.03 | 0.00 | 0.00 | 0.00 | 0.00 | 0.06 | 0.01 | 0.03 | 0.22 | 0.07 | 0.04 | 0.01 | 0.01 | 0.04 | 0.01 | 0.01 | 0.00 | 0.00 | 0.07 | 0.00 | 0.04 | 0.33 | 0.07 | 0.05 | 0.01 | 0.01 |
| >3hr | 15min | 0.01 | 0.07 | 0.03 | 0.00 | 0.01 | 0.03 | 0.01 | 0.03 | 0.01 | 0.04 | 0.00 | 0.00 | 0.01 | 0.01 | 0.01 | 0.02 | 0.00 | 0.01 | 0.02 | 0.00 | 0.02 | 0.01 | 0.03 | 0.00 | 0.00 | 0.00 | 0.01 | 0.03 |
| >3hr | 60min | 0.03 | 0.02 | 0.02 | 0.01 | 0.01 | 0.03 | 0.01 | 0.07 | 0.00 | 0.04 | 0.09 | 0.09 | 0.08 | 0.01 | 0.01 | 0.04 | 0.01 | 0.01 | 0.03 | 0.00 | 0.09 | 0.01 | 0.07 | 0.16 | 0.10 | 0.10 | 0.01 | 0.00 |
| >3hr | 180min | 0.00 | 0.01 | 0.01 | 0.05 | 0.04 | 0.04 | 0.04 | 0.04 | 0.01 | 0.02 | 0.14 | 0.09 | 0.08 | 0.01 | 0.01 | 0.02 | 0.01 | 0.01 | 0.05 | 0.04 | 0.05 | 0.02 | 0.01 | 0.22 | 0.10 | 0.08 | 0.01 | 0.00 |
| | Average | 0.01 | 0.05 | 0.03 | 0.03 | 0.02 | 0.05 | 0.03 | 0.05 | 0.01 | 0.03 | 0.15 | 0.04 | 0.04 | 0.01 | 0.01 | 0.03 | 0.03 | 0.02 | 0.05 | 0.03 | 0.05 | 0.01 | 0.04 | 0.22 | 0.05 | 0.04 | 0.01 | 0.01 |

**Duration**

| Durations | Lead Time | No.Cells | Life.TS | Area | meanPI | medianPI | maxPI | sdPI1 | sdPI2 | GVel | VelX | VelY | Jx | Jy | J.ratio | Phi | Area | meanPi | medPI | maxPI | sdPI1 | sdPI2 | Velg | Velx | Vely | Jx | Jy | Jr | phi |
|---|---|---|---|---|---|---|---|---|---|---|---|---|---|---|---|---|---|---|---|---|---|---|---|---|---|---|---|---|---|
| | Dur <1hr | 0.06 | 0.16 | 0.31 | 0.02 | 0.02 | 0.04 | 0.00 | 0.22 | 0.03 | 0.03 | 0.00 | 0.19 | 0.24 | 0.00 | 0.03 | 0.30 | 0.02 | 0.03 | 0.03 | 0.01 | 0.21 | 0.03 | 0.03 | 0.01 | 0.19 | 0.25 | 0.01 | 0.04 |
| | Dur <3hr | 0.00 | 0.16 | 0.35 | 0.10 | 0.05 | 0.04 | 0.10 | 0.22 | 0.08 | 0.02 | 0.04 | 0.23 | 0.21 | 0.02 | 0.11 | 0.32 | 0.12 | 0.07 | 0.05 | 0.11 | 0.22 | 0.08 | 0.03 | 0.07 | 0.23 | 0.21 | 0.05 | 0.15 |
| | Dur >3hr | 0.07 | 0.02 | 0.43 | 0.20 | 0.11 | 0.18 | 0.21 | 0.21 | 0.15 | 0.04 | 0.04 | 0.25 | 0.16 | 0.14 | 0.01 | 0.40 | 0.22 | 0.14 | 0.20 | 0.23 | 0.20 | 0.18 | 0.07 | 0.07 | 0.23 | 0.14 | 0.18 | 0.01 |
| | Average | 0.04 | 0.11 | 0.36 | 0.11 | 0.06 | 0.09 | 0.10 | 0.22 | 0.09 | 0.03 | 0.03 | 0.22 | 0.20 | 0.05 | 0.05 | 0.34 | 0.12 | 0.08 | 0.09 | 0.12 | 0.21 | 0.10 | 0.04 | 0.05 | 0.22 | 0.20 | 0.08 | 0.07 |

**Appendix 8.2** *Obtained PIC predictors weight for each target variable, lead time and storm groups. The last row at each target variable (average values) are the predictors weights shown in Table 3.*

**Area**

| Durations | Lead Time | Cell | Life | A | avePI | medPI | maxPI | sdPI1 | sdPI2 | Vg | Vx | Vy | Jx | Jy | Jr | Φ | A | avePI | medPI | maxPI | sdPI1 | sdPI2 | Vg | Vx | Vy | Jx | Jy | Jr | phi |
|---|---|---|---|---|---|---|---|---|---|---|---|---|---|---|---|---|---|---|---|---|---|---|---|---|---|---|---|---|---|
| <1hr | 15min | 0.00 | 0.00 | 0.00 | 0.00 | 0.00 | 0.00 | 0.00 | 0.00 | 0.00 | 0.00 | 0.00 | 0.00 | 0.00 | 0.00 | 0.00 | 0.00 | 0.00 | 0.00 | 0.00 | 0.00 | 0.00 | 0.00 | 0.00 | 0.00 | 0.00 | 0.00 | 1.00 | 0.00 |
| | 60min | 0.00 | 0.00 | 0.00 | 0.00 | 0.00 | 0.00 | 0.00 | 0.00 | 0.00 | 0.00 | 0.00 | 0.00 | 0.00 | 0.00 | 0.00 | 0.00 | 0.00 | 0.00 | 0.00 | 0.00 | 0.00 | 0.00 | 0.00 | 0.00 | 0.00 | 0.00 | 1.00 | 0.00 |
| | 180min | 0.00 | 0.00 | 0.00 | 0.00 | 0.00 | 0.00 | 0.00 | 0.00 | 0.00 | 0.00 | 0.00 | 0.00 | 0.00 | 0.00 | 0.00 | 0.00 | 0.00 | 0.00 | 0.00 | 0.00 | 0.00 | 0.00 | 0.00 | 0.00 | 0.00 | 0.00 | 0.00 | 0.00 |
| <3hr | 15min | 0.00 | 0.00 | 0.00 | 0.00 | 0.00 | 0.00 | 0.00 | 0.00 | 0.00 | 0.00 | 0.00 | 0.00 | 0.00 | 0.00 | 0.00 | 0.00 | 0.00 | 0.00 | 0.00 | 0.00 | 0.00 | 1.00 | 0.00 | 0.00 | 0.00 | 0.00 | 0.00 | 0.00 |
| | 60min | 0.00 | 0.00 | 0.00 | 0.00 | 0.00 | 0.00 | 0.00 | 0.00 | 0.00 | 0.00 | 0.00 | 0.00 | 0.00 | 0.00 | 0.00 | 0.00 | 0.00 | 0.00 | 0.00 | 0.00 | 0.00 | 1.00 | 0.00 | 0.00 | 0.00 | 0.00 | 0.00 | 0.00 |
| | 180min | 0.00 | 0.00 | 0.00 | 0.00 | 0.00 | 0.00 | 0.00 | 0.00 | 0.00 | 0.00 | 0.00 | 0.00 | 0.00 | 0.00 | 0.00 | 0.00 | 0.00 | 0.00 | 0.00 | 0.00 | 0.00 | 1.00 | 0.00 | 0.00 | 0.00 | 0.00 | 0.00 | 0.00 |
| >3hr | 15min | 0.00 | 0.10 | 0.25 | 0.00 | 0.00 | 0.00 | 0.00 | 0.00 | 0.57 | 0.00 | 0.00 | 0.00 | 0.00 | 0.00 | 0.00 | 0.08 | 0.00 | 0.00 | 0.00 | 0.00 | 0.00 | 0.00 | 0.00 | 0.00 | 0.00 | 0.00 | 0.00 | 0.00 |
| | 60min | 0.00 | 0.29 | 0.67 | 0.00 | 0.00 | 0.00 | 0.00 | 0.00 | 0.66 | 0.00 | 0.00 | 0.00 | 0.00 | 0.00 | 0.00 | 0.00 | 0.00 | 0.00 | 0.00 | 0.00 | 0.00 | 0.00 | 0.33 | 0.00 | 0.00 | 0.00 | 0.00 | 0.00 |
| | 180min | 0.00 | 0.30 | 0.40 | 0.00 | 0.00 | 0.00 | 0.00 | 0.00 | 0.72 | 0.00 | 0.00 | 0.00 | 0.00 | 0.00 | 0.00 | 0.00 | 0.00 | 0.00 | 0.00 | 0.00 | 0.00 | 0.00 | 0.28 | 0.00 | 0.00 | 0.00 | 0.00 | 0.00 |
| | Average | 0.00 | 0.08 | 0.15 | 0.00 | 0.00 | 0.00 | 0.00 | 0.00 | 0.22 | 0.00 | 0.00 | 0.00 | 0.00 | 0.00 | 0.00 | 0.01 | 0.00 | 0.00 | 0.00 | 0.00 | 0.00 | 0.33 | 0.00 | 0.07 | 0.00 | 0.00 | 0.33 | 0.00 |

**Intensity**

| Durations | Lead Time | No.Cells | Life.TS | Area | meanPI | medianPI | maxPI | sdPI1 | sdPI2 | GVel | VelX | VelY | Jx | Jy | J.ratio | Phi | Area | meanPi | medPI | maxPI | sdPI1 | sdPI2 | Velg | Velx | Vely | Jx | Jy | Jr | phi |
|---|---|---|---|---|---|---|---|---|---|---|---|---|---|---|---|---|---|---|---|---|---|---|---|---|---|---|---|---|---|
| <1hr | 15min | 0.00 | 0.00 | 0.00 | 0.00 | 0.00 | 0.00 | 0.00 | 0.00 | 0.00 | 0.00 | 0.00 | 0.00 | 0.00 | 0.00 | 0.00 | 0.00 | 0.00 | 0.00 | 0.00 | 0.00 | 0.00 | 0.00 | 0.00 | 0.00 | 0.00 | 0.00 | 0.00 | 0.00 |
| | 60min | 0.00 | 0.00 | 0.00 | 0.00 | 0.00 | 0.00 | 0.00 | 0.00 | 0.00 | 0.00 | 0.00 | 0.00 | 0.00 | 0.00 | 0.00 | 0.00 | 0.00 | 0.00 | 0.00 | 0.00 | 0.00 | 0.00 | 0.00 | 0.00 | 0.00 | 0.00 | 0.00 | 0.00 |
| | 180min | 0.00 | 0.00 | 0.00 | 0.00 | 0.00 | 0.00 | 0.00 | 0.00 | 0.00 | 0.00 | 0.00 | 0.00 | 0.00 | 0.00 | 0.00 | 0.00 | 0.00 | 0.00 | 0.00 | 0.00 | 0.00 | 0.00 | 0.00 | 0.00 | 0.00 | 0.00 | 0.00 | 0.00 |
| <3hr | 15min | 0.00 | 0.00 | 0.00 | 0.00 | 0.00 | 0.00 | 0.00 | 0.00 | 0.00 | 0.00 | 0.00 | 1.00 | 0.00 | 0.00 | 0.00 | 0.00 | 0.00 | 0.00 | 0.00 | 0.00 | 0.00 | 0.00 | 0.00 | 0.00 | 0.00 | 0.00 | 0.00 | 0.00 |
| | 60min | 0.00 | 0.00 | 0.00 | 0.00 | 0.00 | 0.00 | 0.00 | 0.00 | 0.00 | 0.00 | 0.00 | 1.00 | 0.00 | 0.00 | 0.00 | 0.00 | 0.00 | 0.00 | 0.00 | 0.00 | 0.00 | 0.00 | 0.00 | 0.00 | 0.00 | 0.00 | 0.00 | 0.00 |
| | 180min | 0.00 | 0.00 | 0.00 | 0.00 | 0.00 | 0.00 | 0.00 | 0.00 | 0.00 | 0.00 | 0.00 | 1.00 | 0.00 | 0.00 | 0.00 | 0.00 | 0.00 | 0.00 | 0.00 | 0.00 | 0.00 | 0.00 | 0.00 | 0.00 | 0.00 | 0.00 | 0.00 | 0.00 |
| >3hr | 15min | 0.00 | 0.00 | 0.00 | 0.00 | 0.00 | 0.00 | 0.00 | 0.00 | 0.00 | 0.00 | 0.00 | 1.00 | 0.00 | 0.00 | 0.00 | 0.00 | 0.00 | 0.00 | 0.00 | 0.00 | 0.00 | 0.00 | 0.00 | 0.00 | 0.00 | 0.00 | 0.00 | 0.00 |
| | 60min | 0.00 | 0.00 | 0.00 | 0.00 | 0.00 | 0.00 | 0.00 | 0.00 | 0.00 | 0.00 | 0.00 | 1.00 | 0.00 | 0.00 | 0.00 | 0.00 | 0.00 | 0.00 | 0.00 | 0.00 | 0.00 | 0.00 | 0.00 | 0.00 | 0.00 | 0.00 | 0.00 | 0.00 |
| | 180min | 0.00 | 0.00 | 0.00 | 0.00 | 0.00 | 0.00 | 0.00 | 0.00 | 0.00 | 0.00 | 0.00 | 1.00 | 0.00 | 0.00 | 0.00 | 0.00 | 0.00 | 0.00 | 0.00 | 0.00 | 0.00 | 0.00 | 0.00 | 0.00 | 0.00 | 0.00 | 0.00 | 0.00 |
| | Average | 0.00 | 0.00 | 0.00 | 0.00 | 0.00 | 0.00 | 0.00 | 0.00 | 0.00 | 0.00 | 0.00 | 1.00 | 0.00 | 0.00 | 0.00 | 0.00 | 0.00 | 0.00 | 0.00 | 0.00 | 0.00 | 0.00 | 0.00 | 0.00 | 0.00 | 0.00 | 0.00 | 0.00 |

**Velocity X**

| Durations | Lead Time | No.Cells | Life.TS | Area | meanPI | medianPI | maxPI | sdPI1 | sdPI2 | GVel | VelX | VelY | Jx | Jy | J.ratio | Phi | Area | meanPi | medPI | maxPI | sdPI1 | sdPI2 | Velg | Velx | Vely | Jx | Jy | Jr | phi |
|---|---|---|---|---|---|---|---|---|---|---|---|---|---|---|---|---|---|---|---|---|---|---|---|---|---|---|---|---|---|
| <1hr | 15min | 0.00 | 0.00 | 0.00 | 0.00 | 0.00 | 0.00 | 0.00 | 0.00 | 0.00 | 0.00 | 0.00 | 0.00 | 0.00 | 0.00 | 0.00 | 0.00 | 0.00 | 0.00 | 0.00 | 0.00 | 0.00 | 0.00 | 0.00 | 0.00 | 0.00 | 0.00 | 0.00 | 0.00 |
| | 60min | 0.00 | 0.00 | 0.00 | 0.00 | 0.00 | 0.00 | 0.00 | 0.00 | 0.00 | 0.00 | 0.00 | 0.00 | 0.00 | 0.00 | 0.00 | 0.00 | 0.00 | 0.00 | 0.00 | 0.00 | 0.00 | 0.00 | 0.00 | 0.00 | 0.00 | 0.00 | 0.00 | 0.00 |
| | 180min | 0.00 | 0.00 | 0.00 | 0.00 | 0.00 | 0.00 | 0.00 | 0.00 | 0.00 | 0.00 | 0.00 | 0.00 | 0.00 | 0.00 | 0.00 | 0.00 | 0.00 | 0.00 | 0.00 | 0.00 | 0.00 | 0.00 | 0.00 | 0.00 | 0.00 | 0.00 | 0.00 | 0.00 |
| <3hr | 15min | 0.00 | 0.00 | 0.00 | 0.00 | 0.00 | 0.00 | 0.00 | 0.00 | 0.00 | 0.00 | 0.00 | 0.00 | 0.00 | 0.00 | 0.00 | 0.00 | 0.00 | 0.00 | 0.00 | 0.00 | 0.00 | 0.00 | 1.00 | 0.00 | 0.00 | 0.00 | 0.00 | 0.00 |
| | 60min | 0.00 | 0.00 | 0.00 | 0.00 | 0.00 | 0.00 | 0.00 | 0.00 | 0.00 | 0.00 | 0.00 | 0.00 | 0.00 | 0.00 | 0.00 | 0.00 | 0.00 | 0.00 | 0.00 | 0.00 | 0.00 | 0.00 | 1.00 | 0.00 | 0.00 | 0.00 | 0.00 | 0.00 |
| | 180min | 0.00 | 0.00 | 0.00 | 0.00 | 0.00 | 0.00 | 0.00 | 0.00 | 0.00 | 0.00 | 0.00 | 0.00 | 0.00 | 0.00 | 0.00 | 0.00 | 0.00 | 0.00 | 0.00 | 0.00 | 0.00 | 0.00 | 1.00 | 0.00 | 0.00 | 0.00 | 0.00 | 0.00 |
| >3hr | 15min | 0.00 | 0.00 | 0.00 | 0.00 | 0.00 | 0.00 | 0.00 | 0.00 | 0.00 | 0.00 | 0.00 | 0.00 | 0.00 | 0.00 | 0.00 | 0.00 | 0.00 | 0.00 | 0.00 | 0.00 | 0.00 | 0.00 | 0.00 | 0.00 | 0.00 | 0.00 | 0.00 | 0.00 |
| | 60min | 0.00 | 0.00 | 0.00 | 0.00 | 0.00 | 0.00 | 0.00 | 0.00 | 0.00 | 0.00 | 0.00 | 0.00 | 0.00 | 0.00 | 0.00 | 0.00 | 0.00 | 0.00 | 0.00 | 0.00 | 0.00 | 0.00 | 0.00 | 0.00 | 0.00 | 0.00 | 0.00 | 0.00 |
| | 180min | 0.00 | 0.00 | 0.00 | 0.00 | 0.00 | 0.00 | 0.00 | 0.00 | 0.00 | 0.00 | 0.00 | 0.00 | 0.00 | 0.00 | 0.00 | 0.00 | 0.00 | 0.00 | 0.00 | 0.00 | 0.00 | 0.00 | 0.00 | 0.00 | 0.00 | 0.00 | 0.00 | 0.00 |
| | Average | 0.00 | 0.00 | 0.00 | 0.00 | 0.00 | 0.00 | 0.00 | 0.00 | 0.00 | 0.00 | 0.00 | 0.00 | 0.00 | 0.00 | 0.00 | 0.00 | 0.00 | 0.00 | 0.00 | 0.00 | 0.00 | 0.00 | 1.00 | 0.00 | 0.00 | 0.00 | 0.00 | 0.00 |

**Velocity Y**

| Durations | Lead Time | No.Cells | Life.TS | Area | meanPI | medianPI | maxPI | sdPI1 | sdPI2 | GVel | VelX | VelY | Jx | Jy | J.ratio | Phi | Area | meanPi | medPI | maxPI | sdPI1 | sdPI2 | Velg | Velx | Vely | Jx | Jy | Jr | phi |
|---|---|---|---|---|---|---|---|---|---|---|---|---|---|---|---|---|---|---|---|---|---|---|---|---|---|---|---|---|---|
| <1hr | 15min | 0.00 | 0.00 | 0.00 | 0.00 | 0.00 | 0.00 | 0.00 | 0.00 | 0.00 | 0.00 | 0.00 | 0.00 | 0.00 | 0.00 | 0.00 | 0.00 | 0.00 | 0.00 | 0.00 | 0.00 | 0.00 | 0.00 | 0.00 | 0.00 | 0.00 | 0.00 | 0.00 | 0.00 |
| | 60min | 0.00 | 0.00 | 0.00 | 0.00 | 0.00 | 0.00 | 0.00 | 0.00 | 0.00 | 0.00 | 0.00 | 0.00 | 0.00 | 0.00 | 0.00 | 0.00 | 0.00 | 0.00 | 0.00 | 0.00 | 0.00 | 0.00 | 0.00 | 0.00 | 0.00 | 0.00 | 0.00 | 0.00 |
| | 180min | 0.00 | 0.00 | 0.00 | 0.00 | 0.00 | 0.00 | 0.00 | 0.00 | 0.00 | 0.00 | 0.00 | 0.00 | 0.00 | 0.00 | 0.00 | 0.00 | 0.00 | 0.00 | 0.00 | 0.00 | 0.00 | 0.00 | 0.00 | 0.00 | 0.00 | 0.00 | 0.00 | 0.00 |
| <3hr | 15min | 0.00 | 0.00 | 0.00 | 0.00 | 0.00 | 0.00 | 0.00 | 0.00 | 0.00 | 0.00 | 0.00 | 0.00 | 0.00 | 0.00 | 0.00 | 0.00 | 0.00 | 0.00 | 0.00 | 0.00 | 0.00 | 0.00 | 0.00 | 0.00 | 0.00 | 0.00 | 0.00 | 0.00 |
| | 60min | 0.00 | 0.00 | 0.00 | 0.00 | 0.00 | 0.00 | 0.00 | 0.00 | 0.00 | 0.00 | 0.00 | 0.00 | 0.00 | 0.00 | 0.00 | 0.00 | 0.00 | 0.00 | 0.00 | 0.00 | 0.00 | 0.00 | 0.00 | 0.00 | 0.00 | 0.00 | 0.00 | 0.00 |
| | 180min | 0.00 | 0.00 | 0.00 | 0.00 | 0.00 | 0.00 | 0.00 | 0.00 | 0.00 | 0.00 | 0.00 | 0.00 | 0.00 | 0.00 | 0.00 | 0.00 | 0.00 | 0.00 | 0.00 | 0.00 | 0.00 | 0.00 | 0.00 | 0.00 | 0.00 | 0.00 | 0.00 | 0.00 |
| >3hr | 15min | 0.00 | 0.00 | 0.00 | 0.00 | 0.00 | 0.00 | 0.00 | 0.00 | 0.00 | 0.00 | 0.00 | 0.00 | 0.00 | 0.00 | 0.00 | 0.00 | 0.00 | 0.00 | 0.00 | 0.00 | 0.00 | 0.00 | 0.00 | 0.00 | 1.00 | 0.00 | 0.00 | 0.00 |
| | 60min | 0.00 | 0.00 | 0.00 | 0.00 | 0.00 | 0.00 | 0.00 | 0.00 | 0.00 | 0.00 | 0.00 | 0.00 | 0.00 | 0.00 | 0.00 | 0.00 | 0.00 | 0.00 | 0.00 | 0.00 | 0.00 | 0.00 | 0.00 | 0.00 | 1.00 | 0.00 | 0.00 | 0.00 |
| | 180min | 0.00 | 0.00 | 0.00 | 0.00 | 0.00 | 0.00 | 0.00 | 0.00 | 0.00 | 0.00 | 0.00 | 0.00 | 0.00 | 0.00 | 0.00 | 0.00 | 0.00 | 0.00 | 0.00 | 0.00 | 0.00 | 0.00 | 0.00 | 0.00 | 1.00 | 0.00 | 0.00 | 0.00 |
| | Average | 0.00 | 0.00 | 0.00 | 0.00 | 0.00 | 0.00 | 0.00 | 0.00 | 0.00 | 0.00 | 0.00 | 0.00 | 0.00 | 0.00 | 0.00 | 0.00 | 0.00 | 0.00 | 0.00 | 0.00 | 0.00 | 0.00 | 0.00 | 0.00 | 1.00 | 0.00 | 0.00 | 0.00 |

**Duration**

| Durations | Lead Time | No.Cells | Life.TS | Area | meanPI | medianPI | maxPI | sdPI1 | sdPI2 | GVel | VelX | VelY | Jx | Jy | J.ratio | Phi | Area | meanPi | medPI | maxPI | sdPI1 | sdPI2 | Velg | Velx | Vely | Jx | Jy | Jr | phi |
|---|---|---|---|---|---|---|---|---|---|---|---|---|---|---|---|---|---|---|---|---|---|---|---|---|---|---|---|---|---|
| Dur <1hr | | 0.00 | 0.00 | 0.00 | 0.00 | 0.00 | 0.00 | 0.00 | 0.00 | 0.00 | 0.00 | 0.00 | 0.00 | 0.00 | 0.00 | 0.00 | 0.00 | 0.00 | 0.00 | 0.00 | 0.00 | 0.00 | 0.00 | 0.00 | 0.00 | 0.00 | 0.00 | 1.00 | 0.00 |
| Dur <3hr | | 0.00 | 0.00 | 0.00 | 0.00 | 0.00 | 0.00 | 0.00 | 0.00 | 0.00 | 0.00 | 0.00 | 0.00 | 0.00 | 0.00 | 0.00 | 0.00 | 0.00 | 0.00 | 0.00 | 0.00 | 0.00 | 0.00 | 1.00 | 0.00 | 0.00 | 0.00 | 0.00 | 0.00 |
| Dur >3hr | | 0.00 | 0.45 | 0.40 | 0.00 | 0.00 | 0.00 | 0.00 | 0.00 | 0.72 | 0.00 | 0.00 | 0.00 | 0.00 | 0.00 | 0.00 | 0.00 | 0.00 | 0.00 | 0.00 | 0.00 | 0.00 | 0.00 | 0.00 | 0.00 | 0.00 | 0.00 | 0.33 | 0.00 |
| | Average | 0.00 | 0.15 | 0.13 | 0.00 | 0.00 | 0.00 | 0.00 | 0.00 | 0.24 | 0.00 | 0.00 | 0.00 | 0.00 | 0.00 | 0.00 | 0.00 | 0.00 | 0.00 | 0.00 | 0.00 | 0.00 | 0.00 | 0.33 | 0.00 | 0.00 | 0.00 | 0.11 | 0.33 | 0.00 |

---

## Author Response (AR1)

Dear Reviewer(s) and Editor,

We would like to thank once again the reviewers for their feedback that has improved the manuscript. Following their comments, the following changes were done in the updated version of the manuscript:

- The Title of the manuscript was changed to: "Improving radar-based rainfall nowcast by a nearest neighbour approach: Part I Storm Characteristics"
- The introduction to the topic and the literature review has been updated to fulfil the comments from two reviewers to:
  - to have a better introduction on the topic: the value of the rainfall forecasting, and why specifically the use of the radar-based nowcasting (Line 34 -59)
  - to give a better overview of the radar based nowcast (both field-based and object based), and the choice of object based is justified (Lines 60-73)
  - to discuss more about the life cycle characteristics of convective storms (Lines 102-112)
  - to discuss the analogous approach implemented in the field based nowcast is and the novelty of the proposed methodology (Lines 128-138)
  - to discuss the main focus of the proposed methodology (Lines 141-151)
- Figure 2 was updated to include more information about the location of the study area in Germany (Figure 2-a), and better representing the study area in Figure 2-b. The description was also updated to meet the comments of the reviewers.
- More information is given about the merging method between radar and rain gauge done prior to the investigation following the comments of reviewer 1 (Lines 170-173).
- More information is given about the selection of the events that form the database following the comments of reviewer 1 (Lines 174-181).
- Figure 3 was introduced in the manuscript to explain the main concepts in this study, the concept of the "leave-one-event-out" cross-validations, and to illustrate the work flow in this study (together with potential sources of information leakage), following mainly the comments from the reviewer 3.
- Describing shortly the choice of the low identification thresholds for the storms following the comments of reviewer 2 (Lines 189-191).
- Describing better the tracking algorithm, and the obtained characteristics from the database (Lines 203-306 and Lines 212-216).
- Figure 4 has been updated, so all information is visible, the intensities are updated to maximum observed as mm/h, the merges are as well included.
- The previous limiting value of 0.5 for selecting the neighbours has been removed, and all the results now include all the neighbours that are closer to the "to-be-nowcasted" storm (previously line 261).
- The description of the Partial Information Correlation has been improved (Lines 285,295).
- How the ensemble members are chosen is discussed in Lines 233-234.
- The mean absolute error (MAE) instead of the average error was used for the optimization and for the assessment of the deterministic nowcast following the comments of Reviewer 2 and 1 (Line 417). The respective Figures 7-10 are changed accordingly.
- Following the comments from reviewer 2, the performance criteria are computed as averages per event (instead of averages for storm groups). This affects the Figure 7-13. The Equation 8, 9 and 10 were updated accordingly together with their description. Please notice that the Figures are new (updated from the response to the reviewers) as the performance is now calculated for each event first and then the median over all events is taken. In The figures 7-

13 we show only the median over the events, thus avoiding the confusion between mean and median (as stated by the three reviewers from the first version of the manuscript).

- The Continuous Rank Probability Score (CRPS) was introduced to asses the performance of the probabilistic nowcast (to enable also the direct comparison with the deterministic one) following the suggestion of Reviewer 2 and comments of Reviewer 1 (Lines 382-289). Figures 11-13 were updated accordingly.
- Appendix 8.1 and 8.2 were added to the manuscript to explain better how the results of the predictors weights were achieved (Line 401). Additionally, the Lines 303-308 were added to explain better how the predictors weights were calculated (following the comments of the reviewer 2 and 3).
- The information leakage on the computation weights is discussed in Lines 452-465 and Appendix 8.3 was added to the manuscript following the comments of reviewer 3.
- The new Figure 8 regarding the optimization of the deterministic k-NN, and the following description and choice of k was updated (Lines 472-488). Additionally, Table 4 is added.
- The information leakage in the optimization of the deterministic k-NN is discussed in Lines 488-495 (following the comments of reviewer 3).
- Table 5 is added in the result section of the deterministic k-NN to explain better where the number (%) in the text are coming from (following the comments of reviewer 2).
- As the Figures 9-10, have been updated to visualise the MAE per event, the discussion of the deterministic 4-NN results has been updated accordingly.
- Section 4.4 showing the results of the probabilistic 30NNs nowcast has been changed entirely since the performance criteria have been changed to the CRPS. A comparison with the deterministic 4-NNs is also shown here to understand which application is better suited for the nowcast of storm characteristics. An outlook paragraph is included in this section to discuss the advantages of the probabilistic approach, and the possible integration of the 30NNs with the Lagrangian persistence following the comments of mainly reviewer 2(Lines 636-645).
- Section 4.5 was introduced together with the Figures 14 and 15 to discuss the role of the unmatched storms in an operational nowcast, and how well the proposed methodology can forecast these storms, following the comments of reviewer 1 (Lines 662-674).
- The choice of other physical predictors was discussed in the outlook of the manuscript and will be subjected to future works, following the comments of reviewer 2 (Lines 731-736).
- The benefit of nowcasting storm characteristics and the future continuation of this work were discussed in Lines 646-660 and Lines 736-741 following the comments of reviewer 3.
- The benefit of the probabilistic nowcast was discussed in Lines 719-726 following the comments of reviewer 3.
- Throughout the text the "death" term was substituted with "dissipation term", "birth" with "initialization", "training" with "optimization", and only the term "nowcast time" was used to refer the time when the nowcast was issued in reference to the storm initiation. Moreover, the term "object-based" instead of "object-oriented" was used in the manuscript to avoid confusion with the programming term (introduced so at Line 59).
- In the respective Figures, the grey lines are a bit darker (for better visualization) and the labels size in the graphs has been increased so it is better distinguishable (following the comments of reviewer 1).
- The problem with the term "optimization of the k" number for the nearest neighbour" is addressed in Lines 488-492 following the comments of reviewer 3. Moreover, since the nearest neighbour is not a proper learner as an artificial neural network (for instance), the term learning has been avoided in the manuscript.

Some of your suggestions or comments that have not been included directly in the manuscript are:

- 1. Visualization of an extreme event from reviewer1. It was not included because the nearest neighbour is not in its final form. The storm characteristics predictions are still to be integrated with the rainfall structure at fine temporal and spatial scales (1km2 and 5min). Moreover, all the events selected are not normal ones, as they have been selected for urban flood purposes. Also, reviewer 2 recommended to explain better the limitations of the k-NN methods for predicting extreme behaviour and how this behaviour is underestimated in case of extreme events (and how the sample size is affecting this prediction). But this is already indirectly included in the calculation of the results, as the dataset is based on extreme events. Lastly, the paper is already too long, and we would like to include this example in the follow up paper. However, if the reviewers think this is very relevant to the study, we could include a small section 4.6 discussion a very extreme event and the data size influence on the performance of the probabilistic 30NNs.
- 2. No comments were made inside the manuscript about the size of the database, and the processing time, because, as already said, the kNN method its still not introduced in the final form. I would prefer to discuss these technical issues once the full kNN is operational, and also to mention what are the running time and memory depending on different cases: for instance, if a single storm or many storms are simultaneously present in the radar image.
- 3. The database and the script for the kNN methods are not yet publish. I will do my best to prepare everything and upload them before *(only if)* the paper is published.

with kind regards, Bora Shehu

---

## Referee Report (RR1)

Review of: Improving radar-based rainfall nowcast by a nearest neighbour approach: Part I – Storm Characteristics by Bora Shehu and Uwe Haberlandt

**Ruben Imhoff**

Ruben.Imhoff@deltares.nl

December 9, 2021

Dear authors, dear Bora and Uwe,

Thank you for the responses to my and the other reviewer's comments. You have done a tremendous amount of work and the manuscript has improved considerably. I only have a few minor suggestions, see below in blue.

I am looking forward to seeing the paper in its final form.

Sincerely,

Ruben Imhoff

**Responses on authors comments**

"Visualization of an extreme event from reviewer1. It was not included because the nearest neighbour is not in its final form. The storm characteristics predictions are still to be integrated with the rainfall structure at fine temporal and spatial scales (1km2 and 5min). Moreover, all the events selected are not normal ones, as they have been selected for urban flood purposes. Also, reviewer 2 recommended to explain better the limitations of the k-NN methods for predicting extreme behaviour and how this behaviour is underestimated in case of extreme events (and how the sample size is affecting this prediction). But this is already indirectly included in the calculation of the results, as the dataset is based on extreme events. Lastly, the paper is already too long, and we would like to include this example in the follow up paper. However, if the reviewers think this is very relevant to the study, we could include a small section 4.6 discussion a very extreme event and the data size influence on the performance of the probabilistic 30NNs."

**I do like the idea of a part 2 of this work, so it makes sense to me to include those examples then and I'm looking forward to seeing them then.**

"No comments were made inside the manuscript about the size of the database, and the processing time, because, as already said, the kNN method its still not introduced in the final form. I would prefer to discuss these technical issues once the full kNN is operational, and also to mention what are the running time and memory depending on different cases: for instance, if a single storm or many storms are simultaneously present in the radar image." The information the authors provided on the size of the database, run times, etc. in the author response, was valuable in my opinion. That said, I'm okay with including it in the part 2 paper of this work.

As response to Lines 146 - 152: How is the storm duration defined? - "The storm is a group of radar pixels that have an intensity higher than a fixed threshold. The duration of the storm is then the lifetime of the radar pixels group as dictated by the threshold used to recognize them and the tracking algorithm that decides if the same storm is observed at continuous time steps."

The authors provide useful information in line 200 and further, but can I ask to also include the given response (see above) in the text? Perhaps something is already there and I've missed it, but if not, I think it is useful information to include.

As response to Lines 367 – 369: Do you have any idea why the result is different for the Total Lifetime? – "My understanding, is that the duration is an easier target to be analysed, which means the values are not zero (because we consider here the total lifetime) and its distribution is not as heavy tailed as the distribution of the other variables. The other variables, depending on the lead time, have more zeros included and have an asymptotic density function. On personal experience, when zeros are not present (or at least not in the frequency of the 4 variables here) the PIC is able to represent quite well the important predictors."

This response may be an interesting discussion point to add to the paper.

**Specific comments**

Lines 70 - 71: "the focus in this study is on object-based nowcast as they are more convenient for convective storms that typically cause urban pluvial floods.": I agree, but feel free to even mention that field-based nowcasting approaches generally have trouble capturing and forecasting this well, which would stress even more the reason to focus on object-based nowcasting methods.

**Figure font sizes and overall size:**

The authors have, indeed, increased the font size of the figures. For some figures, the font size can still be considered somewhat small. Nevertheless, this is also a process of typesetting later on. A suggestion that I would like to make, is to sometimes consider placing the figures (e.g., but not limited to, figures 7 and 8) on multiple rows. That would allow for using larger figures and that would solve the problem, too.

**Technical corrections**

Title - "Improving radar-based rainfall nowcast by a nearest neighbour approach: Part I – Storm Characteristics": I would suggest changing nowcast in nowcasting.

Lines 59 – 60 "object-oriented nowcast (herein as object-based to avoid the confusion with the programming term) and field-based nowcast": in both cases, I think 'nowcast' should be nowcasting

Line 64 "an unique": a unique

---

## Referee Report (RR2)

**Review of "Improving radar-based rainfall nowcast by a nearest neighbour approach: Part I – Storm Characteristics" by Bora Shehu and Uwe Haberlandt**

Seppo Pulkkinen
seppo.pulkkinen@fmi.fi
January 6 2022

**Summary**

I acknowledge that the authors have adequately addressed my previous concerns. I still have a list of mostly minor technical issues related to the presentation of the results. Once these have been addressed, I'm willing to recommend the manuscript for publication.

**General comments**

There is no description about how merging and splitting of storm cells are handled. The way this is done can have a significant impact on the results.

- Are all merged/splitted cells included in the considered storm tracks (so that the storm tracks form a tree and a storm object at a given time may consist of multiple cells)?
- Or do you only consider storms that do not merge or split during their lifetime?

**Specific comments**

- Manuscript title: since you now have "Part I" in the title, you should have more explicit discussion in the conclusions what would "Part II", or even "Part III" include.
- Line 7: erratic ← unpredictable?
- Lines 44 and 57: I'm not sure if these claims are completely true for state of the art rapid-update limited-area NWP models that could be applicable to urban-scale nowcasting. Are there any more recent references about this topic?
- Line 46: You could add that the inability to capture the spatial structure of rainfall is due to the sparsity of the existing rain gauge networks.
- Line 67: I'm not sure if it's necessary to say that you are using a region of size W. In my opinion, this is a technical detail that does not belong to the introduction.
- Line 68: When talking about stratiform rainfall, I would not use the word "storm".
- Line 76: Rather than being observed directly, the velocity vectors are estimated from consecutive storm objects.
- Line 170: Please explain what is the accumulation time of the gauges. Or is the quantity measured by gauges a 5-minute averaged intensity?
- Line 172-173: Taking a gauge-interpolated field instead of a radar image is highly questionable. It does not contain any information about the small-scale features, so why not completely exclude missing time stamps from the dataset?
- Lines 178-179: This is not clearly written. This seems to describe the condition for the end of an event, but it's not clear to me how the start of an event is defined.
- Lines 186-188: This is lacking essential information. Should you also mention that in such a group of pixels (grid cells), the pixels also need to be spatially connected (e.g. they have at least one neighbor in the group).

- Lines 185-194: Estimation of the storm displacements from cross-correlation between two images is described here, but should you also describe the matching of storm objects between different time steps in more detail? And how are merges and splits handled?

- Line 193: "storm is just recognized"? Do you mean that the storm does not yet have previous history?

- Line 198 and the following text: The term "state" is not precisely defined when you first introduce it, which makes its meaning unclear to the reader.
  - At lines 197 and 198, you define the state as the "spatial structure of the rainfall inside the storm boundaries". It is not clear what this means. Please elaborate.
  - I would call all the features of the object together as the state of the storm. Later (e.g. line 313) you in fact use the term in this way because you are comparing the states of the storms against each other. So, could you define the state in this way when you first introduce the term at line 197?

- Line 214: Again, I don't think that it makes much sense to use gauge-only fields as inputs. Could you just exclude time stamps with missing radar data from your dataset?

- Line 315: Table 2: Should this be Table 3?

- Equation (6): The weights Pr for the deterministic nowcast are not explicitly specified. Are they set to 1/k in this case?

- Equation (7): This should be immediately after equation (6), since the weights Pr are mentioned there for the first time.

- Equation (8): Should the MAE terms be the absolute values of the differences, and not the differences of absolute values, as it is currently written?

- Lines 385-386: It is stated that Y' is the finite first moment? Is this correct? Isn't Y' a random variable? And please define the symbol E (expectation).

- Section 4: I'm not able to follow how you compute the MAE when verifying the nowcasts. As in Section 3.2.1 (equation 8), you also need to explicitly define the MAE used for the verification in Section 3.2.2.

- Table 3: I'm not fully able to follow the notation. Why are you using the symbols I in Table 1 but in Table 3 you use PI? And what are PI_sd1 and PI_sd2?

- Lines 398-402: It is confusing that in Table 3 you show the correlation coefficients but then directly move into the predictor weights (the tables in the appendix) without explicitly explaining how the weights are obtained from the correlation coefficients. Are the former directly obtained from the latter? Please make this more clear.

- Lines 416-417: I'm not able to follow this. What rows/columns of Table 3 or the tables in the appendix are you looking at when deciding what predictors are the most important?

- Figure 8 and line 482: It is not clear to me how you minimize the ME. It is a quantity that may have arbitrarily large negative values. Are you taking absolute value somewhere in the minimization process?

- Figure 9: I'm not able to follow how you compute the MAE for a nowcast longer than 30 minutes for a storm, whose lifetime is less than 30 minutes. What are you comparing the nowcast against?

- Line 534: I'm unable to find Figure 10 in the manuscript. Where does this refer to? Should Figure 11 on page 20 be Figure 10?

- Page 22: The figures are in wrong order. Also, are the figure numbers correct?
- Lines 647-661: This discussion is beyond the scope of the current section (verification of the ensemble nowcasts). Could it be moved to Section 5?
- Line 680: It is stated that the predictability limit of the Lagrangian persistence is one hour. Please make clear what type of Lagrangian persistence are you talking about because in the introduction you give different predictability limits for different nowcast types (grid- vs. object-based).
- The appendix: The figures should have caption texts instead of placing the explanations in the subsection titles.

**Figures**

- Figure 1: The notation in the figure caption is inconsistent. In t-delta_t and t+LT, you are using notation with and without subscripts. Please use only one notation. And should the subscript 0 also be included to t in the middle and the right pane?

- Figure 3: Should this be split into separate figures? Only Figure 3a is referred in the same section with the figure. Figures b and c are defined only much later.

- Figure 5 and the caption text: You are using both t_+TL and t_t+LT. Use only either one to avoid confusion.

- Figure 5: It is not clear why you are using I with and without hat. What does the I with hat mean?

- Figure 5: Here you are using psi for the orientation angle, but in Table 1 you use phi. Please use either one to avoid confusion.

- Figure 9: Please explain the meaning of "nowcast time" more clearly. Does it mean the current lifetime of the storm when the nowcast is issued?

- Figures 11 and 12: To me it looks like that the line styles don't match the descriptions in the caption text. It is stated in the text that the probabilistic nowcasts consistently outperform the deterministic ones in terms of CRPS. This would be the case if the probabilistic nowcasts were plotted with solid lines and the deterministic nowcasts with dashed lines, which is the opposite as stated in the caption text. Also note that the labels in the legends inside the plots contradict with the caption texts.

**Technical corrections**

- Line 42: "weather forecast at several days ahead" ← "weather forecast to several days ahead"
- Line 44: "short than an hour" ← "shorter than an hour"
- Line 48: scales ← resolutions?
- Line 55: nowcast ← nowcasts
- Line 62: intermittent? Do you mean continuous?
- Line 70: nowcast ← nowcasts
- Line 101: govern ← dominate?
- Line 102: remove the word "rainfall"?

- Line 103: consulting ← utilizing?
- Line 148: at ← for
- Line 161: check the language
- Line 206: "tracking identification and algorithm" ← "tracking and identification algorithm"
- Line 209: clutters ← clutter
- Line 306: averages ← average
- Line 326: Please add subscripts i to R and Pr.
- Equation (7): In the numerator, you have Rank_i but in the denominator you have Ranki. Should the i be written as subscript?
- Figure 5, caption text: "The nowcast is issued time t_0" ← "The nowcast is issued at time t_0"
- Line 342: "specifically per each variable" ← "separately for each variable"
- line 387: enambles ← enables
- Line 454: "important analysis" ← "importance analysis"
- Line 563: introduces ← introduced
- Line 566: are dependent ← depend
- Page 22: Figure 14 ← Should this be Figure 11?
- Line 643: towards ← over
- Page 25: Figure 154 ← Should this be Figure 14?
- Line 690: Remove "of the".
- Line 699: high ← long
- Line 701: "on two measurements of similarity" ← "based on two similarity metrics"
- Line 712: "combination of the" ← remove the word "the"
- Line 725: works ← improvement
- Line 735: lightening ← lightning
- Line 736: angels ← angles

---

## Author Response (AR2)

Dear Reviewers,

We would like to thank you once again for your feedback that has improved the manuscript. Following your comments, the following changes were done in the updated version of the manuscript:

**➔ Comments from Reviewer RC1**

1. General Comments:

- The authors provide useful information in line 200 and further, but can I ask to also include the given response (see above) in the text? Perhaps something is already there and I've missed it, but if not, I think it is useful information to include.

    Lines 215-217: "The duration (or Total Lifetime) of the storm is then the lifetime of the radar pixels group as dictated by the threshold used to recognize them and the tracking algorithm that decides if the same storm is observed at continuous time steps."

- This response may be an interesting discussion point to add to the paper.

    Lines 450-454:" The Total Lifetime is an easier target to be analysed, which means the values are not zero and its distribution is not as heavy tailed as the distribution of the other variables. The other variables, depending on the lead time, have more zeros included and have an asymptotic density function. It seems that, whenever zeros are not present, like in the case of storms lasting longer than 3 hours, the PIC is able to represent quite well the important predictors."

2. Specific Comments:

- I agree, but feel free to even mention that field-based nowcasting approaches generally have trouble capturing and forecasting this well, which would stress even more the reason to focus on object-based nowcasting methods.

    Lines 70-73:" Even though the field-based approached has gained popularity recently (Ayzel et al., 2020; Imhoff et al., 2020) they still have trouble nowcasting convective storms. Thus, the focus in this study is on object-based nowcasts as they are more convenient for convective storms that typically cause urban pluvial floods."

- The authors have, indeed, increased the font size of the figures. For some figures, the font size can still be considered somewhat small. Nevertheless, this is also a process of typesetting later on. A suggestion that I would like to make, is to sometimes consider placing the figures (e.g., but not limited to, figures 7 and 8) on multiple rows. That would allow for using larger figures and that would solve the problem, too.

    I think I haven't increased the label size of Figure 7 and 8, but I can change them to the same size like Figure 9. I am currently working on that. I would like to keep the same settings (in columns) as in the other following Figures, so it is easy to associate for the reader.

3. Technical corrections:

    All noted and changed!

**➔ **Comments from Reviewer RC2**

1. General Comments:

- There is no description about how merging and splitting of storm cells are handled. The way this is done can have a significant impact on the results.

    o   Are all merged/splitted cells included in the considered storm tracks (so that the storm tracks form a tree and a storm object at a given time may consist of multiple cells)? •

    o   Or do you only consider storms that do not merge or split during their lifetime?

    I suppose you are referring here to the HyRaTrac nowcast algorithm which serve as a base for the tracking and building the storm database. Merged/Splitted Cells are included in the storm tracking, please refer to one of your specific comments to see which line in the text was changed to clarify this process better. Regarding the k-NN, splitting of the cells is (because of HyRaTrac) included indirectly in the forecast. The k-NN is able to forecast splits if past neighbours have splatted during their lifetime. The Merging is also possible with the k-NN but needs an extra step (implemented in part 2 for rainfall intensities): two storms are predicted at the same region, and thus they are superimposed and considered as a merged storm.

2. Specific Comments:

- Manuscript title: since you now have "Part I" in the title, you should have more explicit discussion in the conclusions what would "Part II", or even "Part III" include.

    Lines 753-769: "Improving the nowcasting of storm characteristics is the first step in improving rainfall nowcasting at fine temporal and spatial scales. On a second step, the knowledge about the storm characteristics (as nowcasted by the 30NNs) should be implemented on the spatial structure of the storms to estimate rainfall intensities at fine scales (1km$^2$ and 5min). There are two options to deal with the spatial distribution of the rainfall intensities inside the storm region (which is so far not treated in this study): 1. Increase/Reduce the area by the given nowcasted area (as target variable) for each lead time, scale the average intensity with the nowcasted intensity, and move the position of the storm in the future with the nowcasted velocity in x and y direction. 2. Take the spatial information of the selected neighbours, perform an optimisation in space (such that present storm and the neighbour's storms locations match) and assign this spatial information to the present storm for each lead time. The former is an extension of the target-based 30NNs, while the later an extension of the storm-based 30NNs. So far, the comparison between these two versions, showed that the target-based approach is better suited mainly to nowcast the velocity components, thus a merging of the two could also be reasonable: the storm-based approach is used for nowcasting Area-Intensity-Total Lifetime (features that are co-dependent based on the life cycle characteristics of convective storms), and the target-based approach for the nowcasting of the velocity components. Future works (Part II – Local Intensities) will include the integration of the developed 30NNs application in the object-oriented radar based nowcast to extend the rainfall predictability limit at fine spatial and temporal scales (1km$^2$ and 5min). The main focus of the Part II is to investigate if the methodology applied here can introduce improvements as well at the local scale, i.e. validation with the measurements from the rain gauge observations."

- Line 7: erratic ⟵ unpredictable?
    Noted and changed!

- Lines 44 and 57: I'm not sure if these claims are completely true for state of the art rapid-update limited-area NWP models that could be applicable to urban-scale nowcasting. Are there any more recent references about this topic?

    Lines 41-44: "The Numerical Weather Prediction Models (NWP) are typically used in hydrology for weather forecast to several days ahead, nevertheless they are not suitable for urban modelling as they still cannot produce reliable and accurate intensities for spatial scales smaller than $10km^2$ and temporal time steps shorter than an hour (Kato et al., 2017; Surcel et al., 2015)."

- Line 46: You could add that the inability to capture the spatial structure of rainfall is due to the sparsity of the existing rain gauge networks.

    Lined 44-46: "Ground rainfall measurements (rain-gauges) are considered the true observation of rainfall but they are as well not adequate for QPFs because, due to the sparsity of the existing rain-gauge networks, they cannot capture the spatial structure of rainfall."

- Line 67: I'm not sure if it's necessary to say that you are using a region of size W. In my opinion, this is a technical detail that does not belong to the introduction.

    Noted and changed!

- Line 68: When talking about stratiform rainfall, I would not use the word "storm".

    Noted and changed!

- Line 76: Rather than being observed directly, the velocity vectors are estimated from consecutive storm objects.

    Lines 76-77: "… and velocities are assigned from consecutive storm objects…"

- Line 170: Please explain what is the accumulation time of the gauges. Or is the quantity measured by gauges a 5-minute averaged intensity?

    Lines 171-172: "…while the rain-gauges measure the rainfall intensities at 1min temporal resolution but are aggregated to 5min time steps…"

- Line 172-173: Taking a gauge-interpolated field instead of a radar image is highly questionable. It does not contain any information about the small-scale features, so why not completely exclude missing time stamps from the dataset?

    They are excluded from the application of k-NN, because their total lifetime (or duration) is only 5min. All storms recognized as single time steps are removed from the k-NN application.

- Lines 178-179: This is not clearly written. This seems to describe the condition for the end of an event, but it's not clear to me how the start of an event is defined.

    Lines 181-182: "… The start and the end of the rainfall event is determined when areal mean radar intensity is higher/lower than 0.05mm for more than 4 hours…"

- Lines 186-188: This is lacking essential information. Should you also mention that in such a group of pixels (grid cells), the pixels also need to be spatially connected (e.g. they have at least one neighbour in the group)

    Lines 189-190: "… A storm is initialized if a group of spatially connected radar grid cells (> 64) has a reflectivity higher than Z=20dBz…"

- Lines 185-194: Estimation of the storm displacements from cross-correlation between two images is described here, but should you also describe the matching of storm objects between different time steps in more detail? And how are merges and splits handled?

    Lines 195-203: "…Once storms at different time steps are recognized, they are matched as evolution of a single storm, if the centre of intensity of storm at t=0 falls within the boundary box of the storm at t-5 min. The tracking of individual storms in consecutive images is done by the cross-correlation optimization between the last 2 images (t=0 and t-5 min), and local displacement vectors for each storm are calculated. In case a storm is just recognized (the storm does not yet have previous history), then global displacement vectors based on crosscorrelation of the entire radar image are assigned to them. It is usually the case, that two storms merge together at a certain time, or a single storm splits between several daughter storms. The splitting and merging of the storms is considered here if two criteria are met: a) the minimum distance between the storms that have splatted or merged is smaller than the perimeter of the merged or that-is-splitting storm, and b) the position of the centre of intensity of former/latter storms is within the boundaries of the latter/former storm."

- Line 193: "storm is just recognized"? Do you mean that the storm does not yet have previous history?

  Lines 198-199: "…(the storm does not yet have previous history)…"

- Line 198 and the following text: The term "state" is not precisely defined when you first introduce it, which makes its meaning unclear to the reader.

  Lines 205-207: "…and for each time step of the storm evolution the spatial information is saved and various features are calculated. Here the features computed from the spatial information of the rainfall inside the storm boundaries at a given time step (in 5min) of the storms' life, is referred to as the "state" of the storm…"

- At lines 197 and 198, you define the state as the "spatial structure of the rainfall inside the storm boundaries". It is not clear what this means. Please elaborate.

  Here we meant the spatial information, but the sentence was updated (see above) to avoid the confusion.

- I would call all the features of the object together as the state of the storm. Later (e.g. line 313) you in fact use the term in this way because you are comparing the states of the storms against each other. So, could you define the state in this way when you first introduce the term at line 197?

  Noted and changed! Please see the two points before.

- Line 214: Again, I don't think that it makes much sense to use gauge-only fields as inputs. Could you just exclude time stamps with missing radar data from your dataset?

  They are already excluded from the application of k-NN, because they live only 5min. All storms recognized for single time steps are removed from the k-NN application.

- Line 315: Table 2: Should this be Table 3?

  Noted and changed!

- Equation (6): The weights Pr for the deterministic nowcast are not explicitly specified. Are they set to 1/k in this case?

  Line 338: "… where k is the number of neighbours obtained from optimization, Ri and Pri (from Equation 7) are respectively the response and weight of the ith neighbour…"

- Equation (7): This should be immediately after equation (6), since the weights Pr are mentioned there for the first time.

  Please see the point above.

- Equation (8): Should the MAE terms be the absolute values of the differences, and not the differences of absolute values, as it is currently written?

  I'm sorry for the confusion, but yes you are right and Equation 8 has been updated accordingly.

- Lines 385-386: It is stated that Y' is the finite first moment? Is this correct? Isn't Y' a random variable? And please define the symbol E (expectation).

  Line 339-400: "… where F is a probabilistic forecast, y the observed value, Y and Y´ independent random variables with CDF of F and finite first moment E (Gneiting and Katzfuss, 2014)."

- Section 4: I'm not able to follow how you compute the MAE when verifying the nowcasts. As in Section 3.2.1 (equation 8), you also need to explicitly define the MAE used for the verification in Section 3.2.2.

Lines 389-392; "i) absolute error per lead time and target variable computed for each event and for a specific selected nowcast time

$$MAE_{target} = \sum_{i=1}^{N}(|Pred_{i,+LT} - Obs_{i,+LT}|)/N \ ,$$

where the $P_{red}$ is the predicted response, Obs the observed response for the $i^{th}$ storm, +LT the lead time and N the number of storms considered inside an event."

- Table 3: I'm not fully able to follow the notation. Why are you using the symbols I in Table 1 but in Table 3 you use PI? And what are PI_sd1 and PI_sd2?

  I'm sorry for the confusion. The Table 1 and 3 notations (also in the Appendix) are now changed to be consistent with one another.

- Lines 398-402: It is confusing that in Table 3 you show the correlation coefficients but then directly move into the predictor weights (the tables in the appendix) without explicitly explaining how the weights are obtained from the correlation coefficients. Are the former directly obtained from the latter? Please make this more clear.

  Lines 291-292: "…Here, the Pearson correlation absolute values are used directly as predictors weights in the k-NN application."

- Lines 416-417: I'm not able to follow this. What rows/columns of Table 3 or the tables in the appendix are you looking at when deciding what predictors are the most important?

  Lines 437-440: "Hence based on the Pearson correlation values from Table 3 the following most important predictors were selected: Area –A (as maximum correlation value from first row), Intensity –PIsd1 (as maximum correlation value from second row), - Velocity X – Vx30 (as maximum correlation value from third row), Velocity Y –Vy30 (as maximum correlation value from fourth row),Total Lifetime – A (as maximum correlation value from fifth row)."

- Figure 8 and line 482: It is not clear to me how you minimize the ME. It is a quantity that may have arbitrarily large negative values. Are you taking absolute value somewhere in the minimization process?

  Lines 369-374: "…The objective function is the minimization of the mean absolute error (Equation 8) and of the absolute mean error (Equation 9) between predicted and observed target variables at lead times from +5min to +180 min:

$$MAE_{target} = \sum_{i=1}^{N}(|Pred_{i,+LT} - Obs_{i,+LT}|)/N \ , \qquad (1)$$

$$ME_{target} = |\sum_{i=1}^{N}(Pred_{i,+LT} - Obs_{i,+LT})/N| \ , \qquad (2)$$

where the Pred is the predicted response, Obs the observed response for the ith storm, +LT the lead time and N the number of storms considered inside an event"

- Figure 9: I'm not able to follow how you compute the MAE for a nowcast longer than 30 minutes for a storm, whose lifetime is less than 30 minutes. What are you comparing the nowcast against?

  Lines 413-416: "Lastly, it is important to notice, that the performance criteria can be calculated even for nowcast times longer than the storm lifetime, if the nowcast fails to capture the dissipation of the storms. In this case, Area, Intensity, Velocity in X and Y Direction are compared against zero, while the Total Lifetime against the total observed lifetime of the storms."

- Line 534: I'm unable to find Figure 10 in the manuscript. Where does this refer to? Should Figure 11 on page 20 be Figure 10?

  Yes! The Figure numbering has been correctly updated.

- Page 22: The figures are in wrong order. Also, are the figure numbers correct?

  Yes! The Figure numbering has been correctly updated.

- Lines 647-661: This discussion is beyond the scope of the current section (verification of the ensemble nowcasts). Could it be moved to Section 5?

  Noted and changed!

- Line 680: It is stated that the predictability limit of the Lagrangian persistence is one hour. Please make clear what type of Lagrangian persistence are you talking about because in the introduction you give different predictability limits for different nowcast types (grid- vs. object-based).
  - Lines 691-692: "…and more importantly the extrapolation of the storms in the future based on the Lagrangian persistence, are limiting the forecast horizons of such object-oriented radar based nowcasts to 30-45 min for convective storms and to 1 hour for stratiform events (Shehu & Haberlandt, 2021)."
- The appendix: The figures should have caption texts instead of placing the explanations in the subsection titles.
  - **Appendix 8-1** Strength of relationship between the selected predictors and the target variables averaged for three lead times and storm duration groups computed from Pearson correlation. The green shade indicates the strength of the relationship: with 0 for no relationship at all, and 1 for highest dependency. The averaged computed values for each target variable (last row) are used as input for Table 3. The correlation weights are absolute values of the correlation values between the predictors at specific lead times and target variables.
  - **Appendix 8-2** Strength of relationship between the selected predictors and the target variables averaged for three lead times and storm duration groups computed from PIC method. The green shade indicates the strength of the relationship: with 0 for no relationship at all, and 1 for highest dependency. The averaged computed values for each target variable (last row) are used as input for Table 3. For intensity, velocity in x and y direction, since the PIC recognized only one predictor as important, the average values is fixed as 1 for the respective predictor.
  - **Appendix 8-3** The standard deviation of the Pearson Correlation Weights between predictors and target variables obtained from a cross-sampling of the events (leave one event out). The boxplot for each target variable describes the spread of the standard deviation over all selected predictors.

3. Figures:
- Figure 1: The notation in the figure caption is inconsistent. In t-delta_t and t+LT, you are using notation with and without subscripts. Please use only one notation. And should the subscript 0 also be included to t in the middle and the right pane?
  - Noted and changed!
- Figure 3: Should this be split into separate figures? Only Figure 3a is referred in the same section with the figure. Figures b and c are defined only much later.
  - I understand your concern, but I would like to keep it like this because it saves place, and keeps all the information at one place. I hope this won't bother the reader too much.
- Figure 5 and the caption text: You are using both $t_{+TL}$ and $t_{t+LT}$. Use only either one to avoid confusion.
  - Noted and changed!
- Figure 5: It is not clear why you are using I with and without hat. What does the I with hat mean?
  - I have changed it to simply I.
- Figure 5: Here you are using psi for the orientation angle, but in Table 1 you use phi. Please use either one to avoid confusion.
  - I have updated the Figure and used only phi.
- Figure 9: Please explain the meaning of "nowcast time" more clearly. Does it mean the current lifetime of the storm when the nowcast is issued?
  - The following explanation was added to the caption of the figure: "… Nowcast time dictates when the nowcast is issued relative to storm initiation."

- Figures 11 and 12: To me it looks like that the line styles don't match the descriptions in the caption text. It is stated in the text that the probabilistic nowcasts consistently outperform the deterministic ones in terms of CRPS. This would be the case if the probabilistic nowcasts were plotted with solid lines and the deterministic nowcasts with dashed lines, which is the opposite as stated in the caption text. Also note that the labels in the legends inside the plots contradict with the caption texts.

  Yes indeed, the caption was wrong and it has been updated accordingly.

4. Technical corrections:

   All Noted and changed!